# Revisiting Consensus Error: A Fine-grained Analysis of Local SGD under Second-order Data Heterogeneity

**Kumar Kshitij Patel**[*]
Institute for Foundations of Data Science
Yale University
New Haven, CT 06511
kumarkshitij.patel@yale.edu

**Ali Zindari**
CISPA Helmholtz Center
for Information Security
Saarbrücken, DE 66123
ali.zindari@cispa.de

**Sebastian U. Stich**
CISPA Helmholtz Center
for Information Security
Saarbrücken, DE 66123
stich@cispa.de

**Lingxiao Wang**
Department of Data Science
New Jersey Institute of Technology
Newark, NJ 07102
lw324@njit.edu

## Abstract

Local SGD, or Federated Averaging, is one of the most widely used algorithms for distributed optimization. Although it often outperforms alternatives such as mini-batch SGD, existing theory has not fully explained this advantage under realistic assumptions about data heterogeneity. Recent work has suggested that a second-order heterogeneity assumption may suffice to justify the empirical gains of local SGD. We confirm this conjecture by establishing new upper and lower bounds on the convergence of local SGD. These bounds demonstrate how a low second-order heterogeneity, combined with third-order smoothness, enables local SGD to interpolate between heterogeneous and homogeneous regimes while maintaining communication efficiency. Our main technical contribution is a refined analysis of the consensus error, a central quantity in such results. We validate our theory with experiments on a distributed linear regression task.

## 1 The Unreasonable Effectiveness of Local SGD

We study the following distributed optimization problem over $M$ machines:

$$\min_{x \in \mathbb{R}^d} \left( F(x) := \frac{1}{M} \sum_{m \in [M]} F_m(x) \right), \tag{1}$$

where $F_m := \mathbb{E}_{z_m \sim \mathcal{D}_m}[f(x; z_m)]$ is the a stochastic optimization objective on machine $m$, defined using a loss function $f(\cdot; z \in \mathcal{Z})$ and a data distribution $\mathcal{D}_m \in \Delta(\mathcal{Z})$. Problem (1) appears widely in machine learning—ranging from multi-GPU training in data centers [1] to decentralized training on millions of edge devices [2, 3]. Perhaps the simplest, most basic, and most important distributed setting for solving Problem (1) is that of *intermittent communication* (IC) [4]. In this model, illustrated in Figure 1, $M$ machines optimize the objective across $R$ rounds of communication, and in each round, each machine performs $K$ sequential stochastic gradient updates before communicating.

Several variants of stochastic gradient descent (SGD) have been proposed for the IC setting [5, 6], most of which build on **Local SGD** or **Federated Averaging**. In Local SGD, each machine performs

---

[*]Part of the work was done when the author was a student at Toyota Technological Institute, Chicago (TTIC).

39th Conference on Neural Information Processing Systems (NeurIPS 2025).

$K$ local stochastic updates starting from the last synchronized model and then at the communication round the machines average their models. Specifically, denoting the overall time by $T = KR$, at time step $t \in [0, T-1]^2$, machine $m \in [M]$ samples $z_t^m \sim \mathcal{D}_m$ and performs the update:

$$x_{t+1}^m := x_t^m - \eta \nabla f(x_t^m; z_t^m) \qquad \text{if} \quad t+1 \mod K \neq 0 \ ,$$

$$x_{t+1}^m := \frac{1}{M} \sum_{n \in [M]} (x_t^n - \eta \nabla f(x_t^n; z_t^n)) \quad \text{if} \quad t+1 \mod K = 0 \ , \qquad (2)$$

with initialization $x_0^m = 0$ for all $m \in [M]$. Despite its simplicity, Local SGD consistently outperforms other first-order methods in practice [7, 8], including mini-batch SGD [9, 10, 11]. This strong empirical performance has motivated more than a decade of theoretical work aimed at understanding its advantages [12, 13, 14, 15, 16, 17, 18, 11, 19, 20, 21, 22, 23, 8, 24, 25, 26].

In the homogeneous setting where $\mathcal{D}_m = \mathcal{D}$ for all $m \in [M]$, a sequence of results [16, 11, 21, 23, 27, 25, 28] has shown that to explain the benefit of Local SGD, one must impose higher-order smoothness assumptions—à la quadraticity in linear regression or quasi-self-concordance in logistic regression. However, in the more general and interesting heterogeneous case, most existing works either fail to demonstrate an advantage of Local SGD over mini-batch SGD [17, 19, 18], or require highly restrictive conditions that prevent any

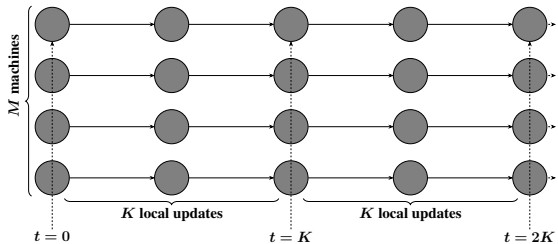

Figure 1: Illustration of the intermittent communication setting.

meaningful data variation across machines (c.f. Assumption 7). Wang et al. [8] described this discrepancy between theory and practice as the *unreasonable effectiveness of Local SGD*.

Recently, Patel et al. [25] conjectured that combining a second-order heterogeneity condition (c.f. Assumption 6) with standard first-order conditions (c.f. Assumptions 4 and 5) can account for the success of Local SGD without requiring overly restrictive assumptions. Concurrently, other works have shown that these second-order conditions sharply characterize the communication complexity of various other distributed optimization algorithms [29, 30, 31, 32, 33, 34, 35]. Motivated by these observations, we aim to validate the conjecture of Patel et al. [25] and show that a low second-order heterogeneity indeed improves the convergence and communication complexity of Local SGD.

Towards this goal, our main contributions are as follows:

**I. New Lower Bound Characterizing Second-order Heterogeneity.** In Theorem 1, we refine the lower bound of Patel et al. [25] to accommodate second-order heterogeneity. Specifically, we show that low second-order heterogeneity reduces the hardness of their lower bound instance. This refinement suggests that the communication complexity of Local SGD depends primarily on the second-order heterogeneity constant, rather than the more conservative second-order smoothness constant (c.f., $\tau$ vs. $H$ in Assumptions 1 and 6).

**II. New Upper Bounds without Restrictive Assumptions.** In Theorems 2 and 3, we establish upper bounds that align with the qualitative predictions of our lower bound: Local SGD converges faster as second-order heterogeneity decreases. These bounds also incorporate the effects of first-order heterogeneity (Assumptions 4 and 5), allowing us to interpolate smoothly between heterogeneous and homogeneous regimes. Our key technical contribution is a new, sharper analysis of the consensus error—a central quantity in distributed optimization—which enables us to avoid the restrictive heterogeneity assumptions used in prior work [20, 25].

**III. Refined Analysis under Third-order Smoothness.** We improve our upper bound for quadratic objectives in Theorems 4 and 5. We then extend the analysis to third-order smooth functions in Theorem 6, requiring control of the fourth moment of the consensus error and careful handling of four coupled recursions without losing the benefits of local updates.

**IV. A Controlled Regression Experiment.** We validate our theory on a synthetic linear regression task in Figure 2 by independently varying first- and second-order heterogeneity and highlighting the role of second-order heterogeneity in dictating Local SGD's communication complexity.

---

[2] Throughout the paper, for integers $i \leq j \in \mathbb{Z}_{\geq 0}$, we will use the notation $[i, j] := \{i, i+1, \ldots, j-1, j+1\}$, and when $i = 1$, we denote it by $[j]$.

Thus, our paper significantly shrinks several gaps in the convergence theory for Local SGD. We summarize some of our results in Table 1, contextualizing them in existing literature. Our techniques may also be useful for other areas of distributed optimization where consensus error-like quantities arise, including communication compression [36, 37, 38], quantization [39, 40], asynchronous updates [41, 42], differential privacy [43, 44], and Byzantine robustness [45, 46].

## 2 Setting and Preliminaries

In this section, we introduce our notation and assumptions while discussing several related works.

**Local SGD Iterates.** For analytical purposes, we will define the average Local SGD iterate at time $t \in [0, T]$ by $x_t := \frac{1}{M} \sum_{m \in [M]} x_t^m$, which may not be computed in practice (when $t \mod K \neq 0$).

**Regularity Assumptions.** We assume that the local objectives are convex and smooth.

**Assumption 1** (Convexity and Smoothness)**.** *For all $m \in [M]$, the function $F_m(\cdot)$ is twice differentiable and satisfies $\mu \cdot I_d \preceq \nabla^2 F_m(\cdot) \preceq H \cdot I_d$ for some $0 \leq \mu \leq H$. When $\mu > 0$, we say $F_m$ is strongly convex and denote its condition number by $\kappa = \frac{H}{\mu}$. Furthermore, there exists $Q \geq 0$ such that for all $x, y \in \mathbb{R}^d$, we have $\left\| \nabla^2 F_m(x) - \nabla^2 F_m(y) \right\|_2 \leq Q \cdot \|x - y\|_2$*[3].

Recall that a strongly convex function admits a unique minimizer. Also, $Q = 0$ implies that $F_m$ is quadratic. We further discuss the role of third-order smoothness in Section 5.

We also assume the stochastic gradients have bounded fourth moments.

**Assumption 2** (Bounded Fourth Moment of Stochastic Gradients)**.** *For all $m \in [M]$ and $x \in \mathbb{R}^d$, we have $\mathbb{E}_{z \sim \mathcal{D}_m}[\nabla f(x; z)] = \nabla F_m(x)$, and*

$$\mathbb{E}_{z \sim \mathcal{D}_m}[\|\nabla f(x; z) - \nabla F_m(x)\|_2^4 \mid x] \leq \sigma^4 \ .$$

Using Jensen's inequality, the above assumption implies the second moment of the stochastic gradients are also bounded, i.e., $\mathbb{E}_{z \sim \mathcal{D}_m}[\|\nabla f(x; z) - \nabla F_m(x)\|_2^2 \mid x] \leq \sigma^2$. We only require the fourth moment bound for Theorem 6, which involves a higher-order moment control of iterates. For all other results (Theorems 2 to 5), the second moment bound suffices.

Finally, we assume that each local function's optima and the global objective are bounded.

**Assumption 3** (Bounded Optima)**.** *For all $m \in [M]$, define the set of optima as $S_m^\star := \arg\min_{x \in \mathbb{R}^d} F_m(x)$. Then $\exists x_m^\star \in S_m^\star$ such that $\|x_m^\star\|_2 \leq B$. Similarly, define the set of optima for the average objective as $S^\star := \arg\min_{x \in \mathbb{R}^d} F(x)$. Then $\exists x^\star \in S^\star$ such that $\|x^\star\|_2 \leq B$.*

These standard regularity assumptions alone are insufficient to establish the utility of a large number of local update steps for optimizing Problem (1) [20, 18, 25]. To address this, we now introduce heterogeneity assumptions that control how the data distributions across machines are related.

**Heterogeneity Assumptions.** The most natural assumption linking the optimization problems across different machines is that their optima are close. In particular, we assume that at least one pair of optima from two machines is close relative to their norms.

**Assumption 4** (Distance between Clients' Optima)**.** *There exists $\zeta_\star \leq 2B$ such that,*

$$\sup_{m,n \in [M]} \inf_{x_m^\star \in S_m^\star, \ x_n^\star \in S_n^\star} \|x_m^\star - x_n^\star\|_2 \leq \zeta_\star \ .$$

When all machines share a common minimizer, we have $\zeta_\star = 0$, and solving Problem (1) recovers this global optimum. However, when machines do not share an optimizer, we must additionally assume that at least one minimizer of the average objective is approximately optimal for each machine. Without this condition, some clients may not benefit from collaboration.

**Assumption 5** (Distance between Clients' and the Average Objective's Optima)**.** *There exists $\phi_\star \leq 2B$ such that*

$$\sup_{m \in [M]} \inf_{x_m^\star \in S_m^\star, \ x^\star \in S^\star} \|x_m^\star - x^\star\|_2 \leq \phi_\star \ .$$

---

[3]We use a single assumption for convexity and smoothness, but in our results we will denote $\mu = 0$ to emphasize when we are considering general convex (as opposed to strongly convex) functions and when we do not want to capture third-order smoothness in our upper bounds, we will interpret the assumption with $Q = \infty$.

| Reference | Function Sub-optimality Bound | Restrictions |
|---|---|---|
| **Strongly Convex Setting** ($\mu > 0$) | | |
| Kolosokova et al. [18] (Upper bound) | $\frac{\sigma^2}{\mu MKR} + \frac{H^3\phi_\star^2}{\mu^2 R^2} + \frac{H\sigma^2}{\mu^2 KR^2}$ | $R = \tilde{\Omega}\left(\frac{H}{\mu}\right)$ |
| Woodworth et al. [20] (Upper Bound) | $\frac{HB^2}{HKR+\mu K^2 R^2} + \frac{\sigma^2}{\mu MKR} + \frac{H^3\zeta^2}{\mu^2 R^2} + \frac{H\sigma^2}{\mu^2 KR^2}$ | - |
| Patel et al. [25] (Upper Bound) | $e^{-\frac{\mu KR}{H}}HB^2 + \frac{\sigma^2}{\mu MKR} + \frac{\tau^2\sigma^2}{\mu^3 KR^2} + \frac{\tau^2 H^2\zeta^2}{\mu^3 R^2} +$ $\frac{Q^2\sigma^4}{\mu^5 K^2 R^4} + \frac{Q^2 H^4 \zeta^4}{\mu^5 R^4}$ | - |
| Theorem 3 (Upper Bound) | $e^{-\frac{\mu KR}{2H}}\mu B^2 + \frac{\sigma^2}{\mu MKR} + \frac{\tau^2 H\phi_\star^2}{\mu^2 R^2} + \frac{H^3\zeta_\star^2}{\mu^2 R^2} + \frac{H\tau^2\sigma^2}{\mu^4 KR^3} +$ $\frac{H\sigma^2}{\mu^2 KR^2}$ | $R = \tilde{\Omega}\left(\frac{\tau\sqrt{\kappa}}{\mu}\right)$ |
| **Convex Setting** ($\mu = 0$) | | |
| Kolosokova et al. [18] (Upper bound) | $\frac{HB^2}{R} + \frac{\sigma B}{\sqrt{MKR}} + \frac{H\phi_\star^{2/3}B^{4/3}}{R^{2/3}} + \frac{H^{1/3}\sigma^{2/3}B^{4/3}}{K^{1/3}R^{2/3}}$ | - |
| Woodworth et al. [20] (Upper Bound) | $\frac{HB^2}{KR} + \frac{\sigma B}{\sqrt{MKR}} + \frac{H\zeta^{2/3}B^{4/3}}{R^{2/3}} + \frac{H^{1/3}\sigma^{2/3}B^{4/3}}{K^{1/3}R^{2/3}}$ | - |
| Patel et al. [25] (Upper Bound) | $\frac{HB^2}{R} + \frac{\sigma B}{\sqrt{MKR}} + \frac{(\tau\sigma B^3)^{1/2}}{K^{1/4}R^{1/2}} + \frac{(\tau H\zeta B^3)^{1/2}}{R^{1/2}} +$ $\frac{(Q\sigma^2 B^5)^{1/3}}{K^{1/3}R^{2/3}} + \frac{(QH^2\zeta^2 B^5)^{1/3}}{R^{2/3}}$ | - |
| Theorem 1 (Lower Bound) | $\frac{\tau B^2}{R} + \frac{HB^2}{KR} + \frac{\sigma B}{\sqrt{MKR}} + \min\left\{\tau\phi_\star^2, \frac{\tau\phi_\star^{2/3}B^{4/3}}{R^{2/3}}\right\} +$ $\min\left\{\frac{\sigma B}{\sqrt{KR}}, \frac{H^{1/3}\sigma^{2/3}B^{4/3}}{K^{1/3}R^{2/3}}\right\}$ | - |

Table 1: Summary of existing and (a subset of) our convergence guarantees for function-value suboptimality for convex and strongly convex settings (ignoring poly-logarithmic factors). The red terms are the ones that determine the communication complexity, as these terms can not be made arbitrarily small even with a very large $K$. Notably, there is no relevant lower bound in the strongly convex setting under the assumptions of our upper bounds, and there are gaps between the upper and lower bounds in the general convex setting.

Most existing first-order heterogeneity conditions are variants of Assumption 5. The quantity $\phi_\star$ captures the notion of *approximate simultaneous realizability* across clients and has also appeared in the literature on collaborative PAC learning and incentives for Federated Learning [47, 48, 49, 50].

**Remark 1** ($\zeta_\star$ vs. $\phi_\star$). With $\mu > 0$, Assumption 5 implies Assumption 4 with $\zeta_\star \leq 2\phi_\star$. However, the reverse is not true in general: there exist problems that satisfy Assumption 4 with $\zeta_\star << B$, yet only satisfy Assumption 5 when $\phi_\star \approx B$. For this reason, we distinguish between these two assumptions. See the discussion in Appendix A for more context on heterogeneity assumptions.

We also impose the following second-order heterogeneity assumption, which bounds how different the Hessians can be across machines.

**Assumption 6** (Bounded Second-order Heterogeneity). *There exists $\tau \leq 2H$ such that,*

$$\sup_{m,n\in[M]} \sup_{x\in\mathbb{R}^d} \left\|\nabla^2 F_m(x) - \nabla^2 F_n(x)\right\|_2 \leq \tau \ .$$

Note that the above assumption can always be satisfied by using $\tau = 2H$ and Assumption 1. $\tau$ corresponds to the second-order smoothness of the function $F_m(\cdot) - F_n(\cdot)$ for any pair of machines $m, n \in [M]$. This observation will enable us to replace $H$ with $\tau$ in several places.

Several recent works have established that Assumption 6 plays a central role in determining the communication complexity of distributed optimization. A prominent line of research has focused on distributed proximal-point methods. Early results in the quadratic setting showed that when $\tau = 0$, these methods can achieve *extreme communication efficiency*, requiring only a constant number of communication rounds [51]. More recent works extended these guarantees to the general case with $\tau > 0$ [32, 33, 34, 35]. In the context of Local SGD, Patel et al. [31] showed that a variance-reduced variant achieves similar communication efficiency in the non-convex setting when $\tau = 0$, and that the optimal communication complexity scales linearly with $\tau$ when the number of local updates $K$ is large. Later, Patel et al. [25] studied Local SGD in the convex setting and showed that it can also be extremely communication-efficient, providing convergence guarantees in terms of $\tau$. However, their analysis relied on the restrictive heterogeneity Assumption 7 [4]. They also conjectured that Assumptions 4 to 6 together are both necessary and sufficient to establish the communication efficiency and dominance of Local SGD over baselines like mini-batch SGD for convex problems. However, convergence guarantees under broader conditions—such as non-quadratic objectives——as well as lower bounds that decay gracefully with $\tau$, have remained open.

**Remark 2** ($\zeta_\star$ and $\tau$ vs. $\phi_\star$). In some settings, the parameter $\phi_\star$ in Assumption 5 can be bounded using $\zeta_\star$ and $\tau$ from Assumptions 4 and 6. For example, if each $F_m$ is a strongly convex quadratic function with Hessian $A_m$ and unique minimizer $x_m^\star$, then the global optimum satisfies

$$x^\star = A^{-1} \cdot \frac{1}{M} \sum_{m \in [M]} A_m x_m^\star \ ,$$

where $A := \frac{1}{M} \sum_{m \in [M]} A_m$. Using this, we can derive [25] (see proof in Appendix A):

$$\|x_m^\star - x^\star\|_2 \le \|x_m^\star - \bar{x}^\star\|_2 + \|\bar{x}^\star - x^\star\|_2 \le \zeta_\star + \frac{\tau \zeta_\star}{\mu} \ , \quad \forall m \in [M] \ ,$$

where $\bar{x}^\star := \frac{1}{M} \sum_m x_m^\star$. Hence, we can set $\phi_\star = \zeta_\star(1 + \tau/\mu)$. For general non-quadratic problems, however, it may not be possible to eliminate the dependence on $\phi_\star$.

## 3  A New General Convex Lower Bound with Second-order Heterogeneity

Our first result is the following convergence lower bound for general convex functions, explicitly capturing second-order heterogeneity $\tau$ along with first-order heterogeneity $\phi_\star$.

**Theorem 1.** *There exists a quadratic problem instance satisfying Assumptions 1 to 6 (with $\mu = 0$) such that for any choice of step-size $\eta$, Local SGD initialized at $x_0 = 0$ outputs a model $x_{KR}$ with:*

$$\mathbb{E}\left[F(x_{KR})\right] - F(x^\star) = \Omega\left( \frac{\tau B^2}{R} + \frac{HB^2}{KR} + \frac{\sigma B}{\sqrt{MKR}} + \min\left\{ \frac{\sigma B}{\sqrt{KR}}, \frac{H^{1/3}\sigma^{2/3}B^{4/3}}{K^{1/3}R^{2/3}} \right\} \right.$$
$$\left. + \min\left\{ \tau\phi_\star^2, \frac{\tau\phi_\star^{2/3}B^{4/3}}{R^{2/3}} \right\} \right) \ .$$

We prove this theorem in Appendix C. When $\phi_\star$ is small and $K$ is large, the lower bound is dominated by the term $\frac{\tau B^2}{R}$, which suggests that the communication complexity of Local SGD should scale as $\frac{\tau B^2}{\epsilon}$—a rate that mirrors known results in the non-convex setting [30, 31].

**Comparison to Existing Lower Bounds.**  The strongest existing lower bound for Local SGD under the first-order heterogeneity assumption $\phi_\star$ is due to Patel et al. [25]. However, their result does not incorporate second-order heterogeneity as in Assumption 6. As the authors note, their hard instance degenerates when $\tau = 0$ because the smoothness constant $H$ also vanishes, making the instance trivial. To address this, we extend the construction of Patel et al. [25] by introducing an additional dimension, which decouples the effects of $\tau$ and $H$ on the convergence behavior of Local SGD. This modification ensures that our lower bound reduces to the bound of Glasgow et al. [23] when $\tau = 0$—a bound that is known to be tight in the homogeneous setting.

---

[4]Patel et al. [25] also showed that Local SGD converges rapidly to its fixed point and exhibits no fixed-point discrepancy for quadratics when $\tau = 0$, yielding extreme communication efficiency (see Appendix B).

**Potential Future Improvements.** We note that our lower bound does not depend on $\zeta_\star$, and $\phi_\star$ alone captures the hardness of first-order heterogeneity. This is because for our hard instance $\zeta_\star \approx \phi_\star$, which is why we choose to state the result in terms of $\phi_\star$, which is usually bigger than $\zeta_\star$ (see Remarks 1 and 2). Deriving a lower bound that can decouple the dependence on $\zeta_\star$ and $\phi_\star$, i.e., decouple the proximity of machines' optima and "fixed-point discrepancy", remains an open question. Also, we suspect that in the last term of the lower bound, $\tau$ can be replaced by $H$. Finally, all quadratic lower bounds for Local SGD [20, 23, 25] do not use unbounded fourth moments. Thus, we do not know if under a weaker second-moment variant of Assumption 2 the lower bound can be improved by using higher moments of the noise to "confuse" the local updates.

In the following section, we will prove new upper bounds that exhibit qualitatively similar behavior to Theorem 1 in regimes with low data heterogeneity and large $K$, reinforcing the role of $\tau$, $\zeta_\star$, and $\phi_\star$ in governing the performance of Local SGD.

## 4    Breaking Down the Consensus Error and New Upper Bounds

In this section, we first present our result on convergence in iterates in the strongly convex setting in Theorem 2, and then on convergence in function values in Theorem 3. Both our results will hold even when $Q = \infty$ in Assumption 1, and as such the goal of this section is to highlight the effect of a low second-order heterogeneity. We will focus here on the key ideas used to derive Theorem 2; the proof of Theorem 3 is morally similar, and deferred to Appendix I.2.

Our analysis proceeds in three stages. We begin by introducing a standard one-step progress result in Lemma 1, which quantifies the improvement of Local SGD in terms of the *consensus error*——a quantity that measures the deviation between local and global iterates and plays a central role in the analysis of many distributed optimization algorithms. We then identify the two main issues in the existing consensus error bounds: (i) they rely on restrictive assumptions [20, 25]; and (ii) they do not characterize the effect of second-order heterogeneity. To address both these issues we establish a new upper bound on the consensus error in Lemma 2, that only depends on Assumptions 4 to 6. Finally, we substitute this bound into the progress lemma and unroll the resulting recursion to obtain convergence guarantees for both strongly convex and general convex objectives. These results reveal how the convergence of Local SGD depends on the data heterogeneity parameters $\tau$, $\zeta_\star$, and $\phi_\star$, and highlight the algorithm's communication efficiency in regimes of low data heterogeneity and large $K$.

**Lemma 1** (Canonical One-step Lemma). *Assume that the problem instance satisfies Assumptions 1, 2 and 6. Then, for step-size $\eta < \frac{1}{H}$ and all $t \in [0, T-1]$, Local-SGD's iterates satisfy:*

$$\mathbb{E}\left[\|x_{t+1} - x^\star\|_2^2\right] \leq (1 - \eta\mu)\,\mathbb{E}\left[\|x_t - x^\star\|_2^2\right] + \frac{\eta H^2}{\mu} \cdot \frac{1}{M} \sum_{m \in [M]} \mathbb{E}\left[\|x_t - x_t^m\|_2^2\right] + \frac{\eta^2 \sigma^2}{M} \ .$$

The above lemma is standard in the analysis of Local SGD [15, 16, 11, 20, 21, 23, 25]; we include a proof in Appendix F for completeness. The blue term is the *consensus error*, which vanishes when all clients communicate at every time step (i.e., in fully synchronous SGD). Early analyses of Local SGD, such as [20], often controlled this term using the following restrictive assumption:

**Assumption 7** (Uniform Bounded First-order Heterogeneity). *There exists $\zeta > 0$ such that*

$$\sup_{m,n \in [M]} \sup_{x \in \mathbb{R}^d} \|\nabla F_m(x) - \nabla F_n(x)\|_2 \leq H \cdot \zeta \ .$$

Under Assumption 7, Woodworth et al. [20] showed that the consensus error can be bounded as:

$$\frac{1}{M} \sum_{m \in [M]} \mathbb{E}\left[\|x_t - x_t^m\|_2^2\right] \leq \eta^2 H^2 \zeta^2 K^2 + 2\eta^2 \sigma^2 K \left(1 + \ln(K)\right) \ . \tag{3}$$

We include a proof of the above statement in Appendix G for completeness. Substituting it into Lemma 1 and unrolling the recursion yields a convergence rate.[5] However, Assumption 7 is very restrictive as it requires the gradient functions across clients to be pointwise similar, allowing only

---

[5] In Appendix G, we also state some iterate convergence results under Assumption 7 that we could not find in a clean form in the existing literature.

limited heterogeneity—essentially in the linear terms. Such mild variation can typically be resolved with constant initial communication rounds (see Appendix A). Notably, Wang et al. [8] criticized the uniform consensus error bound in (3), arguing that contrary to practice it implies an overly conservative step-size $\eta = \mathcal{O}(1/K)$ to prevent consensus error from diverging as $K \to \infty$.

The following result relaxes the need for Assumption 7 by providing a new upper bound on the consensus error that depends on $\zeta_\star$, $\tau$, and the expected iterate error at the most recent communication round—a quantity that decreases over time.

**Lemma 2** (A Coupled Recursion for Consensus Error). *Assume that the problem instance satisfies Assumptions 1 to 6. Then, for step-size $\eta < \frac{1}{H}$ and all $t \in [0, T]$, Local-SGD's iterates satisfy:*

$$\frac{1}{M} \sum_{m \in [M]} \mathbb{E}\left[\|x_t - x_t^m\|_2^2\right] \leq 2\eta^2 H^2 K^2 \zeta_\star^2 + \frac{2\eta^3 \tau^2 K^2 \sigma^2}{\mu} + 2\eta^2 \sigma^2 K \left(1 + \ln(K)\right)$$

$$+ 4\eta^2 \tau^2 (t - \delta(t))^2 (1 - \eta\mu)^{2(t-1-\delta(t))} \left(\mathbb{E}\left[\|x_{\delta(t)} - x^\star\|_2^2\right] + \phi_\star^2\right) ,$$

*where $\delta(t) := t - (t \mod K)$ is the most recent communication round prior to or at time $t$.*

We prove this result in Appendix H. Unlike the earlier bound in (3), our upper bound improves with lower second-order heterogeneity. In the limit $\tau \to 0$, it effectively replaces $\zeta$ with $\zeta_\star$ in (3), and can therefore be significantly smaller. While our bound does require setting $\eta = \mathcal{O}(1/K)$ to prevent blow-up as $K \to \infty$, we provide an alternative bound in Appendix H.3 that avoids this and addresses the concerns raised by Wang et al. [8]. That said, as we explain in Appendix H.3, the regime $\eta = \mathcal{O}(1/K)$ is ultimately the most relevant for our analysis, making Lemma 2 more useful. Finally, we note that similar fine-grained upper bounds on consensus error have also appeared in the literature on decentralized optimization [52, 53, 54]

Combining the coupled recursions in Lemmas 1 and 2 leads to the following convergence guarantee:

**Theorem 2** (Informal, Iterate Error). *Assume a problem instance satisfies Assumptions 1 to 6 (with $Q = \infty$) and $R = \tilde{\Omega}\left(\frac{H\tau}{\mu^2}\right)$. Then, for a suitable $\eta$, and $x_0 = 0$ Local SGD outputs $x_{KR}$ s.t.:*

$$\mathbb{E}\left[\|x_{KR} - x^\star\|_2^2\right] = \tilde{\mathcal{O}}\left(e^{-\frac{\mu KR}{2H}} B^2 + \frac{\sigma^2}{\mu^2 MKR} + \frac{\tau^2 H^2 \phi_\star^2}{\mu^4 R^2} + \frac{H^4 \zeta_\star^2}{\mu^4 R^2} + \frac{H^2 \tau^2 \sigma^2}{\mu^6 KR^3} + \frac{H^2 \sigma^2}{\mu^4 KR^2}\right) .$$

For the complete theorem statement, the precise step-size choice, and the derivation of the bound, see Appendix I.1. As a baseline, we can compare the above rate to the convergence rate of mini-batch SGD in the intermittent communication setting (see e.g., [20]),

$$\mathbb{E}\left[\|x_{KR}^{MB-SGD} - x^\star\|_2^2\right] = \mathcal{O}\left(e^{-\frac{\mu R}{2H}} B^2 + \frac{\sigma^2}{\mu^2 MKR}\right) .$$

It is well known that the convergence rate for mini-batch SGD above is tight and **can not** improve with lower data heterogeneity [20, 25]. Patel et al. [25] proved that local SGD can not beat mini-batch SGD under just Assumptions 4 and 5, leaving open the question of what happens when we additionally have Assumption 6. Theorem 2 answers this question, showing that with a small $\tau$, Local SGD can converge much faster than mini-batch SGD. Notably, when $K \to \infty$, the communication complexity of Local SGD for target accuracy $\epsilon$ and large $K$ satisfies:

$$R^{L-SGD}(\epsilon) = \tilde{\mathcal{O}}\left(\frac{H\tau}{\mu^2} + \frac{\tau H \phi_\star}{\mu^2 \sqrt{\epsilon}} + \frac{H^2 \zeta_\star}{\mu^2 \sqrt{\epsilon}}\right) . \tag{4}$$

The above communication complexity decreases with data heterogeneity, suggesting that Local SGD becomes increasingly communication-efficient when tasks are more aligned. In particular, the convergence rate smoothly interpolates to the behavior on homogeneous problems, for which our bound implies that a constant number of communication rounds suffice. On the other hand, with similar $K$, the communication complexity of mini-batch SGD is $\tilde{\Omega}(\kappa)$, and does not improve with a lower data heterogeneity. We note that this is the first ever result to prove the domination of Local SGD over mini-batch SGD in settings of reasonable heterogeneity, i.e., these rates only depend on $\tau$, $\zeta_\star$, $\phi_\star$, and not on $\zeta$, while also showing a provable benefit of local update steps.

Using a different progress lemma (Appendix F.3), we also derive a corresponding function-value convergence result based on the same consensus error bound in Lemma 2.

**Theorem 3** (Informal, Function Error with Strong Convexity). *Assume a problem instance satisfies Assumptions 1 to 6 (with $Q = \infty$), $R = \tilde{\Omega}\left(\frac{\tau\sqrt{\kappa}}{\mu}\right)$, and $KR = \Omega(\kappa)$. Then, for a suitable $\eta$, Local SGD initialized at $x_0 = 0$ outputs $\hat{x}$, a weighted combination of its iterates, satisfying,*

$$\mathbb{E}\left[F(\hat{x})\right] - F(x^\star) = \tilde{\mathcal{O}}\left(e^{-\frac{\mu KR}{2H}}\mu B^2 + \frac{\sigma^2}{\mu MKR} + \frac{\tau^2 H\phi_\star^2}{\mu^2 R^2} + \frac{H^3\zeta_\star^2}{\mu^2 R^2} + \frac{H\tau^2\sigma^2}{\mu^4 KR^3} + \frac{H\sigma^2}{\mu^2 KR^2}\right) .$$

The proof of the above theorem can be found in Appendix I.2[6]. The above convergence rate also has a desirable dependence on the data heterogeneity constants and improves over mini-batch SGD.

**General Convex Functions.** One might ask, what is the corresponding rate to Theorem 3 for general convex functions (with $\mu$ possibly zero)? A natural approach to get that rate is using a convex to strongly convex reduction, using an appropriate amount of regularization in Theorem 3. This strategy, unfortunately, leads to very stringent constraints on the heterogeneity constants and the number of communication rounds. This is why we omit stating the result here. We suspect that extending the ideas in this section to general convex functions might require more technical innovations. We leave this for future work, and note that this is an important gap as the upper bounds presented in this section are not directly comparable to Theorem 1.

Finally, it is worth noting that the hard instance in Theorem 1 is a quadratic function, and thus has $Q = 0$ while in this section we only present results for general strongly convex objectives. This raises the possibility that, by restricting attention to quadratics, we may be able to improve upon the upper bound in Theorems 2 and 3. In the next section, we explore this direction by deriving tighter upper bounds in regimes where the third-order smoothness constant $Q$ from Assumption 1 is small.

## 5 Incorporating Third-order Smoothness

In the homogeneous setting, Woodworth et al. [22] showed a surprising result: for smooth and convex objectives, Local SGD can outperform mini-batch SGD only when SGD on a single machine also outperforms it. This contradicts empirical findings, where Local SGD consistently outperforms both mini-batch and single machine SGD [55, 7]. However, for certain objective classes with higher-order smoothness—such as quadratics [16, 11] and logistic regression [28]—Local SGD can be provably superior to these two baselines and even be min-max optimal in some scenarios. To alleviate this gap, Yuan and Ma [21] analyzed Local SGD under a third-order smoothness assumption (i.e., bounded $Q$ in Assumption 1) providing convergence guarantees that could interpolate between convex functions and quadratics. Patel et al. [25] extended this to the heterogeneous setting but relied on the restrictive Assumption 7. In this section, equipped with the new consensus error and a modified one-step lemma, we relax that assumption and refine our upper bounds from the previous section to make their dependence on third-order smoothness explicit.

We will begin by stating a modified one-step progress result in Lemma 3 that explicitly captures second-order heterogeneity and third-order smoothness. This directly recovers improved bounds for quadratic objectives by setting $Q = 0$ in Theorems 4 and 5. To handle general third-order smooth functions, we combine this with new bounds on the fourth moment of the consensus error (Appendix H.2) and a corresponding fourth-moment progress lemma (Appendix F.2), resulting in Theorem 6. These results show that when $Q$ and $\tau$ are small, Local SGD can achieve significantly faster convergence, even under substantial first-order heterogeneity.

**Lemma 3** (Modified One-step Lemma). *Assume the problem instance satisfies Assumptions 1, 2 and 6. Then, for step-size $\eta < \frac{1}{H}$ and all $t \in [0, T-1]$, the iterates of Local SGD satisfy:*

$$\mathbb{E}\left[\|x_{t+1} - x^\star\|_2^2\right] \leq (1 - \eta\mu)\,\mathbb{E}\left[\|x_t - x^\star\|_2^2\right] + \frac{\eta^2\sigma^2}{M}$$
$$+ \frac{2\eta Q^2}{\mu} \cdot \frac{1}{M}\sum_{m\in[M]}\mathbb{E}\left[\|x_t - x_t^m\|_2^4\right] + \frac{2\eta\tau^2}{\mu} \cdot \frac{1}{M}\sum_{m\in[M]}\mathbb{E}\left[\|x_t - x_t^m\|_2^2\right] .$$

We prove Lemma 3 in Appendix F. Compared to Lemma 1, this recursion introduces an additional fourth-moment of the consensus error, weighted by the third-order smoothness constant $Q$. While this

---

[6]In the regime $\kappa >> 1$ Theorem 3 is much better than just applying second-order smoothness to Theorem 2.

fourth-moment term can dominate the second-moment term, the decomposition reveals how smoother problems (with small $Q$ and $\tau$) reduce the impact of delayed communication. In particular, when $Q = 0$—when each $F_m$ is quadratic—we obtain significantly sharper bounds than in Theorem 2.

**Assumption 8** (Quadraticity). *For all $m \in [M]$, the objective function $F_m(\cdot)$ is quadratic.*

**Theorem 4** (Informal, Iterate Error for Quadratics). *Assume the problem instance satisfies Assumptions 1 to 6 and 8, $R = \tilde{\Omega}\left(\frac{\tau^2}{\mu^2}\right)$ and $KR = \tilde{\Omega}(1)$. Then, for a suitable choice of step-size $\eta$, Local SGD initialized at $x_0 = 0$ outputs $x_{KR}$ such that:*

$$\mathbb{E}\left[\|x_{KR} - x^\star\|_2^2\right] = \tilde{\mathcal{O}}\left(e^{-\frac{\mu KR}{2H}}B^2 + \frac{\sigma^2}{\mu^2 MKR} + \frac{\tau^4\phi_\star^2}{\mu^4 R^2} + \frac{\tau^2 H^2 \zeta_\star^2}{\mu^4 R^2} + \frac{\tau^4 \sigma^2}{\mu^6 KR^3} + \frac{\tau^2 \sigma^2}{\mu^4 KR^2}\right) \ .$$

We prove this theorem in Appendix I.1. To understand the improvement over Theorem 2, consider the implied communication complexity in the large $K$ regime:

$$R(\epsilon) = \tilde{\mathcal{O}}\left(\frac{\tau^2}{\mu^2} + \frac{\tau^2\phi_\star}{\mu^2\sqrt{\epsilon}} + \frac{\tau H\zeta_\star}{\mu^2\sqrt{\epsilon}}\right) \ , \tag{5}$$

which becomes constant when $\tau = 0$. In contrast, the bound in (4) still depends on $\zeta_\star$ even when $\tau = 0$. This highlights how low third-order smoothness ($Q$) and low second-order heterogeneity ($\tau$) improve Local SGD's performance—especially in settings where first-order heterogeneity is still large. It is also worth noting that the convergence rates for mini-batch SGD do not improve with a lower third-order smoothness, as the hard instances for mini-batch SGD are all quadratic [56].

Using a different modified progress lemma (see Appendix F.3), we also derive the following convergence rate in terms of function values.

**Theorem 5** (Informal, Function Error for Quadratics). *Assume a problem instance satisfies Assumptions 1 to 6 and 8, $R = \tilde{\Omega}\left(\frac{\tau^2}{\mu^2}\right)$, and $KR = \Omega(\kappa)$. Then, for a suitable choice of step-size $\eta$, Local SGD initialized at $x_0 = 0$ outputs $\hat{x}$, a weighted combination of its iterates, satisfying,*

$$\mathbb{E}\left[F(\hat{x})\right] - F(x^\star) = \tilde{\mathcal{O}}\left(e^{-\frac{\mu KR}{2H}}\mu B^2 + \frac{\sigma^2}{\mu MKR} + \frac{\tau^4\phi_\star^2}{\mu^3 R^2} + \frac{\tau^2 H^2 \zeta_\star^2}{\mu^3 R^2} + \frac{\tau^4 \sigma^2}{\mu^5 KR^3} + \frac{\tau^2 \sigma^2}{\mu^3 KR^2}\right) \ .$$

The proof for the above theorem can be found in Appendix I.2. Compared to Theorem 2 we again see an improvement, as all but the first two terms in the convergence rate go to zero when $\tau = 0$.

Finally, we prove the following result for general third-order smooth functions.

**Theorem 6** (Informal, Iterate Error with $Q$). *Assume a problem instance satisfies Assumptions 1 to 6. Then, for a suitable choice of step-size $\eta$, Local SGD initialized at $x_0 = 0$ outputs $x_{KR}$ satisfying:*

$$\mathbb{E}\left[\|x_{KR} - x^\star\|_2^2\right] + \frac{1}{B^2}\mathbb{E}\left[\|x_{KR} - x^\star\|_2^4\right] = \tilde{\mathcal{O}}\Bigg(e^{-\eta\mu KR}B^2 + \frac{\sigma^2}{\mu^2 MKR} + \frac{\sigma^4}{\mu^4 K^3 R^3 M^2 B^2}$$

$$+ \kappa'\left(\frac{\tau^2\phi_\star^2}{\mu^2 R^2} + \frac{\tau^4\sigma^2}{\mu^6 KR^5 B^2}\phi_\star^2 + \frac{\sigma^2\tau^2}{\mu^4 KR^4 B^2}\phi_\star^2 + \frac{\tau^4}{\mu^4 B^2 R^4}\phi_\star^4 + \frac{H^2\zeta_\star^2}{\mu^2 R^2} + \frac{\tau^2\sigma^2}{\mu^4 KR^3}\right)$$

$$+ \kappa'\left(\frac{\sigma^2\ln(K)}{\mu^2 KR^2} + \frac{H^4\zeta_\star^4}{\mu^4 R^3 B^2} + \frac{\tau^4\sigma^4}{\mu^8 K^2 R^5 B^2} + \frac{\sigma^2 H^2\zeta_\star^2}{\mu^4 B^2 R^4} + \frac{\tau^2\sigma^4}{\mu^6 KR^5 B^2} + \frac{\sigma^4\ln(K)}{\mu^4 KB^2 R^4}\right)\Bigg) \ ,$$

*where we assume $R = \tilde{\Omega}\left(\frac{\tau\sqrt{\kappa'}}{\mu}\right)$ and define $\kappa' := 2 + \frac{4Q^2 B^2}{\mu^2} + \frac{6H^4}{\mu^4}$.*

We can see that the above convergence rate improves with smaller $\tau$ and $Q$, via the constant $\kappa'$, and the effect of a low third-order smoothness is most pronounced when $B/\mu^2$ is large relative to $\kappa^4$. To prove the above theorem, we first derive new fourth-moment bounds on the consensus error and one-step progress in Appendices F and H. Solving the resulting four coupled recursions directly is challenging, so we stack the iterate and consensus recursions into two vectors and apply matrix algebra, leading to a cleaner proof in Appendix I.3. A limitation of our analysis is that the final bound is expressed in terms of the norm of a stacked vector that includes both second and fourth-moment errors. Since bounding the fourth moment of the iterate error is not strictly necessary, this may have introduced extraneous terms in the upper bound. We therefore believe that Theorem 6 could be further improved through a more refined analysis of the underlying matrix inequalities.

# 6 Case Study: Distributed Linear Regression

We consider a linear regression task, where for each client $m \in [M]$, the data consists of covariate-label pairs $z_m := (\beta_m, y_m) \sim \mathcal{D}_m$ with Gaussian covariates $\beta_m \sim \mathcal{N}(\mu_m, I_d) \in \mathbb{R}^d$ and labels $y_m \sim \langle x_m^\star, \beta_m \rangle + \mathcal{N}(0, \sigma_{\text{noise}}^2)$ generated using a ground truth model $x_m^\star \in \mathbb{R}^d$. Each client minimizes the mean squared error, $f(x; (\beta_m, y_m)) = \frac{1}{2}(y_m - \langle x, \beta_m \rangle)^2$ leading to an expected loss:

$$F_m(x) = \frac{1}{2}(x - x_m^\star)^\top (\mu_m \mu_m^\top + I_d)(x - x_m^\star) + \frac{1}{2}\sigma_{\text{noise}}^2 \ .$$

Under suitable bounds on $\mu_m$, $\Sigma_m$, and $\sigma_{\text{noise}}$, this problem satisfies Assumptions 1 to 3 for bounded $x$. Furthermore, we have $\left\| \nabla^2 F_m(x) - \nabla^2 F_n(x) \right\|_2 \leq (\|\mu_m\|_2 + \|\mu_n\|_2) \cdot \|\mu_m - \mu_n\|_2$ for any $m, n \in [M]$. So Assumption 6 quantifies the **covariate shift** across clients. Meanwhile, Assumption 4 reflects the **concept shift** via the bound $\|x_m^\star - x_n^\star\|_2 \leq \zeta_\star$.

In Figure 2, we examine the convergence behavior of Local SGD on the synthetic linear regression task. In Figure 2a, we decouple first- and second-order heterogeneity by independently varying the means $\mu_m$ and the ground truths $x_m^\star$. We observe that Local SGD performs well only when both types of heterogeneity are small. This highlights why earlier works that did not account for second-order heterogeneity (Assumption 6) were unable to explain Local SGD's effectiveness fully. In Figure 2b, we fix the first-order heterogeneity and plot the communication complexity required to reach a target accuracy as a function of $\tau$. As expected, we find a monotonic relationship, further reinforcing the connection between second-order heterogeneity and the communication efficiency of Local SGD.

Importantly, when varying the heterogeneity, we ensure we do not inadvertently make the individual optimization problems harder, for example, by increasing the condition number $\kappa$ or the radius $B$. In Appendix J, we describe how we control for this and include additional experiments.

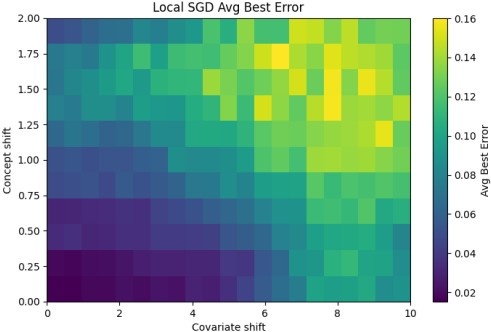

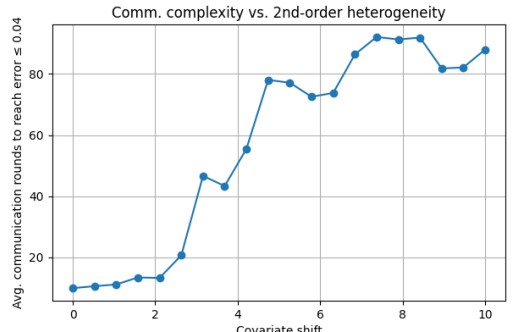

(a) Heatmap of the average best-final $\ell_2$ error of Local SGD after $R = 5$ communication rounds as a function of covariate shift $\tau$ (horizontal axis) and concept shift $\zeta_\star$ (vertical axis).

(b) Communication complexity of Local SGD versus covariate shift $\tau$, for a fixed concept shift $\zeta_\star = 1.0$ to reach an $\ell_2$ error $\leq 0.04$. We allow up to $R_{\max} = 100$ rounds, and plot the mean rounds-to-target.

Figure 2: **Impact of First- and Second-Order Heterogeneity on Local SGD.** In both figures, we use $d = 5$, $M = 20$ clients, $K = 10$ local steps, and a noise level of $\sigma_{\text{noise}} = 0.1$. The step-size is tuned over a logarithmic grid in $[10^{-3}, 10^{-1}]$, and the error is averaged over multiple trials. For (a), we report the mean error over $n_{\text{runs}} = 20$ trials for each $(\tau, \zeta_\star)$ pair, tuning the step-size separately in each trial. Similarly, in (b), we average over $n_{\text{runs}} = 20$ trials for each $\tau$, again tuning the step-size independently per trial. We discuss in Appendix J how to interpret the $\tau, \zeta_\star$ in our plots' axes.

**Practical Implications for Federated Learning.** Our results highlight that the performance of Local SGD depends critically on the structure of data heterogeneity. In practice, this suggests distinguishing between heterogeneity in optimal predictors (first-order, measured by $\zeta_\star$ and $\phi_\star$) and curvature or feature distributions (second-order, measured by $\tau$). For example, $\zeta_\star$ may be small for learning in overparameterized settings while $\tau$ remains significant. Large local steps ($K$) can still yield good performance and communication savings in such cases. But when $\tau$ is very large, aggressive local updates with a fixed step-size can cause instability. We recommend tuning $\eta$ as a function of $K$ and using diagnostic signals—such as consensus error growth or curvature estimates—to adjust training parameters. Estimating $\tau$ from local and running statistics could help guide such choices in practice.

## Acknowledgments and Disclosure of Funding

We thank the anonymous reviewers at NeurIPS, who helped significantly improve the paper. KKP was supported through the NSF TRIPOD Institute on Data, Economics, Algorithms and Learning (IDEAL) and other awards from DARPA and NSF.

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

# Table of Contents

## A  More Discussion on Heterogeneity Assumptions

### A.1  Construction for Remark 1

This sub-section will show a problem with $\zeta_\star << B$ and $\phi_\star \approx B$. As a preliminary remark, note that we must consider high-dimensional examples for this: in a single dimension, $x^\star$ must be in the convex hull of $\{x_1^\star, \ldots, x_M^\star\}$ (which is just a line-segment). The instance we will consider will make use of the following two functions, which take two-dimensional inputs $(x, y) \in \mathbb{R}^2$,

$$f(x,y) = 2\left(x + \frac{\zeta_\star}{2}\right)^2 + \left(x + y + \frac{\zeta_\star}{2}\right)^2 \quad \text{and} \quad g(x,y) = \left(x - \frac{\zeta_\star}{2}\right)^2 + \left(x + y - \frac{\zeta_\star}{2}\right)^2 \ .$$

Note that both these functions are strictly convex with optimizers at $\left(\frac{-\zeta_\star}{2}, 0\right)$ and $\left(\frac{\zeta_\star}{2}, 0\right)$ respectively. However, the optimizer of the average of these functions is given by $\left(-\frac{\zeta_\star}{6}, \frac{\zeta_\star}{6}\right)$, which is notably not on the convex hull of the optimizers of the constituent functions.

Now we define $M$ different objectives on $d$ dimensions (assuming $M$, $d$ are even for simplicity) as follows:

$$F_1(x) = f(x[1], x[2]) + \frac{1}{2}\left\|(0, 0, x[3], \ldots, x[M])\right\|_2^2 \ ,$$

$$F_2(x) = g(x[1], x[2]) + \frac{1}{2}\left\|(0, 0, x[3], \ldots, x[M])\right\|_2^2 \ ,$$

$$F_3(x) = f(x[3], x[4]) + \frac{1}{2}\left\|(x[1], x[2], 0, 0, x[4] \ldots, x[M])\right\|_2^2 \ ,$$

$$F_4(x) = g(x[3], x[4]) + \frac{1}{2}\left\|(x[1], x[2], 0, 0, x[4] \ldots, x[M])\right\|_2^2 \ ,$$

$$\vdots$$

$$F_{M-1}(x) = f(x[M-1], x[M]) + \frac{1}{2}\left\|(0, 0, \ldots, x[M-1], x[M])\right\|_2^2 \ ,$$

$$F_M(x) = g(x[M-1], x[M]) + \frac{1}{2}\left\|(0, 0, \ldots, x[M-1], x[M])\right\|_2^2 \ .$$

Due to the properties of $f, g$ that we discussed above, note that for any two machines $m \neq n$, $\|x_m^\star - x_n^\star\|_2 = \zeta_\star$. Furthermore, note the optimizer of the obejctive is given by $x^\star = \left(-\frac{\zeta_\star}{6}, \frac{\zeta_\star}{6}, -\frac{\zeta_\star}{6}, \frac{\zeta_\star}{6}, \ldots, -\frac{\zeta_\star}{6}, \frac{\zeta_\star}{6}\right)$. This implies that,

$$\|x_m^\star - x^\star\|_2^2 = \left(-\frac{\zeta_\star}{2} + \frac{\zeta_\star}{6}\right)^2 + \frac{(M-1)\zeta_\star^2}{36} = \frac{(M+3)\zeta_\star^2}{36} \qquad m \text{ is odd },$$

$$\|x_m^\star - x^\star\|_2^2 = \left(\frac{\zeta_\star}{2} + \frac{\zeta_\star}{6}\right)^2 + \frac{(M-1)\zeta_\star^2}{36} = \frac{(M+15)\zeta_\star^2}{36} \qquad m \text{ is even },$$

In particular, if we pick $M = 36\frac{B^2}{\zeta_\star^2} - 15$ (assume it is an even number), then we can guarantee that $\phi_\star$ must be at least $B$. This proves the claim we made in the remark.

### A.2  Proof of Remark 2

This is easy to see let us just write $\bar{x}^\star$ more explicitly,

$$\|x_m^\star - x^\star\|_2 \leq \|x_m^\star - \bar{x}^\star\|_2 + \|\bar{x}^\star - x^\star\|_2 ,$$

$$= \left\|\frac{1}{M} \sum_{n \in [M]} (x_m^\star - x_n^\star)\right\|_2 + \left\|\frac{1}{M} \sum_{n \in [M]} (x_n^\star - A^{-1}A_n x_n^\star)\right\|_2 ,$$

$$\leq \frac{1}{M} \sum_{n \in [M]} \|x_m^\star - x_n^\star\|_2 + \left\|\frac{1}{M} \sum_{n \in [M]} A^{-1}(A - A_n)(x_n^\star - \bar{x}^\star)\right\|_2 ,$$

$$\leq^{\text{Assumption 4}} \zeta_\star + \frac{1}{M} \sum_{n \in [M]} \left\|A^{-1}(A - A_n)(x_n^\star - \bar{x}^\star)\right\|_2 ,$$

$$\leq \zeta_\star + \frac{1}{M} \sum_{n \in [M]} \left\|A^{-1}\right\|_2 \|A - A_n\|_2 (x_n^\star - \bar{x}^\star) ,$$

$$\leq \zeta_\star + \frac{\tau\zeta_\star}{\mu} ,$$

which proves the claim of Remark 2.

## B  More Discussion on the Fixed Point Perspective

Several papers have pointed out with varying levels of explicitness [57, 58, 25] that the hardness of analysing Local SGD's convergence comes from a fixed-point discrepancy, i.e., Local SGD in the limit of large $R$ converges to a point different from $x^\star$ whenever $K > 1$. This is an alternative viewpoint to data heterogeneity, and can be useful to provide analyses of Local SGD. For the simple case of strongly convex quadratic functions Patel et al. [25] showed that in the absence of noise, Local GD converges very quickly—with extreme communication efficiency—to its fixed point and they also gave a bound for the fixed point discrepancy in terms of Assumptions 4 to 6.

In this section, we will revisit their analysis, showing that it extends to the stochastic case and also provide a more fine-grained upper bound on the fixed-point discrepancy for the quadratic case. Both these advancements allow us to provide a convergence for strongly convex quadratics in Theorem 7. Then we will make a few comments about the convex quadratic setting, i.e., when $\mu = 0$, commenting on the potential regularization effects of Local-SGD.

Throughout this section, we will consider Local SGD with both an inner step-size $\eta$ and an outer step-size $\beta$. In the main paper, we only analysed and discussed Local SGD with a single step-size,

i.e., we set $\beta = 1$. To make the notation easier to accommodate two step-sizes, we use the following,

$$
\begin{aligned}
x_{r,0}^m &= \bar{x}_{r-1} \ , && \forall\, m \in [M] \\
x_{r,k+1}^m &= x_{r,k}^m - \eta \nabla f(x_{r,k}; z_{r,k}^m), \ z_{r,k}^m \sim \mathcal{D}_m \ , && \forall\, m \in [M], k \in [0, K-1] \\
\bar{x}_r &= \bar{x}_{r-1} + \frac{\beta}{M} \sum_{m \in [M]} \left( x_{r,K}^m - \bar{x}_{r-1} \right) \ .
\end{aligned}
\tag{6}
$$

## B.1 Fast Convergence to Fixed-point in the Quadratic Setting

The following Lemma extends the analysis of Patel et al. [25] for convergence to the fixed point to the stochastic setting.

**Lemma 4.** *For quadratic problems satisfying Assumptions 1 to 3 and 8, with machine $m$'s hessian denoted by $A_m$, with $\eta < \frac{1}{H}$, and $\beta \le \frac{1}{1-(1-\eta H)^K}$ the Local-SGD iterate $\bar{x}_R$ (with initialization $\bar{x}_0 = 0$) satisfies,*

$$
\mathbb{E}\left[ \|\bar{x}_R - x_\infty\|_2^2 \right]
$$

$$
\le \left(1 - \beta\left(1 - (1-\eta\mu)^K\right)\right)^{2R} \|x_\infty\|_2^2 + \eta\beta\left(1 - \left(1 - \beta\left(1 - (1-\eta\mu)^K\right)\right)^R\right)\frac{\sigma^2}{\mu M} \ ,
$$

*where we define $x_\infty := \frac{1}{M}\sum_{m\in[M]} C^{-1} C_m x_m^\star$ for $C_m := I - (I - \eta A_m)^K$ and $C := \frac{1}{M}\sum_{m\in[M]} C_m$. In particular, when $\beta = 1$ we have,*

$$
\mathbb{E}\left[ \|\bar{x}_R - x_\infty\|_2^2 \right] \le (1-\eta\mu)^{2KR} \|x_\infty\|_2^2 + \eta\left(1 - (1-\eta\mu)^{KR}\right)\frac{\sigma^2}{\mu M} \ .
$$

*Proof.* We consider quadratic problems of the form,

$$
F_m(x) = \frac{1}{2}(x - x_m^\star)^T A_m (x - x_m^\star) \ ,
$$

where $A_m \succ 0$ is a positive definite Hessian matrix. We denote the fixed-point of Local SGD by (for a simple intuition see [25]),

$$
x_\infty := \frac{1}{M} \sum_{m \in [M]} C^{-1} C_m x_m^\star \ , \qquad \text{where } C_m := I - (I - \eta A_m)^K \ .
$$

We now note the following about the local-SGD updates between two communication rounds on machine $m \in [M]$,

$$
\begin{aligned}
x_{r,K}^m - x_m^\star &= x_{r,K-1}^m - x_m^\star - \eta A_m(x_{r,K-1}^m - x_m^\star) \\
&\quad + \eta\left(A_m(x_{r,K-1}^m - x_m^\star) - \nabla f(x_{r,K-1}^m; z_{r,K-1}^m)\right) \ , \\
&= (I - \eta A_m)^K \left(x_{r,0}^m - x_m^\star\right) \\
&\quad + \eta \sum_{k=0}^{K-1} (I - \eta A_m)^{K-1-k} \left(A_m(x_{r,k}^m - x_m^\star) - \nabla f(x_{r,k}^m; z_{r,k}^m)\right) \ , \\
&= (I - \eta A_m)^K \left(x_{r-1} - x_m^\star\right) + \eta \sum_{k=0}^{K-1} (I - \eta A_m)^{K-1-k} \xi_{r,k}^m \ .
\end{aligned}
$$

This implies the following

$$
x_{r,K}^m - x_{r-1} = x_m^\star - x_{r-1} + (I - \eta A_m)^K \left(x_{r-1} - x_m^\star\right) + \eta \sum_{k=0}^{K-1} (I - \eta A_m)^{K-1-k} \xi_{r,k}^m \ ,
$$

$$
= -\left(I - (I - \eta A_m)^K\right)(x_{r-1} - x_m^\star) + \eta \sum_{k=0}^{K-1} (I - \eta A_m)^{K-1-k} \xi_{r,k}^m \ ,
$$

$$= -C_m \left(x_{r-1} - x_m^\star\right) + \eta \sum_{k=0}^{K-1} \left(I - \eta A_m\right)^{K-1-k} \xi_{r,k}^m \ ,$$

$$= -C_m x_{r-1} + C_m x_m^\star + \eta \sum_{k=0}^{K-1} \left(I - \eta A_m\right)^{K-1-k} \xi_{r,k}^m \ .$$

This implies for the $r$-th synchronized model,

$$x_r = x_{r-1} + \frac{\beta}{M} \sum_{m \in [M]} \left( -C_m x_{r-1} + C_m x_m^\star + \eta \sum_{k=0}^{K-1} \left(I - \eta A_m\right)^{K-1-k} \xi_{r,k}^m \right) \ ,$$

$$= \left(I - \beta C\right) x_{r-1} + \frac{\beta}{M} \sum_{m \in [M]} C_m x_m^\star + \eta \beta \sum_{k=0}^{K-1} \left(I - \eta A_m\right)^{K-1-k} \left( \frac{1}{M} \sum_{m \in [M]} \xi_{r,k}^m \right) \ ,$$

$$= \left(I - \beta C\right) \left(x_{r-1} - x_\infty\right) + x_\infty - \beta C x_\infty + \frac{\beta}{M} \sum_{m \in [M]} C_m x_m^\star$$

$$+ \eta \beta \sum_{k=0}^{K-1} \left(I - \eta A_m\right)^{K-1-k} \xi_{r,k} \ ,$$

$$= \left(I - \beta C\right) \left(x_{r-1} - x_\infty\right) + x_\infty - \frac{\beta}{M} \sum_{m \in [M]} C_m x_m^\star + \frac{\beta}{M} \sum_{m \in [M]} C_m x_m^\star$$

$$+ \eta \beta \sum_{k=0}^{K-1} \left(I - \eta A_m\right)^{K-1-k} \xi_{r,k} \ .$$

Simplifying and rearranging this, we get for $r = R$,

$$x_R - x_\infty = \left(I - \beta C\right) \left(x_{R-1} - x_\infty\right) + \eta \beta \sum_{k=0}^{K-1} \left(I - \eta A_m\right)^{K-1-k} \xi_{R,k} \ ,$$

$$= \left(I - \beta C\right)^R \left(x_0 - x_\infty\right) + \sum_{r=0}^{R-1} \left(I - \beta C\right)^{R-1-r} \left( \eta \beta \sum_{k=0}^{K-1} \left(I - \eta A_m\right)^{K-1-k} \xi_{r,k} \right)$$

Take the norm, squaring, and taking expectation, we get,

$$\mathbb{E}\left[\|x_r - x_\infty\|_2^2\right] \leq \|I - \beta C\|_2^{2R} \, \mathbb{E}\left[\|(x_0 - x_\infty)\|_2^2\right]$$

$$+ \eta^2 \beta^2 \sum_{r=0}^{R-1} \|I - \beta C\|_2^{2(R-1-r)} \left( \sum_{k=0}^{K-1} \left(I - \eta A_m\right)^{2(K-1-k)} \mathbb{E}\left[\|\xi_{r,k}\|_2^2\right] \right) \ ,$$

$$\leq \|I - \beta C\|_2^{2R} \|x_\infty\|_2^2$$

$$+ \eta^2 \beta^2 \sum_{r=0}^{R-1} \|I - \beta C\|_2^{2(R-1-r)} \left( \sum_{k=0}^{K-1} (1 - \eta \mu)^{K-1-k} \frac{\sigma^2}{M} \right) \ ,$$

$$\leq \left(1 - \beta \lambda_{min}(C)\right)^{2R} \|x_\infty\|_2^2$$

$$+ \eta^2 \beta^2 \sum_{r=0}^{R-1} \left(1 - \beta \lambda_{min}(C)\right)^{R-1-r} \left( \frac{1 - (1 - \eta \mu)^K}{\eta \mu} \cdot \frac{\sigma^2}{M} \right) \ ,$$

We now need upper and lower bounds on the minimum eigenvalue of $C$. For this, note that,

$$\lambda_{min}(C) = \lambda_{min} \left( \frac{1}{M} \sum_{m \in [M]} \left(I - (I - \eta A_m)^K\right) \right) \ ,$$

$$= \min_{\|v\|_2 = 1} \frac{1}{M} \sum_{m \in [M]} v^T \left(I - (I - \eta A_m)^K\right) v \ ,$$

$$= 1 - \max_{\|v\|_2 = 1} \frac{1}{M} \sum_{m \in [M]} v^T (I - \eta A_m)^K v \ ,$$

$$\in 1 - \left( (1 - \eta\mu)^K, (1 - \eta H)^K \right) \ ,$$

$$\in \left( 1 - (1 - \eta\mu)^K, 1 - (1 - \eta H)^K \right) \ .$$

Plugging these bounds in the above inequality leads to,

$$\mathbb{E}\left[ \|x_r - x_\infty\|_2^2 \right] \leq \left( 1 - \beta \left( 1 - (1 - \eta\mu)^K \right) \right)^{2R} \|x_\infty\|_2^2$$

$$+ \eta\beta \frac{1 - \left( 1 - \beta \left( 1 - (1 - \eta\mu)^K \right) \right)^R}{1 - (1 - \eta\mu)^K} \cdot \frac{1 - (1 - \eta\mu)^K}{\mu} \cdot \frac{\sigma^2}{M} \ ,$$

$$\leq \left( 1 - \beta \left( 1 - (1 - \eta\mu)^K \right) \right)^{2R} \|x_\infty\|_2^2$$

$$+ \eta\beta \left( 1 - (1 - \beta (1 - (1 - \eta\mu)^K))^R \right) \frac{\sigma^2}{\mu M} \ .$$

Note that the range of $\lambda_{min}(C)$ is what suggests the upper bound on $\beta$ of $\frac{1}{1-(1-\eta H)^K}$. $\qquad \square$

Next we will establish an upper bound on $\|x_\infty\|_2$, which would allow us to provide the upper bound in terms of $B$ from Assumption 3.

**Lemma 5.** *In the setting of the previous lemma,*

$$\|x_\infty\|_2 \leq \min \left\{ \eta\tau K \kappa \zeta_\star + B, \kappa B \right\} \ .$$

*Proof.* Recall the definition of $x_\infty$,

$$\|x_\infty\|_2 = \left\| C^{-1} \left( \frac{1}{M} \sum_{m \in [M]} C_m x_m^\star \right) \right\|_2 \ ,$$

$$= \left\| C^{-1} \left( \frac{1}{M} \sum_{m \in [M]} (C_m - C + C)(x_m^\star - \bar{x}^\star + \bar{x}^\star) \right) \right\|_2 \ ,$$

$$= \left\| C^{-1} \left( \frac{1}{M} \sum_{m \in [M]} (C_m - C)(x_m^\star - \bar{x}^\star) \right) + \bar{x}^\star \right\|_2 \ ,$$

$$\leq \frac{1}{M^2} \sum_{m,n \in [M]} \left\| C^{-1}(C_m - C_n) \right\|_2 + \frac{1}{M} \sum_{m \in [M]} \|x_m^\star\|_2 \ ,$$

$$= \frac{1}{M^2} \sum_{m,n \in [M]} \left\| C^{-1} \right\|_2 \left\| (I - \eta A_n)^K - (I - \eta A_m)^K \right\|_2 \|x_m^\star - x_n^\star\|_2 + \frac{1}{M} \sum_{m \in [M]} \|x_m^\star\|_2 \ ,$$

$$\overset{\text{(Lemma 19 and Assumptions 3, 4 and 6)}}{\leq} \frac{\eta\tau K \left( 1 - (1 - \eta H)^{K-1} \right)}{1 - (1 - \eta\mu)^K} \zeta_\star + B \ ,$$

$$\leq \eta\tau K \cdot \frac{1 - (1 - \eta H)^K}{1 - (1 - \eta\mu)^K} \zeta_\star + B \ .$$

Now we will show that the factor $g(K) = \frac{1-(1-\eta H)^K}{1-(1-\eta\mu)^K}$ can be upper bounded by $\kappa = g(1)$ for any choice of step-size $\eta$. To do this we show that $g(K)$ is a non-increasing function and thus can be upper bounded by $g(1)$. To see this note for $k \in \mathbb{Z}_{\geq 1}$, while denoting $0 < a := 1 - \eta H \leq 1 - \eta\mu =: b < 1$,

$$g(k) = \frac{1 - a^K}{1 - b^K} \ ,$$

$$= \frac{1 - a}{1 - b} \cdot \frac{1 + a + \cdots + a^{k-1}}{1 + b + \cdots + b^{k-1}} \ ,$$

$$=: \frac{1-a}{1-b} \cdot \frac{S_k(a)}{S_k(b)} ,$$

where we defined the geometric sum $S_k(\cdot)$ for ease of notation. Using this we get that,

$$
\begin{aligned}
(g(k) - g(k+1)) \frac{1-b}{1-a} &= \frac{S_k(a)}{S_k(b)} - \frac{S_{k+1}(a)}{S_{k+1}(b)} , \\
&= \frac{S_k(a)}{S_k(b)} - \frac{S_k(a) + a^k}{S_k(b) + b^k} , \\
&= \frac{S_k(a)(S_k(b) + b^k) - (S_k(a) + a^k)S_k(b)}{S_k(b)(S_k(b) + b^k)} , \\
&= \frac{a^k b^k}{S_k(b)(S_k(b) + b^k)} \left( \frac{S_k(a)}{a^k} - \frac{S_k(b)}{b^k} \right) , \\
&= \frac{a^k b^k}{S_k(b)(S_k(b) + b^k)} \sum_{i=0}^{k-1} \left( \frac{a^i}{a^k} - \frac{b^i}{b^k} \right) , \\
&= \frac{a^k b^k}{S_k(b)(S_k(b) + b^k)} \sum_{i=0}^{k-1} \left( a^{i-k} - b^{i-k} \right) , \\
&\geq^{(a<b)} 0 .
\end{aligned}
$$

Thus $g(\cdot)$ is a non-increasing function proving our earlier claim. Plugging this above gives us,

$$\|x_\infty\|_2 \leq \eta \tau K \kappa \zeta_\star + B .$$

Note that in the very first step of the proof we could also upper bound $\|x_\infty\|_2$ by $g(K)B$, thus we can also get the trivial upper bound $\kappa B$, following the result of the proof. This proves the lemma. □

Combining the previous two lemmas and simplifying we get the following convergence rate to the fixed point.

**Proposition 1** (Fast Convergence to Fixed Point). *For quadratic problems satisfying Assumptions 1 to 3 and 8, with machine $m$'s hessian denoted by $A_m$, with $\eta < \frac{1}{H}$, and $\beta = 1$ the Local-SGD iterate $\bar{x}_R$ (with initialization $\bar{x}_0 = 0$) satisfies,*

$$\mathbb{E}\left[ \|\bar{x}_R - x_\infty\|_2^2 \right] \leq e^{-2\eta\mu KR} \cdot \min\{\eta\tau K\kappa\zeta_\star + B, \kappa B\}^2 + \frac{\eta\sigma^2}{\mu M} .$$

Note that while in Assumption 3 we assume that $\|x^\star\|_2 \leq B$, but in general if we only assume the norms of the individual machines' optimizers are bounded by $B$, then the most natural upper bound on $x^\star$ is $\frac{\tau\zeta_\star}{\mu} + B$.

## B.2 Improved Fixed-point Discrepancy Upper Bound for Quadratics

We will need the following lemma about the Lipschitzness of a specific matrix polynomial.

**Lemma 6.** *Let $A_m, A_n \in \mathbb{R}^{d \times d}$ be symmetric positive-definite matrices whose spectra lie inside the interval $[\mu, H] \subset (0, 1/\eta)$, with $0 < \mu \leq H$ and $0 < \eta < 1/H$. Fix an integer $K \geq 1$ and define the polynomial*

$$R(\lambda) = 1 - (1 - \eta\lambda)^K - \eta K\lambda, \qquad \lambda \in \mathbb{R}.$$

*Extend $R$ to symmetric matrices by functional calculus, $R(X) = I - (I - \eta X)^K - \eta KX$. Then*

$$\|R(A_m) - R(A_n)\|_2 \leq L \|A_m - A_n\|_2, \qquad L = \eta K \left[ 1 - (1 - \eta H)^{K-1} \right].$$

*Proof. Step 1: A scalar Lipschitz constant.* Direct differentiation gives

$$R'(\lambda) = \eta K \left[ (1 - \eta\lambda)^{K-1} - 1 \right],$$

which is non-positive and increasing on $[\mu, H]$. Hence

$$L = \sup_{\lambda \in [\mu, H]} |R'(\lambda)| = \eta K \left[ 1 - (1 - \eta H)^{K-1} \right].$$

*Step 2: Fréchet derivative.* Write $X = U \operatorname{diag}(\lambda_1, \ldots, \lambda_d) U^\top$ and set $F = U^\top E U$ for any symmetric perturbation $E$. The Daleckii–Krein formula yields

$$DR[X](E) = U(M \odot F) U^\top, \qquad M_{ij} = \frac{R(\lambda_i) - R(\lambda_j)}{\lambda_i - \lambda_j}.$$

Because $-R$ is operator-monotone on $[\mu, H]$, the matrix $M$ is positive-semidefinite and its entries satisfy $|M_{ij}| \leq L$.

*Step 3: Schur-multiplier estimate.* If a PSD matrix $M$ has entries bounded by $L$, then for every $G \in \mathbb{R}^{d \times d}$

$$\|M \odot G\|_2 \leq (\max_i M_{ii}) \|G\|_2 \leq L \|G\|_2.$$

Applying this with $G = F$ gives

$$\|DR[X](E)\|_2 \leq L \|E\|_2.$$

*Step 4: Integration along a line segment.* Set $\Delta := A_m - A_n$ and $A(t) := A_n + t\Delta$ for $t \in [0, 1]$. Define $\Phi(t) := R\big(A(t)\big)$. Step 3 implies $\|\Phi'(t)\|_2 \leq L\|\Delta\|_2$ for all $t$, so

$$\big\| R(A_m) - R(A_n) \big\|_2 = \big\| \Phi(1) - \Phi(0) \big\|_2 \leq \int_0^1 \|\Phi'(t)\|_2 \, dt \leq L \|\Delta\|_2.$$

This is precisely the claimed bound. $\qquad\square$

Now we are ready to prove the following lemma which improves upon the result of Patel et al. [25]

**Lemma 7** (Fixed Point Discrepancy for Quadratics). *For quadratic functions satisfying Assumptions 1, 4 and 6 we can guarantee the following for $\eta < 1/H$,*

$$\|x^\star - x_\infty\|_2 \leq \frac{\zeta_\star \tau}{\mu} \cdot \frac{(1 - \eta H)^K - 1 + \eta H K + \eta \mu K \left( 1 - (1 - \eta H)^{K-1} \right)}{1 - (1 - \eta \mu)^K} \ .$$

**Remark 3.** Note that the above bound goes to zero when $\tau$ or $\zeta_\star$ is zero, which matches the behavior of the bound due to Patel et al. [25]. However, the bound also goes to zero when $K = 1$ or $\eta \to 0$, a behavior their bound did not capture.

*Proof.* Note the following,

$$\|x^\star - x_\infty\|_2 = \left\| \frac{1}{M^2} \sum_{m,n \in [M]} \left( A^{-1} A_m - C^{-1} C_m \right) (x_m^\star - x_n^\star) \right\|_2 ,$$

$$\leq \frac{1}{M^2} \sum_{m,n \in [M]} \left\| A^{-1} A_m - C^{-1} C_m \right\|_2 \|x_m^\star - x_n^\star\|_2 ,$$

$$\leq \frac{1}{M} \sum_{m \in [M]} \left\| A^{-1} A_m - C^{-1} C_m \right\|_2 \zeta_\star ,$$

Let us denote the following,

$$C_m := I - (I - \eta A_m)^K =: \eta K A_m + R_m \quad \text{and} \quad R := \frac{1}{M} \sum_{m \in [M]} R_m .$$

In particular, note that when $K = 1$, then $R_m = 0$, which implies that $R = 0$. Using this notation, we have the following,

$$\left\| A^{-1} A_m - C^{-1} C_m \right\|_2 = \left\| C^{-1} \left( C A^{-1} A_m - C_m \right) \right\|_2 ,$$

$$
\begin{aligned}
&= \left\| C^{-1}\left(\left(\eta K A + R\right) A^{-1} A_m - \eta K A_m - R_m\right)\right\|_2 \ , \\
&= \left\| C^{-1}\left(R A^{-1} A_m - R_m\right)\right\|_2 \ , \\
&= \left\| C^{-1}\left(R A^{-1} A_m - R + R - R_m\right)\right\|_2 \ , \\
&\leq \left\| C^{-1}\right\|_2 \left(\|R\|_2 \left\| A^{-1} A_m - I\right\|_2 + \|R - R_m\|_2\right) \ , \\
&\leq \frac{1}{1 - (1 - \eta\mu)^K} \frac{1}{M} \sum_{n \in [M]} \left(\frac{\tau}{\mu}\|R_n\|_2 + \|R_m - R_n\|_2\right) \ .
\end{aligned}
$$

Now it suffices to upper bound the two terms $\|R_m\|_2$ and $\|R_m - R_n\|_2$. As a sanity check, note that when $K = 1$ and $\tau = 0$, the upper bounds are still zero. For the first term, note the following using the diagonalization of the matrix $A_m = V_m \Sigma_m V_m^{-1}$,

$$
\begin{aligned}
\textcolor{blue}{\|R_n\|_2} &= \left\| I - \eta K A_n - (I - \eta A_n)^K\right\|_2 \ , \\
&= \left\| I - \eta K \Sigma_n - (I - \eta \Sigma_n)^K\right\|_2 \ , \\
&\leq \sup_{\lambda \in [\mu, H]} \left| 1 - \eta K \lambda - (1 - \eta\lambda)^K\right| \ , \\
&= (1 - \eta H)^K - 1 + \eta H K \ ,
\end{aligned}
$$

where we use the fact that $\eta < \frac{1}{H}$ which implies that $\eta\lambda < 1$ in the above function, which in turn implies that $\left|1 - \eta K \lambda - (1 - \eta\lambda)^K\right|$ is an increasing function in the range $\lambda \in [\mu, H]$. Now we need to bound the second term $\|R_m - R_n\|_2$. Note that ideally we would like the upper bound to also vanish with $K = 1$ and $\tau = 0$. We cannot use the strategy from above because we do not know if the matrices $A_m$ and $A_n$ commute. Instead we will use the following property (see Lemma 6 that follows),

$$
\|R(A_m) - R(A_n)\|_2 \leq \sup_{\lambda \in [\mu, H]} |R'(\lambda)| \|A_m - A_n\|_2 \ ,
$$

where we define $R(\lambda) := 1 - (1 - \eta\lambda)^K - \eta K \lambda$. Note the following,

$$
\begin{aligned}
|R'(\lambda)| &= \left| \eta K (1 - \eta\lambda)^{K-1} - \eta K\right| \ , \\
&= \left| - \eta K \left(1 - (1 - \eta\lambda)^{K-1}\right)\right| \ , \\
&= \eta K \cdot \left|1 - (1 - \eta\lambda)^{K-1}\right| \ .
\end{aligned}
$$

Plugging this in the above bound gives us,

$$
\begin{aligned}
\textcolor{blue}{\|R_m - R_n\|_2} &= \|R(A_m) - R(A_n)\|_2 \ , \\
&\leq \sup_{\lambda \in [\mu, H]} \eta K \|A_m - A_n\|_2 \cdot \left|1 - (1 - \eta\lambda)^{K-1}\right| \ , \\
&\leq \textcolor{cyan}{\eta K \tau \left(1 - (1 - \eta H)^{K-1}\right)} \ .
\end{aligned}
$$

For a sanity check, note that when $K = 1$ or $\tau = 0$, this bound is zero. Plugging the $\textcolor{blue}{\text{blue}}$ and $\textcolor{cyan}{\text{cyan}}$ upper bounds back into $\textcolor{red}{\text{the original bound}}$ on fixed-point discrepancy, we get,

$$
\begin{aligned}
\|x^\star - x_\infty\|_2 &\leq \frac{\zeta_\star}{1 - (1 - \eta\mu)^K} \frac{1}{M^2} \sum_{m,n \in [M]} \left(\frac{\tau}{\mu}\|R_n\|_2 + \|R_m - R_n\|_2\right) \ , \\
&\leq \frac{\zeta_\star}{1 - (1 - \eta\mu)^K} \\
&\quad \times \frac{1}{M^2} \sum_{m,n \in [M]} \left(\frac{\tau}{\mu}\left((1 - \eta H)^K - 1 + \eta H K\right) + \eta K \tau \left(1 - (1 - \eta H)^{K-1}\right)\right) \ , \\
&= \frac{\zeta_\star \tau}{\mu} \cdot \frac{(1 - \eta H)^K - 1 + \eta H K + \eta\mu K \left(1 - (1 - \eta H)^{K-1}\right)}{1 - (1 - \eta\mu)^K} \ .
\end{aligned}
$$

This proves the lemma. $\qquad\square$

While the general behavior of the fixed-point discrepancy upper bound can be quite complex, we can study its effect for a specific step-size, which leads to the following convergence guarantee.

**Theorem 7.** *Assume we are optimizing a problem instance satisfying Assumptions 1 to 4, 6 and 8,* $R \geq 2\ln\left(\frac{B^2}{\epsilon}\right)$, $B^2 > \epsilon$ *and* $KR \geq 2\kappa\ln\left(\frac{B^2}{\epsilon}\right)$ *for some target accuracy* $\epsilon$. *Then using* $\eta = \frac{1}{\mu KR}\ln\left(\frac{B^2}{\epsilon}\right)$ *and* $\beta = 1$, *Local SGD initialized at* $x_0 = 0$ *outputs* $\bar{x}_r$ *such that,*

$$\mathbb{E}\left[\|\bar{x}_R - x^\star\|_2^2\right] \leq \frac{\epsilon^2}{B^2}\left(\frac{2\tau^2\kappa^2\zeta_\star^2}{\mu^2 B^2 R^2}\ln^2\left(\frac{B^2}{\epsilon}\right) + 2\right) + \frac{\sigma^2}{\mu^2 MKR} + \frac{9\zeta_\star^2\tau^2\kappa^2}{\mu^2 R^2}\ln^2\left(\frac{B^2}{\epsilon}\right) \ .$$

**Remark 4.** As we will see in Appendix I.3, while analysing Local SGD for third-order smooth functions, we derive a slightly worse convergence rate which contains all the terms in the above upper bound upto constant factors.

*Proof.* We will first simplify the upper bound by noting that $\frac{1-(1-\eta H)^K}{1-(1-\eta\mu)^K}$ is non-increasing in $K$, as was proved in Lemma 5,

$$\frac{(1-\eta H)^K - 1 + \eta HK + \eta\mu K\left(1-(1-\eta H)^{K-1}\right)}{1-(1-\eta\mu)^K}$$

$$\leq \frac{\eta HK}{1-(1-\eta\mu)^K} + \eta\mu K \cdot \frac{1-(1-\eta H)^K}{1-(1-\eta\mu)^K} \ ,$$

$$\leq \frac{\eta HK}{1-(1-\eta\mu)^K} + \eta\mu K \cdot \frac{1-(1-\eta H)}{1-(1-\eta\mu)} \ ,$$

$$= \frac{\eta HK}{1-(1-\eta\mu)^K} + \eta HK \ .$$

Now, assume that $2\kappa\ln\left(\frac{B^2}{\epsilon}\right) \leq KR$ for target accuracy $\epsilon$ so that we can pick $\eta = \frac{1}{\mu KR}\ln\left(\frac{B^2}{\epsilon}\right) \leq \frac{1}{2H} < \frac{1}{H}$. For this choice of $\eta$ the above upper bound reduces to,

$$\|x^\star - x_\infty\|_2 \leq \frac{\zeta_\star\tau}{\mu}\left(\frac{\frac{\kappa}{R}\ln\left(\frac{B^2}{\epsilon}\right)}{1-(1-1/(KR)\ln\left(\frac{B^2}{\epsilon}\right))^K} + \frac{\kappa}{R}\ln\left(\frac{B^2}{\epsilon}\right)\right) \ ,$$

$$\leq \frac{\zeta_\star\tau\kappa}{\mu R}\ln\left(\frac{B^2}{\epsilon}\right)\left(1 + \frac{1}{1-e^{-\frac{1}{R}\ln\left(\frac{B^2}{\epsilon}\right)}}\right) \ .$$

Now further assuming $R \geq 2\ln\left(\frac{B^2}{\epsilon}\right)$, we can upper $e^{-\frac{1}{R}\ln\left(\frac{B^2}{\epsilon}\right)}$ by $\frac{1}{2}$ which implies,

$$\|x^\star - x_\infty\|_2 \leq \frac{3\zeta_\star\tau\kappa}{\mu R}\ln\left(\frac{B^2}{\epsilon}\right) \ .$$

Let us also upper bound the rate of convergence to the fixed point for this choice of step-size and under these assumptions,

$$\mathbb{E}\left[\|\bar{x}_R - x_\infty\|_2^2\right] \leq (1-\eta\mu)^{2KR}\min\{\eta\tau K\kappa\zeta_\star + B, \kappa B\}^2 + \eta\left(1-(1-\eta\mu)^{KR}\right)\frac{\sigma^2}{\mu M} \ ,$$

$$\leq \frac{\epsilon^2}{B^4}\min\left\{\frac{2\tau^2\kappa^2\zeta_\star^2}{\mu^2 R^2}\ln^2\left(\frac{B^2}{\epsilon}\right) + 2B^2, \kappa^2 B^2\right\} + \frac{\sigma^2}{\mu^2 MKR} \ ,$$

$$\leq \epsilon \cdot \frac{\epsilon}{B^2}\min\left\{\frac{2\tau^2\kappa^2\zeta_\star^2}{\mu^2 B^2 R^2}\ln^2\left(\frac{B^2}{\epsilon}\right) + 2, \kappa^2\right\} + \frac{\sigma^2}{\mu^2 MKR} \ .$$

The above term is under standard conditions of the order $\epsilon$, so we can ignore it. Cobining this with the fixed point discrepancy upper bound above proves the statement of the lemma. $\qquad\square$

### B.3 On the Nature of Local SGD's Fixed Point for Quadratics

While in the general convex setting we can not write an explicit formula for the fixed point $x_\infty$, we can characterize it as the mini-mum norm solution of a certain leas-squares problem, where the geometry for each machine is defined by the matrices $C_m$.

**Proposition 2** (Fixed Point for Convex Quadratics). *Assume we have a problem instance satisfying Assumptions 1 to 3 and 8 with $\sigma = 0$, $\eta < 1/H$. Further define $C_m := I - (I - \eta A_m)^K$, $C := \frac{1}{M} \sum_{m \in [M]} C_m$ and $c := \frac{1}{M} \sum_{m \in [M]} C_m x_m^\star$ for some $x_m^\star \in S_m^\star$ for each $m \in [M]$. If $c \neq 0$ and $c \in im(C) = ker(C)^\perp$, then Local GD converges to the following solution in the limit of $R \to \infty$,*

$$x_\infty = \arg\min \quad \|x\|_2 \quad , \quad s.t. \quad x \in \min_{x \in \mathbb{R}^d} \frac{1}{M} \sum_{m \in [M]} \|x - x_m^\star\|_{C_m}^2 \quad .$$

*If on the other hand $c \neq 0$ and $c \notin im(C)$, the the iterates do not converge, but if we define the sequence $y_R = C\bar{x}_R$, then*

$$\lim_{R \to \infty} y_R = \sum_{i \in [l]} v_i v_i^T c \lim_{R \to \infty} \left(1 - (1 - \lambda_i)^R\right) \quad ,$$

*where $C = \sum_{i \in [l]} \lambda_i v_i v_i^T$ is the eigen-value decomposition of $C$ for orthonormal vectors $\{v_1, \ldots, v_l\}$. If $c = 0$, the iterates of Local-GD do not move from $\bar{x}_0 = 0$.*

**Remark 5.** When the objectives on each machine are strongly convex, then we always have $c \in im(C) = \mathbb{R}^d$. In general when $im(C) = \mathbb{R}^d$, we can guarantee convergence to a fixed point. An even weaker sufficient condition is to assume that $\cap_{m \in [M]} ker(A_m) = \{0\}$, which guarantees that $ker(C) = \{0\}$ and hence $im(C) = \mathbb{R}^d$. We prove this last condition in Lemma 8. The condition $\cap_{m=1}^M ker(A_m) = \{0\}$ is equivalent to the average Hessian $A$ being positive definite, i.e., $A \succ 0$, or in the global objective being strongly convex. This condition ensures that local curvature from different clients collectively constrains all directions and the machines are no simultaneously blind to some direction.

*Proof.* We recall that even in the convex setting (i.e., with $\mu = 0$) we can write the following for the Local SGD iterate $\bar{x}_R$ in the noise-less setting with $\beta = 1$ and initialization $\bar{x}_0 = 0$,

$$\bar{x}_R = \frac{1}{M} \sum_{m \in [M]} \left((I - \eta A_m)^K (\bar{x}_{R-1} - x_m^\star) + x_m^\star\right) \quad ,$$

$$= \frac{1}{M} \sum_{m \in [M]} \left((I - \eta A_m)^K\right) \bar{x}_{R-1} + \frac{1}{M} \sum_{m \in [M]} \left(I - (I - \eta A_m)^K\right) x_m^\star \quad ,$$

$$= (I - C) \bar{x}_{R-1} + \frac{1}{M} \sum_{m \in [M]} C_m x_m^\star \quad ,$$

$$\stackrel{(x_0 = 0)}{=} \sum_{j=0}^{R-1} (I - C)^j c \quad .$$

Now let us assume an orthonormal basis for the span of $C$ is given by $\{v_1, \ldots, v_l\}$ where $l \leq d$. This allows us to write,

$$C = \sum_{i \in [l]} \lambda_i v_i v_i^T \quad ,$$

where $0 < \lambda_i \leq 1 - (1 - \eta H)^K < 1$ as our step-size is $\eta < 1/H$. Let us extend this to an orthonormal basis for the entire vector space $\mathbb{R}^d$ as $\{v_1, \ldots, v_l, v_{l+1}, \ldots, v_d\}$ so that $v_{l+1}, \ldots, v_d \in ker(C)$. This also implies for $j \in \mathbb{Z}_{\geq 0}$,

$$(I - C)^j = \left(\sum_{i \in [l]} (1 - \lambda_i) v_i v_i^T + \sum_{i \in [l+1,d]} v_i v_i^T\right)^j \quad ,$$

$$= \sum_{i \in [l]} (1 - \lambda_i)^j v_i v_i^T + \sum_{i \in [l+1,d]} v_i v_i^T \quad .$$

Now we will inspect how $\bar{x}_R$ evolves in each direction, $i \in [d]$.

First let us consider $i \in [l]$,

$$v_i^T \bar{x}_R = \sum_{j=0}^{R-1} v_i^T (I - C)^j c \ ,$$

$$= \sum_{j=0}^{R-1} (1 - \lambda_i)^j v_i^T c \ ,$$

$$= \frac{1 - (1 - \lambda_i)^R}{\lambda_i} v_i^T c \ .$$

No matter how we pick $\eta, \ K$, this would converge as $R \to \infty$, to some quantity proportional to $v_i^T c$.

Now let us consider $i \in [l + 1, d]$,

$$v_i^T \bar{x}_R = \sum_{j=0}^{R-1} v_i^T (I - C)^j c \ ,$$

$$= \sum_{j=0}^{R-1} v_i^T c \ ,$$

$$= R v_i^T c \ .$$

Notably, this does not converge unless $v_i^T c = 0$.

In particular, the iterates of local GD converge iff $v_i^T c = 0$ for all $i \in [l + 1, d]$. Or in other words, $c \in \mathrm{im}(C) = \ker(A)^\perp$. First let us assume, this is true, then we can conclude that the local GD iterates only evolve in the sub-space $\mathrm{im}(C)$. Where do they converge? Solving the fixed-point equation in the limit of large $R$ gives us,

$$x_\infty = (I - C) x_\infty + c \ ,$$
$$\Rightarrow C x_\infty = c \ .$$

Summarizing the two key findings so far we get that assuming $c \in \mathrm{im}(C)$, $C x_\infty = c$ and $x_\infty \notin \ker(C)$. This is equivalent to saying that $x_\infty$ is the minimum norm solution of the linear system $Cx = c$. In other words,

$$x_\infty = \arg \min_{x \text{ s.t. } Cx = c} \|x\|_2 \ .$$

Further, note that using the least square formulation we can write the solutions of the linear system $Cx = c$ as,

$$\min_{x \in \mathbb{R}^d} \frac{1}{M} \sum_{m \in [M]} \|x - x_m^\star\|_{C_m}^2 \ .$$

This implies that $x_\infty$ (when it exists) is the solution of the following optimization problem,

$$\min \quad \|x\|_2 \ , \quad \text{s.t.} \quad x \in \min_{x \in \mathbb{R}^d} \frac{1}{M} \sum_{m \in [M]} \|x - x_m^\star\|_{C_m}^2 \ .$$

In the case, that $c \notin \mathrm{im}(A)$, there exists $i \in [l + 1, d]$ such that $v_i^T c \neq 0$. The iterates will explode in this direction, but still notably, the sequence $C \bar{x}_R$ does converge, because

$$\lim_{R \to \infty} C \bar{x}_R = \lim_{R \to \infty} \sum_{i \in [l]} \lambda_i v_i v_i^T \bar{x}_R \ ,$$

$$= \lim_{R \to \infty} \sum_{i \in [l]} v_i v_i^T c \left(1 - (1 - \lambda_i)^R\right) \ ,$$

No matter how we pick $\eta, \ K$ this limit exists.

$\square$

**Lemma 8.** *Let $A_1, \ldots, A_M \in \mathbb{R}^{d \times d}$ be symmetric positive semidefinite matrices, and let $1/H > \eta > 0$ and $K \in \mathbb{N}$. Define*

$$C_m := I - (I - \eta A_m)^K, \quad C := \frac{1}{M} \sum_{m=1}^{M} C_m \ .$$

*Suppose the kernel intersection is trivial:*

$$\bigcap_{m=1}^{M} \ker(A_m) = \{0\} \ .$$

*Assume further that $\eta < 1/\lambda_{\max}(A_m)$ for all $m$ (where $\lambda_{\max}(A_m) \leq H$ denotes the largest eigenvalue of $A_m$.*

*Then:*

1. *For each $m$, $\ker(C_m) = \ker(A_m)$.*

2. *The matrix $C$ is full rank: $\operatorname{im}(C) = \mathbb{R}^d$, i.e., $\ker(C) = \{0\}$.*

*Proof.* **Part (1):** Since $A_m \succeq 0$, its eigenvalues lie in $[0, \lambda_{\max}(A_m)]$. Then $I - \eta A_m$ has eigenvalues in $[1 - \eta\lambda_{\max}(A_m), 1] \subset (0, 1]$, so:

$$C_m = I - (I - \eta A_m)^K = \sum_{j=1}^{K} \binom{K}{j} (-\eta A_m)^j \ ,$$

a matrix polynomial in $A_m$. Due to the polynomial structure it is easy to see that,

$$\ker(C_m) \supseteq \ker(A_m) \ .$$

To see the other side note, we will prove the contrapositive. Suppose that $v \notin \ker(A_m)$, but $v \in \ker(C_m)$, then

$$C_m v = v - (I - \eta A_m)^K v = 0 \ ,$$
$$\Rightarrow \|v\|_2 = \left\| (I - \eta A_m)^K v \right\|_2 < \|v\|_2 \ ,$$

which is a contradition. Thus $v \notin \ker(A_m)$ implies that, $v \notin \ker(C_m)$, or in other words,

$$\ker(C_m) \subseteq \ker(A_m) \ .$$

This proves the first part of the statement that $\ker(A_m) = \ker(C_m)$.

**Part (2):** Now suppose for contradiction that $Cv = 0$ for some $v \neq 0$. Then:

$$\sum_{m=1}^{M} C_m v = 0 \quad \Rightarrow \quad \langle Cv, v \rangle = \frac{1}{M} \sum_{m=1}^{M} \langle C_m v, v \rangle = 0 \ .$$

Since each $C_m \succeq 0$, it must be that $\langle C_m v, v \rangle = 0 \Rightarrow C_m v = 0 \Rightarrow v \in \ker(C_m) = \ker(A_m)$ for all $m$. So:

$$v \in \bigcap_{m=1}^{M} \ker(A_m) = \{0\} \ ,$$

contradicting $v \neq 0$. Hence $\ker(C) = \{0\}$, and since $C$ is symmetric, $\operatorname{im}(C) = \mathbb{R}^d$. $\qquad\square$

### B.4 Implicit Regularization due to Local SGD

Several works have tried to understand the effectiveness of Local-SGD from a different perspective, i.e., by arguing that the solution obtained by Local-SGD is somehow better. In other words, these works have tried to characterize the implicit regularization of using local update steps. On such notable work is due to Gu et al. [59].

For convex quadratic problems, the fixed-point perspective can also be used to understand the implicit regularization of Local SGD. Specifically, recall that under the assumption we discussed in

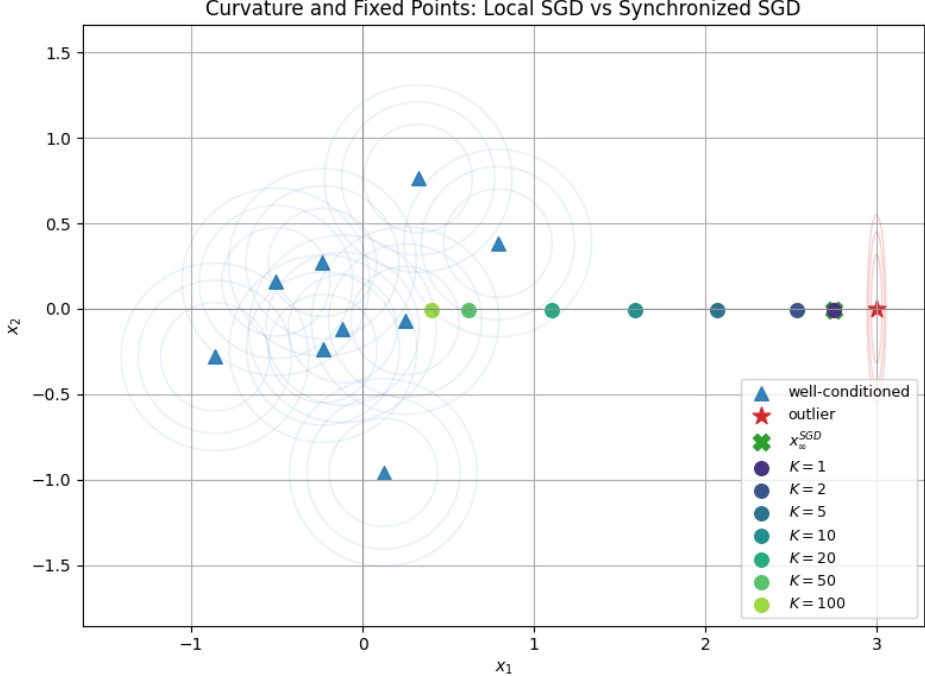

Figure 3: The effect of having an outlier with a sharp curvature on Local SGD's fixed point with progressively higher local update steps.

the previous sub-section, i.e., $\cap_{m \in [M]} \ker(A_m) = \{0\}$ we can also characterize the fixed-point of synchronized SGD as follows,

$$x_\infty^{SGD} = \arg\min \quad \|x\|_2 \quad , \quad \text{s.t.} \quad x \in \min_{x \in \mathbb{R}^d} \frac{1}{M} \sum_{m \in [M]} \|x - x_m^\star\|_{A_m}^2 \quad .$$

Thus the main difference with respect to Local SGD with $K > 1$, is a different geometry on each machine defined by $A_m$ as opposed to $C_m$ of Local-SGD,

$$x_\infty^{L-SGD} = \arg\min \quad \|x\|_2 \quad , \quad \text{s.t.} \quad x \in \min_{x \in \mathbb{R}^d} \frac{1}{M} \sum_{m \in [M]} \|x - x_m^\star\|_{C_m}^2 \quad .$$

One natural question is: **under what conditions is the geometry endowed by Local-SGD better?**

When $\eta$ is "large enough," then for larger $K$, the matrix polynomial $C_m = I - (I - \eta A_m)^K$ increasingly flattens the influence of high-curvature (i.e., high-eigenvalue) directions in $A_m$. In other words, Local SGD implicitly applies a spectral filter that downweights directions where the local objective is sharply curved. This has a regularization effect: machines with highly ill-conditioned losses or extremely sharp curvature (possibly due to overfitting, poor conditioning, or adversarial data) contribute less in those sensitive directions. Instead, Local SGD emphasizes agreement in directions where curvature is more moderate or shared across machines.

As a result, the fixed point $x_\infty^{L\text{-}SGD}$ avoids overreacting to any single client's extreme curvature and instead biases the solution toward directions of consensus and smoothness. In this sense, Local SGD can be interpreted as interpolating between machine-specific optimization (via $A_m$) and a more uniform averaging of preferences (via $C_m$), particularly in settings with heterogeneous curvature. This implicit regularization may lead to better generalization in practice, especially when the global objective inherits pathological structure from just a few problematic machines.

In Figure 3 we simulate the effect of having an outlier with a sharp curvature, showing how progressively more local update steps regularize the geometry.

## B.5 Extension to Non-quadratics?

The biggest issue with extending the above analysis to non-quadratic, is that it becomes hard to even write the expression for the fixed point in a closed form. As we will see in Appendix I it is much easier to use the usual consensus error based analysis in these settings.

## C Proof of the Lower Bound in Theorem 1

To prove Theorem 1, the main result we need is Lemma 10, which we then combine with the lower bound due to Glasgow et al. [23]. Notably, Lemma 10 is an improvement of the lower bound due to Patel et al. and like them we also borrow the folklore Lemma 9 about gradient descent.

We first state the following standard Lemma which in the context of Local-SGD was previously also used by Patel et al. [25].

**Lemma 9.** *There exists $F(x)$ a convex quadratic function for $x \in \mathbb{R}^2$ satisfying which is $H$-smooth, $\mu$-strongly convex with $\kappa = 12R$, and a bounded optima $x^\star$ with $\|x^\star\|_2 \leq B$ such that $\hat{x}_R$ the $R^{th}$ gradient descent iterate initialized at zero and for any step size $\eta > 0$, $F(\hat{x}_R) - F(x^\star) \geq \frac{HB^2}{8R}$.*

*Proof.* Let $A$ be the Hessian of $F$. Observe that we have $F(x) - F(x^\star) = \frac{1}{2}(x - x^\star)^T A(x - x^\star)$. Let $v_1$ and $v_2$ be the eigenvectors of norm 1 of $A$ with the greatest and least eigenvalues, respectively. Assume $x^\star := -B\left(\frac{v_1 + v_2}{\sqrt{2}}\right)$, which ensures $\|x^\star\|_2 = B$. Then, solving for the GD iterates in closed form, we have

$$
\begin{aligned}
x_R - x^\star &= x_{R-1} - x^\star - \eta A (x_{R-1} - x^\star) \ , \\
&= (I - \eta A)(x_{R-1} - x^\star) \ , \\
&=^{(a)} \left(v_1 v_1^T + v_2 v_2^T - \eta H v_1 v_1^T - \eta \mu v_2 v_2^T\right)^R (x_0 - x^\star) \ , \\
&= \left((1 - \eta H)v_1 v_1^T + (1 - \eta\mu)v_2 v_2^T\right)^R (x_0 - x^\star) \ , \\
&= \left((1 - \eta H)^R v_1 v_1^T + (1 - \eta\mu)^R v_2 v_2^T\right)(-x^\star) \ , \\
&= \frac{B}{\sqrt{2}}(1 - \eta H)^R v_1 + \frac{B}{\sqrt{2}}\left(1 - \eta\frac{H}{\kappa}\right)^R v_2 \ .
\end{aligned}
$$

where in (a) we ue the eigenvalue decomposition of $A = H v_1 v_1^T + \mu v_2 v_2^T$ and the fact that for orthonormal vectors $v_1$, $v_2$ we have $I_2 = v_1 v_1^T + v_2 v_2^T$. Observe that if $\eta \geq \frac{3}{H}$, then the iterates explode and we have $F(x_R) \geq F(x_0) \geq \Omega\left(HB^2\right)$.

If $\eta \leq \frac{3}{H}$, then using the fact that $\kappa \geq 6$, we have

$$
\begin{aligned}
F(x_R) - F(x^\star) &\geq^{(a)} \frac{1}{2}\left(\frac{B}{\sqrt{2}}\left(1 - \frac{3}{\kappa}\right)^R v_2\right)^T A \left(\frac{B}{\sqrt{2}}\left(1 - \frac{3}{\kappa}\right)^R v_2\right) \ , \\
&= \frac{B^2}{4}\left(1 - \frac{3}{\kappa}\right)^{2R} v_2^T A v_2 \ , \\
&= \frac{B^2}{4}\left(1 - \frac{3}{\kappa}\right)^{2R} \frac{H}{\kappa} \ , \\
&\geq^{(b)} \frac{HB^2}{4R}\left(1 - \frac{6R}{\kappa}\right) \ ,
\end{aligned}
$$

where in (a) we lower bound by the function sub-optimality only in the second component corresponding to $v_2$; and in (b) we assume $\kappa \geq 3$ and Bernoulli's inequality. Finally using $\kappa = 12R$ we get the lower bound $\frac{HB^2}{8R}$. The result follows. $\qquad\square$

Note that the following proof actually works for Local SGD with both an inner step-size $\eta$ and an outer step-size $\beta$. In the main paper we only analysed and discussed Local SGD with a single

step-size, i.e., we set $\beta = 1$. This makes the lower bound below stronger and not weaker, as it works for a more general algorithm. To make the notation easier to accomodate two step-sizes, we use the following,

$$
\begin{aligned}
x_{r,0}^m &= \bar{x}_{r-1} \ , && \forall \, m \in [M] \\
x_{r,k+1}^m &= x_{r,k}^m - \eta \nabla f(x_{r,k}; z_{r,k}^m), \ z_{r,k}^m \sim \mathcal{D}_m \ , && \forall \, m \in [M], k \in [0, K-1] \\
\bar{x}_r &= \bar{x}_{r-1} + \frac{\beta}{M} \sum_{m \in [M]} \left( x_{r,K}^m - \bar{x}_{r-1} \right) \ .
\end{aligned}
\tag{7}
$$

**Lemma 10.** *There exists a convex quadratic function for $x \in \mathbb{R}^3$ satisfying Assumptions 1, 3 and 6, such the Local SGD iterate $\bar{x}_R$, when initialized at zero and for any choice of step-sizes $\eta$, $\beta > 0$ must have $F(\bar{x}_R) - F(x^\star) = \Omega \left( \frac{\tau B^2}{R} \right)$.*

*Proof.* We consider the quadratic functions defined by the following two Hessians for $\tau \leq H$,

$$
A_1 = \begin{bmatrix} \tau \hat{A}_1 & 0 \\ 0 & H \end{bmatrix} \quad \text{and} \quad A_2 = \begin{bmatrix} \tau \hat{A}_2 & 0 \\ 0 & H \end{bmatrix} \ ,
$$

where we for some $\alpha \in (0, 1)$,

$$
\hat{A}_1 := \begin{bmatrix} 1 & 0 \\ 0 & 0 \end{bmatrix} \ , \quad \text{and}
$$

$$
\hat{A}_2 := vv^T = (\alpha, \sqrt{1 - \alpha^2})(\alpha, \sqrt{1 - \alpha^2})^T = \begin{bmatrix} \alpha^2 & \alpha\sqrt{1 - \alpha^2} \\ \alpha\sqrt{1 - \alpha^2} & 1 - \alpha^2 \end{bmatrix} \ .
$$

Note about the spectrum of $A_1$,

$$
\text{Spec} \, (A_1) = \{0, \tau, H\} \ .
$$

Similarly for $A_2$ we note that,

$$
\begin{aligned}
\det \left( \hat{A}_2 - \lambda I_2 \right) &= 0 \ , \\
\Rightarrow (\lambda - \alpha^2)(\lambda - 1 + \alpha^2) &= \alpha^2(1 - \alpha^2) \ , \\
\Rightarrow \lambda^2 - \left( \alpha^2 + 1 - \alpha^2 \right) \lambda &= 0 \ , \\
\Rightarrow \lambda &\in \{0, \ 1\} \ .
\end{aligned}
$$

which implies that also for,

$$
\text{Spec} \, (A_2) = \{0, \tau, H\} \ .
$$

Thus objectives defined by both these Hessians $A_1$ and $A_2$ are $H$-smooth. Further, we can notice the following about the difference between these Hessians,

$$
\begin{aligned}
\text{Spec} \, (A_1 - A_2) &= \tau \cdot \text{Spec} \left( \hat{A}_1 - \hat{A}_2 \right) \cup \{0\} \ , \\
&= \tau \cdot \text{Spec} \left( \begin{bmatrix} 1 - \alpha^2 & -\alpha\sqrt{1 - \alpha^2} \\ -\alpha\sqrt{1 - \alpha^2} & -(1 - \alpha^2) \end{bmatrix} \right) \cup \{0\} \ , \\
&= \left\{ -\tau\sqrt{1 - \alpha^2}, 0, \tau\sqrt{1 - \alpha^2} \right\} \cup \{0\} \ ,
\end{aligned}
$$

which implies that,

$$
\|A_1 - A_2\|_2 = \tau\sqrt{1 - \alpha^2} \leq \tau \ .
$$

Now we shall split the objectives on each machine as follows, For even $m$, let

$$
F_m(x) := \frac{1}{2}(x - x^*)^T A_1 (x - x^*) \ .
$$

For odd $m$, let

$$
F_m(x) := \frac{1}{2}(x - x^*)^T A_2 (x - x^*) \ .
$$

Note that the iterates after $K$ local updates leading up to communication round $r$ on machine $m$ gives,

$$\tilde{x}^m_{r,K} = (I - \eta A_i)^K \tilde{x}_{r-1} \ ,$$

where we denote $\tilde{x}^m_{r,k} = x^m_{r,k} - x^\star$ for all $k \in [0, K]$ and $\tilde{x}_r = \bar{x}_r - x^\star$ for all $r \in [0, R]$. For odd machines it is straightforward that,

$$(I - \eta A_1)^K = \begin{bmatrix} (1-\eta\tau)^K & 0 & 0 \\ 0 & 1 & 0 \\ 0 & 0 & (1-\eta H)^K \end{bmatrix} = \begin{bmatrix} I_2 - \left(\frac{1-(1-\eta\tau)^K}{\tau}\right)\tau\hat{A}_1 & 0 \\ 0 & (1-\eta H)^K \end{bmatrix} \ .$$

For even machines, the above can also be noted using $v_\perp$ as the unit vector orthogonal to $v$,

$$(I - \eta A_2)^K = \begin{bmatrix} I_2 - \eta\tau\hat{A}_2 & 0 \\ 0 & 1-\eta H \end{bmatrix}^K = \begin{bmatrix} (I_2 - \eta\tau vv^T)^K & 0 \\ 0 & (1-\eta H)^K \end{bmatrix} \ ,$$

$$= \begin{bmatrix} (vv^T + v_\perp v_\perp^T - \eta\tau vv^T)^K & 0 \\ 0 & (1-\eta H)^K \end{bmatrix} \ ,$$

$$\overset{(a)}{=} \begin{bmatrix} (1-\eta\tau)^K vv^T + v_\perp v_\perp^T & 0 \\ 0 & (1-\eta H)^K \end{bmatrix} \ ,$$

$$= \begin{bmatrix} I_2 - vv^T + (1-\eta\tau)^K vv^T & 0 \\ 0 & (1-\eta H)^K \end{bmatrix} \ ,$$

$$= \begin{bmatrix} I_2 - \left(\frac{1-(1-\eta\tau)^K}{\tau}\right)\tau\hat{A}_2 & 0 \\ 0 & (1-\eta H)^K \end{bmatrix} \ ,$$

where in (a) we note that $\left((1-\eta\tau)vv^T + v_\perp v_\perp^T\right)^2 = (1-\eta\tau)^2 vv^T + v_\perp v_\perp^T$. This implies for the local updates with $\tilde{\eta} := \left(\frac{1-(1-\eta\tau)^K}{\tau}\right)$ for all $m \in [M]$,

$$\tilde{x}^m_{r,K} = \begin{bmatrix} I_2 - \tilde{\eta}\tau\hat{A}_m & 0 \\ 0 & (1-\eta H)^K \end{bmatrix} \tilde{x}_{r-1} = \begin{bmatrix} \left(I_2 - \tilde{\eta}\tau\hat{A}_i\right)\tilde{x}_{r-1}[1:2] \\ (1-\eta H)^K \tilde{x}_{r-1}[3] \end{bmatrix} \ .$$

Now, using the calculations so far, we can write the updates between two communication rounds as,

$$\tilde{x}_r = \tilde{x}_{r-1} + \frac{\beta}{M} \sum_{m\in[M]} \left(\tilde{x}^m_{r,K} - \tilde{x}_{r-1}\right) \ ,$$

$$= \tilde{x}_{r-1} - \frac{\beta}{M} \sum_{m\in[M]} \begin{bmatrix} \tilde{\eta}\tau\hat{A}_{(m-1)\bmod(2)+1}\tilde{x}_{r-1}[1:2] \\ (1-(1-\eta H)^K)\tilde{x}_{r-1}[3] \end{bmatrix} \ ,$$

$$= \begin{bmatrix} (I_2 - \beta\tilde{\eta}A[1:2;1:2])\tilde{x}_{r-1}[1:2] \\ \left(1 - \beta\left(1-(1-\eta H)^K\right)\right)\tilde{x}_{r-1}[3] \end{bmatrix} \ .$$

The above calculation implies that the third coordinate evolves as synchronized gradient descent with $KR$ iterations, while the first two coordinates evolve with step size $\beta\tilde{\eta}$ and a hessian matrix of $A[1:2;1:2]$ (i.e., the top-left $2 \times 2$ block of $A$ the average Hessian) for $R$ iterations[7]. Now note that $A[1:2;1:2] = \tau(1-a)\hat{A}_1 + \tau a\hat{A}_2$ and

$$a := \begin{cases} 1/2 & \text{if } M \text{ is even,} \\ (M+1)/2M & \text{otherwise.} \end{cases}$$

Now, all we need to do is apply Lemma 9 to the first two dimensions. To be able to do so we need to be able to choose a condition number $\kappa = \Omega(R)$ for $A[1:2;1:2]$, in particular $\Omega(\kappa)$. Let us first consider the case with even machines, i.e., when $a = 1/2$. Then note that,

$$A[1:2;1:2] = \tau\frac{\hat{A}_1 + \hat{A}_2}{2} = \frac{\tau}{2}\begin{bmatrix} 1+\alpha^2 & \alpha\sqrt{1-\alpha^2} \\ \alpha\sqrt{1-\alpha^2} & 1-\alpha^2 \end{bmatrix} \ ,$$

---

[7]We don't need to restrict the step-sizes because Lemma 9 works for any step-size.

which implies for the spectrum of the matrix,

$$\text{Spec}(A[1:2;1:2]) = \frac{\tau}{2}\{1-\alpha, 1+\alpha\} \ ,$$

which in turn guarantees that,

$$\kappa\left(A[1:2;1:2]\right) = \frac{1+\alpha}{1-\alpha} \ ,$$

which can indeed be made $\Omega(R)$ by picking an $\alpha$ close enough to 1. Now let us look at the case when $M$ is odd and $a = \frac{M+1}{2M}$,

$$A[1:2;1:2] = \frac{\tau}{2M}\begin{bmatrix} M-1+(M+1)\alpha^2 & (M+1)\alpha\sqrt{1-\alpha^2} \\ (M+1)\alpha\sqrt{1-\alpha^2} & (M+1)\left(1-\alpha^2\right) \end{bmatrix} \ ,$$

which using simple calculations as before implies for the spectrum of the matrix,

$$\text{Spec}\left(A[1:2;1:2]\right) = \frac{\tau}{2M}\left\{M-\sqrt{1-\alpha^2+M^2\alpha^2}, M+\sqrt{1-\alpha^2+M^2\alpha^2}\right\} \ ,$$

which implies that,

$$\kappa\left(A[1:2;1:2]\right) = \frac{M+\sqrt{1-\alpha^2+M^2\alpha^2}}{M-\sqrt{1-\alpha^2+M^2\alpha^2}} \ ,$$

which can which can indeed be made $\Omega(R)$ by picking an $\alpha$ close enough to 1. Finally this allows us to use Lemma 9 which implies that the progress on the first two coordinates is lower bounded by $\frac{\tau B^2}{R}$ for any choice of hyperparameters. To make this more explicit, note the following for any model $\hat{x}$,

$$\begin{aligned}
F(\hat{x}) - F(x^\star) &= \frac{1}{2}x^T A x - (Ax^\star)^T x \ , \\
&= \frac{1}{2}x^T A x - (x^\star)^T A x \ , \\
&= \frac{1}{2}x[1:2]^T A[1:2;1:2]x[1:2] - (x^\star[1:2])^T A[1:2;1:2]x[1:2] \\
&\quad + \frac{H}{2}x[3]^2 - Hx^\star[3]x[3] \ , \\
&\geq \frac{1}{2}x[1:2]^T A[1:2;1:2]x[1:2] - (x^\star[1:2])^T A[1:2;1:2]x[1:2] \ , \\
&=: F_{1:2}(\hat{x}) - F_{1:2}(x^\star) \ ,
\end{aligned}$$

where we define a different quadratic objective $F_{1:2} : \mathbb{R}^2 \to \mathbb{R}^2$ using the top left two-dimensional block of the Hessian $A$. This implies that we can lower bound the sub-optimality $F(x_R) - F(x^\star)$ by $\frac{\tau B^2}{R}$, which finishes the proof of the lemma. $\qquad\square$

Now to conclude the proof, we first note the following tight lower bound for the homogeneous setting due to Glasgow et al. [23] for the local SGD iterate $\bar{x}_R$, which we recall also uses a quadratic hard instance satisfying Assumptions 1 to 3,

$$F(\bar{x}_R) - F(x^\star) = \Omega\left(\frac{HB^2}{KR} + \frac{\sigma B}{\sqrt{MKR}} + \min\left\{\frac{\sigma B}{\sqrt{KR}}, \frac{H^{1/3}\sigma^{2/3}B^{4/3}}{K^{1/3}R^{2/3}}\right\}\right) \ .$$

We also recall the heterogeneous lower bound due to Glasgow et al. [23] using a quadratic hard instance satisfying Assumptions 1 to 3, and apply it on $\tau$-smooth problems (instead of $H$-smooth in their construction, as they do not decouple $\tau$ and $H$ in their construction),

$$F(\bar{x}_R) - F(x^\star) = \Omega\left(\min\left\{\tau\phi_\star^2, \frac{\tau\phi_\star^{2/3}B^{4/3}}{R^{2/3}}\right\}\right) \ .$$

To translate their bound to our setting we also set $\zeta_\star$ (in their lower bound, not to be confused with our Assumption 4) as $\phi_\star\tau$ to account for the different definitions of first-order heterogeneity in their paper and ours (c.f., Assumption 5). Combining Lemma 10 with the above two lower bounds from Glasgow et al. [23] by placing different hard instances on disjoint co-ordinates and noting the independent evolution in the gradient descent iterates completes the proof of Theorem 1.

# D Notation and Outline of the Upper Bounds' Proofs

Recall that the algorithm we would like to analyze is local SGD in the intermittent communication setting. In particular, we assume the algorithm runs over $R \in \mathbb{N}$ communication rounds, with $K \in \mathbb{N}$ local update steps between each communication round and total $T = KR$ time steps. We also assume we have $M \in \mathbb{N}$ machines/clients/agents with each agent $m \in [M]$ sampling from their data distribution $\mathcal{D}_m \in \Delta(\mathcal{Z})$. These samples from the data distribution are used to calculate the stochastic gradients for each machine for each time step. In particular, at time $t \in [0, T]$ agent $m$ calculated $g_t^m := \nabla f(x_t^m; z_t^m)$ where $z_t^m \sim \mathcal{D}_m$. We recall the local SGD updates that use these stochastic gradients for all $t \in [0, T-1]$ and $m \in [M]$,

$$
\begin{aligned}
x_{t+1}^m &:= x_t^m - \eta g_t^m && \text{if} && t+1 \mod K \neq 0 \ , \\
x_{t+1}^m &:= \frac{1}{M} \sum_{n \in [M]} (x_t^n - \eta g_t^n) && \text{if} && t+1 \mod K = 0 \ .
\end{aligned}
$$

We will also define the "ghost iterate" for all times $t \in [0, T]$ which may or may not be physically computed depending on the time $t$,

$$
x_t := \frac{1}{M} \sum_{m \in [M]} x_t^m \ .
$$

Considering these iterations, we will define several quantities in the analyses throughout the appendix. We include this notation in Table 2 for ease of reference. With the above notation in mind, our analysis aims to provide upper bounds for $A(KR)$ and $E(KR)$ as a function of problem-dependent parameters that appear in all our assumptions. To do this:

- We will first state some technical lemmas in Appendix E.

- Then in Appendix F we state recursions across communication rounds for the sequences $A(\cdot)$, $B(\cdot)$, and $E(\cdot)$ in terms of the consensus error sequences $C(\cdot)$ and $D(\cdot)$. These recursions[8] highlight the need to control the consensus error sequences $C(\cdot)$ and $D(\cdot)$.

- In Appendix G we first control the consensus error by relying on the strongest Assumption 7. In the following sections, we relax this need for the $\zeta$ assumption and do a more fine-grained analysis of the consensus error.

- In Appendix H we provide more fine-grained recursions for $C(\cdot)$ and $D(\cdot)$, which depend on $A(\cdot)$ and $B(\cdot)$, i.e., they leads to coupled recursions. The main technical contribution of this paper is providing these coupled recursions and using them to provide new upper bounds.

- Appendix I then brings together the results from Appendix F and Appendix H and provides convergence guarantees in terms of the step size $\eta$. Then we tune the step-size and obtain all the upper bounds from the main body of the paper.

# E Useful Technical Lemmas

## E.1 Useful Facts about Stochastic Noise

Throughout this sub-section, we will assume Assumption 2. Recall the following standard lemmas about the stochastic gradient noise,

**Lemma 11** (Averaged Stochastic Noise Second Moment). *For $t \in [0, T-1]$ we have,*

$$
\mathbb{E}\left[\|\xi_t\|_2^2\right] \leq \frac{\sigma^2}{M} \ .
$$

---

[8]We note that we are less explicit about randomness in the proof of these recursions and the following results. In particular, we often omit repetitive steps using the tower rule and conditional expectations to shorten the already complex proofs. We urge the reader to familiarize themselves with applying these techniques by first reading the proof of Lemma 20.

| Symbol | Definition |
|--------|-----------|
| $A(t)$ | $\mathbb{E}\left[\|x_t - x^\star\|_2^2\right],\ \forall\, t \in [0, T]$ |
| $B(t)$ | $\mathbb{E}\left[\|x_t - x^\star\|_2^4\right],\ \forall\, t \in [0, T]$ |
| $C(t)$ | $\frac{1}{M^2}\sum_{m,n \in [M]} \mathbb{E}\left[\|x_t^m - x_t^n\|_2^2\right],\ \forall\, t \in [0, T]$ |
| $D(t)$ | $\frac{1}{M^2}\sum_{m,n \in [M]} \mathbb{E}\left[\|x_t^m - x_t^n\|_2^4\right],\ \forall\, t \in [0, T]$ |
| $E(t)$ | $\mathbb{E}\left[F(x_t)\right] - \min_{x^\star \in \mathbb{R}^d} F(x^\star),\ \forall\, t \in [0, T]$ |
| $\delta(t)$ | $t - t \mod (K),\ \forall\, t \in [0, T]$ |
| $g_t^m$ | $\nabla f(x_t^m; z_t^m),\ z_t^m \sim \mathcal{D}_m,\ \forall\, t \in [0, T],\ m \in [M]$ |
| $\xi_t^m$ | $\nabla F_m(x_t^m) - \nabla f(x_t^m; z_t^m),\ z_t^m \sim \mathcal{D}_m,\ \forall\, t \in [0, T],\ m \in [M]$ |
| $g_t$ | $g_t := \frac{1}{M}\sum_{m \in [M]} g_t^m,\ \forall\, t \in [0, T]$ |
| $\xi_t$ | $\xi_t := \frac{1}{M}\sum_{m \in [M]} \xi_t^m,\ \forall\, t \in [0, T]$ |
| $\mathcal{H}_t$ | $\sigma\left(\{z_0^m\}_{m=1}^M, \ldots, \{z_{t-1}^m\}_{m=1}^M\right),\ \forall\, t \in [1, T]$ |

Table 2: Summary of the notation used in the appendix. In the definition of the filtration $\mathcal{H}_t$, we abuse the notation (c.f., Assumption 2) and use $\sigma(X)$ to denote the sigma algebra defined by the set of random variables $X$.

*Proof.* Recall that at any time step $t \in [0, T-1]$,

$$
\mathbb{E}\left[\|\xi_t\|_2^2\right] = \mathbb{E}\left[\left\|\frac{1}{M}\sum_{m \in [M]}\xi_t^m\right\|_2^2\right]\,,
$$

$$
\overset{\text{(Tower rule)}}{=} \mathbb{E}\left[\mathbb{E}\left[\left\|\frac{1}{M}\sum_{m \in [M]}(g_t^m - \nabla f(x_t^m; z_t^m))\right\|_2^2 \Big| \mathcal{H}_t\right]\right]\,,
$$

$$
\overset{\text{(a)}}{=} \mathbb{E}\left[\frac{1}{M^2}\sum_{m \in [M]}\mathbb{E}\left[\|(g_t^m - \nabla f(x_t^m; z_t^m))\|_2^2 | \mathcal{H}_t\right]\right]\,,
$$

$$
\overset{\text{(Assumption 2)}}{\leq} \frac{1}{M^2}\sum_{m \in [M]}\sigma^2 = \frac{\sigma^2}{M}\,,
$$

where (a) uses the fact that for all $m \neq n$, $z_t^m \perp z_t^n | \mathcal{H}_t$, i.e., $\xi_t^1, \ldots, \xi_t^M$ are independent conditioned on the history $\mathcal{H}_t$. $\qquad \square$

We can also give the following stronger bound on the fourth moment of the stochastic noise.

**Lemma 12** (Averaged Stochastic Noise Fourth Moment). *For $t \in [0, T-1]$ we have,*

$$\mathbb{E} \left[ \|\xi_t\|_2^4 \right] \leq \frac{3\sigma^4}{M^2} .$$

*Proof.* Recall that at any time step $t \in [0, T-1]$,

$$\mathbb{E} \left[ \|\xi_t\|_2^4 \right] = \mathbb{E} \left[ \left\| \frac{1}{M} \sum_{m \in [M]} \xi_t^m \right\|_2^4 \right] ,$$

$$= \mathbb{E} \left[ \left( \left\| \frac{1}{M} \sum_{m \in [M]} \xi_t^m \right\|_2^2 \right)^2 \right] = \mathbb{E} \left[ \left( \frac{1}{M^2} \sum_{m,n \in [M]} \langle \xi_t^m, \xi_t^n \rangle \right)^2 \right] ,$$

$$= \frac{1}{M^4} \sum_{l,m,n,o \in [M]} \mathbb{E} \left[ \langle \xi_t^l, \xi_t^m \rangle \langle \xi_t^n, \xi_t^o \rangle \right] ,$$

$$\overset{\text{(Tower Rule)}}{=} \frac{1}{M^4} \sum_{l,m,n,o \in [M]} \mathbb{E} \left[ \mathbb{E} \left[ \langle \xi_t^l, \xi_t^m \rangle \langle \xi_t^n, \xi_t^o \rangle | \mathcal{H}_t \right] \right] ,$$

Recall that for all $m \neq n$, $z_t^m \perp z_t^n | \mathcal{H}_t$, i.e., $\xi_t^1, \ldots, \xi_t^M$ are independent conditioned on the history $\mathcal{H}_t$. In the above sum, the only non-zero terms are the ones where either $l = m = n = o$, or where the set $\{l, m, n, o\}$ has two distinct values, each repeated twice. There are $M$ terms of the first kind, and $3M(M-1)$ terms of the second kind (first choose two colours out of $M$, then choose two indices out of $\{l, m, n, o\}$ which divides into two groups, i.e., total $\frac{M(M-1)}{2} \times \frac{4!}{2!2!}$). Using this we get,

$$\mathbb{E} \left[ \|\xi_t\|_2^4 \right] = \frac{1}{M^4} \left( \sum_{l \in [M]} \mathbb{E} \left[ \|\xi_t^l\|_2^4 \right] + 3 \sum_{l \neq m \in [M]} \mathbb{E} \left[ \|\xi_t^l\|_2^2 \right] \mathbb{E} \left[ \|\xi_t^m\|_2^2 \right] \right) ,$$

$$\leq \frac{1}{M^4} \left( M\sigma^4 + 3M(M-1)\sigma^4 \right) ,$$

$$\leq \frac{3\sigma^4}{M^2} ,$$

which proves the lemma. $\qquad \square$

**Lemma 13** (Averaged Stochastic Noise Third Moment). *For $t \in [0, T-1]$ we have,*

$$\mathbb{E} \left[ \|\xi_t\|_2^3 \right] \leq \frac{\sqrt{3}\sigma^3}{M^{3/2}} .$$

*Proof.* This result follows from simply noting the previous two lemmas, and the fact that,

$$\mathbb{E} \left[ \|\xi_t\|_2^3 \right] = \mathbb{E} \left[ \|\xi_t\|_2^2 \|\xi_t\|_2 \right] ,$$

$$\overset{\text{Cauchy Shwartz}}{\leq} \sqrt{\mathbb{E} \left[ \|\xi_t\|_2^4 \right]} \sqrt{\mathbb{E} \left[ \|\xi_t\|_2^2 \right]} ,$$

$$\overset{\text{Lemmas 11 and 12}}{\leq} \sqrt{\frac{\sigma^2}{M}} \sqrt{\frac{3\sigma^4}{M^2}} ,$$

$$\leq \frac{\sqrt{3}\sigma^3}{M^{3/2}} ,$$

which proves the lemma. $\qquad \square$

We can also note the following about the difference of the stochastic noise in two machines.

**Lemma 14** (Second Moment of Difference). *For $t \in [0, T-1]$ and for $m \neq n \in [M]$ we have,*

$$\mathbb{E}\left[\|\xi_t^m - \xi_t^n\|_2^2\right] \leq 2\sigma^2 \ .$$

*Proof.* Note the following for $m \neq n \in [M]$, and for $t \in [0, T-1]$

$$\mathbb{E}\left[\|\xi_t^m - \xi_t^n\|_2^2\right] = \mathbb{E}\left[\|\xi_t^m\|_2^2 + \|\xi_t^m\|_2^2 - 2\langle\xi_t^m, \xi_t^n\rangle\right] \ ,$$

$$=^{\text{(a), (Tower Rule)}} \mathbb{E}\left[\|\xi_t^m\|_2^2\right] + \mathbb{E}\left[\|\xi_t^n\|_2^2\right] - 2\mathbb{E}\left[\langle\mathbb{E}\left[\xi_t^m|\mathcal{H}_t\right], \mathbb{E}\left[\xi_t^n|\mathcal{H}_t\right]\rangle\right] \ ,$$

$$\leq^{\text{(Assumption 2), (b)}} 2\sigma^2 \ ,$$

where in (a) we used that $\xi_t^m \perp \xi_t^n|\mathcal{H}_t$; and in (b) we used that $\mathbb{E}\left[\xi_t^m|\mathcal{H}_t\right] = \mathbb{E}\left[\xi_t^n|\mathcal{H}_t\right] = 0$. This proves the lemma. $\qquad\square$

**Lemma 15** (Fourth Moment of Difference). *For $t \in [0, T-1]$ and for $m \neq n \in [M]$ we have,*

$$\mathbb{E}\left[\|\xi_t^m - \xi_t^n\|_2^4\right] \leq 8\sigma^4 \ .$$

*Proof.* Note the following for $m \neq n \in [M]$, and for $t \in [0, T-1]$

$$\mathbb{E}\left[\|\xi_t^m - \xi_t^n\|_2^4\right] = \mathbb{E}\left[\left(\|\xi_t^m\|_2^2 + \|\xi_t^m\|_2^2 - 2\langle\xi_t^m, \xi_t^n\rangle\right)^2\right] \ ,$$

$$=^{\text{(a)}} \mathbb{E}\left[\|\xi_t^m\|_2^4\right] + \mathbb{E}\left[\|\xi_t^n\|_2^4\right] + 4\mathbb{E}\left[(\langle\xi_t^m, \xi_t^n\rangle)^2\right]$$

$$+ 2\mathbb{E}\left[\|\xi_t^m\|_2^2\right]\mathbb{E}\left[\|\xi_t^n\|_2^2\right] - 2\mathbb{E}\left[\|\xi_t^m\|_2^2 \xi_t^m\right]^T \mathbb{E}[\xi_t^n]^{\cancel{0}}$$

$$- 2\mathbb{E}\left[\|\xi_t^n\|_2^2 \xi_t^n\right]^T \mathbb{E}[\xi_t^m]^{\cancel{0}},$$

$$= \mathbb{E}\left[\|\xi_t^m\|_2^4\right] + \mathbb{E}\left[\|\xi_t^n\|_2^4\right] + 6\mathbb{E}\left[\|\xi_t^m\|_2^2\right]\mathbb{E}\left[\|\xi_t^n\|_2^2\right] \ ,$$

$$\leq^{\text{(Assumption 2)}} 8\sigma^4 \ ,$$

where in (a) we used that $\xi_t^m \perp \xi_t^n|\mathcal{H}_t$ along with tower rule several times like in previous lemmas. This finishes the proof. $\qquad\square$

### E.2 Other Analytical Lemmas

We will also use the following inequality several times, essentially a variant of the A.M.-G.M. inequality.

**Lemma 16.** *For any $a, b \in \mathbb{R}$ and $\gamma > 0$ we have,*

$$(a+b)^2 \leq \left(1 + \frac{1}{\gamma}\right)a^2 + (1+\gamma)b^2 \ ,$$

$$(a+b)^4 \leq \left(1 + \frac{1}{\gamma}\right)^3 a^4 + (1+\gamma)^3 b^4 \ .$$

*Proof.* Note the following,

$$(a+b)^2 = a^2 + b^2 + 2ab \ ,$$

$$= a^2 + b^2 + 2\left(\frac{a}{\sqrt{\gamma}}\right)(\sqrt{\gamma}b) \ ,$$

$$\leq^{\text{(A.M.-G.M. Inequality)}} a^2 + b^2 + \frac{a^2}{\gamma} + \gamma b^2 \ ,$$

$$\leq \left(1 + \frac{1}{\gamma}\right)a^2 + (1+\gamma)b^2 \ ,$$

which proves the first statement of the lemma. To get the second statement we will just apply the first statement twice as follows,

$$(a+b)^4 \leq \left( \left(1 + \frac{1}{\gamma}\right) a^2 + (1+\gamma) b^2 \right)^2 ,$$

$$\leq \left(1 + \frac{1}{\gamma}\right) \left( \left(1 + \frac{1}{\gamma}\right) a^2 \right)^2 + (1+\gamma) \left( (1+\gamma) b^2 \right)^2 ,$$

$$= \left(1 + \frac{1}{\gamma}\right)^3 a^4 + (1+\gamma)^3 b^4 ,$$

which proves the second statement of the lemma. □

**Lemma 17.** *For any $a, b, c \in \mathbb{R}$ we have,*

$$(a+b+c)^2 \leq 3a^2 + 3b^2 + 3c^2 ,$$
$$(a+b+c)^4 \leq 27a^4 + 27b^4 + 27c^4 .$$

*Proof.* We note the following,

$$(a+b+c)^2 = a^2 + b^2 + c^2 + 2ab + 2bc + 2ca ,$$
$$\leq^{\text{(A.M.-G.M. inequality)}} a^2 + b^2 + c^2 + (a^2 + b^2) + (b^2 + c^2) + (c^2 + a^2) ,$$
$$= 3(a^2 + b^2 + c^2) ,$$

which proves the first statement. For the second statement using the first statement note the following,

$$(a+b+c)^4 \leq \left(3a^2 + 3b^2 + 3c^2\right)^2 ,$$
$$\leq 3 \left(3a^2\right)^2 + 3 \left(3b^2\right)^2 + 3 \left(3c^2\right)^2 ,$$
$$= 27a^4 + 27b^4 + 27c^4 ,$$

which proves the lemma. □

**Lemma 18.** *Let $x \in (0,1)$ and $K > 1$ then we have*

$$\sum_{i=1}^{K-1} x^{i-1} i^2 \leq \frac{K}{(1-x)^2} .$$

*Proof.* Note the following,

$$\sum_{i=1}^{K-1} x^{i-1} i^2 \leq K \sum_{i=1}^{K-1} i x^{i-1} ,$$

$$= K \nabla_x \left( \sum_{i=1}^{K-1} x^i \right) ,$$

$$= K \nabla_x \left( x \frac{1 - x^K}{1 - x} \right) ,$$

$$= K \frac{1 - x^K}{1 - x} + K x \frac{1 - K x^{K-1} + (K-1) x^K}{(1-x)^2} ,$$

$$= K \frac{1 - x^K - x + x^{K+1}}{(1-x)^2} + K \frac{x - K x^K + (K-1) x^{K+1}}{(1-x)^2} ,$$

$$= K \frac{1 - (K+1) x^K + K x^{K+1}}{(1-x)^2} ,$$

$$\leq \frac{K}{(1-x)^2} ,$$

where in the last inequality we just note that $1 - (K+1)x^K + Kx^{K+1} \leq 1$. This proves the lemma. □

**Lemma 19.** *Let $A$ and $B$ be two positive-semi definite matrices. We have:*

$$A^k - B^k = \sum_{j=0}^{k-1} A^{k-1-j}(A-B)B^j$$

*Proof.* we prove by induction. For $k = 1$ we have:

$$A - B = \sum_{j=0}^{0} A^{-j}(A-B)B^j = A - B$$

for $k + 1$ we have:

$$A^{k+1} - B^{k+1} = AA^k - BB^k = AA^k - AB^k + AB^k - BB^k = A(A^k - B^k) + (A-B)B^k$$

for the first term in the above equality we have:

$$A(A^k - B^k) = A\sum_{j=0}^{k-1} A^{k-1-j}(A-B)B^j = \sum_{j=0}^{k-1} A^{k-j}(A-B)B^j$$

By adding the second term we have:

$$A^{k+1} - B^{k+1} = \sum_{j=0}^{k-1} A^{k-j}(A-B)B^j + (A-B)B^k = \sum_{j=0}^{k} A^{k-j}(A-B)B^j$$

which completes the proof. $\square$

# F   Deriving Round-wise Recursions for Errors

In this section, we derive several recursions that prove useful later in the analysis and form the core of our proof. An informed reader would note that the ideas and in some cases the entire recursions occur in previous works [25, 20, 18, 15].

## F.1   Second Moment of the Error in Iterates

The main result of this sub-section is the following result, which relates $A(\cdot)$ to $C(\cdot)$ and $D(\cdot)$.

**Lemma 20.** *Assume we have a problem instance satisfying Assumptions 1, 2 and 6. Then assuming $\eta < \frac{1}{H}$ we have for all $t \in [0, T-1]$,*

$$A(t+1) \leq (1 - \eta\mu)\, A(t) + \frac{\eta}{\mu} \cdot \min\left\{ {\color{red}2Q^2 D(t) + 2\tau^2 C(t)}, {\color{blue}H^2 C(t)} \right\} + \frac{\eta^2\sigma^2}{M}\ .$$

*This also implies that for all $r \in [R]$,*

$$A(Kr) \leq (1 - \eta\mu)^K A(K(r-1)) + \left(1 - (1-\eta\mu)^K\right) \frac{\eta\sigma^2}{\mu M}$$

$$+ \frac{\eta}{\mu} \sum_{j=K(r-1)}^{Kr-1} (1-\eta\mu)^{Kr-1-j} \min\left\{ {\color{red}2Q^2 D(j) + 2\tau^2 C(j)}, {\color{blue}H^2 C(j)} \right\}\ .$$

**Remark 6.** Note that the above lemma implies that if $Q$ and $\tau$ are both zero—i.e., we have a quadratic problem with no second-order heterogeneity—then we will achieve extreme communication efficiency, matching the convergence rate of mini-batch SGD, with $KR$ communication rounds. As such, the trade-off between the {\color{red}red} and {\color{blue}blue} upper bounds is that the former allows us to exploit higher-order assumptions, but we need to be able to bound the fourth moment of the consensus error. In contrast, the latter only requires a bound on the second moment of the consensus error.

*Proof.* We note the following about the progress made in a single iteration for $t \in [0, T-1]$,

$$A(t+1) = \mathbb{E}\left[\|x_{t+1} - x^\star\|_2^2\right]\ ,$$

$$\overset{\text{(Tower rule)}}{=} \mathbb{E}\left[\mathbb{E}\left[\left\|x_t - x^\star - \frac{\eta}{M}\sum_{m\in[M]} g_t^m\right\|_2^2 \Bigg| \mathcal{H}_t\right]\right] \, ,$$

$$\overset{\text{(a)}}{=} \mathbb{E}\left[\left\|x_t - x^\star - \frac{\eta}{M}\sum_{m\in[M]}\nabla F_m(x_t^m)\right\|_2^2\right] + \eta^2\mathbb{E}\left[\left\|\frac{1}{M}\sum_{m\in[M]}\xi_t^m\right\|_2^2\right] \, ,$$

$$\overset{\text{(Lemma 11)}}{\leq} \mathbb{E}\left[\left\|x_t - \eta\nabla F(x_t) - x^\star + \eta\nabla F(x_t) - \frac{\eta}{M}\sum_{m\in[M]}\nabla F_m(x_t^m)\right\|_2^2\right] + \frac{\eta^2\sigma^2}{M} \, ,$$

$$\overset{\text{(Lemma 16), (b)}}{\leq} \left(1 + \frac{\eta\mu}{1-\eta\mu}\right)(1-\eta\mu)^2\,\mathbb{E}\left[\|x_t - x^\star\|_2^2\right]$$

$$+ \left(1 + \frac{1-\eta\mu}{\eta\mu}\right)\eta^2\mathbb{E}\left[\left\|\frac{1}{M}\sum_{m\in[M]}(\nabla F_m(x_t) - \nabla F_m(x_t^m))\right\|_2^2\right] + \frac{\eta^2\sigma^2}{M} \, ,$$

$$= (1-\eta\mu)\,\mathbb{E}\left[\|x_t - x^\star\|_2^2\right] + \frac{\eta}{\mu}\cdot\mathbb{E}\left[\left\|\frac{1}{M}\sum_{m\in[M]}(\nabla F_m(x_t) - \nabla F_m(x_t^m))\right\|_2^2\right] + \frac{\eta^2\sigma^2}{M} \, ,$$

$$\overset{\text{(c)}}{\leq} (1-\eta\mu)\,\mathbb{E}\left[\|x_t - x^\star\|_2^2\right] + \frac{\eta}{\mu}\cdot\mathbb{E}\left[\left\|\frac{1}{M}\sum_{m\in[M]}\nabla^2 F_m(\hat{x}_t^m)(x_t - x_t^m)\right\|_2^2\right] + \frac{\eta^2\sigma^2}{M} \, ,$$

where in (a) we used the fact that $\xi_t^1,\ldots,\xi_t^1 \in m\mathcal{H}_t$ i.e., they are measurable/"non-random" under $\mathcal{H}_t$ and zero-mean, which allows us to ignore the cross-terms while squaring; in (b) we use the fact that $\eta < 1/H$ which implies that $0 \preceq I - \eta\nabla^2 F(\cdot) \preceq (1-\eta\mu)\cdot I_d$ and also that $(1-\eta\mu) > 0$; and in (c) we note that due to the mean-value theorem there exists some $\hat{x}_t^m$ which is a convex combination of $x_t^m$ and $x_t$. From this point, we can proceed in two different ways. First, to get the blue upper bound we just use smoothness as follows,

$$A(t+1)$$

$$\leq (1-\eta\mu)\,\mathbb{E}\left[\|x_t - x^\star\|_2^2\right] + \frac{\eta}{\mu}\cdot\mathbb{E}\left[\left\|\frac{1}{M}\sum_{m\in[M]}\nabla^2 F_m(\hat{x}_t^m)(x_t - x_t^m)\right\|_2^2\right] + \frac{\eta^2\sigma^2}{M} \, ,$$

$$\overset{\text{(Jensen's Inequality)}}{\leq} (1-\eta\mu)\,\mathbb{E}\left[\|x_t - x^\star\|_2^2\right] + \frac{\eta}{\mu}\cdot\frac{1}{M}\sum_{m\in[M]}\mathbb{E}\left[\|\nabla^2 F_m(\hat{x}_t^m)(x_t - x_t^m)\|_2^2\right]$$

$$+ \frac{\eta^2\sigma^2}{M} \, ,$$

$$\leq (1-\eta\mu)\,\mathbb{E}\left[\|x_t - x^\star\|_2^2\right] + \frac{\eta H^2}{\mu}\cdot\frac{1}{M}\sum_{m\in[M]}\mathbb{E}\left[\|x_t^m - x_t\|_2^2\right] + \frac{\eta^2\sigma^2}{M} \, ,$$

$$\overset{\text{(Jensen's Inequality)}}{\leq} (1-\eta\mu)\,\mathbb{E}\left[\|x_t - x^\star\|_2^2\right] + \frac{\eta H^2}{\mu}\cdot\frac{1}{M^2}\sum_{m,n\in[M]}\mathbb{E}\left[\|x_t^m - x_t^n\|_2^2\right] + \frac{\eta^2\sigma^2}{M} \, ,$$

$$= (1-\eta\mu)\,A(t) + \frac{\eta H^2}{\mu}C(t) + \frac{\eta^2\sigma^2}{M} \, ,$$

which proves one part of the lemma. To get the red upper bound, we will use second-order heterogeneity and third-order smoothness as follows,

$$A(t+1)$$

$$\leq (1 - \eta\mu) \, \mathbb{E}\left[\|x_t - x^\star\|_2^2\right] + \frac{\eta}{\mu} \cdot \mathbb{E}\left[\left\|\frac{1}{M}\sum_{m\in[M]} \nabla^2 F_m(\hat{x}_t^m)(x_t - x_t^m)\right\|_2^2\right] + \frac{\eta^2\sigma^2}{M} \quad,$$

$$= (1 - \eta\mu) \, \mathbb{E}\left[\|x_t - x^\star\|_2^2\right]$$

$$+ \frac{\eta}{\mu} \cdot \mathbb{E}\left[\left\|\frac{1}{M}\sum_{m\in[M]} \left(\nabla^2 F_m(\hat{x}_t^m) - \nabla^2 F_m(x_t) + \nabla^2 F_m(x_t) - \nabla^2 F(x_t)\right)(x_t - x_t^m)\right\|_2^2\right]$$

$$+ \frac{\eta^2\sigma^2}{M} \quad,$$

$$\leq^{\text{(Jensen's Inequality), (Lemma 16)}} (1 - \eta\mu) \, \mathbb{E}\left[\|x_t - x^\star\|_2^2\right]$$

$$+ \frac{2\eta}{\mu M}\sum_{m\in[M]} \mathbb{E}\left[\left\|\left(\nabla^2 F_m(\hat{x}_t^m) - \nabla^2 F_m(x_t)\right)(x_t - x_t^m)\right\|_2^2\right]$$

$$+ \frac{2\eta}{\mu M}\sum_{m\in[M]} \mathbb{E}\left[\left\|\left(\nabla^2 F(x_t) - \nabla^2 F_m(x_t)\right)(x_t - x_t^m)\right\|_2^2\right] + \frac{\eta^2\sigma^2}{M} \quad,$$

$$\leq^{\text{(Assumptions 1 and 6)}} (1 - \eta\mu) \, \mathbb{E}\left[\|x_t - x^\star\|_2^2\right] + \frac{2\eta Q^2}{\mu M}\sum_{m\in[M]} \mathbb{E}\left[\|\hat{x}_t^m - x_t\|_2^2 \, \|x_t - x_t^m\|_2^2\right]$$

$$+ \frac{2\eta\tau^2}{\mu M}\sum_{m\in[M]} \mathbb{E}\left[\|x_t - x_t^m\|_2^2\right] + \frac{\eta^2\sigma^2}{M} \quad,$$

$$\leq^{\text{(a)}} (1 - \eta\mu) \, \mathbb{E}\left[\|x_t - x^\star\|_2^2\right] + \frac{2\eta Q^2}{\mu M}\sum_{m\in[M]} \mathbb{E}\left[\|x_t - x_t^m\|_2^4\right] + \frac{2\eta\tau^2}{\mu M}\sum_{m\in[M]} \mathbb{E}\left[\|x_t - x_t^m\|_2^2\right]$$

$$+ \frac{\eta^2\sigma^2}{M} \quad,$$

$$\leq^{\text{(Jensen's Inequality)}} (1 - \eta\mu) \, \mathbb{E}\left[\|x_t - x^\star\|_2^2\right] + \frac{2\eta Q^2}{\mu M^2}\sum_{m,n\in[M]} \mathbb{E}\left[\|x_t^n - x_t^m\|_2^4\right]$$

$$+ \frac{2\eta\tau^2}{\mu M^2}\sum_{m,n\in[M]} \mathbb{E}\left[\|x_t^n - x_t^m\|_2^2\right] + \frac{\eta^2\sigma^2}{M} \quad,$$

$$= (1 - \eta\mu) \, A(t) + \frac{2\eta Q^2 D(t)}{\mu} + \frac{2\eta\tau^2 C(t)}{\mu} + \frac{\eta^2\sigma^2}{M} \quad,$$

where in (a) we use that $\|\hat{x}_t^m - x_t\|_2 \leq \|x_t^m - x_t\|_2$ for all $m \in [M]$. This proves the second part of the upper bound, thus finishing the proof for the first statement of the lemma. Note that for $r \in [R]$ we can re-write this result as follows,

$$A(Kr) \leq (1 - \eta\mu) \, A(Kr - 1)$$

$$+ \frac{\eta}{\mu} \cdot \min\left\{2Q^2 D(Kr - 1) + 2\tau^2 C(Kr - 1), H^2 C(Kr - 1)\right\} + \frac{\eta^2\sigma^2}{M} \quad,$$

$$\leq (1 - \eta\mu)^K A(K(r - 1))$$

$$+ \frac{\eta}{\mu} \sum_{j=K(r-1)}^{Kr-1} (1 - \eta\mu)^{Kr-1-j} \min\left\{2Q^2 D(j) + 2\tau^2 C(j), H^2 C(j)\right\} + \frac{\eta\sigma^2}{\mu M} \quad,$$

where in the second inequality we just unrolled the recursion till the time-step of the previous communication. This finishes the proof of the lemma. $\qquad \square$

It would also be helpful to state the following lemma, which talks about the convergence on individual machines between two communication rounds.

**Lemma 21** (Single Machine SGD Second Moment). *For any machine $m \in [M]$, for $t \in [0, T]$, and for $k \geq 0$ we have the following for $\eta < \frac{1}{H}$,*

$$\mathbb{E}\left[\left\|x_{\delta(t)+k}^m - x^\star\right\|_2^2\right] \leq (1 - \eta\mu)^{2k}\mathbb{E}\left[\left\|x_{\delta(t)} - x_m^\star\right\|_2^2\right] + \frac{\eta\sigma^2}{\mu} \ .$$

The above lemma follows the usual strongly convex analysis of SGD (see, for instance, [60]), since we can rely on that between two communication rounds.

## F.2 Fourth Moment of the Error in Iterates

**Lemma 22.** *Assume we have a problem instance satisfying Assumptions 1, 2 and 6. Then assuming $\eta < \frac{1}{H}$ we have for all $t \in [0, T-1]$,*

$$B(t+1) \leq (1 - \eta\mu)B(t) + \left(\frac{\eta H^4}{\mu^3} + \frac{16\eta^3\sigma^2 Q^2}{\mu M}\right)D(t) + \frac{8\eta^2\sigma^2(1 - \eta\mu)}{M}A(t)$$

$$+ \frac{16\eta^3\sigma^2\tau^2}{\mu M}C(t) + \frac{9\eta^4\sigma^4}{M^2} \ .$$

*We can also get the following simpler upper bound,*

$$B(t+1) \leq (1 - \eta\mu)B(t) + \frac{\eta H^4}{\mu^3}D(t) + \frac{8\eta^2\sigma^2(1 - \eta\mu)}{M}A(t) + \frac{8\eta^3\sigma^2 H^2}{\mu M}C(t) + \frac{9\eta^4\sigma^4}{M^2} \ .$$

*This also implies that for $r \in [R]$ we have,*

$$B(Kr) \leq (1 - \eta\mu)^K B(K(r-1)) + \left(\frac{\eta H^4}{\mu^3} + \frac{16\eta^3\sigma^2 Q^2}{\mu M}\right) \sum_{j=K(r-1)}^{Kr-1} (1 - \eta\mu)^{Kr-1-j}D(j)$$

$$+ \frac{8\eta^2\sigma^2}{M} \sum_{j=K(r-1)}^{Kr-1} (1 - \eta\mu)^{Kr-j}A(j) + \frac{16\eta^3\sigma^2\tau^2}{\mu M} \sum_{j=K(r-1)}^{Kr-1} (1 - \eta\mu)^{Kr-1-j}C(j)$$

$$+ \frac{9\eta^3\sigma^4}{\mu M^2} \ .$$

*Proof.* For $t \in [0, T-1]$ we note the following,

$$\mathbb{E}\left[\|x_{t+1} - x^\star\|_2^4\right]$$

$$= \mathbb{E}\left[\left\|x_t - x^\star - \frac{\eta}{M}\sum_{m\in[M]}\nabla F_m(x_t^m) + \frac{\eta}{M}\sum_{m\in[M]}\xi_t^m\right\|_2^4\right] \ ,$$

$$= \mathbb{E}\left[\left(\left\|x_t - x^\star - \frac{\eta}{M}\sum_{m\in[M]}\nabla F_m(x_t^m)\right\|_2^2 + \left\|\frac{\eta}{M}\sum_{m\in[M]}\xi_t^m\right\|_2^2\right.\right.$$

$$\left.\left. + 2\eta\left\langle x_t - x^\star - \frac{\eta}{M}\sum_{m\in[M]}\nabla F_m(x_t^m), \frac{1}{M}\sum_{m\in[M]}\xi_t^m\right\rangle\right)^2\right] \ ,$$

$$= \mathbb{E}\left[\left\|x_t - x^\star - \frac{\eta}{M}\sum_{m\in[M]}\nabla F_m(x_t^m)\right\|_2^4\right] + \eta^4\mathbb{E}\left[\left\|\frac{1}{M}\sum_{m\in[M]}\xi_t^m\right\|_2^4\right]$$

$$+ 4\eta^2\mathbb{E}\left[\left(\left\langle x_t - x^\star - \frac{\eta}{M}\sum_{m\in[M]}\nabla F_m(x_t^m), \frac{1}{M}\sum_{m\in[M]}\xi_t^m\right\rangle\right)^2\right]$$

$$+ 2\eta^2 \mathbb{E}\left[\left\|x_t - x^\star - \frac{\eta}{M}\sum_{m\in[M]} \nabla F_m(x_t^m)\right\|_2^2 \left\|\frac{1}{M}\sum_{m\in[M]}\xi_t^m\right\|_2^2\right]$$

$$+ 4\eta \mathbb{E}\left[\left\|x_t - x^\star - \frac{\eta}{M}\sum_{m\in[M]} \nabla F_m(x_t^m)\right\|_2^2 \left(x_t - x^\star - \frac{\eta}{M}\sum_{m\in[M]} \nabla F_m(x_t^m)\right)\right]^T$$

$$\times\ \mathbb{E}\left[\frac{1}{M}\sum_{m\in[M]}\xi_t^m\right]^{\ 0}$$

$$+ 4\eta^3 \mathbb{E}\left[\left\|x_t - x^\star - \frac{\eta}{M}\sum_{m\in[M]} \nabla F_m(x_t^m)\right\|_2 \left\|\frac{1}{M}\sum_{m\in[M]}\xi_t^m\right\|_2^3\right],$$

$$\overset{\text{(Cauchy Shwartz)}}{\leq} \mathbb{E}\left[\left\|x_t - x^\star - \frac{\eta}{M}\sum_{m\in[M]} \nabla F_m(x_t^m)\right\|_2^4\right] + \eta^4 \mathbb{E}\left[\left\|\frac{1}{M}\sum_{m\in[M]}\xi_t^m\right\|_2^4\right]$$

$$+ 4\eta^2 \mathbb{E}\left[\left(\left\|x_t - x^\star - \frac{\eta}{M}\sum_{m\in[M]} \nabla F_m(x_t^m)\right\|_2 \left\|\frac{1}{M}\sum_{m\in[M]}\xi_t^m\right\|_2\right)^2\right]$$

$$+ 2\eta^2 \mathbb{E}\left[\left\|x_t - x^\star - \frac{\eta}{M}\sum_{m\in[M]} \nabla F_m(x_t^m)\right\|_2^2 \left\|\frac{1}{M}\sum_{m\in[M]}\xi_t^m\right\|_2^2\right]$$

$$+ 4\eta^3 \mathbb{E}\left[\left\|x_t - x^\star - \frac{\eta}{M}\sum_{m\in[M]} \nabla F_m(x_t^m)\right\|_2 \left\|\frac{1}{M}\sum_{m\in[M]}\xi_t^m\right\|_2^3\right],$$

$$\overset{\text{(Tower Rule)}}{=} \mathbb{E}\left[\left\|x_t - x^\star - \frac{\eta}{M}\sum_{m\in[M]} \nabla F_m(x_t^m)\right\|_2^4\right] + \eta^4 \mathbb{E}\left[\left\|\frac{1}{M}\sum_{m\in[M]}\xi_t^m\right\|_2^4\right]$$

$$+ 6\eta^2 \mathbb{E}\left[\mathbb{E}\left[\left\|x_t - x^\star - \frac{\eta}{M}\sum_{m\in[M]} \nabla F_m(x_t^m)\right\|_2^2 \left\|\frac{1}{M}\sum_{m\in[M]}\xi_t^m\right\|_2^2 \Big| \mathcal{H}_t\right]\right]$$

$$+ 4\eta^3 \mathbb{E}\left[\mathbb{E}\left[\left\|x_t - x^\star - \frac{\eta}{M}\sum_{m\in[M]} \nabla F_m(x_t^m)\right\|_2 \left\|\frac{1}{M}\sum_{m\in[M]}\xi_t^m\right\|_2^3 \Big| \mathcal{H}_t\right]\right],$$

$$\overset{\text{(a)}}{=} \mathbb{E}\left[\left\|x_t - x^\star - \frac{\eta}{M}\sum_{m\in[M]} \nabla F_m(x_t^m)\right\|_2^4\right] + \eta^4 \mathbb{E}\left[\left\|\frac{1}{M}\sum_{m\in[M]}\xi_t^m\right\|_2^4\right]$$

$$+ 6\eta^2 \mathbb{E}\left[\left\|x_t - x^\star - \frac{\eta}{M}\sum_{m\in[M]} \nabla F_m(x_t^m)\right\|_2^2 \mathbb{E}\left[\left\|\frac{1}{M}\sum_{m\in[M]}\xi_t^m\right\|_2^2 \Big| \mathcal{H}_t\right]\right]$$

$$+ 4\eta^3 \mathbb{E}\left[\left\|x_t - x^\star - \frac{\eta}{M}\sum_{m\in[M]} \nabla F_m(x_t^m)\right\|_2 \mathbb{E}\left[\left\|\frac{1}{M}\sum_{m\in[M]}\xi_t^m\right\|_2^3 \Big| \mathcal{H}_t\right]\right],$$

$$\leq^{\text{(Lemmas 11 to 13), (b)}} \mathbb{E}\left[\left\|x_t - x^\star - \frac{\eta}{M}\sum_{m\in[M]}\nabla F_m(x_t^m)\right\|_2^4\right] + \frac{3\eta^4\sigma^4}{M^2}$$

$$+ \frac{6\eta^2\sigma^2}{M}\mathbb{E}\left[\left\|x_t - x^\star - \frac{\eta}{M}\sum_{m\in[M]}\nabla F_m(x_t^m)\right\|_2^2\right]$$

$$+ \frac{4\sqrt{3}\eta^3\sigma^3}{M^{3/2}}\sqrt{\mathbb{E}\left[\left\|x_t - x^\star - \frac{\eta}{M}\sum_{m\in[M]}\nabla F_m(x_t^m)\right\|_2^2\right]}\;,$$

$$= \mathbb{E}\left[\left\|x_t - x^\star - \frac{\eta}{M}\sum_{m\in[M]}\nabla F_m(x_t^m)\right\|_2^4\right] + \frac{3\eta^4\sigma^4}{M^2}$$

$$+ \frac{6\eta^2\sigma^2}{M}\mathbb{E}\left[\left\|x_t - x^\star - \frac{\eta}{M}\sum_{m\in[M]}\nabla F_m(x_t^m)\right\|_2^2\right]$$

$$+ 4\sqrt{\left(\frac{3\eta^4\sigma^4}{M^2}\right)\left(\frac{\eta^2\sigma^2}{M}\mathbb{E}\left[\left\|x_t - x^\star - \frac{\eta}{M}\sum_{m\in[M]}\nabla F_m(x_t^m)\right\|_2^2\right]\right)}\;,$$

$$\leq^{\text{(A.M.-G.M. Inequality)}} \mathbb{E}\left[\left\|x_t - x^\star - \frac{\eta}{M}\sum_{m\in[M]}\nabla F_m(x_t^m)\right\|_2^4\right] + \frac{9\eta^4\sigma^4}{M^2}$$

$$+ \frac{8\eta^2\sigma^2}{M}\mathbb{E}\left[\left\|x_t - x^\star - \frac{\eta}{M}\sum_{m\in[M]}\nabla F_m(x_t^m)\right\|_2^2\right]\;,$$

$$= \mathbb{E}\left[\left\|x_t - x^\star - \eta\nabla F(x_t) + \eta\nabla F(x_t) - \frac{\eta}{M}\sum_{m\in[M]}\nabla F_m(x_t^m)\right\|_2^4\right] + \frac{9\eta^4\sigma^4}{M^2}$$

$$+ \frac{8\eta^2\sigma^2}{M}\mathbb{E}\left[\left\|x_t - x^\star - \eta\nabla F(x_t) + \eta\nabla F(x_t) - \frac{\eta}{M}\sum_{m\in[M]}\nabla F_m(x_t^m)\right\|_2^2\right]\;,$$

$$=^{\text{(c)}} \mathbb{E}\left[\left\|\left(I - \eta\nabla^2 F(\hat{x}_t)\right)(x_t - x^\star) + \eta\nabla F(x_t) - \frac{\eta}{M}\sum_{m\in[M]}\nabla F_m(x_t^m)\right\|_2^4\right] + \frac{9\eta^4\sigma^4}{M^2}$$

$$+ \frac{8\eta^2\sigma^2}{M}\mathbb{E}\left[\left\|\left(I - \eta\nabla^2 F(\hat{x}_t)\right)(x_t - x^\star) + \eta\nabla F(x_t) - \frac{\eta}{M}\sum_{m\in[M]}\nabla F_m(x_t^m)\right\|_2^2\right]\;,$$

$$\leq^{\text{(Lemma 16), (d)}} \left(1 + \frac{\eta\mu}{1 - \eta\mu}\right)^3 (1 - \eta\mu)^4 \mathbb{E}\left[\|x_t - x^\star\|_2^4\right]$$

$$+ \left(1 + \frac{1 - \eta\mu}{\eta\mu}\right)^3 \eta^4 \mathbb{E}\left[\left\|\frac{1}{M}\sum_{m\in[M]}\left(\nabla F_m(x_t) - \nabla F_m(x_t^m)\right)\right\|_2^4\right] + \frac{9\eta^4\sigma^4}{M^2}$$

$$+ \frac{8\eta^2\sigma^2}{M}\left(1+\frac{\eta\mu}{1-\eta\mu}\right)(1-\eta\mu)^2\mathbb{E}\left[\|x_t-x^\star\|_2^2\right]$$

$$+ \frac{8\eta^2\sigma^2}{M}\left(1+\frac{1-\eta\mu}{\eta\mu}\right)\eta^2\mathbb{E}\left[\left\|\frac{1}{M}\sum_{m\in[M]}(\nabla F_m(x_t)-\nabla F_m(x_t^m))\right\|_2^2\right] \quad,$$

$$\overset{\text{(Jensen's Inequality)}}{\leq} (1-\eta\mu)\mathbb{E}\left[\|x_t-x^\star\|_2^4\right]$$

$$+ \frac{\eta}{\mu^3 M}\sum_{m\in[M]}\mathbb{E}\left[\|(\nabla F_m(x_t)-\nabla F_m(x_t^m))\|_2^4\right]+\frac{9\eta^4\sigma^4}{M^2}$$

$$+ \frac{8\eta^2\sigma^2(1-\eta\mu)}{M}\mathbb{E}\left[\|x_t-x^\star\|_2^2\right]+\frac{8\eta^3\sigma^2}{\mu M}\mathbb{E}\left[\left\|\frac{1}{M}\sum_{m\in[M]}(\nabla F_m(x_t)-\nabla F_m(x_t^m))\right\|_2^2\right] \quad,$$

$$\overset{\text{(Assumption 1), (d)}}{\leq} (1-\eta\mu)B(t)+\frac{\eta H^4}{\mu^3}D(t)+\frac{8\eta^2\sigma^2(1-\eta\mu)}{M}A(t)+\frac{9\eta^4\sigma^4}{M^2}$$

$$+ \frac{8\eta^3\sigma^2}{\mu M}\mathbb{E}\left[\left\|\frac{1}{M}\sum_{m\in[M]}\left(\nabla^2 F_m(\hat{x}_t^m)-\nabla^2 F_m(x_t)+\nabla^2 F_m(x_t)-\nabla^2 F(x_t)\right)(x_t-x_t^m)\right\|_2^2\right] \quad,$$

$$\overset{\text{(Assumptions 1 and 6), (e)}}{\leq} (1-\eta\mu)B(t)+\frac{\eta H^4}{\mu^3}D(t)+\frac{8\eta^2\sigma^2(1-\eta\mu)}{M}A(t)+\frac{9\eta^4\sigma^4}{M^2}$$

$$+ \frac{8\eta^3\sigma^2}{\mu M}\left(\frac{2Q^2}{M}\sum_{m\in[M]}\mathbb{E}\left[\|x_t-x_t^m\|_2^4\right]+\frac{2\tau^2}{M}\sum_{m\in[M]}\mathbb{E}\left[\|x_t-x_t^m\|_2^2\right]\right) \quad,$$

$$\overset{\text{(Jensen's Inequality)}}{\leq} (1-\eta\mu)B(t)+\frac{\eta H^4}{\mu^3}D(t)+\frac{8\eta^2\sigma^2(1-\eta\mu)}{M}A(t)+\frac{9\eta^4\sigma^4}{M^2}$$

$$+ \frac{8\eta^3\sigma^2}{\mu M}\left(2Q^2 D(t)+2\tau^2 C(t)\right) \quad,$$

$$= (1-\eta\mu)B(t)+\left(\frac{\eta H^4}{\mu^3}+\frac{16\eta^3\sigma^2 Q^2}{\mu M}\right)D(t)+\frac{8\eta^2\sigma^2(1-\eta\mu)}{M}A(t)+\frac{16\eta^3\sigma^2\tau^2}{\mu M}C(t)$$

$$+ \frac{9\eta^4\sigma^4}{M^2} \quad,$$

where in (a) we used the fact that $x_t-x^\star-\frac{\eta}{M}\sum_{m\in[M]}\nabla F_m(x_t^m)\in m\mathcal{H}_t$; in (b) we used the Jensen's inequality $\mathbb{E}\left[\|y\|_2\right]\leq\sqrt{\mathbb{E}\left[\|y\|_2^2\right]}$; in (c) we use mean value theorem to conclude that there exists some $\hat{x}_t$ which is a convex combination of $x_t$ and $x^\star$ such that $\nabla F(x_t)=\nabla F(x^\star)+\nabla^2 F(\hat{x}_t)(x_t-x_\star)$; in (d) we apply mean value theorem to find a $\hat{x}_t^m$ which is a convex combination of $x_t$ and $x_t^m$ such that $\nabla F_m(x_t)=\nabla F_m(x_t^m)+\nabla^2 F_m(\hat{x}_t^m)\cdot(x_t-x_t^m)$; and in (e) we used the fact that $\|\hat{x}_t^m-x_t\|_2\leq\|x_t^m-x_t\|_2$. This finishes the proof of the first statement of the lemma. By letting $t+1=Kr$ for some $r\in[R]$, and unrolling till the previous communication round we get,

$$B(Kr)$$

$$\leq (1-\eta\mu)B(Kr-1)+\left(\frac{\eta H^4}{\mu^3}+\frac{16\eta^3\sigma^2 Q^2}{\mu M}\right)D(Kr-1)$$

$$+ \frac{8\eta^2\sigma^2(1-\eta\mu)}{M}A(Kr-1)+\frac{16\eta^3\sigma^2\tau^2}{\mu M}C(Kr-1)+\frac{9\eta^4\sigma^4}{M^2} \quad,$$

$$\leq (1-\eta\mu)^K B(K(r-1))+\left(\frac{\eta H^4}{\mu^3}+\frac{16\eta^3\sigma^2 Q^2}{\mu M}\right)\sum_{j=K(r-1)}^{Kr-1}(1-\eta\mu)^{Kr-1-j}D(j)$$

$$+ \frac{8\eta^2\sigma^2}{M} \sum_{j=K(r-1)}^{Kr-1} (1-\eta\mu)^{Kr-j} A(j) + \frac{16\eta^3\sigma^2\tau^2}{\mu M} \sum_{j=K(r-1)}^{Kr-1} (1-\eta\mu)^{Kr-1-j} C(j) + \frac{9\eta^3\sigma^4}{\mu M^2} \ ,$$

where in the last inequality for the last term we used that $\sum_{j=K(r-1)}^{Kr-1}(1-\eta\mu)^{Kr-1-j} \leq \frac{1-(1-\eta\mu)^K}{\eta\mu} \leq \frac{1}{\eta\mu}$. This proves the second statement of the lemma. $\qquad\square$

It would also be helpful to state the following lemma, which talks about the convergence on individual machines between two communication rounds.

**Lemma 23** (Single Machine SGD Fourth Moment). *For any machine $m \in [M]$, for $t \in [0, T]$, and for $k \geq 0$ we have the following for $\eta < \frac{1}{H}$,*

$$\mathbb{E}\left[\left\|x^m_{\delta(t)+k} - x^\star\right\|^4_2\right] \leq (1-\eta\mu)^{4k}\mathbb{E}\left[\left\|x_{\delta(t)} - x^\star_m\right\|^4_2\right] + 8\eta^2\sigma^2 k(1-\eta\mu)^{2k}\mathbb{E}\left[\left\|x_{\delta(t)} - x^\star_m\right\|^2_2\right]$$
$$+ \frac{11\eta^2\sigma^4}{\mu^2} \ .$$

*We can also get the following simpler bound,*

$$\mathbb{E}\left[\left\|x^m_{\delta(t)+k} - x^\star\right\|^4_2\right] \leq (1-\eta\mu)^{3k}\mathbb{E}\left[\left\|x_{\delta(t)} - x^\star_m\right\|^4_2\right] + \frac{16\eta\sigma^4}{\mu^3} \ .$$

*Proof.* For any machine $m \in [M]$ note the following for $t \geq \delta(t)$,

$$\mathbb{E}\left[\left\|x^m_{t+1} - x^\star_m\right\|^4_2\right]$$
$$= \mathbb{E}\left[\left\|x^m_t - x^\star_m - \eta\nabla F_m(x^m_t) + \eta\xi^m_t\right\|^4_2\right] \ ,$$
$$= \mathbb{E}\left[\left(\left\|x^m_t - x^\star_m - \eta\nabla F_m(x^m_t)\right\|^2_2 + \eta^2\left\|\xi^m_t\right\|^2_2 + 2\eta\left\langle x^m_t - x^\star_m - \eta\nabla F^m_t(x^m_t), \xi^m_t\right\rangle\right)^2\right] \ ,$$
$$\leq \mathbb{E}\left[\left\|x^m_t - x^\star_m - \eta\nabla F_m(x^m_t)\right\|^4_2\right]$$
$$\quad + \eta^4\mathbb{E}\left[\left\|\xi^m_t\right\|^4_2\right]$$
$$\quad + 4\eta^2\mathbb{E}\left[\left\|x^m_t - x^\star_m - \eta\nabla F_m(x^m_t)\right\|^2_2\right]\mathbb{E}\left[\left\|\xi^m_t\right\|^2_2\right]$$
$$\quad + 2\eta^2\mathbb{E}\left[\left\|x^m_t - x^\star_m - \eta\nabla F_m(x^m_t)\right\|^2_2\right]\mathbb{E}\left[\left\|\xi^m_t\right\|^2_2\right]$$
$$\quad + 4\eta^3\mathbb{E}\left[\left\|x^m_t - x^\star_m - \eta\nabla F_m(x^m_t)\right\|_2\right]\mathbb{E}\left[\left\|\xi^m_t\right\|^3_2\right]$$
$$\quad + 4\eta\mathbb{E}\left[\left\|x^m_t - x^\star_m - \eta\nabla F_m(x^m_t)\right\|^2_2 (x^m_t - x^\star_m - \eta\nabla F_m(x^m_t))^T\right]\underbrace{\mathbb{E}\left[\xi^m_t\right]}_{0} \ ,$$
$$\leq^{(a)} \mathbb{E}\left[\left\|x^m_t - x^\star_m - \eta\nabla F_m(x^m_t)\right\|^4_2\right] + \eta^4\sigma^4 + 6\eta^2\sigma^2\mathbb{E}\left[\left\|x^m_t - x^\star_m - \eta\nabla F_m(x^m_t)\right\|^2_2\right]$$
$$\quad + 4\eta^3\sigma^3\mathbb{E}\left[\left\|x^m_t - x^\star_m - \eta\nabla F_m(x^m_t)\right\|_2\right] \ ,$$
$$\leq^{\text{(Jensen's Inequality)}} \mathbb{E}\left[\left\|x^m_t - x^\star_m - \eta\nabla F_m(x^m_t)\right\|^4_2\right] + 6\eta^2\sigma^2\mathbb{E}\left[\left\|x^m_t - x^\star_m - \eta\nabla F_m(x^m_t)\right\|^2_2\right]$$
$$\quad + 4\eta^3\sigma^3\sqrt{\mathbb{E}\left[\left\|x^m_t - x^\star_m - \eta\nabla F_m(x^m_t)\right\|^2_2\right]} + \eta^4\sigma^4 \ ,$$
$$\leq^{\text{(A.M.-G.M. Inequality)}} \mathbb{E}\left[\left\|x^m_t - x^\star_m - \eta\nabla F_m(x^m_t)\right\|^4_2\right] + 6\eta^2\sigma^2\mathbb{E}\left[\left\|x^m_t - x^\star_m - \eta\nabla F_m(x^m_t)\right\|^2_2\right]$$
$$\quad + 4\eta^3\left(\frac{\eta\sigma^4}{2} + \frac{\sigma^2}{2\eta}\mathbb{E}\left[\left\|x^m_t - x^\star_m - \eta\nabla F_m(x^m_t)\right\|_2\right]^2\right) + \eta^4\sigma^4 \ ,$$
$$= \mathbb{E}\left[\left\|x^m_t - x^\star_m - \eta\nabla F_m(x^m_t)\right\|^4_2\right] + 3\eta^4\sigma^4 + 8\eta^2\sigma^2\mathbb{E}\left[\left\|x^m_t - x^\star_m - \eta\nabla F_m(x^m_t)\right\|^2_2\right] \ ,$$
$$\leq^{(b)} (1-\eta\mu)^4\mathbb{E}\left[\left\|x^m_t - x^\star_m\right\|^4_2\right] + 3\eta^4\sigma^4 + 8\eta^2\sigma^2(1-\eta\mu)^2\mathbb{E}\left[\left\|x^m_t - x^\star_m\right\|^2_2\right] \ ,$$

$$\leq^{\text{(Lemma 21)}} (1-\eta\mu)^4 \mathbb{E}\left[\|x_t^m - x_m^\star\|_2^4\right] + 3\eta^4\sigma^4$$

$$+ 8\eta^2\sigma^2(1-\eta\mu)^2\left((1-\eta\mu)^{2(t-\delta(t))}\mathbb{E}\left[\|x_{\delta(t)} - x_m^\star\|_2^2\right] + \frac{\eta\sigma^2}{\mu}\right) \;,$$

$$= (1-\eta\mu)^4\mathbb{E}\left[\|x_t^m - x_m^\star\|_2^4\right] + 3\eta^4\sigma^4$$

$$+ 8\eta^2\sigma^2(1-\eta\mu)^{2(t+1-\delta(t))}\mathbb{E}\left[\|x_{\delta(t)} - x_m^\star\|_2^2\right] + \frac{8\eta^3\sigma^4(1-\eta\mu)^2}{\mu} \;,$$

$$\leq^{\text{(c)}} (1-\eta\mu)^4\mathbb{E}\left[\|x_t^m - x_m^\star\|_2^4\right] + 8\eta^2\sigma^2(1-\eta\mu)^{2(t+1-\delta(t))}\mathbb{E}\left[\|x_{\delta(t)} - x_m^\star\|_2^2\right] + \frac{11\eta^3\sigma^4}{\mu} \;,$$

$$= (1-\eta\mu)^{4(t+1-\delta(t))}\mathbb{E}\left[\left\|x_{\delta(t)}^m - x_m^\star\right\|_2^4\right] + \frac{11\eta^2\sigma^4}{\mu^2}$$

$$+ 8\eta^2\sigma^2\sum_{j=\delta(t)}^t (1-\eta\mu)^{4(t-j)}(1-\eta\mu)^{2(j+1-\delta(t))}\mathbb{E}\left[\|x_{\delta(t)} - x_m^\star\|_2^2\right] \;,$$

$$\leq (1-\eta\mu)^{4(t+1-\delta(t))}\mathbb{E}\left[\left\|x_{\delta(t)} - x_m^\star\right\|_2^4\right]$$

$$+ 8\eta^2\sigma^2\mathbb{E}\left[\|x_{\delta(t)} - x_m^\star\|_2^2\right]\sum_{j=\delta(t)}^t (1-\eta\mu)^{2(t+1-\delta(t))} + \frac{11\eta^2\sigma^4}{\mu^2} \;,$$

$$\leq (1-\eta\mu)^{4(t+1-\delta(t))}\mathbb{E}\left[\left\|x_{\delta(t)} - x_m^\star\right\|_2^4\right]$$

$$+ 8\eta^2\sigma^2\left(t+1-\delta(t)\right)(1-\eta\mu)^{2(t+1-\delta(t))}\mathbb{E}\left[\|x_{\delta(t)} - x_m^\star\|_2^2\right] + \frac{11\eta^2\sigma^4}{\mu^2} \;,$$

where in (a) we used the fact that $\mathbb{E}\left[\|\xi_t^m\|_2^3\right] \leq \sqrt{\mathbb{E}\left[\|\xi_t^m\|_2^2\right]\mathbb{E}\left[\|\xi_t^m\|_2^4\right]}$ and Assumption 2; in (b) we used that $\|x_t^m - \eta\nabla F_m(x_t^m) - x_m^\star\| \leq (1-\eta\mu)\|x_t^m - x_m^\star\|_2$ for $\eta < \frac{1}{H}$; in (c) we use that $\eta < \frac{1}{H} \leq \frac{1}{\mu}$ which implies that $3\eta^4\sigma^4 \leq \frac{3\eta^3\sigma^4}{\mu}$. We gave the above analysis for $t+1 > \delta(t)$, thus it can be translated for $k > 0$ as follows,

$$\mathbb{E}\left[\left\|x_{\delta(t)+k}^m - x_m^\star\right\|_2^4\right] \leq (1-\eta\mu)^{4k}\mathbb{E}\left[\left\|x_{\delta(t)} - x_m^\star\right\|_2^4\right]$$

$$+ 8\eta^2\sigma^2 k(1-\eta\mu)^{2k}\mathbb{E}\left[\left\|x_{\delta(t)} - x_m^\star\right\|_2^2\right] + \frac{11\eta^2\sigma^4}{\mu^2} \;.$$

Since when $k = 0$, this is still a valid upper bound, we have proven the first lemma statement. To get the simpler upper bound, we will complete the square, proceeding from the red term in the above analysis as follows,

$$\mathbb{E}\left[\|x_{t+1}^m - x_m^\star\|_2^4\right]$$

$$\leq (1-\eta\mu)^4\mathbb{E}\left[\|x_t^m - x_m^\star\|_2^4\right] + 3\eta^4\sigma^4 + 8\eta^2\sigma^2(1-\eta\mu)^2\mathbb{E}\left[\|x_t^m - x_m^\star\|_2^2\right] \;,$$

$$\leq^{\text{Jensen's Inequality}} (1-\eta\mu)^4\mathbb{E}\left[\|x_t^m - x_m^\star\|_2^4\right] + 16\eta^4\sigma^4 + 8\eta^2\sigma^2(1-\eta\mu)^2\sqrt{\mathbb{E}\left[\|x_t^m - x_m^\star\|_2^4\right]} \;,$$

$$\leq \left((1-\eta\mu)^2\sqrt{\mathbb{E}\left[\|x_t^m - x_m^\star\|_2^4\right]} + 4\eta^2\sigma^2\right)^2 \;.$$

Taking the square root of both sides, we get,

$$\sqrt{\mathbb{E}\left[\|x_{t+1}^m - x_m^\star\|_2^4\right]} \leq (1-\eta\mu)^2\sqrt{\mathbb{E}\left[\|x_t^m - x_m^\star\|_2^4\right]} + 4\eta^2\sigma^2 \;,$$

$$\leq (1-\eta\mu)^{2(t+1-\delta(t))}\sqrt{\mathbb{E}\left[\left\|x_{\delta(t)}^m - x_m^\star\right\|_2^4\right]} + \frac{4\eta\sigma^2}{\mu} \;.$$

Finally whole squaring and using Lemma 16 we get,

$$
\mathbb{E}\left[\left\|x_{t+1}^m - x_m^\star\right\|_2^4\right] \le \left(1 + \frac{\eta\mu}{1 - \eta\mu}\right)(1 - \eta\mu)^{4(t+1-\delta(t))}\mathbb{E}\left[\left\|x_{\delta(t)}^m - x_m^\star\right\|_2^4\right]
$$
$$
+ \left(1 + \frac{1 - \eta\mu}{\eta\mu}\right)\frac{16\eta^2\sigma^4}{\mu^2} \ ,
$$
$$
\le (1 - \eta\mu)^{3(t+1-\delta(t))}\mathbb{E}\left[\left\|x_{\delta(t)}^m - x_m^\star\right\|_2^4\right] + \frac{16\eta\sigma^4}{\mu^3} \ .
$$

We proved this for $t + 1 > \delta(t)$, but clearly it also holds when $t + 1 = \delta(t)$, which implies that for all $k \ge 0$,

$$
\mathbb{E}\left[\left\|x_{\delta(t)+k}^m - x^\star\right\|_2^4\right] \le (1 - \eta\mu)^{3k}\mathbb{E}\left[\left\|x_{\delta(t)} - x_m^\star\right\|_2^4\right] + \frac{16\eta\sigma^4}{\mu^3} \ ,
$$

thus finishing the proof of the lemma. $\qquad\square$

### F.3   Function Value Error

The main result of this sub-section comes from the work [25], which relates $E(\cdot)$ to $C(\cdot)$ and $D(\cdot)$.

**Lemma 24** (Section D.4, [25]). *Assume we have a problem instance satisfying Assumptions 1 and 2. Then assuming $\eta < \frac{1}{H}$ we have for all $t \in [0, T-1]$,*

$$
E(t) \le \left(\frac{1}{\eta} - \frac{\mu}{2}\right)\mathbb{E}\left[\|x_t - x^\star\|_2^2\right] - \frac{1}{\eta}\mathbb{E}\left[\|x_{t+1} - x^\star\|_2^2\right] + \frac{8\tau^2}{\mu}C(t) + \frac{2Q^2}{\mu}D(t) + \frac{\eta\sigma^2}{M} \ .
$$

**Remark 7.** The above result is known from the paper [25], and we do not claim any novelty here. Our contribution is improving the bound on the second and fourth moment consensus error ($C(t)$ and $D(t)$). Later, we will put our improved bound in this lemma and provide a tighter convergence guarantee for local SGD under the $\zeta$ assumption.

We will also recall the more straightforward recursion, which does not explicitly depend on $Q$, used in several existing results, including the following due to Woodworth et al. [20].

**Lemma 25** (Lemma 7, [20]). *Assume we have a problem instance satisfying Assumptions 1 and 2. Then assuming $\eta \le \frac{1}{10H}$ we have for all $t \in [0, T-1]$,*

$$
E(t) \le \left(\frac{1}{\eta} - \mu\right)\mathbb{E}\left[\|x_t - x^\star\|_2^2\right] - \frac{1}{\eta}\mathbb{E}\left[\|x_{t+1} - x^\star\|_2^2\right] + 2HC(t) + \frac{3\eta\sigma^2}{M} \ .
$$

## G   Uniform Control over the Consensus Error and Analysis using Assumption 7

### G.1   Upper Bound on Second Moment of Consensus Error

In this subsection, we re-state the upper bound on the second moment of consensus error from the work [20]. We do not claim any novelty and include this lemma for completeness.

**Lemma 26** (Lemma 8 from [20]). *For all $t \in [0, T]$ under Assumptions 2 and 7 with a stepsize $\eta \le \frac{1}{2H}$ we have,*

$$
C(t) \le 6K^2\eta^2H^2\zeta^2 + 6K\sigma^2\eta^2 \ . \tag{8}
$$

*Proof.* Note the following about the second moment of the difference between the iterates on two machines $m, n \in [M]$ when $t > \delta(t)$,

$$
\mathbb{E}\left[\|x_t^m - x_t^n\|_2^2\right]
$$
$$
= \mathbb{E}\left[\left\|x_{t-1}^m - x_{t-1}^n - \eta g_{t-1}^m + \eta g_{t-1}^n\right\|_2^2\right] \ ,
$$
$$
\le \mathbb{E}\left[\left\|x_{t-1}^m - x_{t-1}^n - \eta\nabla F_m(x_{t-1}^m) + \eta\nabla F_n(x_{t-1}^n)\right\|_2^2\right] + 2\eta^2\sigma^2 \ ,
$$

$$= \mathbb{E}\left[\left\|x_{t-1}^m - x_{t-1}^n - \eta\left(\nabla F_m(x_{t-1}^m) - \nabla F_m(x_{t-1}^n)\right) - \eta\left(\nabla F_m(x_{t-1}^n) - \nabla F_n(x_{t-1}^n)\right)\right\|_2^2\right]$$
$$+ 2\eta^2\sigma^2 \ ,$$

$$\leq^{(a)} \mathbb{E}\left[\left\|x_{t-1}^m - x_{t-1}^n - \eta\nabla^2 F_m(c)(x_{t-1}^m - x_{t-1}^n) - \eta\left(\nabla F_m(x_{t-1}^n) - \nabla F_n(x_{t-1}^n)\right)\right\|_2^2\right]$$
$$+ 2\eta^2\sigma^2 \ ,$$

$$= \mathbb{E}\left[\left\|\left(I - \eta\nabla^2 F_m(c)\right)(x_{t-1}^m - x_{t-1}^n) - \eta\left(\nabla F_m(x_{t-1}^n) - \nabla F_n(x_{t-1}^n)\right)\right\|_2^2\right] + 2\eta^2\sigma^2 \ ,$$

$$\leq \left(1 + \frac{1}{K-1}\right)\mathbb{E}\left[\left\|\left(I - \eta\nabla^2 F_m(c)\right)(x_{t-1}^m - x_{t-1}^n)\right\|_2^2\right]$$
$$+ 2K\eta^2\mathbb{E}\left[\left\|\nabla F_m(x_{t-1}^n) - \nabla F_n(x_{t-1}^n)\right\|_2^2\right] + 2\eta^2\sigma^2 \ ,$$

$$\leq \left(1 + \frac{1}{K-1}\right)(1 - \eta\mu)^2\mathbb{E}\left[\left\|x_{t-1}^m - x_{t-1}^n\right\|_2^2\right] + 2K\eta^2 H^2\zeta^2 + 2\eta^2\sigma^2 \ ,$$

$$\leq \left(1 + \frac{1}{K-1}\right)\mathbb{E}\left[\left\|x_{t-1}^m - x_{t-1}^n\right\|_2^2\right] + 2K\eta^2 H^2\zeta^2 + 2\eta^2\sigma^2 \ ,$$

where in (a) we use the mean value theorem to find a $c$ between $x_{t-1}^m$ and $x_{t-1}^n$ such that $\nabla F_m(x_{t-1}^m) - \nabla F_m(x_{t-1}^n) = \nabla^2 F(c) \cdot (x_{t-1}^m - x_{t-1}^n)$. Unrolling the recursion for $K - 1$ steps givse us,

$$\mathbb{E}\left[\left\|x_t^m - x_t^n\right\|_2^2\right] \leq 6K^2\eta^2 H^2\zeta^2 + 6\eta^2 K\sigma^2 \ ,$$

$\square$

## G.2 Upper Bound on Fourth Moment of Consensus Error

In this sub-section, we re-state the fourth moment upper bound on consensus error from the work [25]. Here we do not claim any novelty and we include it for completeness.

**Lemma 27** (Lemma 12 from [25]). *For all $t \in [0, T]$ under Assumptions 2 and 7 with a stepsize $\eta \leq \frac{1}{2H}$ we have,*

$$D(t) \leq 3840\eta^4 K^4 H^4\zeta^4 + 5920\eta^4 K^2\sigma^4 \ .$$

*Proof.* Note the following about the fourth moment of the difference between the iterates on two machines $m, n \in [M]$,

$$\mathbb{E}\left[\left\|x_t^m - x_t^n\right\|_2^4\right]$$
$$= \mathbb{E}\left[\left\|x_{t-1}^m - x_{t-1}^n - \eta g_{t-1}^m + \eta g_{t-1}^n\right\|_2^4\right] \ ,$$
$$= \mathbb{E}\left[\left(\left\|x_{t-1}^m - x_{t-1}^n - \eta\nabla F_m(x_{t-1}^m) + \eta\nabla F_n(x_{t-1}^n) + \eta\xi_{t-1}^m - \eta\xi_{t-1}^n\right\|_2^2\right)^2\right] \ ,$$
$$= \mathbb{E}\left[\left(\left\|x_{t-1}^m - x_{t-1}^n - \eta\nabla F_m(x_{t-1}^m) + \eta\nabla F_n(x_{t-1}^n)\right\|_2^2 + \eta^2\left\|\xi_{t-1}^m - \xi_{t-1}^n\right\|_2^2\right.\right.$$
$$\left.\left. + 2\eta\left\langle x_{t-1}^m - x_{t-1}^n - \eta\nabla F_m(x_{t-1}^m) + \eta\nabla F_n(x_{t-1}^n), \xi_{t-1}^m - \xi_{t-1}^n\right\rangle\right)^2\right] \ ,$$
$$= \mathbb{E}\left[\left\|x_{t-1}^m - x_{t-1}^n - \eta\nabla F_m(x_{t-1}^m) + \eta\nabla F_n(x_{t-1}^n)\right\|_2^4\right] + \eta^4\mathbb{E}\left[\left\|\xi_{t-1}^m - \xi_{t-1}^n\right\|_2^4\right]$$
$$+ 4\eta^2\mathbb{E}\left[\left(\left\langle x_{t-1}^m - x_{t-1}^n - \eta\nabla F_m(x_{t-1}^m) + \eta\nabla F_n(x_{t-1}^n), \xi_{t-1}^m - \xi_{t-1}^n\right\rangle\right)^2\right]$$
$$+ 2\eta^2\mathbb{E}\left[\left\|x_{t-1}^m - x_{t-1}^n - \eta\nabla F_m(x_{t-1}^m) + \eta\nabla F_n(x_{t-1}^n)\right\|_2^2\left\|\xi_{t-1}^m - \xi_{t-1}^n\right\|_2^2\right]$$
$$+ 4\eta^3\mathbb{E}\left[\left\langle x_{t-1}^m - x_{t-1}^n - \eta\nabla F_m(x_{t-1}^m) + \eta\nabla F_n(x_{t-1}^n), \xi_{t-1}^m - \xi_{t-1}^n\right\rangle\left\|\xi_{t-1}^m - \xi_{t-1}^n\right\|_2^2\right]$$

$$+ 4\eta \mathbb{E}\left[ \left\| x_{t-1}^m - x_{t-1}^n - \eta \nabla F_m(x_{t-1}^m) + \eta \nabla F_n(x_{t-1}^n) \right\|_2^2 \right.$$

$$\left. \left( x_{t-1}^m - x_{t-1}^n - \eta \nabla F_m(x_{t-1}^m) + \eta \nabla F_n(x_{t-1}^n) \right) \right] \cdot \mathbb{E}\left[ \xi_{t-1}^m - \xi_{t-1}^n \right]^{\nearrow 0},$$

$$\overset{\text{(C.S. Inequality, Assumption 2)}}{\leq} \mathbb{E}\left[ \left\| x_{t-1}^m - x_{t-1}^n - \eta \nabla F_m(x_{t-1}^m) + \eta \nabla F_n(x_{t-1}^n) \right\|_2^4 \right] + 8\sigma^4 \eta^4$$

$$+ 6\eta^2 \mathbb{E}\left[ \left\| x_{t-1}^m - x_{t-1}^n - \eta \nabla F_m(x_{t-1}^m) + \eta \nabla F_n(x_{t-1}^n) \right\|_2^2 \right] \mathbb{E}\left[ \left\| \xi_{t-1}^m - \xi_{t-1}^n \right\|_2^2 \right]$$

$$+ 4\eta^3 \mathbb{E}\left[ \left\| x_{t-1}^m - x_{t-1}^n - \eta \nabla F_m(x_{t-1}^m) + \eta \nabla F_n(x_{t-1}^n) \right\|_2 \right] \mathbb{E}\left[ \left\| \xi_{t-1}^m - \xi_{t-1}^n \right\|_2^3 \right], \qquad \text{(a)}$$

In order to bound the term $\mathbb{E}\left[ \left\| \xi_{t-1}^m - \xi_{t-1}^n \right\|_2^3 \right]$ we use Cauchy-Schwarz Inequality:

$$\mathbb{E}\left[ \left\| \xi_{t-1}^m - \xi_{t-1}^n \right\|_2^3 \right] = \mathbb{E}\left[ \left\| \xi_{t-1}^m - \xi_{t-1}^n \right\|_2 \cdot \left\| \xi_{t-1}^m - \xi_{t-1}^n \right\|_2^2 \right]$$

$$\leq \sqrt{ \mathbb{E}\left[ \left\| \xi_{t-1}^m - \xi_{t-1}^n \right\|_2^2 \right] \mathbb{E}\left[ \left\| \xi_{t-1}^m - \xi_{t-1}^n \right\|_2^4 \right] } \overset{\text{Assumption 2}}{\leq} 4\sqrt{\sigma^6}$$

Also the term $4\eta^3 \mathbb{E}\left[ \left\| x_{t-1}^m - x_{t-1}^n - \eta \nabla F_m(x_{t-1}^m) + \eta \nabla F_n(x_{t-1}^n) \right\|_2 \right]$ can be bounded as:

$$\mathbb{E}\left[ \left\| x_{t-1}^m - x_{t-1}^n - \eta \nabla F_m(x_{t-1}^m) + \eta \nabla F_n(x_{t-1}^n) \right\|_2 \right]$$

$$\overset{\text{Jensen's Inequality}}{\leq} \sqrt{ \mathbb{E}\left[ \left\| x_{t-1}^m - x_{t-1}^n - \eta \nabla F_m(x_{t-1}^m) + \eta \nabla F_n(x_{t-1}^n) \right\|_2^2 \right] }$$

Putting everything back into (a) gives us:

$$\mathbb{E}\left[ \left\| x_t^n - x_t^m \right\|_2^4 \right]$$

$$\overset{\text{(Assumption in (2))}}{\leq} \mathbb{E}\left[ \left\| x_{t-1}^m - x_{t-1}^n - \eta \nabla F_m(x_{t-1}^m) + \eta \nabla F_n(x_{t-1}^n) \right\|_2^4 \right] + 8\eta^4 \sigma^4$$

$$+ 12\eta^2 \sigma^2 \mathbb{E}\left[ \left\| x_{t-1}^m - x_{t-1}^n - \eta \nabla F_m(x_{t-1}^m) + \eta \nabla F_n(x_{t-1}^n) \right\|_2^2 \right]$$

$$+ 16\eta^3 \sqrt{ \sigma^6 \mathbb{E}\left[ \left\| x_{t-1}^m - x_{t-1}^n - \eta \nabla F_m(x_{t-1}^m) + \eta \nabla F_n(x_{t-1}^n) \right\|_2^2 \right] },$$

To bound the third term in the above inequality, we use the A.M. - G.M. Inequality $\sqrt{ab} \leq \frac{a}{2\gamma} + \frac{\gamma b}{2}$ for $\gamma > 0$. Let $\gamma = \eta, a = \sigma^2 \mathbb{E}\left[ \left\| x_{t-1}^m - x_{t-1}^n - \eta \nabla F_m(x_{t-1}^m) + \eta \nabla F_n(x_{t-1}^n) \right\|_2^2 \right], b = \sigma^4$. We have:

$$16\eta^3 \sqrt{ \sigma^6 \mathbb{E}\left[ \left\| x_{t-1}^m - x_{t-1}^n - \eta \nabla F_m(x_{t-1}^m) + \eta \nabla F_n(x_{t-1}^n) \right\|_2^2 \right] }$$

$$= 16\eta^3 \sqrt{ (\sigma^4) \left( \sigma^2 \mathbb{E}\left[ \left\| x_{t-1}^m - x_{t-1}^n - \eta \nabla F_m(x_{t-1}^m) + \eta \nabla F_n(x_{t-1}^n) \right\|_2^2 \right] \right) },$$

$$\leq 16\eta^3 \left( \frac{\eta \sigma^4}{2} + \frac{\sigma^2}{2\eta} \mathbb{E}\left[ \left\| x_{t-1}^m - x_{t-1}^n - \eta \nabla F_m(x_{t-1}^m) + \eta \nabla F_n(x_{t-1}^n) \right\|_2^2 \right] \right),$$

So we have:

$$\mathbb{E}\left[ \left\| x_t^n - x_t^m \right\|_2^4 \right]$$

$$\leq \mathbb{E}\left[ \left\| x_{t-1}^m - x_{t-1}^n - \eta \nabla F_m(x_{t-1}^m) + \eta \nabla F_n(x_{t-1}^n) \right\|_2^4 \right] + 8\eta^4 \sigma^4$$

$$+ 12\eta^2 \sigma^2 \mathbb{E}\left[ \left\| x_{t-1}^m - x_{t-1}^n - \eta \nabla F_m(x_{t-1}^m) + \eta \nabla F_n(x_{t-1}^n) \right\|_2^2 \right]$$

$$+ 16\eta^3 \left( \frac{\eta \sigma^4}{2} + \frac{\sigma^2}{2\eta} \mathbb{E}\left[ \left\| x_{t-1}^m - x_{t-1}^n - \eta \nabla F_m(x_{t-1}^m) + \eta \nabla F_n(x_{t-1}^n) \right\|_2^2 \right] \right),$$

$$= \mathbb{E}\left[\left\|x_{t-1}^m - x_{t-1}^n - \eta\nabla F_m(x_{t-1}^m) + \eta\nabla F_n(x_{t-1}^n)\right\|_2^4\right]$$
$$+ 20\eta^2\sigma^2\mathbb{E}\left[\left\|x_{t-1}^m - x_{t-1}^n - \eta\nabla F_m(x_{t-1}^m) + \eta\nabla F_n(x_{t-1}^n)\right\|_2^2\right] + 16\eta^4\sigma^4 \;,$$

$$= \mathbb{E}\left[\left\|x_{t-1}^m - x_{t-1}^n - \eta\left(\nabla F_m(x_{t-1}^m) - \nabla F_m(x_{t-1}^n)\right) + \eta\left(\nabla F_n(x_{t-1}^n) - \nabla F_m(x_{t-1}^n)\right)\right\|_2^4\right]$$
$$+ 20\eta^2\sigma^2\mathbb{E}\left[\left\|x_{t-1}^m - x_{t-1}^n - \eta\left(\nabla F_m(x_{t-1}^m) - \nabla F_m(x_{t-1}^n)\right) + \eta\left(\nabla F_n(x_{t-1}^n) - \nabla F_m(x_{t-1}^n)\right)\right\|_2^2\right]$$
$$+ 16\eta^4\sigma^4 \;,$$

Now by using Lemma 16 we have:

$$\mathbb{E}\left[\left\|x_t^m - x_t^n\right\|_2^4\right]$$
$$\leq \left(1 + \frac{1}{\gamma_{t-1}}\right)^3 \mathbb{E}\left[\left\|x_{t-1}^m - x_{t-1}^n - \eta\left(\nabla F_m(x_{t-1}^m) - \nabla F_m(x_{t-1}^n)\right)\right\|_2^4\right]$$
$$+ \eta^4(1 + \gamma_{t-1})^3\mathbb{E}\left[\left\|\nabla F_n(x_{t-1}^n) - \nabla F_m(x_{t-1}^n)\right\|_2^4\right]$$
$$\textcolor{red}{+ 20\eta^2\sigma^2\left(1 + \frac{1}{\gamma_{t-1}}\right)\mathbb{E}\left[\left\|x_{t-1}^m - x_{t-1}^n - \eta\left(\nabla F_m(x_{t-1}^m) - \nabla F_m(x_{t-1}^n)\right)\right\|_2^2\right]}$$
$$+ 20\eta^4\sigma^2(1 + \gamma_{t-1})\mathbb{E}\left[\left\|\nabla F_n(x_{t-1}^n) - \nabla F_m(x_{t-1}^n)\right\|_2^2\right] + 16\eta^4\sigma^4 \;,$$

From the mean-value theorem we know that $\nabla F(x) - \nabla F(y) = \nabla^2 F(c)(x - y)$ for some $c = \lambda x + (1 - \lambda)y$. By applying this theorem to the $\textcolor{blue}{\text{first}}$ and $\textcolor{red}{\text{third}}$ term of the above inequality we have:

$$\left(1 + \frac{1}{\gamma_{t-1}}\right)^3 \mathbb{E}\left[\left\|x_{t-1}^m - x_{t-1}^n - (\eta\nabla F_m(x_{t-1}^m) - \eta\nabla F_m(x_{t-1}^n))\right\|_2^4\right]$$
$$= \left(1 + \frac{1}{\gamma_{t-1}}\right)^3 \mathbb{E}\left[\left\|x_{t-1}^m - x_{t-1}^n - \eta\nabla^2 F_m(c)(x_{t-1}^m - x_{t-1}^n)\right\|_2^4\right] \;,$$
$$= \left(1 + \frac{1}{\gamma_{t-1}}\right)^3 \mathbb{E}\left[\left\|(I - \eta\nabla^2 F_m(c))(x_{t-1}^m - x_{t-1}^n)\right\|_2^4\right] \;,$$
$$\leq \left(1 + \frac{1}{\gamma_{t-1}}\right)^3 (1 - \eta\mu)^4\mathbb{E}\left[\left\|x_{t-1}^m - x_{t-1}^n\right\|_2^4\right] \;,$$

With the same approach for the third term we have:

$$20\eta^2\sigma^2\left(1 + \frac{1}{\gamma_{t-1}}\right)\mathbb{E}\left[\left\|x_{t-1}^m - x_{t-1}^n - \eta\left(\nabla F_m(x_{t-1}^m) - \eta\nabla F_m(x_{t-1}^n)\right)\right\|_2^2\right]$$
$$\leq 20\eta^2\sigma^2\left(1 + \frac{1}{\gamma_{t-1}}\right)(1 - \eta\mu)^2\mathbb{E}\left[\left\|x_{t-1}^m - x_{t-1}^n\right\|_2^2\right] \;,$$

Putting all together gives us:

$$\mathbb{E}\left[\left\|x_t^n - x_t^m\right\|_2^4\right]$$
$$\leq \left(1 + \frac{1}{K-1}\right)^3 (1 - \eta\mu)^4\mathbb{E}\left[\left\|x_{t-1}^m - x_{t-1}^n\right\|_2^4\right]$$
$$+ 16\eta^4 K^3\left(\mathbb{E}\left[\left\|\nabla F(x_{t-1}^m) - \nabla F_m(x_{t-1}^m)\right\|_2^4\right] + \mathbb{E}\left[\left\|\nabla F(x_{t-1}^n) - \nabla F_n(x_{t-1}^n)\right\|_2^4\right]\right)$$
$$+ 20\eta^2\sigma^2\left(1 + \frac{1}{K-1}\right)(1 - \eta\mu)^2\mathbb{E}\left[\left\|x_{t-1}^m - x_{t-1}^n\right\|_2^2\right]$$
$$+ 40\eta^4\sigma^2 K\left(\mathbb{E}\left[\left\|\nabla F(x_{t-1}^m) - \nabla F_m(x_{t-1}^m)\right\|_2^2\right] + \mathbb{E}\left[\left\|\nabla F(x_{t-1}^n) - \nabla F_n(x_{t-1}^n)\right\|_2^2\right]\right)$$
$$+ 16\eta^4\sigma^4 \;,$$
$$\leq \left(1 + \frac{1}{K-1}\right)^3 (1 - \eta\mu)^4\mathbb{E}\left[\left\|x_{t-1}^m - x_{t-1}^n\right\|_2^4\right] + 32\eta^4 K^3 H^4\zeta^4$$

$$+ 20\eta^2\sigma^2 \left(1 + \frac{1}{K-1}\right) (1-\eta\mu)^2 \mathbb{E}\left[\left\|x_{t-1}^m - x_{t-1}^n\right\|_2^2\right] + 80\eta^4\sigma^2 K H^2 \zeta^2 + 16\eta^4\sigma^4 \ ,$$

$$\leq \left(1 + \frac{1}{K-1}\right)^3 \mathbb{E}\left[\left\|x_{t-1}^m - x_{t-1}^n\right\|_2^4\right] + 32\eta^4 K^3 H^4 \zeta^4$$
$$+ 40\eta^2\sigma^2 \left(3K\sigma^2\eta^2 + 6K^2\eta^2 H^2 \zeta^2\right) + 80\eta^4\sigma^2 K H^2 \zeta^2 + 16\eta^4\sigma^4 \ ,$$

$$\leq \left(1 + \frac{1}{K-1}\right)^3 \mathbb{E}\left[\left\|x_{t-1}^m - x_{t-1}^n\right\|_2^4\right] + 32\eta^4 K^3 H^4 \zeta^4 + 136\eta^4 K \sigma^4 + 320\eta^4\sigma^2 K^2 H^2 \zeta^2 \ ,$$

$$\leq \left(1 + \frac{1}{K-1}\right)^{3(K-1)} \left(32\eta^4 K^4 H^4 \zeta^4 + 136\eta^4 K^2 \sigma^4 + 320\eta^4\sigma^2 K^3 H^2 \zeta^2\right) \ ,$$

$$\leq^{(a)} 20 \left(32\eta^4 K^4 H^4 \zeta^4 + 136\eta^4 K^2 \sigma^4 + 160 \cdot 2 \cdot \left(\eta^2\sigma^2 K\right) \cdot \left(\eta^2 H^2 \zeta^2 K^2\right)\right) \ ,$$

$$\leq 20 \left(192\eta^4 K^4 H^4 \zeta^4 + 296\eta^4 K^2 \sigma^4\right) \ ,$$

$$\leq 3840\eta^4 K^4 H^4 \zeta^4 + 5920\eta^4 K^2 \sigma^4 \ ,$$

where in (a) we used that $(1 + 1/x)^x \leq 20$ for all $x \geq 0$. Finally averaging this over $m, n \in [M]$ implies,

$$\frac{1}{M} \sum_{m \in [M]} \mathbb{E}\left[\left\|x_t - x_t^m\right\|_2^4\right] \leq \frac{1}{M^2} \sum_{m,n \in [M]} \mathbb{E}\left[\left\|x_t^n - x_t^m\right\|_2^4\right] \ ,$$
$$\leq 3840\eta^4 K^4 H^4 \zeta^4 + 5920\eta^4 K^2 \sigma^4 \ ,$$

which proves the lemma. $\qquad\square$

### G.3 Convergence in Iterates

In this sub-section, we provide a convergence guarantee for the iterates of local SGD under Assumptions 1, 6 and 7. We do so by using the red upper bound from Lemma 20.

**Lemma 28** (Convergence with $\zeta$, $\tau$ and $Q$). *We can prove the following convergence guarantee assuming $\eta < 1/H$:*

$$A(T) \leq (1-\eta\mu)^{KR} B^2 + \frac{15360 Q^2 \eta^4 K^4 H^4 \zeta^4}{\mu^2} + \frac{23680 Q^2 \eta^4 K^2 \sigma^4}{\mu^2} + \frac{24\tau^2\eta^2 K^2 H^2 \zeta^2}{\mu^2}$$
$$+ \frac{12\tau^2\eta^2 K \sigma^2}{\mu^2} + \frac{\eta\sigma^2}{\mu M} \ .$$

*Proof.* Use the red upper bound for one-step progress from Lemma 20. We first restate the one-step lemma using the red upper bound,

$$A(KR)$$
$$\leq (1-\eta\mu) A(KR-1) + \frac{2\eta Q^2}{\mu} D(KR-1) + \frac{2\eta\tau^2}{\mu} C(KR-1) + \frac{\eta^2\sigma^2}{M} \ ,$$

$$\leq (1-\eta\mu)^K A(K(R-1)) + \frac{2\eta}{\mu} \sum_{j=K(R-1)}^{KR-1} (1-\eta\mu)^{KR-1-j} \left(Q^2 D(j) + \tau^2 C(j)\right)$$
$$+ \left(1 - (1-\eta\mu)^K\right) \frac{\eta\sigma^2}{\mu M} \ ,$$

$$\leq (1-\eta\mu)^K A(K(R-1))$$
$$+ \frac{2Q^2\eta}{\mu} \sum_{j=K(R-1)}^{KR-1} (1-\eta\mu)^{KR-1-j} \left(3840\eta^4 K^4 H^4 \zeta^4 + 5920\eta^4 K^2 \sigma^4\right)$$
$$+ \frac{2\tau^2\eta}{\mu} \sum_{j=K(R-1)}^{KR-1} (1-\eta\mu)^{KR-1-j} \left(6K^2\eta^2 H^2 \zeta^2 + 3K\eta^2\sigma^2\right)$$

$$+ \left(1 - (1 - \eta\mu)^K\right) \frac{\eta\sigma^2}{\mu M} \ ,$$

$$\leq (1 - \eta\mu)^K A(K(R-1))$$

$$+ \left(\frac{7680Q^2\eta^5 K^4 H^4 \zeta^4}{\mu} + \frac{11840Q^2\eta^5 K^2 \sigma^4}{\mu}\right) \sum_{j=K(R-1)}^{KR-1} (1 - \eta\mu)^{KR-1-j}$$

$$+ \left(\frac{12\tau^2\eta^3 K^2 H^2 \zeta^2}{\mu} + \frac{6\tau^2\eta^3 K\sigma^2}{\mu}\right) \sum_{j=K(R-1)}^{KR-1} (1 - \eta\mu)^{KR-1-j}$$

$$+ \left(1 - (1 - \eta\mu)^K\right) \frac{\eta\sigma^2}{\mu M} \ . \tag{a}$$

Note that we can simplify the summation as follows,

$$\sum_{j=K(R-1)}^{KR-1} (1 - \eta\mu)^{KR-1-j} = \sum_{i=0}^{K-1} (1 - \eta\mu)^i = \frac{1 - (1 - \eta\mu)^K}{\eta\mu} \ .$$

Plugging the above result back into (a) gives us,

$$A(KR) \leq (1 - \eta\mu)^K A(K(R-1)) + \left(1 - (1 - \eta\mu)^K\right) \left(\frac{7680Q^2\eta^4 K^4 H^4 \zeta^4}{\mu^2}\right)$$

$$+ \quad \left(1 - (1 - \eta\mu)^K\right) \left(\frac{11840Q^2\eta^4 K^2 \sigma^4}{\mu^2} + \frac{12\tau^2\eta^2 K^2 H^2 \zeta^2}{\mu^2} + \frac{6\tau^2\eta^2 K\sigma^2}{\mu^2} + \frac{\eta\sigma^2}{\mu M}\right) \ ,$$

Now we unroll the above inequality over $R$ rounds and we have,

$$A(KR) \leq (1 - \eta\mu)^{KR} B^2 + \frac{7680Q^2\eta^4 K^4 H^4 \zeta^4}{\mu^2} + \frac{11840Q^2\eta^4 K^2 \sigma^4}{\mu^2} + \frac{12\tau^2\eta^2 K^2 H^2 \zeta^2}{\mu^2}$$

$$+ \frac{6\tau^2\eta^2 K\sigma^2}{\mu^2} + \frac{\eta\sigma^2}{\mu M} \ ,$$

which finishes the proof. $\qquad\square$

### G.4 Tuning the Step-size

In the previous subsection, we provided the following bound on the iterates of local SGD,

$$A(KR) \leq e^{-\eta\mu KR} B^2 + \frac{7680Q^2\eta^4 K^4 H^4 \zeta^4}{\mu^2} + \frac{11840Q^2\eta^4 K^2 \sigma^4}{\mu^2} + \frac{12\tau^2\eta^2 K^2 H^2 \zeta^2}{\mu^2}$$

$$+ \frac{6\tau^2\eta^2 K\sigma^2}{\mu^2} + \frac{\eta\sigma^2}{\mu M} \ .$$

To achieve the final bound, we need to tune the step-size to have the following Theorem,

**Lemma 29.** *For all $t \in [0, T]$ under Assumptions 1, 2, 6 and 7 assuming $\eta \leq \frac{1}{2H}$ we have,*

$$A(KR) \leq e^{-\frac{\mu KR}{2H}} + \tilde{\mathcal{O}}\left(\frac{7680Q^2 K^4 \zeta^4}{\mu^6 R^4} + \frac{11840Q^2 \sigma^4}{\mu^6 K^2 R^4} + \frac{12\tau^2 \zeta^2}{\mu^4 R^2} + \frac{6\tau^2 \sigma^2}{\mu^4 KR^2} + \frac{\sigma^2}{\mu^2 MKR}\right) \ .$$

*Proof.* We pick the step-size as follows:

- if $\frac{1}{2H} \geq \frac{1}{\mu KR} \ln(\max\{2, \mu^2 B^2 T/\sigma^2\})$, we choose $\eta = \frac{1}{\mu KR} \ln(\max\{2, \mu^2 B^2 T/\sigma^2\})$.

- if $\frac{1}{2H} \leq \frac{1}{\mu KR} \ln(\max\{2, \mu^2 B^2 T/\sigma^2\})$, we choose $\eta = \frac{1}{2H}$.

Which gives us the following result,

$$A(KR) \leq e^{-\frac{\mu KR}{2H}} + + \tilde{\mathcal{O}}\left(\frac{7680Q^2 K^4 \zeta^4}{\mu^6 R^4} + \frac{11840Q^2 \sigma^4}{\mu^6 K^2 R^4} + \frac{12\tau^2 \zeta^2}{\mu^4 R^2} + \frac{6\tau^2 \sigma^2}{\mu^4 KR^2} + \frac{\sigma^2}{\mu^2 MKR}\right) \ ,$$

where the notation $\tilde{\mathcal{O}}(\cdot)$ hides the logarithmic terms. This proves the lemma. $\qquad\square$

# H Double Recursions for Consensus Error

In this section, we will relate the consensus error to the iterate errors of the previous communication round. This would allow us to get more fine-grained upper bounds on consensus error, which would decay with time and more communication.

## H.1 Second Moment of the Consensus Error

We can prove the following bound on the second moment of the consensus error using Assumptions 4 and 6.

**Lemma 30.** *For all $t \in [0, T]$ assuming $\eta < \frac{1}{H}$ we have,*

$$
\begin{aligned}
C(t) &\leq 2\eta^2 H^2 K^2 \zeta_\star^2 + \frac{2\eta^3 \tau^2 K^2 \sigma^2}{\mu} + 2\eta^2 \sigma^2 K \left(1 + \ln(K)\right) \\
&\quad + 4\eta^2 \tau^2 \left(t - \delta(t)\right)^2 \left(1 - \eta\mu\right)^{2(t-1-\delta(t))} \left(A(\delta(t)) + \phi_\star^2\right) \ , \\
&\leq 2\eta^2 H^2 K^2 \zeta_\star^2 + \frac{2\eta^3 \tau^2 K^2 \sigma^2}{\mu} + 2\eta^2 \sigma^2 K \left(1 + \ln(K)\right) + 4\eta^2 \tau^2 K^2 \left(A(\delta(t)) + \phi_\star^2\right) \ .
\end{aligned}
$$

*This also implies that for $r \in [R]$,*

$$
\sum_{j=K(r-1)}^{Kr-1} (1 - \eta\mu)^{Kr-1-j} C(j)
$$

$$
\begin{aligned}
&\leq \frac{1 - (1 - \eta\mu)^K}{\eta\mu} \left(2\eta^2 H^2 K^2 \zeta_\star^2 + \frac{2\eta^3 \tau^2 K^2 \sigma^2}{\mu} + 2\eta^2 \sigma^2 K \left(1 + \ln(K)\right)\right) \\
&\quad + \frac{1 - (1 - \eta\mu)^K}{\eta\mu} 4\eta^2 \tau^2 K^2 (1 - \eta\mu)^{K-2} \left(A(K(r-1)) + \phi_\star^2\right) \ .
\end{aligned}
$$

*Proof.* Note the following about the difference of iterates on two machines $m, n \in [M]$ for some time $t > \delta(t)$ (for $t = \delta(t)$ the l.h.s. is zero),

$$
\mathbb{E}\left[\|x_t^m - x_t^n\|_2^2\right]
$$

$$
= \mathbb{E}\left[\|x_{t-1}^m - x_{t-1}^n - \eta g_{t-1}^m + \eta g_{t-1}^n\|_2^2\right] \ ,
$$

$$
\leq^{\text{(Assumption 2), (a)}} \mathbb{E}\left[\|x_{t-1}^m - x_{t-1}^n - \eta \left(\nabla F_m(x_{t-1}^m) - \nabla F_n(x_{t-1}^n)\right)\|_2^2\right] + 2\eta^2 \sigma^2 \ ,
$$

$$
=^{\text{(b)}} \mathbb{E}\left[\|x_{t-1}^m - x_{t-1}^n - \eta \left(\nabla F_m(x_{t-1}^m) - \nabla F_m(x_{t-1}^n)\right) - \eta \left(\nabla F_m(x_{t-1}^n) - \nabla F_n(x_{t-1}^n)\right)\|_2^2\right]
$$

$$
+ 2\eta^2 \sigma^2 \ ,
$$

where in (a) we exploited the fact that $\xi_t^m \perp \xi_t^n | \mathcal{H}_t$ and $x_{t-1}^m, x_{t-1}^n \in m\mathcal{H}_t$ as well as used tower rule to introduce conditional expectation; and in (b) we added and subtracted the term $\nabla F_m(x_{t-1}^n)$. By mean value theorem we know that there exists a $c = x_{t-1}^n + \theta(x_{t-1}^m - x_{t-1}^n)$ for some $\theta \in [0, 1]$ such that:

$$
\nabla F_m(x_{t-1}^m) - \nabla F_m(x_{t-1}^n) = \nabla^2 F_m(c)(x_{t-1}^m - x_{t-1}^n)
$$

Using this in the above inequality, we get:

$$
\mathbb{E}\left[\|x_t^m - x_t^n\|_2^2\right]
$$

$$
\leq \mathbb{E}\left[\|x_{t-1}^m - x_{t-1}^n - \eta \nabla^2 F_m(c)(x_{t-1}^m - x_{t-1}^n) - \eta \left(\nabla F_m(x_{t-1}^n) - \nabla F_n(x_{t-1}^n)\right)\|_2^2\right] + 2\eta^2 \sigma^2 \ ,
$$

$$
= \mathbb{E}\left[\|\left(I - \eta\nabla^2 F_m(c)\right)(x_{t-1}^m - x_{t-1}^n) - \eta \left(\nabla F_m(x_{t-1}^n) - \nabla F_n(x_{t-1}^n)\right)\|_2^2\right] + 2\eta^2 \sigma^2 \ ,
$$

$$
\leq^{\text{(a)}} \left(1 + \frac{1}{\gamma_{t-1}}\right) \mathbb{E}\left[\|\left(I - \eta\nabla^2 F_m(c)\right)(x_{t-1}^m - x_{t-1}^n)\|_2^2\right]
$$

$$+ \left(1 + \gamma_{t-1}\right) \eta^2 \mathbb{E}\left[\left\|\nabla F_m(x_{t-1}^n) - \nabla F_n(x_{t-1}^n)\right\|_2^2\right] + 2\eta^2\sigma^2 \ ,$$

$$\leq^{(b)} \left(1 + \frac{1}{\gamma_{t-1}}\right)(1 - \eta\mu)^2 \mathbb{E}\left[\left\|x_{t-1}^m - x_{t-1}^n\right\|_2^2\right]$$
$$+ \left(1 + \gamma_{t-1}\right)\eta^2 \mathbb{E}\left[\left\|\nabla F_m(x_{t-1}^n) - \nabla F_n(x_{t-1}^n)\right\|_2^2\right] + 2\eta^2\sigma^2 \ ,$$

$$= \left(1 + \frac{1}{\gamma_{t-1}}\right)(1 - \eta\mu)^2 \mathbb{E}\left[\left\|x_{t-1}^m - x_{t-1}^n\right\|_2^2\right] + 2\eta^2\sigma^2$$
$$+ \left(1 + \gamma_{t-1}\right)\eta^2 \mathbb{E}\left[\left\|\nabla F_m(x_{t-1}^n) - \nabla F_m(x_n^\star) - \nabla F_n(x_{t-1}^n) + \nabla F_m(x_n^\star)\right\|_2^2\right] \ ,$$

$$\leq^{(c)} \left(1 + \frac{1}{\gamma_{t-1}}\right)(1 - \eta\mu)^2 \mathbb{E}\left[\left\|x_{t-1}^m - x_{t-1}^n\right\|_2^2\right]$$
$$+ 2\left(1 + \gamma_{t-1}\right)\eta^2 \mathbb{E}\left[\left\|\nabla F_m(x_{t-1}^n) - \nabla F_m(x_n^\star) - \nabla F_n(x_{t-1}^n) + \nabla F_n(x_n^\star)\right\|_2^2\right]$$
$$+ 2\left(1 + \gamma_{t-1}\right)\eta^2 \mathbb{E}\left[\left\|\nabla F_m(x_n^\star) - \nabla F_m(x_m^\star)\right\|_2^2\right] + 2\eta^2\sigma^2 \ ,$$

$$\leq^{\text{(Assumptions 1 and 4)}} \left(1 + \frac{1}{\gamma_{t-1}}\right)(1 - \eta\mu)^2 \mathbb{E}\left[\left\|x_{t-1}^m - x_{t-1}^n\right\|_2^2\right]$$
$$\color{blue}{+ 2\left(1 + \gamma_{t-1}\right)\eta^2 \mathbb{E}\left[\left\|\nabla F_m(x_{t-1}^n) - \nabla F_n(x_{t-1}^n) - \left(\nabla F_m(x_n^\star) - \nabla F_n(x_n^\star)\right)\right\|_2^2\right]}$$
$$+ 2\left(1 + \gamma_{t-1}\right)\eta^2 H^2 \zeta_\star^2 + 2\eta^2\sigma^2 \ ,$$

where in (a) and (c) we used Lemma 16; and in (b) we used Assumption 1 and the fact that $\eta < 1/H$. We will again use the mean value theorem for the blue term in the above inequality. For $v := x_{t-1}^n - x_n^\star$ we have:

$$\nabla F_m(x_{t-1}^n) - \nabla F_n(x_{t-1}^n) - \left(\nabla F_m(x_n^\star) - \nabla F_n(x_n^\star)\right)$$
$$= \int_0^1 \left(\nabla^2 F_m(x_n^\star + tv) - \nabla^2 F_n(x_n^\star + tv)\right) v\, dt \ ,$$
$$\Rightarrow \left\|\nabla F_m(x_{t-1}^n) - \nabla F_n(x_{t-1}^n) - \left(\nabla F_m(x_n^\star) - \nabla F_n(x_n^\star)\right)\right\|_2$$
$$\leq \int_0^1 \left\|\nabla^2 F_m(x_n^\star + tv) - \nabla^2 F_n(x_n^\star + tv)\right\|_2 \|v\|_2\, dt \ ,$$
$$\Rightarrow \left\|\nabla F_m(x_{t-1}^n) - \nabla F_n(x_{t-1}^n) - \left(\nabla F_m(x_n^\star) - \nabla F_n(x_n^\star)\right)\right\|_2 \leq \tau \left\|x_{t-1}^n - x_n^\star\right\|_2 \ .$$

Plugging this in the inequality above gives the following,

$$\mathbb{E}\left[\left\|x_t^m - x_t^n\right\|_2^2\right]$$
$$\leq \left(1 + \frac{1}{\gamma_{t-1}}\right)(1 - \eta\mu)^2 \mathbb{E}\left[\left\|x_{t-1}^m - x_{t-1}^n\right\|_2^2\right]$$
$$+ 2\left(1 + \gamma_{t-1}\right)\eta^2\tau^2 \mathbb{E}\left[\left\|x_{t-1}^n - x_n^\star\right\|_2^2\right] + 2\left(1 + \gamma_{t-1}\right)\eta^2 H^2 \zeta_\star^2 + 2\eta^2\sigma^2 \ ,$$
$$\leq \left(1 + \frac{1}{\gamma_{t-1}}\right)(1 - \eta\mu)^2 \mathbb{E}\left[\left\|x_{t-1}^m - x_{t-1}^n\right\|_2^2\right] + 2\left(1 + \gamma_{t-1}\right)\eta^2 H^2 \zeta_\star^2 + 2\eta^2\sigma^2$$
$$+ 2\left(1 + \gamma_{t-1}\right)\eta^2\tau^2$$
$$\times \left((1 - \eta\mu)^{2(t-1-\delta(t))} \mathbb{E}\left[\left\|x_{\delta(t)} - x_n^\star\right\|_2^2\right] + \left(1 - (1 - \eta\mu)^{2(t-1-\delta(t))}\right)\frac{\eta\sigma^2}{\mu}\right) \ ,$$

where in the last inequality above we just used an upper bound for the convergence of SGD on a single machine $n \in [M]$. As a sanity check note that if $t - 1 = \delta(t)$ then the red term becomes $\mathbb{E}\left[\left\|x_{\delta(t)} - x_n^\star\right\|_2^2\right]$. Continuing further and choosing $\gamma_j = j - \delta(j)$ (note that the term with $1/\gamma_{t-1}$ becomes 1 when $t - 1 = \delta(t)$, making this choice well defined), this leads to,

$$\mathbb{E}\left[\left\|x_t^m - x_t^n\right\|_2^2\right]$$

$$
\leq \prod_{j=\delta(t)}^{t-1} \left( 1 + \frac{1}{\gamma_j} \right) (1 - \eta\mu)^2 \mathbb{E}\left[ \left\| x_{\delta(t)} - x_{\delta(t)} \right\|_2^2 \right]
$$

$$
+ 2\eta^2 \sum_{j=\delta(t)}^{t-1} \left( \prod_{i=j+1}^{t-1} \left( 1 + \frac{1}{\gamma_i} \right) (1 - \eta\mu)^2 \right) \left( (1 + \gamma_j) H^2 \zeta_\star^2 + \sigma^2 \right)
$$

$$
+ 2\eta^2\tau^2 \sum_{j=\delta(t)}^{t-1} \left( \prod_{i=j+1}^{t-1} \left( 1 + \frac{1}{\gamma_i} \right) (1 - \eta\mu)^2 \right) (1 + \gamma_j)(1 - \eta\mu)^{2(j-\delta(t))} \mathbb{E}\left[ \left\| x_{\delta(t)} - x_n^\star \right\|_2^2 \right]
$$

$$
+ \frac{2\eta^3\tau^2\sigma^2}{\mu} \sum_{j=\delta(t)}^{t-1} \left( \prod_{i=j+1}^{t-1} \left( 1 + \frac{1}{\gamma_i} \right) (1 - \eta\mu)^2 \right) (1 + \gamma_j) \left( 1 - (1 - \eta\mu)^{2(j-\delta(t))} \right) \ ,
$$

$$
= 2\eta^2 \sum_{j=\delta(t)}^{t-1} \left( \prod_{i=j+1}^{t-1} \left( 1 + \frac{1}{\gamma_i} \right) \right) (1 - \eta\mu)^{2(t-1-j)} \left( (1 + \gamma_j) H^2 \zeta_\star^2 + \sigma^2 \right)
$$

$$
+ 2\eta^2\tau^2 \sum_{j=\delta(t)}^{t-1} \left( \prod_{i=j+1}^{t-1} \left( 1 + \frac{1}{\gamma_i} \right) \right) (1 - \eta\mu)^{2(t-1-j)} (1 + \gamma_j)(1 - \eta\mu)^{2(j-\delta(t))} \mathbb{E}\left[ \left\| x_{\delta(t)} - x_n^\star \right\|_2^2 \right]
$$

$$
+ \frac{2\eta^3\tau^2\sigma^2}{\mu} \sum_{j=\delta(t)}^{t-1} \left( \prod_{i=j+1}^{t-1} \left( 1 + \frac{1}{\gamma_i} \right) \right) (1 - \eta\mu)^{2(t-1-j)} (1 + \gamma_j) \left( 1 - (1 - \eta\mu)^{2(j-\delta(t))} \right) \ ,
$$

$$
= 2\eta^2 \sum_{j=\delta(t)}^{t-1} \left( \prod_{i=j+1}^{t-1} \frac{i+1-\delta(t)}{i-\delta(t)} \right) (1 - \eta\mu)^{2(t-1-j)} \left( (j + 1 - \delta(t)) H^2 \zeta_\star^2 + \sigma^2 \right)
$$

$$
+ 2\eta^2\tau^2 \sum_{j=\delta(t)}^{t-1} \left( \prod_{i=j+1}^{t-1} \frac{i+1-\delta(t)}{i-\delta(t)} \right) (j + 1 - \delta(t))(1 - \eta\mu)^{2(t-1-\delta(t))} \mathbb{E}\left[ \left\| x_{\delta(t)} - x_n^\star \right\|_2^2 \right]
$$

$$
+ \frac{2\eta^3\tau^2\sigma^2}{\mu} \sum_{j=\delta(t)}^{t-1} \left( \prod_{i=j+1}^{t-1} \frac{i+1-\delta(t)}{i-\delta(t)} \right) (j + 1 - \delta(t)) \left( (1 - \eta\mu)^{2(t-1-j)} - (1 - \eta\mu)^{2(t-1-\delta(t))} \right) \ ,
$$

$$
= 2\eta^2 \sum_{j=\delta(t)}^{t-1} \frac{t - \delta(t)}{j + 1 - \delta(t)} (1 - \eta\mu)^{2(t-1-j)} \left( (j + 1 - \delta(t)) H^2 \zeta_\star^2 + \sigma^2 \right)
$$

$$
+ 2\eta^2\tau^2 \sum_{j=\delta(t)}^{t-1} (t - \delta(t))(1 - \eta\mu)^{2(t-1-\delta(t))} \mathbb{E}\left[ \left\| x_{\delta(t)} - x_n^\star \right\|_2^2 \right]
$$

$$
+ \frac{2\eta^3\tau^2\sigma^2}{\mu} \sum_{j=\delta(t)}^{t-1} (t - \delta(t)) \left( (1 - \eta\mu)^{2(t-1-j)} - (1 - \eta\mu)^{2(t-1-\delta(t))} \right) \ ,
$$

$$
= 2\eta^2 (t - \delta(t)) \sum_{j=\delta(t)}^{t-1} (1 - \eta\mu)^{2(t-1-j)} \left( H^2 \zeta_\star^2 + \frac{\sigma^2}{j + 1 - \delta(t)} \right)
$$

$$
+ 2\eta^2\tau^2 (t - \delta(t))^2 (1 - \eta\mu)^{2(t-1-\delta(t))} \mathbb{E}\left[ \left\| x_{\delta(t)} - x_n^\star \right\|_2^2 \right]
$$

$$
+ \frac{2\eta^3\tau^2\sigma^2}{\mu} \sum_{j=\delta(t)}^{t-1} (t - \delta(t)) \left( (1 - \eta\mu)^{2(t-1-j)} - (1 - \eta\mu)^{2(t-1-\delta(t))} \right) \ ,
$$

$$
\textcolor{red}{\leq 2\eta^2 (t - \delta(t))^2 H^2 \zeta_\star^2 + 2\eta^2 (t - \delta(t))\sigma^2 \sum_{j=\delta(t)}^{t-1} \frac{1}{j + 1 - \delta(t)}}
$$

$$+ 2\eta^2\tau^2 \left(t - \delta(t)\right)^2 (1 - \eta\mu)^{2(t-1-\delta(t))} \mathbb{E}\left[\left\|x_{\delta(t)} - x_n^\star\right\|_2^2\right]$$

$$+ \frac{2\eta^3\tau^2\sigma^2 \left(t - \delta(t)\right)^2}{\mu} \quad ,$$

$$\leq^{(a)} \left(t - \delta(t)\right)^2 \left(2\eta^2 H^2\zeta_\star^2 + \frac{2\eta^3\tau^2\sigma^2}{\mu}\right) + 2\eta^2\sigma^2(t - \delta(t))\left(1 + \ln(t - \delta(t))\right)$$

$$+ 4\eta^2\tau^2 \left(t - \delta(t)\right)^2 (1 - \eta\mu)^{2(t-1-\delta(t))} \left(\mathbb{E}\left[\left\|x_{\delta(t)} - x^\star\right\|_2^2\right] + \|x^\star - x_n^\star\|_2^2\right) \quad ,$$

$$\leq \left(t - \delta(t)\right) \left(2\eta^2 H^2 K\zeta_\star^2 + \frac{2\eta^3\tau^2 K\sigma^2}{\mu} + 2\eta^2\sigma^2 \left(1 + \ln(K)\right)\right)$$

$$+ 4\eta^2\tau^2 \left(t - \delta(t)\right)^2 (1 - \eta\mu)^{2(t-1-\delta(t))} \left(A(\delta(t)) + \phi_\star^2\right) \quad , \tag{9}$$

$$\leq 2\eta^2 H^2 K^2\zeta_\star^2 + \frac{2\eta^3\tau^2 K^2\sigma^2}{\mu} + 2\eta^2\sigma^2 K \left(1 + \ln(K)\right)$$

$$+ 4\eta^2\tau^2 \left(t - \delta(t)\right)^2 (1 - \eta\mu)^{2(t-1-\delta(t))} \left(A(\delta(t)) + \phi_\star^2\right) \quad ,$$

where in (a) we combined the red terms into one, used the fact that

$$\frac{1 - (1 - \eta\mu)^{2(t-\delta(t))}}{1 - (1 - \eta\mu)^2} \leq \frac{1}{\eta\mu(2 - \eta\mu)} \leq \frac{1}{\eta\mu}$$

because $\eta < 1/H$ and used Lemma 16. As a sanity check, note that the above bound has the property that when $t = \delta(t)$, it automatically becomes zero (we adopt the notation that $0 \cdot (-\infty)$ in the second term becomes 0). Thus, we can safely drop the assumption that $t > \delta(t)$, making the above bound valid for all values of $t$. Finally, averaging the upper bound over $m, n \in [M]$ proves the lemma's main upper bound. To get the other result, we will simply use this upper bound. In particular noting that $\delta(j) = K(r - 1)$ we get,

$$\sum_{j=K(r-1)}^{Kr-1} (1 - \eta\mu)^{Kr-1-j} C(j)$$

$$\leq \sum_{j=K(r-1)}^{Kr-1} (1 - \eta\mu)^{Kr-1-j} \left(2\eta^2 H^2 K^2\zeta_\star^2 + \frac{2\eta^3\tau^2 K^2\sigma^2}{\mu} + 2\eta^2\sigma^2 K \left(1 + \ln(K)\right)\right)$$

$$+ \sum_{j=K(r-1)}^{Kr-1} (1 - \eta\mu)^{Kr-1-j} \left(4\eta^2\tau^2 \left(j - K(r-1)\right)^2 (1 - \eta\mu)^{2(j-1-K(r-1))} \left(A(K(r-1)) + \phi_\star^2\right)\right) \quad ,$$

$$= \frac{1 - (1 - \eta\mu)^K}{\eta\mu} \left(2\eta^2 H^2 K^2\zeta_\star^2 + \frac{2\eta^3\tau^2 K^2\sigma^2}{\mu} + 2\eta^2\sigma^2 K \left(1 + \ln(K)\right)\right)$$

$$+ 4\eta^2\tau^2(1 - \eta\mu)^{K-2} \left(A(K(r-1)) + \phi_\star^2\right) \sum_{j=K(r-1)}^{Kr-1} (1 - \eta\mu)^{j-1-K(r-1)} \left(j - K(r-1)\right)^2 \quad ,$$

$$\leq \frac{1 - (1 - \eta\mu)^K}{\eta\mu} \left(2\eta^2 H^2 K^2\zeta_\star^2 + \frac{2\eta^3\tau^2 K^2\sigma^2}{\mu} + 2\eta^2\sigma^2 K \left(1 + \ln(K)\right)\right)$$

$$+ \frac{1 - (1 - \eta\mu)^K}{\eta\mu} 4\eta^2\tau^2 K^2(1 - \eta\mu)^{K-2} \left(A(K(r-1)) + \phi_\star^2\right) \quad .$$

This finishes the proof. $\qquad \square$

## H.2 Fourth Moment of the Consensus Error

**Lemma 31.** *For all $t \in [0, T]$ assuming $\eta < 1/H$ we have,*

$$D(t) \leq \left(\frac{128\eta^5\tau^4\sigma^2}{\mu}(t - \delta(t)) + 320\eta^4\sigma^2\tau^2\right)(t - \delta(t))^3(1 - \eta\mu)^{t-1-\delta(t)} \left(A(\delta(t)) + \phi_\star^2\right)$$

$$+ 64\eta^4\tau^4(t - \delta(t))^4(1 - \eta\mu)^{t-1-\delta(t)} \left(B(\delta(t)) + \phi_\star^4\right)$$

$$+ \left( \frac{8\eta^3 H^4 \zeta_\star^4}{\mu} + \frac{88\eta^5 \tau^4 \sigma^4}{\mu^3} + 160\eta^4 K \sigma^2 H^2 \zeta_\star^2 + \frac{160\eta^5 \tau^2 K \sigma^4}{\mu} + 112\eta^4 \sigma^4 \left(1 + \ln(K)\right) \right) (t - \delta(t))^3 \ ,$$

$$\leq \left( \frac{128\eta^5 \tau^4 K^4 \sigma^2}{\mu} + 320\eta^4 \sigma^2 \tau^2 K^3 \right) \left( A(\delta(t)) + \phi_\star^2 \right) + 64\eta^4 \tau^4 K^4 \left( B(\delta(t)) + \phi_\star^4 \right)$$

$$+ \frac{8\eta^3 K^3 H^4 \zeta_\star^4}{\mu} + \frac{88\eta^5 K^3 \tau^4 \sigma^4}{\mu^3} + 160\eta^4 K^4 \sigma^2 H^2 \zeta_\star^2 + \frac{160\eta^5 \tau^2 K^4 \sigma^4}{\mu} + 112\eta^4 K^3 \sigma^4 \left(1 + \ln(K)\right) \ .$$

*This also implies that for $r \in [R]$,*

$$\sum_{j=K(r-1)}^{Kr-1} (1 - \eta\mu)^{Kr-1-j} D(j)$$

$$\leq \left(1 - (1 - \eta\mu)^K\right) \left( \frac{128\eta^4 K^4 \tau^4 \sigma^2}{\mu^2} + \frac{320\eta^3 K^3 \sigma^2 \tau^2}{\mu} \right) (1 - \eta\mu)^{K-4} \left( A(K(r-1)) + \phi_\star^2 \right)$$

$$+ \left(1 - (1 - \eta\mu)^K\right) \frac{64\eta^3 K^4 \tau^4}{\mu} (1 - \eta\mu)^{K-5} \left( B(K(r-1)) + \phi_\star^4 \right)$$

$$\left(1 - (1 - \eta\mu)^K\right)$$

$$\times \left( \frac{8\eta^2 K^3 H^4 \zeta_\star^4}{\mu^2} + \frac{88\eta^4 K^3 \tau^4 \sigma^4}{\mu^4} + \frac{160\eta^3 K^4 \sigma^2 H^2 \zeta_\star^2}{\mu} + \frac{160\eta^4 \tau^2 K^4 \sigma^4}{\mu^2} + \frac{112\eta^3 K^3 \sigma^4 \left(1 + \ln(K)\right)}{\mu} \right) \ .$$

*Proof.* Note the following about the fourth moment of the difference between the iterates on two machines $m, n \in [M]$ for $t > \delta(t)$ (for $t = \delta(t)$ the l.h.s. is zero),

$$\mathbb{E}\left[ \left\| x_t^m - x_t^n \right\|_2^4 \right]$$

$$= \mathbb{E}\left[ \left\| x_{t-1}^m - x_{t-1}^n - \eta g_{t-1}^m + \eta g_{t-1}^n \right\|_2^4 \right] \ ,$$

$$= \mathbb{E}\left[ \left( \left\| x_{t-1}^m - x_{t-1}^n - \eta \nabla F_m(x_{t-1}^m) + \eta \nabla F_n(x_{t-1}^n) + \eta \xi_{t-1}^m - \eta \xi_{t-1}^n \right\|_2^2 \right)^2 \right] \ ,$$

$$= \mathbb{E}\left[ \left( \left\| x_{t-1}^m - x_{t-1}^n - \eta \nabla F_m(x_{t-1}^m) + \eta \nabla F_n(x_{t-1}^n) \right\|_2^2 + \eta^2 \left\| \xi_{t-1}^m - \xi_{t-1}^n \right\|_2^2 \right. \right.$$

$$\left. \left. + 2\eta \left\langle x_{t-1}^m - x_{t-1}^n - \eta \nabla F_m(x_{t-1}^m) + \eta \nabla F_n(x_{t-1}^n), \xi_{t-1}^m - \xi_{t-1}^n \right\rangle \right)^2 \right] \ ,$$

$$=^{(a)} \mathbb{E}\left[ \left\| x_{t-1}^m - x_{t-1}^n - \eta \nabla F_m(x_{t-1}^m) + \eta \nabla F_n(x_{t-1}^n) \right\|_2^4 \right] + \eta^4 \mathbb{E}\left[ \left\| \xi_{t-1}^m - \xi_{t-1}^n \right\|_2^4 \right]$$

$$+ 4\eta^2 \mathbb{E}\left[ \left( \left\langle x_{t-1}^m - x_{t-1}^n - \eta \nabla F_m(x_{t-1}^m) + \eta \nabla F_n(x_{t-1}^n), \xi_{t-1}^m - \xi_{t-1}^n \right\rangle \right)^2 \right]$$

$$+ 2\eta^2 \mathbb{E}\left[ \left\| x_{t-1}^m - x_{t-1}^n - \eta \nabla F_m(x_{t-1}^m) + \eta \nabla F_n(x_{t-1}^n) \right\|_2^2 \left\| \xi_{t-1}^m - \xi_{t-1}^n \right\|_2^2 \right]$$

$$+ 4\eta^3 \mathbb{E}\left[ \left\langle x_{t-1}^m - x_{t-1}^n - \eta \nabla F_m(x_{t-1}^m) + \eta \nabla F_n(x_{t-1}^n), \xi_{t-1}^m - \xi_{t-1}^n \right\rangle \left\| \xi_{t-1}^m - \xi_{t-1}^n \right\|_2^2 \right] \ ,$$

$$\leq^{\text{(Lemma 15), (b)}} \mathbb{E}\left[ \left\| x_{t-1}^m - x_{t-1}^n - \eta \nabla F_m(x_{t-1}^m) + \eta \nabla F_n(x_{t-1}^n) \right\|_2^4 \right] + 8\sigma^4 \eta^4$$

$$+ 6\eta^2 \mathbb{E}\left[ \left\| x_{t-1}^m - x_{t-1}^n - \eta \nabla F_m(x_{t-1}^m) + \eta \nabla F_n(x_{t-1}^n) \right\|_2^2 \right] \mathbb{E}\left[ \left\| \xi_{t-1}^m - \xi_{t-1}^n \right\|_2^2 \right]$$

$$+ 4\eta^3 \mathbb{E}\left[ \left\| x_{t-1}^m - x_{t-1}^n - \eta \nabla F_m(x_{t-1}^m) + \eta \nabla F_n(x_{t-1}^n) \right\|_2 \right] \mathbb{E}\left[ \left\| \xi_{t-1}^m - \xi_{t-1}^n \right\|_2^3 \right] \ ,$$

where in (a) we use the fact that $\mathbb{E}\left[\xi_{t-1}^m - \xi_{t-1}^n | \mathcal{H}_{t-1}\right] = 0$ and the conditional indepence of stochastic noise i.e., $\{\xi_{t-1}^m, \xi_{t-1}^n\} \perp \{x_{t-1}^m, x_{t-1}^n\} \mid \mathcal{H}_{t-1}$ allowing us to ignore one of the terms while expanding the square; and in (b) we again used this fact along with an application of Cauchy Shwartz inequality. In order to bound the term $\mathbb{E}\left[\left\|\xi_{t-1}^m - \xi_{t-1}^n\right\|_2^3\right]$ we use Cauchy-Schwarz Inequality:

$$\mathbb{E}\left[ \left\| \xi_{t-1}^m - \xi_{t-1}^n \right\|_2^3 \right] = \mathbb{E}\left[ \left\| \xi_{t-1}^m - \xi_{t-1}^n \right\|_2 \cdot \left\| \xi_{t-1}^m - \xi_{t-1}^n \right\|_2^2 \right]$$

$$\leq \sqrt{\mathbb{E}\left[\left\|\xi_{t-1}^m - \xi_{t-1}^n\right\|_2^2\right] \mathbb{E}\left[\left\|\xi_{t-1}^m - \xi_{t-1}^n\right\|_2^4\right]} \ ,$$

$$\leq^{\text{(Lemmas 12 and 14)}} 4\sqrt{\sigma^6} = 4\sigma^3 \ .$$

Also the term $\mathbb{E}\left[\left\|x_{t-1}^m - x_{t-1}^n - \eta\nabla F_m(x_{t-1}^m) + \eta\nabla F_n(x_{t-1}^n)\right\|_2\right]$ can be bounded as:

$$\mathbb{E}\left[\left\|x_{t-1}^m - x_{t-1}^n - \eta\nabla F_m(x_{t-1}^m) + \eta\nabla F_n(x_{t-1}^n)\right\|_2\right]$$

$$\overset{\text{(Jensen's Inequality)}}{\leq} \sqrt{\mathbb{E}\left[\left\|x_{t-1}^m - x_{t-1}^n - \eta\nabla F_m(x_{t-1}^m) + \eta\nabla F_n(x_{t-1}^n)\right\|_2^2\right]}$$

Putting everything back together gives us:

$$\mathbb{E}\left[\left\|x_t^n - x_t^m\right\|_2^4\right]$$

$$\overset{\text{(Assumption 2)}}{\leq} \mathbb{E}\left[\left\|x_{t-1}^m - x_{t-1}^n - \eta\nabla F_m(x_{t-1}^m) + \eta\nabla F_n(x_{t-1}^n)\right\|_2^4\right] + 8\eta^4\sigma^4$$

$$+ 12\eta^2\sigma^2 \mathbb{E}\left[\left\|x_{t-1}^m - x_{t-1}^n - \eta\nabla F_m(x_{t-1}^m) + \eta\nabla F_n(x_{t-1}^n)\right\|_2^2\right]$$

$$+ 16\eta^3 \sqrt{\sigma^6 \mathbb{E}\left[\left\|x_{t-1}^m - x_{t-1}^n - \eta\nabla F_m(x_{t-1}^m) + \eta\nabla F_n(x_{t-1}^n)\right\|_2^2\right]} \ ,$$

To bound the third term in the above inequality, we use the A.M. - G.M. Inequality $\sqrt{ab} \leq \frac{a}{2\gamma} + \frac{\gamma b}{2}$ for $\gamma > 0$. Let $\gamma = \eta, a = \sigma^2 \mathbb{E}\left[\left\|x_{t-1}^m - x_{t-1}^n - \eta\nabla F_m(x_{t-1}^m) + \eta\nabla F_n(x_{t-1}^n)\right\|_2^2\right], b = \sigma^4$. We have:

$$16\eta^3 \sqrt{\sigma^6 \mathbb{E}\left[\left\|x_{t-1}^m - x_{t-1}^n - \eta\nabla F_m(x_{t-1}^m) + \eta\nabla F_n(x_{t-1}^n)\right\|_2^2\right]}$$

$$= 16\eta^3 \sqrt{(\sigma^4)\left(\sigma^2 \mathbb{E}\left[\left\|x_{t-1}^m - x_{t-1}^n - \eta\nabla F_m(x_{t-1}^m) + \eta\nabla F_n(x_{t-1}^n)\right\|_2^2\right]\right)}$$

$$\leq 16\eta^3 \left(\frac{\eta\sigma^4}{2} + \frac{\sigma^2}{2\eta}\mathbb{E}\left[\left\|x_{t-1}^m - x_{t-1}^n - \eta\nabla F_m(x_{t-1}^m) + \eta\nabla F_n(x_{t-1}^n)\right\|_2^2\right]\right)$$

Plugging this upper bound and following a similar strategy as in Lemma 30 we get

$$\mathbb{E}\left[\left\|x_t^n - x_t^m\right\|_2^4\right]$$

$$\leq \mathbb{E}\left[\left\|x_{t-1}^m - x_{t-1}^n - \eta\nabla F_m(x_{t-1}^m) + \eta\nabla F_n(x_{t-1}^n)\right\|_2^4\right] + 8\eta^4\sigma^4$$

$$+ 12\eta^2\sigma^2 \mathbb{E}\left[\left\|x_{t-1}^m - x_{t-1}^n - \eta\nabla F_m(x_{t-1}^m) + \eta\nabla F_n(x_{t-1}^n)\right\|_2^2\right]$$

$$+ 16\eta^3 \left(\frac{\eta\sigma^4}{2} + \frac{\sigma^2}{2\eta}\mathbb{E}\left[\left\|x_{t-1}^m - x_{t-1}^n - \eta\nabla F_m(x_{t-1}^m) + \eta\nabla F_n(x_{t-1}^n)\right\|_2^2\right]\right) \ ,$$

$$= \mathbb{E}\left[\left\|x_{t-1}^m - x_{t-1}^n - \eta\nabla F_m(x_{t-1}^m) + \eta\nabla F_n(x_{t-1}^n)\right\|_2^4\right]$$

$$+ 20\eta^2\sigma^2 \mathbb{E}\left[\left\|x_{t-1}^m - x_{t-1}^n - \eta\nabla F_m(x_{t-1}^m) + \eta\nabla F_n(x_{t-1}^n)\right\|_2^2\right] + 16\eta^4\sigma^4 \ ,$$

$$= \mathbb{E}\left[\left\|x_{t-1}^m - x_{t-1}^n - \eta\nabla F_m(x_{t-1}^m) + \eta\nabla F_m(x_{t-1}^n) - \eta\nabla F_m(x_{t-1}^n) + \eta\nabla F_n(x_{t-1}^n)\right\|_2^4\right] + 16\eta^4\sigma^4$$

$$+ 20\eta^2\sigma^2 \mathbb{E}\left[\left\|x_{t-1}^m - x_{t-1}^n - \eta\nabla F_m(x_{t-1}^m) + \eta\nabla F_m(x_{t-1}^n) - \eta\nabla F_m(x_{t-1}^n) + \eta\nabla F_n(x_{t-1}^n)\right\|_2^2\right] \ ,$$

$$\leq^{\text{(Lemma 16)}} \left(1 + \frac{1}{\gamma_{t-1}}\right)^3 (1 - \eta\mu)^4 \mathbb{E}\left[\left\|x_{t-1}^m - x_{t-1}^n\right\|_2^4\right]$$

$$+ (1 + \gamma_{t-1})^3 \eta^4 \mathbb{E}\left[\left\|\nabla F_m(x_{t-1}^n) - \nabla F_n(x_{t-1}^n) - \nabla F_m(x_n^\star) + \nabla F_m(x_n^\star)\right\|_2^4\right]$$

$$+ 20\eta^2\sigma^2 \left(1 + \frac{1}{\gamma_{t-1}}\right)(1 - \eta\mu)^2 \mathbb{E}\left[\left\|x_{t-1}^m - x_{t-1}^n\right\|_2^2\right]$$

$$+ 20\eta^4\sigma^2(1+\gamma_{t-1})\mathbb{E}\left[\left\|\nabla F_m(x_{t-1}^n) - \nabla F_n(x_{t-1}^n) - \nabla F_m(x_n^\star) + \nabla F_m(x_n^\star)\right\|_2^2\right] + 16\eta^4\sigma^4 \ ,$$

$$\overset{\text{(Lemma 16 and Assumptions 4 and 6)}}{\leq} \left(1 + \frac{1}{\gamma_{t-1}}\right)^3 (1-\eta\mu)^4 \mathbb{E}\left[\left\|x_{t-1}^m - x_{t-1}^n\right\|_2^4\right]$$

$$+ 8\left(1+\gamma_{t-1}\right)^3 \eta^4 \left(\tau^4 \mathbb{E}\left[\left\|x_{t-1}^n - x_n^\star\right\|_2^4\right] + H^4\zeta_\star^4\right)$$

$$+ 20\eta^2\sigma^2 \left(1 + \frac{1}{\gamma_{t-1}}\right)(1-\eta\mu)^2 \mathbb{E}\left[\left\|x_{t-1}^m - x_{t-1}^n\right\|_2^2\right]$$

$$+ 40\eta^4\sigma^2(1+\gamma_{t-1})\left(\tau^2\mathbb{E}\left[\left\|x_{t-1}^n - x_n^\star\right\|_2^2\right] + H^2\zeta_\star^2\right) + 16\eta^4\sigma^4 \ .$$

Averaging this over $m, n \in [M]$ we have for all $t > \delta(t)$,

$$D(t) \leq \left(1 + \frac{1}{\gamma_{t-1}}\right)^3 (1-\eta\mu)^4 D(t-1)$$

$$+ 8\left(1+\gamma_{t-1}\right)^3 \eta^4\tau^4 \frac{1}{M}\sum_{n\in[M]} \mathbb{E}\left[\left\|x_{t-1}^n - x_n^\star\right\|_2^4\right] + 8\left(1+\gamma_{t-1}\right)^3\eta^4 H^4\zeta_\star^4$$

$$+ 20\eta^2\sigma^2 \left(1 + \frac{1}{\gamma_{t-1}}\right)(1-\eta\mu)^2 C(t-1)$$

$$+ 40\eta^4\sigma^2\tau^2(1+\gamma_{t-1})\frac{1}{M}\sum_{n\in[M]} \mathbb{E}\left[\left\|x_{t-1}^n - x_n^\star\right\|_2^2\right]$$

$$+ 40\eta^4\sigma^2(1+\gamma_{t-1})H^2\zeta_\star^2 + 16\eta^4\sigma^4 \ ,$$

Now we will use a couple of upper bounds that we already have for $\mathbb{E}\left[\left\|x_{t-1}^n - x_n^\star\right\|_2^4\right]$ from Lemma 23, $\mathbb{E}\left[\left\|x_{t-1}^n - x_n^\star\right\|_2^2\right]$ from Lemma 21 and $C(t-1)$ for $t-1 \geq \delta(t)$ from (9) in the proof of Lemma 30. This gives us the following with $\gamma_j = j - \delta(j) = j - \delta(t)$ for $j \geq \delta(t)$,

$$D(t)$$

$$\leq \left(1 + \frac{1}{\gamma_{t-1}}\right)^3 (1-\eta\mu)^4 D(t-1) + 8\left(1+\gamma_{t-1}\right)^3 \eta^4 H^4\zeta_\star^4 + (1+\gamma_{t-1})^3 \frac{88\eta^6\tau^4\sigma^4}{\mu^2}$$

$$+ 8\left(1+\gamma_{t-1}\right)^3 \eta^4\tau^4 \frac{1}{M}\sum_{n\in[M]} \left((1-\eta\mu)^{4(t-1-\delta(t))}\mathbb{E}\left[\left\|x_{\delta(t)} - x_n^\star\right\|_2^4\right]\right)$$

$$+ 8\left(1+\gamma_{t-1}\right)^3 \eta^4\tau^4 \frac{1}{M}\sum_{n\in[M]} \left(8\eta^2\sigma^2(t-1-\delta(t))(1-\eta\mu)^{2(t-1-\delta(t))}\mathbb{E}\left[\left\|x_{\delta(t)} - x_n^\star\right\|_2^2\right]\right)$$

$$+ 20\eta^2\sigma^2 \left(1 + \frac{1}{\gamma_{t-1}}\right)(1-\eta\mu)^2 C(t-1)$$

$$+ 40\eta^4\sigma^2\tau^2(1+\gamma_{t-1})\frac{1}{M}\sum_{n\in[M]} \left((1-\eta\mu)^{2(t-1-\delta(t))}\mathbb{E}\left[\left\|x_{\delta(t)} - x_n^\star\right\|_2^2\right] + \frac{\eta\sigma^2}{\mu}\right)$$

$$+ 40\eta^4\sigma^2(1+\gamma_{t-1})H^2\zeta_\star^2 + 16\eta^4\sigma^4 \ ,$$

$$\overset{\text{(Lemma 16 and Assumption 5)}}{\leq} \left(1 + \frac{1}{\gamma_{t-1}}\right)^3 (1-\eta\mu)^4 D(t-1) + 8\left(1+\gamma_{t-1}\right)^3 \eta^4 H^4\zeta_\star^4$$

$$+ 64\left(1+\gamma_{t-1}\right)^3 \eta^4\tau^4 \left((1-\eta\mu)^{4(t-1-\delta(t))}\left(B(\delta(t)) + \phi_\star^4\right)\right) + (1+\gamma_{t-1})^3 \frac{88\eta^6\tau^4\sigma^4}{\mu^2}$$

$$+ 128\left(1+\gamma_{t-1}\right)^3 \eta^4\tau^4 \left(\eta^2\sigma^2(t-1-\delta(t))(1-\eta\mu)^{2(t-1-\delta(t))}\left(A(\delta(t)) + \phi_\star^2\right)\right)$$

$$+ 20\eta^2\sigma^2 \left(1 + \frac{1}{\gamma_{t-1}}\right)(1-\eta\mu)^2 \left(4\eta^2\tau^2\left(t-1-\delta(t)\right)^2(1-\eta\mu)^{2(t-2-\delta(t))}\left(A(\delta(t)) + \phi_\star^2\right)\right)$$

$$+ 20\eta^2\sigma^2 \left(1 + \frac{1}{\gamma_{t-1}}\right)(1-\eta\mu)^2 \left(2(t-1-\delta(t))\left(\eta^2 KH^2\zeta_\star^2 + \frac{\eta^3\tau^2 K\sigma^2}{\mu} + \eta^2\sigma^2\left(1+\ln(K)\right)\right)\right)$$

$$+ 40\eta^4\sigma^2\tau^2(1+\gamma_{t-1})\left(2(1-\eta\mu)^{2(t-1-\delta(t))}\left(A(\delta(t))+\phi_\star^2\right)+\frac{\eta\sigma^2}{\mu}\right)$$

$$+ 40\eta^4\sigma^2(1+\gamma_{t-1})H^2\zeta_\star^2 + 16\eta^4\sigma^4 \ ,$$

$$\leq \prod_{j=\delta(t)}^{t-1}\left(1+\frac{1}{\gamma_j}\right)^3(1-\eta\mu)^4 \cancelto{0}{\mathbb{E}\left[\left\|x_{\delta(t)}-\bar{x}_{\delta(t)}\right\|_2^4\right]}$$

$$+ \left(8\eta^4 H^4\zeta_\star^4 + \frac{88\eta^6\tau^4\sigma^4}{\mu^2}\right)\sum_{j=\delta(t)}^{t-1}\left(\prod_{i=j+1}^{t-1}\left(1+\frac{1}{\gamma_i}\right)^3(1-\eta\mu)^4\right)(1+\gamma_j)^3$$

$$+ 64\eta^4\tau^4\left(B(\delta(t))+\phi_\star^4\right)$$

$$\times \sum_{j=\delta(t)}^{t-1}\left(\prod_{i=j+1}^{t-1}\left(1+\frac{1}{\gamma_i}\right)^3(1-\eta\mu)^4\right)(1+\gamma_j)^3(1-\eta\mu)^{4(j-\delta(t))}$$

$$+ 128\eta^6\tau^4\sigma^2\left(A(\delta(t))+\phi_\star^2\right)$$

$$\times \sum_{j=\delta(t)}^{t-1}\left(\prod_{i=j+1}^{t-1}\left(1+\frac{1}{\gamma_i}\right)^3(1-\eta\mu)^4\right)(1+\gamma_j)^3(j-\delta(t))(1-\eta\mu)^{2(j-\delta(t))}$$

$$+ 80\eta^4\sigma^2\tau^2\left(A(\delta(t))+\phi_\star^2\right)$$

$$\times \sum_{j=\delta(t)}^{t-1}\left(\prod_{i=j+1}^{t-1}\left(1+\frac{1}{\gamma_i}\right)^3(1-\eta\mu)^4\right)\left(1+\frac{1}{\gamma_j}\right)(j-\delta(t))(1-\eta\mu)^{2(j-\delta(t))}$$

$$+ 40\eta^2\sigma^2\left(\eta^2 KH^2\zeta_\star^2 + \frac{\eta^3\tau^2 K\sigma^2}{\mu} + \eta^2\sigma^2\left(1+\ln(K)\right)\right)$$

$$\times \sum_{j=\delta(t)}^{t-1}\left(\prod_{i=j+1}^{t-1}\left(1+\frac{1}{\gamma_i}\right)^3(1-\eta\mu)^4\right)(1-\eta\mu)^2\left(1+\frac{1}{\gamma_j}\right)(j-\delta(t))$$

$$+ 80\eta^4\sigma^2\tau^2\left(A(\delta(t))+\phi_\star^2\right)$$

$$\times \sum_{j=\delta(t)}^{t-1}\left(\prod_{i=j+1}^{t-1}\left(1+\frac{1}{\gamma_i}\right)^3(1-\eta\mu)^4\right)(1+\gamma_j)(1-\eta\mu)^{2(j-\delta(t))}$$

$$+ 40\eta^4\sigma^2\left(\frac{\eta\tau^2\sigma^2}{\mu}+H^2\zeta_\star^2\right)\sum_{j=\delta(t)}^{t-1}\left(\prod_{i=j+1}^{t-1}\left(1+\frac{1}{\gamma_i}\right)^3(1-\eta\mu)^4\right)(1+\gamma_j)$$

$$+ 16\eta^4\sigma^4\sum_{j=\delta(t)}^{t-1}\left(\prod_{i=j+1}^{t-1}\left(1+\frac{1}{\gamma_i}\right)^3(1-\eta\mu)^4\right) \ ,$$

$$= \left(8\eta^4 H^4\zeta_\star^4 + \frac{88\eta^6\tau^4\sigma^4}{\mu^2}\right)\sum_{j=\delta(t)}^{t-1}(t-\delta(t))^3(1-\eta\mu)^{4(t-1-j)}$$

$$+ 64\eta^4\tau^4\left(B(\delta(t))+\phi_\star^4\right)\sum_{j=\delta(t)}^{t-1}(t-\delta(t))^3(1-\eta\mu)^{4(t-1-\delta(t))}$$

$$+ 128\eta^6\tau^4\sigma^2\left(A(\delta(t))+\phi_\star^2\right)\sum_{j=\delta(t)}^{t-1}(t-\delta(t))^3(j-\delta(t))(1-\eta\mu)^{4(t-1)-2j-2\delta(t)}$$

$$+ 80\eta^4\sigma^2\tau^2\left(A(\delta(t))+\phi_\star^2\right)\sum_{j=\delta(t)}^{t-1}\frac{(t-\delta(t))^3}{(j+1-\delta(t))^2}(1-\eta\mu)^{4(t-1)-2j-2\delta(t)}$$

$$+ 40\eta^2\sigma^2 \left( \eta^2 K H^2 \zeta_\star^2 + \frac{\eta^3 \tau^2 K \sigma^2}{\mu} + \eta^2\sigma^2 \left(1 + \ln(K)\right) \right) \sum_{j=\delta(t)}^{t-1} \frac{(t-\delta(t))^3}{(j+1-\delta(t))^2} (1-\eta\mu)^{4(t-j)-2}$$

$$+ 80\eta^4\sigma^2\tau^2 \left( A(\delta(t)) + \phi_\star^2 \right) \sum_{j=\delta(t)}^{t-1} \frac{(t-\delta(t))^3}{(j+1-\delta(t))^2} (1-\eta\mu)^{4(t-1)-2j-2\delta(t)}$$

$$+ 40\eta^4\sigma^2 \left( \frac{\eta\tau^2\sigma^2}{\mu} + H^2\zeta_\star^2 \right) \sum_{j=\delta(t)}^{t-1} \frac{(t-\delta(t))^3}{(j+1-\delta(t))^2} (1-\eta\mu)^{4(t-1-j)}$$

$$+ 16\eta^4\sigma^4 \sum_{j=\delta(t)}^{t-1} \frac{(t-\delta(t))^3}{(j+1-\delta(t))^3} (1-\eta\mu)^{4(t-1-j)} \ ,$$

$$\leq^{(a)} \left( \frac{8\eta^3 H^4 \zeta_\star^4}{\mu} + \frac{88\eta^5\tau^4\sigma^4}{\mu^3} \right) (t-\delta(t))^3$$

$$+ 64\eta^4\tau^4 \left( B(\delta(t)) + \phi_\star^4 \right) (t-\delta(t))^4 (1-\eta\mu)^{4(t-1-\delta(t))}$$

$$+ \frac{128\eta^5\tau^4\sigma^2}{\mu} \left( A(\delta(t)) + \phi_\star^2 \right) (t-\delta(t))^4 (1-\eta\mu)^{2(t-1-\delta(t))}$$

$$+ {\color{teal}160\eta^4\sigma^2\tau^2 \left( A(\delta(t)) + \phi_\star^2 \right) (t-\delta(t))^3 (1-\eta\mu)^{2(t-1-\delta(t))}}$$

$$+ 80\eta^2\sigma^2 \left( {\color{orange}\eta^2 H^2 K \zeta_\star^2} + \frac{{\color{orange}\eta^3\tau^2 K\sigma^2}}{\mu} + {\color{orange}\eta^2\sigma^2 \left(1 + \ln(K)\right)} \right) (t-\delta(t))^3$$

$$+ {\color{teal}160\eta^4\sigma^2\tau^2 \left( A(\delta(t)) + \phi_\star^2 \right) (t-\delta(t))^3 (1-\eta\mu)^{2(t-1-\delta(t))}}$$

$$+ 80\eta^4\sigma^2 \left( \frac{\eta\tau^2\sigma^2}{\mu} + {\color{red}H^2\zeta_\star^2} \right) (t-\delta(t))^3 + {\color{orange}32\eta^4\sigma^4 (t-\delta(t))^3} \ ,$$

$$\leq^{(b)} \left( \frac{8\eta^3 H^4 \zeta_\star^4}{\mu} + \frac{88\eta^5\tau^4\sigma^4}{\mu^3} + {\color{red}160\eta^4 K\sigma^2 H^2 \zeta_\star^2} + \frac{{\color{orange}160\eta^5\tau^2 K\sigma^4}}{\mu} + {\color{orange}112\eta^4\sigma^4 \left(1 + \ln(K)\right)} \right) (t-\delta(t))^3$$

$$+ 64\eta^4\tau^4 (t-\delta(t))^4 (1-\eta\mu)^{4(t-1-\delta(t))} \left( B(\delta(t)) + \phi_\star^4 \right)$$

$$+ \left( \frac{128\eta^5\tau^4\sigma^2}{\mu} (t-\delta(t)) + {\color{teal}320\eta^4\sigma^2\tau^2} \right) (t-\delta(t))^3 (1-\eta\mu)^{2(t-1-\delta(t))} \left( A(\delta(t)) + \phi_\star^2 \right) \ ,$$

where in (a) we used that $\sum_{j=\delta(t)}^{t-1} \frac{1}{(j+1-\delta(t))^3} < \sum_{j=\delta(t)}^{t-1} \frac{1}{(j+1-\delta(t))^2} \leq \frac{\pi^2}{6} < 2$; in (b) we used that $\eta < 1/H \leq 1/\mu$ to get the {\color{red}red} and {\color{teal}blue} terms. This finishes the proof of the lemma, once we note that when $t = \delta(t)$, the upper bound is zero, which means we can extend the proof to $t \geq \delta(t)$, which essentially means all $t$.

We can now use this bound to give the following bound for $r \in [R]$,

$$\sum_{j=K(r-1)}^{Kr-1} (1-\eta\mu)^{Kr-1-j} D(j)$$

$$\leq \frac{128\eta^5\tau^4\sigma^2}{\mu} \left( A(K(r-1)) + \phi_\star^2 \right) \sum_{j=K(r-1)}^{Kr-1} (1-\eta\mu)^{Kr-1-j} (j-K(r-1))^4 (1-\eta\mu)^{2(j-1-K(r-1))}$$

$$+ 320\eta^4\sigma^2\tau^2 \left( A(K(r-1)) + \phi_\star^2 \right) \sum_{j=K(r-1)}^{Kr-1} (1-\eta\mu)^{Kr-1-j} (j-K(r-1))^3 (1-\eta\mu)^{2(j-1-K(r-1))}$$

$$+ 64\eta^4\tau^4 \left( B(K(r-1)) + \phi_\star^4 \right) \sum_{j=K(r-1)}^{Kr-1} (1-\eta\mu)^{Kr-1-j} (j-K(r-1))^4 (1-\eta\mu)^{4(j-1-K(r-1))}$$

$$\left( \frac{8\eta^3 H^4 \zeta_\star^4}{\mu} + \frac{88\eta^5\tau^4\sigma^4}{\mu^3} + 160\eta^4 K\sigma^2 H^2 \zeta_\star^2 + \frac{160\eta^5\tau^2 K\sigma^4}{\mu} + 112\eta^4\sigma^4 \left(1 + \ln(K)\right) \right)$$

$$\times \sum_{j=K(r-1)}^{Kr-1} (1-\eta\mu)^{Kr-1-j}(j-K(r-1))^3 \ ,$$

$$\leq \frac{128\eta^5 K^4\tau^4\sigma^2}{\mu}(1-\eta\mu)^{K-4}\left(A(K(r-1))+\phi_\star^2\right)\sum_{j=K(r-1)}^{Kr-1}(1-\eta\mu)^{j-K(r-1)}$$

$$+320\eta^4 K^3\sigma^2\tau^2(1-\eta\mu)^{K-4}\left(A(K(r-1))+\phi_\star^2\right)\sum_{j=K(r-1)}^{Kr-1}(1-\eta\mu)^{j-K(r-1)}$$

$$+64\eta^4 K^4\tau^4(1-\eta\mu)^{K-5}\left(B(K(r-1))+\phi_\star^4\right)\sum_{j=K(r-1)}^{Kr-1}(1-\eta\mu)^{3(j-K(r-1))}$$

$$\left(\frac{8\eta^3 K^3 H^4\zeta_\star^4}{\mu}+\frac{88\eta^5 K^3\tau^4\sigma^4}{\mu^3}+160\eta^4 K^4\sigma^2 H^2\zeta_\star^2+\frac{160\eta^5\tau^2 K^4\sigma^4}{\mu}+112\eta^4 K^3\sigma^4\left(1+\ln(K)\right)\right)$$

$$\times \sum_{j=K(r-1)}^{Kr-1}(1-\eta\mu)^{Kr-1-j} \ ,$$

$$\leq \left(1-(1-\eta\mu)^K\right)\frac{128\eta^4 K^4\tau^4\sigma^2}{\mu^2}(1-\eta\mu)^{K-4}\left(A(K(r-1))+\phi_\star^2\right)$$

$$+\left(1-(1-\eta\mu)^K\right)\frac{320\eta^3 K^3\sigma^2\tau^2}{\mu}(1-\eta\mu)^{K-4}\left(A(K(r-1))+\phi_\star^2\right)$$

$$+\left(1-(1-\eta\mu)^K\right)\frac{64\eta^3 K^4\tau^4}{\mu}(1-\eta\mu)^{K-5}\left(B(K(r-1))+\phi_\star^4\right)$$

$$\left(1-(1-\eta\mu)^K\right)$$

$$\times \left(\frac{8\eta^2 K^3 H^4\zeta_\star^4}{\mu^2}+\frac{88\eta^4 K^3\tau^4\sigma^4}{\mu^4}+\frac{160\eta^3 K^4\sigma^2 H^2\zeta_\star^2}{\mu}+\frac{160\eta^4\tau^2 K^4\sigma^4}{\mu^2}+\frac{112\eta^3 K^3\sigma^4\left(1+\ln(K)\right)}{\mu}\right) \ ,$$

which proves the claim. $\qquad\square$

## H.3 Should Consensus Error Explode for a Large Step-size?

Note that the results in Lemmas 30 and 31 suggest that when $K\to\infty$ we must pick $\eta=\mathcal{O}\left(\frac{1}{K}\right)$ so that the consensus error does not explode. This small step-size was criticized by Wang et al. [8] through experiments showing that even without such a small step-size, consensus error did not blow up in the regime of large $K$. In the following lemma we show that even with $\eta=\theta\left(\frac{1}{H}\right)$, consensus error does not blow up, and actually saturates to a value that depends on the data heterogeneity Assumptions 4 to 6. The lemma relies on just the evolution of iterates on a single machine, and the fact that it is decoupled between communication rounds.

**Lemma 32** (Alternative Bounds on the Consensus Error ). *We have the following for any $t\geq\delta(t)$ for $\eta<1/H$ when optimizing a problem that satisfies Assumptions 1, 2, 4 and 5,*

$$C(t)\leq 12(1-\eta\mu)^{2(t-\delta(t))}\left(A(\delta(t))+\phi_\star^2\right)+\frac{6\eta\sigma^2}{\mu}+3\zeta_\star^2 \ ,$$

$$D(t)\leq 432(1-\eta\mu)^{3(t-\delta(t))}\left(B(\delta(t))+\phi_\star^4\right)+\frac{864\eta\sigma^4}{\mu^3}+27\zeta_\star^4 \ .$$

*In particular, when $t-\delta(t)\to\infty$ the upper bounds converge to $\frac{6\eta\sigma^2}{\mu}+3\zeta_\star^2$ and $\frac{864\eta\sigma^4}{\mu^3}+27\zeta_\star^4$ respectively.*

*Proof.* We note that for any and $m,n\in[M]$

$$\mathbb{E}\left[\|x_t^m-x_t^n\|_2^2\right]$$

$$=\mathbb{E}\left[\|x_t^m-x_m^\star-x_t^n+x_n^\star+x_m^\star-x_n^\star\|_2^2\right] \ ,$$

$$\overset{\text{(Lemma 17 and Assumption 4)}}{\leq} 3\mathbb{E}\left[\|x_t^m - x_m^\star\|_2^2\right] + 3\mathbb{E}\left[\|x_t^n - x_n^\star\|_2^2\right] + 3\zeta_\star^2 \ ,$$

$$\overset{\text{(Lemma 21)}}{\leq} 3\left((1-\eta\mu)^{2(t-\delta(t))}\mathbb{E}\left[\|x_{\delta(t)} - x_m^\star\|_2^2\right] + \frac{\eta\sigma^2}{\mu}\right)$$

$$+ 3\left((1-\eta\mu)^{2(t-\delta(t))}\mathbb{E}\left[\|x_{\delta(t)} - x_n^\star\|_2^4\right] + \frac{\eta\sigma^2}{\mu}\right) + 3\zeta_\star^2 \ .$$

Averaging this over $m, n \in [M]$,

$$C(t) \leq 6(1-\eta\mu)^{2(t-\delta(t))}\frac{1}{M}\sum_{m\in[M]}\mathbb{E}\left[\|x_{\delta(t)} - x_m^\star\|_2^2\right] + \frac{6\eta\sigma^2}{\mu} + 3\zeta_\star^2 \ ,$$

$$\leq 12(1-\eta\mu)^{2(t-\delta(t))}\left(\mathbb{E}\left[\|x_{\delta(t)} - x^\star\|_2^2\right] + \phi_\star^2\right) + \frac{6\eta\sigma^2}{\mu} + 3\zeta_\star^2 \ ,$$

$$= 12(1-\eta\mu)^{2(t-\delta(t))}\left(A(\delta(t)) + \phi_\star^2\right) + \frac{6\eta\sigma^2}{\mu} + 3\zeta_\star^2 \ ,$$

which proves the first statement. For the second result we similarly note that for any and $m, n \in [M]$ and $t \in [0, T]$,

$$\mathbb{E}\left[\|x_t^m - x_t^n\|_2^4\right]$$

$$= \mathbb{E}\left[\|x_t^m - x_m^\star - x_t^n + x_n^\star + x_m^\star - x_n^\star\|_2^4\right] \ ,$$

$$\overset{\text{(Lemma 17 and Assumption 4)}}{\leq} 27\mathbb{E}\left[\|x_t^m - x_m^\star\|_2^4\right] + 27\mathbb{E}\left[\|x_t^n - x_n^\star\|_2^4\right] + 27\zeta_\star^4 \ ,$$

$$\overset{\text{(Lemma 23)}}{\leq} 27\left((1-\eta\mu)^{3(t-\delta(t))}\mathbb{E}\left[\|x_{\delta(t)} - x_m^\star\|_2^4\right] + \frac{16\eta\sigma^4}{\mu^3}\right)$$

$$+ 27\left((1-\eta\mu)^{3(t-\delta(t))}\mathbb{E}\left[\|x_{\delta(t)} - x_n^\star\|_2^4\right] + \frac{16\eta\sigma^4}{\mu^3}\right) + 27\zeta_\star^4 \ .$$

Averaging this over $m, n \in [M]$,

$$D(t) \leq 54(1-\eta\mu)^{3(t-\delta(t))}\frac{1}{M}\sum_{m\in[M]}\mathbb{E}\left[\|x_{\delta(t)} - x_m^\star\|_2^4\right] + 27\zeta_\star^4 + \frac{864\eta\sigma^4}{\mu^3} \ ,$$

$$\overset{\text{(Lemma 16 and Assumption 5)}}{\leq} 432(1-\eta\mu)^{3(t-\delta(t))}\left(\mathbb{E}\left[\|x_{\delta(t)} - x^\star\|_2^4\right] + \phi_\star^4\right)$$

$$+ 27\zeta_\star^4 + \frac{864\eta\sigma^4}{\mu^3} \ ,$$

$$= 432(1-\eta\mu)^{3(t-\delta(t))}\left(A(\delta(t)) + \phi_\star^4\right) + 27\zeta_\star^4 + \frac{864\eta\sigma^4}{\mu^3} \ ,$$

which proves the second statement of the lemma. $\qquad\square$

The reason we do not use the above lemma over Lemmas 30 and 31, is that our step-size tuning in Appendix I dictates that we anyways need to use $\eta = \mathcal{O}\left(\frac{1}{\mu KR}\right)$ to get our convergence guarantees which puts the issue of an exploding consensus error to rest. Having said that the above lemma offers reconciliation with the observations by Wang et al. [8] in the regime when $\eta = \theta\left(\frac{1}{H}\right)$.

# I  Putting it All Together

## I.1  Convergence in Iterates without Third-order Smoothness

This subsection will essentially combine the weaker blue upper bound from Lemma 20 with the consensus error upper bound from Lemma 30. This would lead to an inequality that we can unroll across communication rounds.

**Lemma 33.** *Under Assumptions 1 to 6 using $\eta < 1/H$ and such that*

$$\rho_1 = (1 - \eta\mu)^K + \left(1 - (1 - \eta\mu)^K\right) \frac{4\eta^2 H^2 \tau^2}{\mu^2} K^2 (1 - \eta\mu)^{K-2} < 1$$

*we can get the following convergence guarantee with initialization $x_0 = 0$,*

$$A(KR) \le \rho_1^R B^2 + \frac{1 - (1 - \eta\mu)^K}{1 - \rho_1} \cdot \frac{\eta\sigma^2}{\mu M} + \frac{1 - (1 - \eta\mu)^K}{1 - \rho_1} \cdot \frac{4\eta^2 \tau^2 H^2 K^2 (1 - \eta\mu)^{K-2} \phi_\star^2}{\mu^2}$$

$$+ \frac{1 - (1 - \eta\mu)^K}{1 - \rho_1} \left(\frac{2\eta^2 H^4 K^2 \zeta_\star^2}{\mu^2} + \frac{2\eta^3 H^2 \tau^2 K^2 \sigma^2}{\mu^3} + \frac{2\eta^2 H^2 \sigma^2 K \left(1 + \ln(K)\right)}{\mu^2}\right) \quad .$$

*Proof.* First recall the round-wise recursion from Lemma 20 for $r = R$,

$$A(KR) \le (1 - \eta\mu)^K A(K(R-1)) + \frac{\eta H^2}{\mu} \sum_{j=K(R-1)}^{KR-1} (1 - \eta\mu)^{KR-1-j} C(j)$$

$$+ \left(1 - (1 - \eta\mu)^K\right) \frac{\eta\sigma^2}{\mu M} \quad ,$$

$$\le^{\text{(Lemma 30)}} (1 - \eta\mu)^K A(K(R-1)) + \left(1 - (1 - \eta\mu)^K\right) \frac{\eta\sigma^2}{\mu M}$$

$$\frac{1 - (1 - \eta\mu)^K}{\mu^2} \left(2\eta^2 H^4 K^2 \zeta_\star^2 + \frac{2\eta^3 H^2 \tau^2 K^2 \sigma^2}{\mu} + 2\eta^2 H^2 \sigma^2 K \left(1 + \ln(K)\right)\right)$$

$$+ \frac{1 - (1 - \eta\mu)^K}{\mu^2} 4\eta^2 \tau^2 H^2 K^2 (1 - \eta\mu)^{K-2} \left(A(K(r-1)) + \phi_\star^2\right) \quad ,$$

$$= \left((1 - \eta\mu)^K + \left(1 - (1 - \eta\mu)^K\right) \frac{4\eta^2 H^2 \tau^2}{\mu^2} K^2 (1 - \eta\mu)^{K-2}\right) A(K(R-1))$$

$$+ \left(1 - (1 - \eta\mu)^K\right) \frac{\eta\sigma^2}{\mu M} + \frac{1 - (1 - \eta\mu)^K}{\mu^2} 4\eta^2 \tau^2 H^2 K^2 (1 - \eta\mu)^{K-2} \phi_\star^2$$

$$+ \frac{1 - (1 - \eta\mu)^K}{\mu^2} \left(2\eta^2 H^4 K^2 \zeta_\star^2 + \frac{2\eta^3 H^2 \tau^2 K^2 \sigma^2}{\mu} + 2\eta^2 H^2 \sigma^2 K \left(1 + \ln(K)\right)\right) \quad ,$$

$$\le \rho_1^R B^2 + \frac{1 - (1 - \eta\mu)^K}{1 - \rho_1} \cdot \frac{\eta\sigma^2}{\mu M} + \frac{1 - (1 - \eta\mu)^K}{1 - \rho_1} \cdot \frac{4\eta^2 \tau^2 H^2 K^2 (1 - \eta\mu)^{K-2} \phi_\star^2}{\mu^2}$$

$$+ \frac{1 - (1 - \eta\mu)^K}{1 - \rho_1} \left(\frac{2\eta^2 H^4 K^2 \zeta_\star^2}{\mu^2} + \frac{2\eta^3 H^2 \tau^2 K^2 \sigma^2}{\mu^3} + \frac{2\eta^2 H^2 \sigma^2 K \left(1 + \ln(K)\right)}{\mu^2}\right) \quad ,$$

where we defined $\rho_1 = (1 - \eta\mu)^K + \left(1 - (1 - \eta\mu)^K\right) \frac{4\eta^2 H^2 \tau^2}{\mu^2} K^2 (1 - \eta\mu)^{K-2}$. This proves the lemma. $\square$

We can tune the step-size in the above guarantee, using standard techniques while making sure that $\tau$ is small enough and $K$ is large enough. This gives the following result,

**Lemma 34** (Strongly Convex Functions Iterate Convergence with $\tau, \zeta_\star, \phi_\star$)**.** *Assuming*

$$R \ge \max\left\{\frac{3H\tau}{\mu^2} \ln\left(\frac{B^2}{\epsilon}\right), \frac{2H\tau}{\mu^2} \ln^{3/2}\left(\frac{B^2}{\epsilon}\right)\right\}$$

*we can get the following convergence guarantee for local SGD, initializing at $x_0 = 0$ and optimizing functions satisfying Assumptions 1 to 6,*

$$A(KR) = \tilde{\mathcal{O}}\left(e^{-\frac{\mu KR}{2H}} B^2 + \frac{\sigma^2}{\mu^2 MKR} + \frac{\tau^2 H^2 \phi_\star^2}{\mu^4 R^2} + \frac{H^4 \zeta_\star^2}{\mu^4 R^2} + \frac{H^2 \tau^2 \sigma^2}{\mu^6 KR^3} + \frac{H^2 \sigma^2 \left(1 + \ln(K)\right)}{\mu^4 KR^2}\right) \quad ,$$

*where we pick the step-size,*

$$\eta = \min\left\{\frac{1}{2H}, \frac{1}{\mu KR} \ln\left(\frac{B^2}{\epsilon}\right)\right\} \quad ,$$

*for the choice of $\epsilon$,*

$$\epsilon := \max \left\{ \frac{2\sigma^2}{\mu^2 MKR}, \frac{8\tau^2 H^2 \phi_\star^2}{\mu^4 R^2}, \frac{4H^4 \zeta_\star^2}{\mu^4 R^2}, \frac{4H^2 \tau^2 \sigma^2}{\mu^6 KR^3}, \frac{4H^2 \sigma^2 (1 + \ln(K))}{\mu^4 KR^2}, \epsilon_{target} \right\} \ ,$$

*where $\epsilon_{target}$ is a target, which is greater than or equal to the machine precision.*

*Proof.* We will pick our step-size as follows, where we will specify some $\epsilon$ bigger than machine-precision in the end:

$$\eta = \min \left\{ \frac{1}{2H}, \frac{1}{\mu KR} \ln \left( \frac{B^2}{\epsilon} \right) \right\} \ .$$

We will now derive conditions that are enough to bound $\frac{1-(1-\eta\mu)^K}{1-\rho_1}$ by 2. Note the following,

$$\frac{1 - (1 - \eta\mu)^K}{1 - \rho_1} \leq 2 \ ,$$

$$\Leftrightarrow \rho_1 \leq \frac{1 + (1 - \eta\mu)^K}{2} \ ,$$

$$\Leftrightarrow \left( 1 - (1 - \eta\mu)^K \right) \frac{4\eta^2 H^2 \tau^2}{\mu^2} K^2 (1 - \eta\mu)^{K-2} \leq \frac{1 - (1 - \eta\mu)^K}{2} \ ,$$

$$\Leftrightarrow \frac{4\eta^2 H^2 \tau^2}{\mu^2} K^2 (1 - \eta\mu)^{K-2} \leq \frac{1}{2} \ ,$$

$$\Leftarrow \frac{4H^2 \tau^2}{\mu^4 R^2} \ln^2 \left( \frac{B^2}{\epsilon} \right) \leq \frac{1}{2} \ ,$$

$$\Leftarrow R \geq \frac{3H\tau}{\mu^2} \ln \left( \frac{B^2}{\epsilon} \right) \ ,$$

Hence it is sufficient to assume that $R \geq \frac{3H\tau}{\mu^2} \ln \left( \frac{B^2}{\epsilon} \right)$. This allows us to simplify the convergence rate from the previous lemma as follows,

$$A(KR) \leq \rho_1^R B^2 + \frac{2\eta\sigma^2}{\mu M} + \frac{8\eta^2 \tau^2 H^2 K^2 (1 - \eta\mu)^{K-2} \phi_\star^2}{\mu^2}$$

$$+ \frac{4\eta^2 H^4 K^2 \zeta_\star^2}{\mu^2} + \frac{4\eta^3 H^2 \tau^2 K^2 \sigma^2}{\mu^3} + \frac{4\eta^2 H^2 \sigma^2 K (1 + \ln(K))}{\mu^2} \ ,$$

$$\leq \rho_1^R B^2 + \frac{2\eta\sigma^2}{\mu M} + \frac{8\eta^2 \tau^2 H^2 K^2 \phi_\star^2}{\mu^2} + \frac{4\eta^2 H^4 K^2 \zeta_\star^2}{\mu^2} + \frac{4\eta^3 H^2 \tau^2 K^2 \sigma^2}{\mu^3}$$

$$+ \frac{4\eta^2 H^2 \sigma^2 K (1 + \ln(K))}{\mu^2} \ .$$

Now, let us upper bound the exponential term more carefully. Recall that due to the choice of our step-size, as we used this before,

$$\rho_1 = (1 - \eta\mu)^K + \left( 1 - (1 - \eta\mu)^K \right) \frac{4\eta^2 H^2 \tau^2}{\mu^2} K^2 (1 - \eta\mu)^{K-2} \ ,$$

$$\leq^{(a)} (1 - \eta\mu)^K + \eta\mu K \frac{4\eta^2 H^2 \tau^2}{\mu^2} K^2 (1 - \eta\mu)^{K-2} \ ,$$

$$\leq (1 - \eta\mu)^K + \frac{4H^2 \tau^2}{\mu^4 R^3} \ln^3 \left( \frac{B^2}{\epsilon} \right) (1 - \eta\mu)^{K-2} \ ,$$

$$\leq (1 - \eta\mu)^{K-2} + \frac{4H^2 \tau^2}{\mu^4 R^3} \ln^3 \left( \frac{B^2}{\epsilon} \right) (1 - \eta\mu)^{K-2} \ ,$$

$$\leq \left( 1 + \frac{4H^2 \tau^2}{\mu^4 R^3} \ln^3 \left( \frac{B^2}{\epsilon} \right) \right) (1 - \eta\mu)^{K-2} \ ,$$

$$\leq e^{-\eta\mu(K-2) + \frac{4H^2 \tau^2}{\mu^4 R^3} \ln^3 \left( \frac{B^2}{\epsilon} \right)} \ .$$

where in (a) we use Bernoulli's inequality, and the choice of the step-size which implies that $\eta\mu < 1$. Assuming $K \geq 4$, and raising the power of both sides to $R$ gives us,

$$\rho_1^R \leq e^{-\frac{\eta\mu KR}{2} + \frac{4H^2\tau^2}{\mu^4 R^2}\ln^3\left(\frac{B^2}{\epsilon}\right)},$$

$$\leq^{(a)} e^{-\frac{\eta\mu KR}{2} + 1},$$

where in (a) we assumed that $R \geq \frac{2H\tau}{\mu^2}\ln^{3/2}\left(\frac{B^2}{\epsilon}\right)$. Finally, we will pick the $\epsilon$ as follows,

$$\epsilon := \max\left\{\frac{2\sigma^2}{\mu^2 MKR}, \frac{8\tau^2 H^2\phi_\star^2}{\mu^4 R^2}, \frac{4H^4\zeta_\star^2}{\mu^4 R^2}, \frac{4H^2\tau^2\sigma^2}{\mu^6 KR^3}, \frac{4H^2\sigma^2\left(1 + \ln(K)\right)}{\mu^4 KR^2}, \epsilon_{target}\right\},$$

where $\epsilon_{target}$ is a target, which is greater than or equal to the machine precision (say, floating point precision), thus implying that $\ln\left(\frac{B^2}{\epsilon}\right)$ is a numerical constant. We note two things that justify that choice of this step-size,

- The largest $\epsilon$ will lead to the step size we end up using, an in particular determine govern which term dominates the convergence rate. For instance, let us assume that $\epsilon = \frac{2\sigma^2}{\mu^2 MKR}$. Furthermore, let $\frac{1}{2H} \geq \frac{1}{\mu KR}\ln\left(\frac{B^2}{\epsilon}\right)$ which implies that $e^{-\frac{\mu KR}{2H}} \leq \frac{2\sigma^2}{\mu^2 MKR}$. With $\eta = \frac{1}{\mu KR}\ln\left(\frac{B^2}{\epsilon}\right)$, this makes the convergence rate,

$$A(KR) \leq \frac{2\sigma^2}{\mu^2 MKR} + \frac{2\sigma^2}{\mu^2 MKR}\ln\left(\frac{B^2}{\frac{2\sigma^2}{\mu^2 MKR}}\right) = \tilde{\mathcal{O}}\left(e^{-\frac{\mu KR}{2H}} + \frac{\sigma^2}{\mu^2 MKR}\right).$$

- On the other hand if $\frac{1}{2H} \leq \frac{1}{\mu KR}\ln\left(\frac{B^2}{\epsilon}\right)$, then it implies that, $e^{-\frac{\mu KR}{2H}} \geq \frac{2\sigma^2}{\mu^2 MKR}$, which makes the convergence rate,

$$A(KR) \leq e^{-\frac{\mu KR}{2H}} + \frac{\sigma^2}{\mu HM} = \tilde{\mathcal{O}}\left(e^{-\frac{\mu KR}{2H}} + \frac{\sigma^2}{\mu^2 MKR}\right).$$

Using the above logic for all possible choices of $\epsilon$ (ant thus $\eta$) allows us to give the following convergence rate,

$$A(KR) = \tilde{\mathcal{O}}\left(e^{-\frac{\mu KR}{2H}}B^2 + \frac{\sigma^2}{\mu^2 MKR} + \frac{\tau^2 H^2\phi_\star^2}{\mu^4 R^2} + \frac{H^4\zeta_\star^2}{\mu^4 R^2} + \frac{H^2\tau^2\sigma^2}{\mu^6 KR^3} + \frac{H^2\sigma^2\left(1 + \ln(K)\right)}{\mu^4 KR^2}\right).$$

$\square$

Furthermore, assuming the functions are quadratic we can replace some of the smoothness constants with $\tau$, by relying on the better red upper bound of Lemma 20, as with $Q = 0$ we do not need to bound the fourth moment of consensus error. The proof more or less follows the above lemma, and results in the following rate for quadratics.

**Lemma 35.** *Under Assumptions 1 to 6 and 8 using $\eta < 1/H$ and such that $\rho_2 = (1 - \eta\mu)^K + \left(1 - (1 - \eta\mu)^K\right)\frac{4\eta^2\tau^4}{\mu^2}K^2(1 - \eta\mu)^{K-2} < 1$ we can get the following convergence guarantee with initialization $x_0 = 0$,*

$$A(KR) \leq \rho_2^R B^2 + \frac{1 - (1 - \eta\mu)^K}{1 - \rho_2} \cdot \frac{\eta\sigma^2}{\mu M} + \frac{1 - (1 - \eta\mu)^K}{1 - \rho_2} \cdot \frac{4\eta^2\tau^4 K^2(1 - \eta\mu)^{K-2}\phi_\star^2}{\mu^2}$$

$$+ \frac{1 - (1 - \eta\mu)^K}{1 - \rho_2}\left(\frac{2\eta^2 H^2\tau^2 K^2\zeta_\star^2}{\mu^2} + \frac{2\eta^3\tau^4 K^2\sigma^2}{\mu^3} + \frac{2\eta^2\tau^2\sigma^2 K\left(1 + \ln(K)\right)}{\mu^2}\right).$$

One notable thing is that for quadratics, when $\tau = 0$, we can get the fast convergence guarantee for dense mini-batch SGD, i.e., with $KR$ communication rounds. We do not get this for non-quadratics, which highlights the need to understand the effect of third-order smoothness. This is not surprising because third-order smoothness is known to play a vital role in the convergence of local SGD even in a homogeneous setting. Just like the strongly convex case we can tune the step-size to get the following convergence rate for quadratics,

**Lemma 36** (Quadratics Iterate Convergence with $\tau, \zeta_\star, \phi_\star$). *Assuming* $R \geq \max\left\{\frac{3\tau^2}{\mu^2}\ln\left(\frac{B^2}{\epsilon}\right), \frac{2\tau^2}{\mu^2}\ln^{3/2}\left(\frac{B^2}{\epsilon}\right)\right\}$ *we can get the following convergence guarantee for local SGD, initializing at $x_0 = 0$ and optimizing functions satisfying Assumptions 1 to 6 and 8,*

$$A(KR) = \tilde{O}\left(e^{-\frac{\mu KR}{2H}}B^2 + \frac{\sigma^2}{\mu^2 MKR} + \frac{\tau^4\phi_\star^2}{\mu^4 R^2} + \frac{\tau^2 H^2\zeta_\star^2}{\mu^4 R^2} + \frac{\tau^4\sigma^2}{\mu^6 KR^3} + \frac{\tau^2\sigma^2\left(1 + \ln(K)\right)}{\mu^4 KR^2}\right) ,$$

*where we pick the step-size,*

$$\eta = \min\left\{\frac{1}{2H}, \frac{1}{\mu KR}\ln\left(\frac{B^2}{\epsilon}\right)\right\} ,$$

*for the choice of $\epsilon$,*

$$\epsilon := \max\left\{\frac{2\sigma^2}{\mu^2 MKR}, \frac{8\tau^4\phi_\star^2}{\mu^4 R^2}, \frac{4\tau^2 H^2\zeta_\star^2}{\mu^4 R^2}, \frac{4\tau^4\sigma^2}{\mu^6 KR^3}, \frac{4\tau^2\sigma^2\left(1 + \ln(K)\right)}{\mu^4 KR^2}, \epsilon_{target}\right\} ,$$

*where $\epsilon_{target}$ is a target, which is greater than or equal to the machine precision.*

It can be noted in the above convergence rate than when $\tau = 0$ we recover the fast convergence rate of dense mini-batch SGD.

## I.2   Convergence in Function Value without Third-order Smoothness

**Lemma 37** (Strongly Convex Function Convergence with $\tau$, $\zeta_\star$, $\phi_\star$). *Assuming*

$$R \geq \frac{4\tau\sqrt{\kappa}}{\mu}\max\left\{\ln\left(\frac{\mu B^2}{\epsilon}\right), \sqrt{2\ln^3\left(\frac{\mu B^2}{\epsilon}\right)}\right\} ,$$

$KR \geq 4\kappa$ *we can get the following convergence guarantee for local SGD, initializing at $x_0 = 0$ and optimizing functions satisfying Assumptions 1 to 6,*

$$\mathbb{E}\left[F(\hat{x})\right] - F(x^\star) = \tilde{O}\left(e^{-\frac{\mu KR}{2H}}\mu B^2 + \frac{H^3\zeta_\star^2}{\mu^2 R^2} + \frac{H\tau^2\sigma^2}{\mu^4 KR^3} + \frac{H\sigma^2\left(1 + \ln(K)\right)}{\mu^2 KR^2} + \frac{H\tau^2\phi_\star^2}{\mu^2 R^2} + \frac{\sigma^2}{\mu MKR}\right) ,$$

*where we define $\hat{x} = \sum_{t=0}^{T-1} w_t x_t$ for the choice of weights*

$$w_t := \frac{\rho_4^{R-1-\delta(t)/K}(1 - \eta\mu)^{\delta(t)+K-1-t}}{W}$$

*for $W = \frac{1-\rho_4^R}{1-\rho_4} \cdot \frac{1-(1-\eta\mu)^K}{\eta\mu}$ and $\rho_4 = (1 - \eta\mu)^K + \left(1 - (1 - \eta\mu)^K\right)\frac{8\eta^2 H\tau^2 K^2}{\mu}(1 - \eta\mu)^{K-2}$. And we pick the step-size as,*

$$\eta = \min\left\{\frac{1}{2H}, \frac{1}{\mu KR}\ln\left(\frac{\mu B^2}{\epsilon}\right)\right\} ,$$

*for the choice of $\epsilon$,*

$$\epsilon = \min\left\{\max\left\{\frac{4H^3\zeta_\star^2}{\mu^2 R^2}, \frac{4H\tau^2\sigma^2}{\mu^4 KR^3}, \frac{4H\sigma^2\left(1 + \ln(K)\right)}{\mu^2 KR^2}, \frac{8H\tau^2\phi_\star^2}{\mu^2 R^2}, \frac{3\sigma^2}{\mu MKR}, \epsilon_{target}\right\}, \frac{\mu B^2}{6}\right\} ,$$

*where $\epsilon_{target}$ is a target, which is greater than or equal to the machine precision.*

*Proof.* The main task in this subsection is to combine Lemmas 25 and 30. Recall Lemma 25 implies for all $t \in [0, T-1]$,

$$A(t+1) \leq (1 - \eta\mu)A(t) - \eta E(t) + 2\eta HC(t) + \frac{3\eta^2\sigma^2}{M} . \tag{$\star$}$$

Also recall the upper bound on the consensus error from Lemma 30 for all $t \in [0, T]$,

$$C(t) \leq 2\eta^2 H^2 K^2\zeta_\star^2 + \frac{2\eta^3\tau^2 K^2\sigma^2}{\mu} + 2\eta^2\sigma^2 K\left(1 + \ln(K)\right)$$

$$+ 4\eta^2\tau^2 \left(t - \delta(t)\right)^2 (1 - \eta\mu)^{2(t-1-\delta(t))} \left(A(\delta(t)) + \phi_\star^2\right) \ .$$

Plugging this upper bound into $(\star)$ gives us,

$$A(t+1) \le (1 - \eta\mu) A(t) - \eta E(t) + 4\eta^3 H^3 K^2 \zeta_\star^2 + \frac{4\eta^4 H\tau^2 K^2 \sigma^2}{\mu} + 4\eta^3 H\sigma^2 K \left(1 + \ln(K)\right)$$

$$+ 8\eta^3 H\tau^2 K^2 (1 - \eta\mu)^{2(t-1-\delta(t))} \left(A(\delta(t)) + \phi_\star^2\right) + \frac{3\eta^2\sigma^2}{M} \ .$$

Unrolling the above recursion for over an arbitrary round $r \in [0, R-1]$ gives us (c.f., the calculations in Lemma 30),

$$A(K(r+1)) \le (1 - \eta\mu)^K A(Kr) - \eta \sum_{t=Kr}^{Kr+K-1} (1 - \eta\mu)^{Kr+K-1-t} E(t)$$

$$+ \left(1 - (1 - \eta\mu)^K\right) \frac{8\eta^2 H\tau^2 K^2}{\mu} (1 - \eta\mu)^{K-2} A(Kr) + \frac{1 - (1 - \eta\mu)^K}{\eta\mu} C_1 \ .$$

Where $C_1$ is the sum of constant terms in the upper bound which do not depend on $t$ and is defined as,

$$C_1 := 4\eta^3 H^3 K^2 \zeta_\star^2 + \frac{4\eta^4 H\tau^2 K^2 \sigma^2}{\mu} + 4\eta^3 H\sigma^2 K \left(1 + \ln(K)\right) + 8\eta^3 H\tau^2 K^2 \phi_\star^2 + \frac{3\eta^2\sigma^2}{M} \ .$$

We also define the following constant,

$$\rho_4 := (1 - \eta\mu)^K + \left(1 - (1 - \eta\mu)^K\right) \frac{8\eta^2 H\tau^2 K^2}{\mu} (1 - \eta\mu)^{K-2} \ .$$

These notations allows us to re-write the above recursion as follows for $r \in [0, R-1]$,

$$A(K(r+1)) \le \rho_4 A(Kr) - \eta \sum_{t=Kr}^{Kr+K-1} (1 - \eta\mu)^{Kr+K-1-t} E(t) + \frac{1 - (1 - \eta\mu)^K}{\eta\mu} C_1 \ .$$

Now unrolling the recursion over $R$ rounds gives us,

$$A(KR) \le \rho_4^R A(0) - \eta \sum_{r=0}^{R-1} \rho_4^{R-1-r} \sum_{t=Kr}^{Kr+K-1} (1 - \eta\mu)^{Kr+K-1-t} E(t)$$

$$+ \frac{1 - (1 - \eta\mu)^K}{\eta\mu} \sum_{r=0}^{R-1} \rho_4^{R-1-r} C_1 \ ,$$

$$\le \rho_4^R A(0) - \eta \sum_{r=0}^{R-1} \rho_4^{R-1-r} \sum_{t=Kr}^{Kr+K-1} (1 - \eta\mu)^{Kr+K-1-t} E(t)$$

$$+ \frac{1 - (1 - \eta\mu)^K}{\eta\mu} \cdot \frac{1 - \rho_4^R}{1 - \rho_4} \cdot C_1 \ .$$

We will now define the following sum of weights,

$$W := \sum_{r=0}^{R-1} \rho_4^{R-1-r} \sum_{t=Kr}^{Kr+K-1} (1 - \eta\mu)^{Kr+K-1-t} \ ,$$

$$= \sum_{r=0}^{R-1} \rho_4^{R-1-r} \cdot \frac{1 - (1 - \eta\mu)^K}{\eta\mu} \ ,$$

$$= \frac{1 - \rho_4^R}{1 - \rho_4} \cdot \frac{1 - (1 - \eta\mu)^K}{\eta\mu} \ .$$

Dividing by $\eta W$ in the above recursion and re-arranging gives us the following,

$$\frac{1}{W} \sum_{r=0}^{R-1} \rho_4^{R-1-r} \sum_{t=Kr}^{Kr+K-1} (1 - \eta\mu)^{Kr+K-1-t} E(t)$$

$$
\leq \frac{\rho_4^R}{\eta W} A(0) - \frac{A(KR)}{\eta W} + \frac{1}{\eta W} \cdot \frac{1 - (1 - \eta\mu)^K}{\eta\mu} \cdot \frac{1 - \rho_4^R}{1 - \rho_4} \cdot C_1 \ ,
$$

$$
\leq \frac{\rho_4^R}{1 - \rho_4^R} \cdot \frac{1 - \rho_4}{1 - (1 - \eta\mu)^K} \mu B^2 + \frac{C_1}{\eta} \ ,
$$

$$
= \frac{\rho_4^R}{1 - \rho_4^R} \left( 1 - \frac{8\eta^2 H \tau^2 K^2}{\mu} (1 - \eta\mu)^{K-2} \right) \mu B^2 + 4\eta^2 H^3 K^2 \zeta_\star^2 + \frac{4\eta^3 H \tau^2 K^2 \sigma^2}{\mu}
$$

$$
+ 4\eta^2 H \sigma^2 K \left( 1 + \ln(K) \right) + 8\eta^2 H \tau^2 K^2 \phi_\star^2 + \frac{3\eta\sigma^2}{M} \ .
$$

Now similar to the proof in the previous section we will pick the step-size as follows,

$$
\eta := \min \left\{ \frac{1}{2H}, \frac{1}{\mu KR} \ln\left( \frac{\mu B^2}{\epsilon} \right) \right\} \ ,
$$

where we will define $\epsilon$ later in the proof. Our goal now is to bound the term $\frac{\rho_4^R}{1-\rho_4^R} \left( 1 - \frac{8\eta^2 H \tau^2 K^2}{\mu}(1 - \eta\mu)^{K-2} \right)$ so that it looks more like the exponential decay in usual convergence analyses. We first note the following,

$$
\frac{8H\tau^2}{\mu^3 R^2} \ln^2\left( \frac{\mu B^2}{\epsilon} \right) \leq \frac{1}{2} \ ,
$$

by assuming $R \geq \frac{4\tau}{\mu} \sqrt{\kappa} \ln\left( \frac{\mu B^2}{\epsilon} \right)$. This allows us to upper bound $\left( 1 - \frac{8\eta^2 H \tau^2 K^2}{\mu}(1 - \eta\mu)^{K-2} \right)$ by 1. Now we will upper bound $\frac{\rho_4^R}{1-\rho_4^R}$. To do this we first note the following,

$$
\rho_4^R = (1 - \eta\mu)^{KR} \left( 1 + \left( 1 - (1 - \eta\mu)^K \right) \frac{8\eta^2 H \tau^2 K^2}{\mu(1 - \eta\mu)} \right)^R \ ,
$$

$$
\leq^{(a)} e^{-\eta\mu KR} \left( 1 + \eta\mu K \frac{8\eta^2 H \tau^2 K^2}{\mu(1 - \eta\mu)^2} \right)^R \ ,
$$

$$
\leq e^{-\eta\mu KR} \left( 1 + \frac{1}{R^3} \ln^3\left( \frac{\mu B^2}{\epsilon} \right) \frac{8H\tau^2}{\mu^3(1 - \mu/2H)^2} \right)^R \ ,
$$

$$
\leq e^{-\eta\mu KR + \frac{1}{R^2} \ln^3\left( \frac{\mu B^2}{\epsilon} \right) \frac{8H\tau^2}{\mu^3(1 - 1/(2\kappa)^2)}} \ ,
$$

$$
\leq^{(\kappa \geq 1)} e^{-\eta\mu KR + \frac{1}{R^2} \ln^3\left( \frac{\mu B^2}{\epsilon} \right) \frac{32H\tau^2}{\mu^3}} \ ,
$$

$$
\leq^{(b)} e^{-\eta\mu KR + 1} \ ,
$$

where in (a) we use the Bernoulli's inequality after noting that $\eta\mu < 1$ for our choice of step-size; and in (b) we used $R \geq \frac{\tau}{\mu} \sqrt{\ln^3\left( \frac{\mu B^2}{\epsilon} \right) 32\kappa}$. Now using this upper bound we get,

$$
\frac{\rho_4^R}{1 - \rho_4^R} \leq \frac{e^{-\eta\mu KR + 1}}{1 - e^{-\eta\mu KR + 1}} \ ,
$$

$$
\leq^{(a)} 2e^{-\eta\mu KR + 1} \leq 6e^{-\eta\mu KR} \ ,
$$

where in (a) we assume that $e^{-\eta\mu KR + 1} \leq \frac{1}{2}$ which can be verified to be true for both choices of step-sizes as follows,

$$
(i) \ e^{-\frac{\mu KR}{2H} + 1} \leq \frac{1}{2} \Leftarrow 2e \leq e^{\frac{\mu KR}{2H}} \Leftarrow 4\kappa \leq KR \ ;
$$

$$
(ii) \ e^{-\ln\left( \mu B^2/\epsilon \right) + 1} \leq \frac{1}{2} \Leftarrow \frac{e\epsilon}{\mu B^2} \leq \frac{1}{2} \Leftarrow \epsilon \leq \frac{\mu B^2}{6} \ .
$$

We are almost done, but we still need to choose an $\epsilon$. We do this the same way as in the previous section: we pick $\epsilon$ as the maximum of the target accuracy $\epsilon_{target}$ and the value of the convergence rate terms which are an increasing function of $\eta$, at $\eta' = \frac{1}{\mu KR}$. In particular we pick $\epsilon$ as,

$$
\epsilon = \min \left\{ \max \left\{ \frac{4H^3 \zeta_\star^2}{\mu^2 R^2}, \frac{4H\tau^2 \sigma^2}{\mu^4 KR^3}, \frac{4H\sigma^2 \left( 1 + \ln(K) \right)}{\mu^2 KR^2}, \frac{8H\tau^2 \phi_\star^2}{\mu^2 R^2}, \frac{3\sigma^2}{\mu MKR}, \epsilon_{target} \right\}, \frac{\mu B^2}{6} \right\} \ .
$$

Finally, note that the we have essentially used the weights on the models $\{x_0, \ldots, x_{KR-1}\}$ defined by the blue term. Rigorously for time step $t \in [0, T-1]$ we use the following weight,

$$w_t = \frac{\rho_4^{R-1-\delta(t)/K}(1-\eta\mu)^{\delta(t)+K-1-t}}{W} \quad ,$$

and we bound the function sub-optimality of the point $\sum_{t=0}^{T-1} w_t x_t$ by using Jensen's inequality as follows,

$$\mathbb{E}\left[F\left(\sum_{t=0}^{T-1} w_t x_t\right)\right] - F(x^\star) \leq \sum_{t=0}^{T-1} w_t \left(\mathbb{E}\left[F\left(x_t\right)\right] - F(x^\star)\right) \quad .$$

Thus, our choice of $\epsilon$, $\eta$, and averaging weights proves the lemma statement, assuming the highlighted required conditions. $\qquad\square$

In the following lemma, we state the result for strongly convex quadratics, by noting that in the proof of the above lemma, we simply replace the usage of Lemma 25 by Lemma 24 and note that $Q = 0$ for quadratics, which allows us to replace several smoothness constants $H$ in the convergence rate by $\tau$.

**Lemma 38** (Strongly Convex Function Convergence with $\tau$, $\zeta_\star$, $\phi_\star$ for Quadratics). *Assuming $R \geq \frac{4\tau^2}{\mu^2} \max\left\{\ln\left(\frac{\mu B^2}{\epsilon}\right), \sqrt{2\ln^3\left(\frac{\mu B^2}{\epsilon}\right)}\right\}$, $KR \geq 4\kappa$ we can get the following convergence guarantee for local SGD, initializing at $x_0 = 0$ and optimizing functions satisfying Assumptions 1 to 6,*

$$\mathbb{E}[F(\hat{x})] - F(x^\star) = \tilde{\mathcal{O}}\left(e^{-\frac{\mu KR}{2H}}\mu B^2 + \frac{\tau^2 H^2 \zeta_\star^2}{\mu^3 R^2} + \frac{\tau^4\sigma^2}{\mu^5 KR^3} + \frac{\tau^2\sigma^2\left(1+\ln(K)\right)}{\mu^3 KR^2} + \frac{\tau^4\phi_\star^2}{\mu^3 R^2} + \frac{\sigma^2}{\mu MKR}\right) \quad ,$$

*where we define $\hat{x} = \sum_{t=0}^{T-1} w_t x_t$ for the choice of weights*

$$w_t := \frac{\rho_4^{R-1-\delta(t)/K}(1-\eta\mu)^{\delta(t)+K-1-t}}{W}$$

*for $W = \frac{1-\rho_4^R}{1-\rho_4} \cdot \frac{1-(1-\eta\mu)^K}{\eta\mu}$ and $\rho_4 = (1-\eta\mu)^K + \left(1-(1-\eta\mu)^K\right)\frac{8\eta^2\tau^4 K^2}{\mu^2}(1-\eta\mu)^{K-2}$. And we pick the step-size as,*

$$\eta = \min\left\{\frac{1}{2H}, \frac{1}{\mu KR}\ln\left(\frac{\mu B^2}{\epsilon}\right)\right\} \quad ,$$

*for the choice of $\epsilon$,*

$$\epsilon = \min\left\{\max\left\{\frac{4\tau^2 H^2\zeta_\star^2}{\mu^3 R^2}, \frac{4\tau^4\sigma^2}{\mu^5 KR^3}, \frac{4\tau^2\sigma^2\left(1+\ln(K)\right)}{\mu^3 KR^2}, \frac{8\tau^4\phi_\star^2}{\mu^3 R^2}, \frac{3\sigma^2}{\mu MKR}, \epsilon_{target}\right\}, \frac{\mu B^2}{6}\right\} \quad ,$$

*where $\epsilon_{target}$ is a target, which is greater than or equal to the machine precision.*

## I.3 Convergence in Iterates with Third-order Smoothness

The main technical challenge in incorporating third-order smoothness (c.f., Assumption 1) in our upper bounds lies in bounding the sequence $D(\cdot)$ while working with the upper bound in Lemma 20. One natural approach is to mirror the analysis in the previous section: unroll the consensus error recursion back to the previous communication round, substitute that into the upper bound for $A(\cdot)$, and then iterate across rounds. However, this strategy quickly encounters difficulties. We need to control the fourth moment of the iterate error, $B(\cdot)$, and we lack a uniform upper bound for it.

To overcome this, we adopt a different strategy. As the following lemma shows, we analyze the pair $(A(\cdot), B(\cdot))$ together in terms of the pair $(C(\cdot), D(\cdot))$, treating them as components of a two-dimensional recursion. Once we do this, we can more or less use ideas similar to those before.

**Lemma 39.** *Under Assumptions 1 to 6 using $\eta < 1/H$ and defining*

$$\rho_3 := (1-\eta\mu)^K + \left(\left(1-(1-\eta\mu)^K\right)\right.$$

$$\times \left( \frac{2\tau^2}{\mu^2} + \frac{2Q^2 B^2}{\mu^2} + \frac{16\eta^2\sigma^2\tau^2}{\mu^2 MB^2} + \frac{H^4}{\mu^4} + \frac{16\eta^2\sigma^2 Q^2}{\mu^2 M} \right) \left( 4\eta^2\tau^2 K^2 + 64\eta^4\tau^4 K^4 \right) \right) \ ,$$

$$\Psi := 4\eta^2\tau^2 K^2\phi_\star^2 + \frac{128\eta^5\tau^4 K^4\sigma^2}{\mu B^2}\phi_\star^2 + \frac{320\eta^4\sigma^2\tau^2 K^3}{B^2}\phi_\star^2 + \frac{64\eta^4\tau^4 K^4}{B^2}\phi_\star^4$$
$$+ 2\eta^2 H^2 K^2\zeta_\star^2 + \frac{2\eta^3\tau^2 K^2\sigma^2}{\mu} + 2\eta^2\sigma^2 K \left(1 + \ln(K)\right) + \frac{8\eta^3 K^3 H^4\zeta_\star^4}{\mu B^2} + \frac{88\eta^5 K^3\tau^4\sigma^4}{\mu^3 B^2}$$
$$+ \frac{160\eta^4 K^4\sigma^2 H^2\zeta_\star^2}{B^2} + \frac{160\eta^5\tau^2 K^4\sigma^4}{\mu B^2} + \frac{112\eta^4 K^3\sigma^4 \left(1 + \ln(K)\right)}{B^2} \ .$$

*we can get the following convergence guarantee with initialization $x_0 = 0$,*

$$\max\left\{ A(KR), \frac{B(KR)}{B^2} \right\} \le 2\rho_3^R B^2 + \frac{1 - (1 - \eta\mu)^K}{1 - \rho_3} \left( \frac{\eta\sigma^2}{\mu M} + \frac{9\eta^3\sigma^4}{\mu M^2 B^2} \right)$$
$$+ \frac{1 - (1 - \eta\mu)^K}{1 - \rho_3} \left( \frac{2\tau^2}{\mu^2} + \frac{2Q^2 B^2}{\mu^2} + \frac{16\eta^2\sigma^2\tau^2}{\mu^2 MB^2} + \frac{H^4}{\mu^4} + \frac{16\eta^2\sigma^2 Q^2}{\mu^2 M} \right) \Psi \ .$$

*Proof.* We will denote the following vectors for all $t \in [0, T]$,

$$\mathbb{A}(t) := \begin{bmatrix} A(t) \\ B(t)/B^2 \end{bmatrix} \qquad \text{and} \qquad \mathbb{C}(t) := \begin{bmatrix} C(t) \\ D(t)/B^2 \end{bmatrix} \ ,$$

where note that $B$ comes from Assumption 3 and we divide the sequences $B(t)$, $C(t)$ by $B^2$ to make them "dimensionally consistent" or similarly scale-variant as the sequences $A(t)$, $C(t)$. Based on the recursions we have developed in Lemmas 20 and 22 we get the following vector recursion,

$$\mathbb{A}(t+1) \le (1 - \eta\mu) \begin{bmatrix} 1 & 0 \\ \frac{8\eta^2\sigma^2}{MB^2} & 1 \end{bmatrix} \mathbb{A}(t) + \begin{bmatrix} \frac{2\eta\tau^2}{\mu} & \frac{2\eta Q^2 B^2}{\mu} \\ \frac{16\eta^3\sigma^2\tau^2}{\mu MB^2} & \frac{\eta H^4}{\mu^3} + \frac{16\eta^3\sigma^2 Q^2}{\mu M} \end{bmatrix} \mathbb{C}(t) + \begin{bmatrix} \frac{\eta^2\sigma^2}{M} \\ \frac{9\eta^4\sigma^4}{M^2 B^2} \end{bmatrix} \ ,$$
$$=: P\mathbb{A}(t) + Q\mathbb{C}(t) + N \ ,$$
$$\le P^{t+1-\delta(t)}\mathbb{A}(\delta(t)) + \sum_{j=\delta(t)}^{t} P^{t-j} \left( Q\mathbb{C}(j) + N \right) \ ,$$

where we define $P, Q \in \mathbb{R}^{2\times 2}$ and $N \in \mathbb{R}^2$ to simplify the calculations. Let us also recall the recursion we get for the consensus error terms based on Lemmas 30 and 31,

$$\mathbb{C}(t) \le \begin{bmatrix} 4\eta^2\tau^2 K^2 & 0 \\ \frac{128\eta^5\tau^4 K^4\sigma^2}{\mu B^2} + \frac{320\eta^4\sigma^2\tau^2 K^3}{B^2} & 64\eta^4\tau^4 K^4 \end{bmatrix} \mathbb{A}(\delta(t))$$
$$+ \begin{bmatrix} 4\eta^2\tau^2 K^2\phi_\star^2 \\ \frac{128\eta^5\tau^4 K^4\sigma^2}{\mu B^2}\phi_\star^2 + \frac{320\eta^4\sigma^2\tau^2 K^3}{B^2}\phi_\star^2 + \frac{64\eta^4\tau^4 K^4}{B^2}\phi_\star^4 \end{bmatrix}$$
$$+ \begin{bmatrix} 2\eta^2 H^2 K^2\zeta_\star^2 + \frac{2\eta^3\tau^2 K^2\sigma^2}{\mu} + 2\eta^2\sigma^2 K \left(1 + \ln(K)\right) \\ \frac{8\eta^3 K^3 H^4\zeta_\star^4}{\mu B^2} + \frac{88\eta^5 K^3\tau^4\sigma^4}{\mu^3 B^2} + \frac{160\eta^4 K^4\sigma^2 H^2\zeta_\star^2}{B^2} + \frac{160\eta^5\tau^2 K^4\sigma^4}{\mu B^2} + \frac{112\eta^4 K^3\sigma^4(1+\ln(K))}{B^2} \end{bmatrix} \ ,$$
$$=: U\mathbb{A}(\delta(t)) + V \ ,$$

where we define $U \in \mathbb{R}^{2\times 2}$ and $V \in \mathbb{R}^2$. Now we can plug in this upper bound in the inequality above, which gives us,

$$\mathbb{A}(t+1) \le P^{t+1-\delta(t)}\mathbb{A}(\delta(t)) + \sum_{j=\delta(t)}^{t} P^{t-j} \left( QU\mathbb{A}(\delta(t)) + QV + N \right) \ .$$

Now, let us denote $t = KR - 1$ and unroll across communication rounds to get the following,

$$\mathbb{A}(KR) \le P^K \mathbb{A}(K(R-1)) + \sum_{j=K(R-1)}^{KR-1} P^{KR-1-j} \left( QU\mathbb{A}(K(R-1)) + QV + N \right) \ ,$$

$$=: P^K \mathbb{A}(K(R-1)) + \bar{P}\left(QU\mathbb{A}(K(R-1)) + QV + N\right) \ ,$$
$$= \left(P^K + \bar{P}QU\right)\mathbb{A}(K(R-1)) + \bar{P}\left(QV + N\right) \ ,$$

where we define $\bar{P} = \sum_{j=K(R-1)}^{KR-1} P^{KR-1-j} \in \mathbb{R}^{2\times 2}$. Taking the norm on both sides and using the triangle inequality, we get,

$$\|\mathbb{A}(KR)\|_2 \le \left\|\left(P^K + \bar{P}QU\right)\right\|_2 \|\mathbb{A}(K(R-1))\|_2 + \left\|\bar{P}Q\right\|_2 \|V\|_2 + \left\|\bar{P}\right\|_2 \|N\|_2 \ ,$$

$$\textcolor{red}{\le \left(\left\|P^K\right\|_2 + \left\|\bar{P}\right\|_2 \|Q\|_2 \|U\|_2\right) \|\mathbb{A}(K(R-1))\|_2 + \left\|\bar{P}\right\|_2 \|Q\|_2 \|V\|_2 + \left\|\bar{P}\right\|_2 \|N\|_2 \ .}$$

We will not individually upper bound these spectral norms. First note that due to $P$ being a lower triangular matrix,

$$P^K = (1-\eta\mu)^K \begin{bmatrix} 1 & 0 \\ \frac{8\eta^2\sigma^2 K}{MB^2} & 1 \end{bmatrix} \ .$$

Since $P^K$ is a lower triangular matrix, its eigenvalues can be read off its diagonal. In particular, we note that $\left\|P^K\right\|_2 = (1-\eta\mu)^K$. We can use a similar idea to upper bound $\left\|\bar{P}\right\|_2$ as follows,

$$\bar{P} = \begin{bmatrix} \sum_{i=0}^{K-1}(1-\eta\mu)^i & 0 \\ \frac{8\eta^2\sigma^2}{MB^2}\sum_{i=0}^{K-1} i(1-\eta\mu)^i & \sum_{i=0}^{K-1}(1-\eta\mu)^i \end{bmatrix} \ .$$

This implies $\left\|\bar{P}\right\|_2 = \frac{1-(1-\eta\mu)^K}{\eta\mu}$. We also note the following about $Q$, noting that the spectral norm is upper-bounded by the Frobenius norm,

$$\|Q\|_2 \le \frac{2\eta\tau^2}{\mu} + \frac{2\eta Q^2 B^2}{\mu} + \frac{16\eta^3\sigma^2\tau^2}{\mu MB^2} + \frac{\eta H^4}{\mu^3} + \frac{16\eta^3\sigma^2 Q^2}{\mu M} \ .$$

Finally, noting that $U$ is also lower diagonal, we note that,

$$\|U\|_2 = \max\left\{4\eta^2\tau^2 K^2, 64\eta^4\tau^4 K^4\right\} \ ,$$
$$\le 4\eta^2\tau^2 K^2 + 64\eta^4\tau^4 K^4 \ .$$

Combining the upper bounds for $P^K, \bar{P}, Q, U$ we get,

$$\left\|P^K\right\|_2 + \left\|\bar{P}\right\|_2 \|Q\|_2 \|U\|_2$$
$$\le (1-\eta\mu)^K$$
$$+ \left(1 - (1-\eta\mu)^K\right)\left(\frac{2\tau^2}{\mu^2} + \frac{2Q^2 B^2}{\mu^2} + \frac{16\eta^2\sigma^2\tau^2}{\mu^2 MB^2} + \frac{H^4}{\mu^4} + \frac{16\eta^2\sigma^2 Q^2}{\mu^2 M}\right)\left(4\eta^2\tau^2 K^2 + 64\eta^4\tau^4 K^4\right) \ ,$$
$$=: \rho_3 \ .$$

Note that when $\tau = 0$, then $\rho_3 = (1-\eta\mu)^K$, which will lead to the fast exponential decay we do get in the homogeneous setting. Using the above calculation, we can also conclude that,

$$\left\|\bar{P}\right\|_2 \|Q\|_2 \|V\|_2 \le \left(1 - (1-\eta\mu)^K\right)\left(\frac{2\tau^2}{\mu^2} + \frac{2Q^2 B^2}{\mu^2} + \frac{16\eta^2\sigma^2\tau^2}{\mu^2 MB^2} + \frac{H^4}{\mu^4} + \frac{16\eta^2\sigma^2 Q^2}{\mu^2 M}\right)\|V\|_2 \ ,$$

$$\left\|\bar{P}\right\|_2 \|N\|_2 \le \left(1 - (1-\eta\mu)^K\right)\left(\frac{\eta\sigma^2}{\mu M} + \frac{9\eta^3\sigma^4}{\mu M^2 B^2}\right) \ .$$

Plugging this back into the red inequality and then unrolling the recursion, we get,

$$\|\mathbb{A}(KR)\|_2 \le \rho_3^R \|\mathbb{A}(K(R-1))\|_2$$
$$+ \frac{1-(1-\eta\mu)^K}{1-\rho_3}\left(\frac{2\tau^2}{\mu^2} + \frac{2Q^2 B^2}{\mu^2} + \frac{16\eta^2\sigma^2\tau^2}{\mu^2 MB^2} + \frac{H^4}{\mu^4} + \frac{16\eta^2\sigma^2 Q^2}{\mu^2 M}\right)\|V\|_2$$
$$+ \frac{1-(1-\eta\mu)^K}{1-\rho_3}\left(\frac{\eta\sigma^2}{\mu M} + \frac{9\eta^3\sigma^4}{\mu M^2 B^2}\right),$$

which proves our convergence rate upon applying the triangle inequality to note that,

$$\|V\|_2 \le 4\eta^2\tau^2 K^2\phi_\star^2 + \frac{128\eta^5\tau^4 K^4\sigma^2}{\mu B^2}\phi_\star^2 + \frac{320\eta^4\sigma^2\tau^2 K^3}{B^2}\phi_\star^2 + \frac{64\eta^4\tau^4 K^4}{B^2}\phi_\star^4$$

$$+ 2\eta^2 H^2 K^2 \zeta_\star^2 + \frac{2\eta^3 \tau^2 K^2 \sigma^2}{\mu} + 2\eta^2 \sigma^2 K \left(1 + \ln(K)\right) + \frac{8\eta^3 K^3 H^4 \zeta_\star^4}{\mu B^2} + \frac{88\eta^5 K^3 \tau^4 \sigma^4}{\mu^3 B^2}$$

$$+ \frac{160\eta^4 K^4 \sigma^2 H^2 \zeta_\star^2}{B^2} + \frac{160\eta^5 \tau^2 K^4 \sigma^4}{\mu B^2} + \frac{112\eta^4 K^3 \sigma^4 \left(1 + \ln(K)\right)}{B^2} \quad .$$

This proves the lemma. $\qquad\square$

We will now tune the step-size following a similar idea to the previous section to achieve the following convergence rate.

**Lemma 40.** *Assuming sufficiently many communication rounds,*

$$R \geq \frac{8\tau}{\mu} \max \left\{ \ln^2 \left(\frac{B^2}{\epsilon}\right) \cdot \left(\frac{4QB}{\mu} + \frac{5H^2}{\mu^2}\right), \ln^{3/2}\left(\frac{B^2}{\epsilon}\right) \left(1 + \sqrt{\frac{QB}{\mu} + \frac{H}{\mu}}\right), \frac{\ln(B^2/\epsilon)}{\ln(\ln(B^2/\epsilon))} \right\},$$

$B^2 > e\epsilon$, *and* $KR \geq \frac{8\sigma}{\mu^2 \sqrt{M}} \ln\left(\frac{B^2}{\epsilon}\right) \cdot \max\left\{\frac{\tau}{B}, Q\right\}$ *we can get the following convergence guarantee for local SGD, initializing at $x_0 = 0$ and optimizing functions satisfying Assumptions 1 to 6,*

$$\|\mathbb{A}(KR)\|_2 = \tilde{\mathcal{O}}\left( e^{-\eta \mu KR} B^2 + \frac{\sigma^2}{\mu^2 MKR} + \frac{\sigma^4}{\mu^4 K^3 R^3 M^2 B^2} + \kappa'\left(\frac{\tau^2 \phi_\star^2}{\mu^2 R^2} + \frac{\tau^4 \sigma^2}{\mu^6 KR^5 B^2}\phi_\star^2\right)\right.$$

$$+ \kappa'\left(\frac{\sigma^2 \tau^2}{\mu^4 KR^4 B^2}\phi_\star^2 + \frac{\tau^4}{\mu^4 B^2 R^4}\phi_\star^4 + \frac{H^2 \zeta_\star^2}{\mu^2 R^2} + \frac{\tau^2 \sigma^2}{\mu^4 KR^3} + \frac{\sigma^2\left(1 + \ln(K)\right)}{\mu^2 KR^2}\right)$$

$$\left.+ \kappa'\left(\frac{H^4 \zeta_\star^4}{\mu^4 R^3 B^2} + \frac{\tau^4 \sigma^4}{\mu^8 K^2 R^5 B^2} + \frac{\sigma^2 H^2 \zeta_\star^2}{\mu^4 B^2 R^4} + \frac{\tau^2 \sigma^4}{\mu^6 KR^5 B^2} + \frac{\sigma^4\left(1 + \ln(K)\right)}{\mu^4 KB^2 R^4}\right)\right) \quad,$$

*where we define $\kappa' := 2 + \frac{4Q^2 B^2}{\mu^2} + \frac{6H^4}{\mu^4}$ and we pick the step-size,*

$$\eta = \min\left\{\frac{1}{2H}, \frac{1}{\mu KR}\ln\left(\frac{B^2}{\epsilon}\right)\right\} \quad,$$

*with the choice of $\epsilon$ is given by*

$$\epsilon := \max\left\{ \frac{\sigma^2}{\mu^2 MKR}, \frac{\sigma^4}{\mu^4 K^3 R^3 M^2 B^2}, \frac{\tau^2 \phi_\star^2 \kappa'}{\mu^2 R^2}, \frac{\tau^4 \sigma^2 \kappa' \phi_\star^2}{\mu^6 KR^5 B^2}, \frac{\sigma^2 \tau^2 \kappa' \phi_\star^2}{\mu^4 KR^4 B^2}, \frac{\tau^4 \kappa' \phi_\star^4}{\mu^4 B^2 R^4}, \right.$$

$$\frac{H^2 \zeta_\star^2 \kappa'}{\mu^2 R^2} + \frac{\tau^2 \sigma^2}{\mu^4 KR^3} + \frac{\sigma^2\left(1 + \ln(K)\right)}{\mu^2 KR^2}, \frac{H^4 \zeta_\star^4 \kappa'}{\mu^4 R^3 B^2}, \frac{\tau^4 \sigma^4 \kappa'}{\mu^8 K^2 R^5 B^2}, \frac{\sigma^2 H^2 \zeta_\star^2 \kappa'}{\mu^4 B^2 R^4},$$

$$\left.\frac{\tau^2 \sigma^4 \kappa'}{\mu^6 KR^5 B^2}, \frac{\sigma^4\left(1 + \ln(K)\right)\kappa'}{\mu^4 KB^2 R^4}, \epsilon_{target}\right\}$$

*where $\epsilon_{target}$ is the target accuracy, greater than or equal to the machine precision.*

*Proof.* We will pick the following step-size,

$$\eta = \min\left\{\frac{1}{2H}, \frac{1}{\mu KR}\ln\left(\frac{B^2}{\epsilon}\right)\right\} \quad,$$

where the choice of $\epsilon > 0$ will be made explicit later. We will first identify the requirements on problem parameters to guarantee that,

$$\frac{1 - (1 - \eta\mu)^K}{1 - \rho_3} \leq 2 \quad,$$

$$\Leftrightarrow \frac{1 - (1 - \eta\mu)^K}{2} \leq (1 - \rho_3) \quad,$$

$$\Leftrightarrow \frac{1}{2} \leq 1 - \left(\frac{2\tau^2}{\mu^2} + \frac{2Q^2 B^2}{\mu^2} + \frac{16\eta^2 \sigma^2 \tau^2}{\mu^2 MB^2} + \frac{H^4}{\mu^4} + \frac{16\eta^2 \sigma^2 Q^2}{\mu^2 M}\right)\left(4\eta^2 \tau^2 K^2 + 64\eta^4 \tau^4 K^4\right) \quad,$$

$$\Leftrightarrow \left( \frac{2\tau^2}{\mu^2} + \frac{2Q^2B^2}{\mu^2} + \frac{16\eta^2\sigma^2\tau^2}{\mu^2MB^2} + \frac{H^4}{\mu^4} + \frac{16\eta^2\sigma^2Q^2}{\mu^2M} \right) \left( 4\eta^2\tau^2K^2 + 64\eta^4\tau^4K^4 \right) \le \frac{1}{2} \;,$$

$$\Leftarrow^{(a)} \left( \frac{2Q^2B^2}{\mu^2} + \frac{16\sigma^2\tau^2}{\mu^4K^2R^2MB^2} \ln^2\left(\frac{B^2}{\epsilon}\right) + \frac{3H^4}{\mu^4} + \frac{16\sigma^2Q^2}{\mu^4K^2R^2M} \ln^2\left(\frac{B^2}{\epsilon}\right) \right)$$
$$\times \left( \frac{4\tau^2}{\mu^2R^2} \ln^2\left(\frac{B^2}{\epsilon}\right) + \frac{64\tau^4}{\mu^4R^4} \ln^4\left(\frac{B^2}{\epsilon}\right) \right) \le \frac{1}{2} \;,$$

$$\Leftarrow (i) \; \left( \frac{2Q^2B^2}{\mu^2} + \frac{3H^4}{\mu^4} \right) \left( \frac{4\tau^2}{\mu^2R^2} \ln^2\left(\frac{B^2}{\epsilon}\right) + \frac{64\tau^4}{\mu^4R^4} \ln^4\left(\frac{B^2}{\epsilon}\right) \right) \le \frac{1}{4} \;; \quad \text{and}$$

$$(ii) \; \left( \frac{16\sigma^2\tau^2}{\mu^4K^2R^2MB^2} \ln^2\left(\frac{B^2}{\epsilon}\right) + \frac{16\sigma^2Q^2}{\mu^4K^2R^2M} \ln^2\left(\frac{B^2}{\epsilon}\right) \right)$$
$$\times \left( \frac{4\tau^2}{\mu^2R^2} \ln^2\left(\frac{B^2}{\epsilon}\right) + \frac{64\tau^4}{\mu^4R^4} \ln^4\left(\frac{B^2}{\epsilon}\right) \right) \le \frac{1}{4} \;,$$

$$\Leftarrow (i) \; \sqrt{\frac{16Q^2B^2}{\mu^2} + \frac{24H^4}{\mu^4}} \cdot \frac{2\tau}{\mu} \ln\left(\frac{B^2}{\epsilon}\right) \le R \;;$$

$$(ii) \; \sqrt[4]{\frac{16Q^2B^2}{\mu^2} + \frac{24H^4}{\mu^4}} \cdot \frac{\sqrt[4]{64}\,\tau}{\mu} \ln\left(\frac{B^2}{\epsilon}\right) \le R \;;$$

$$(iii) \; \frac{8\sigma\tau}{\mu^2\sqrt{M}B} \ln\left(\frac{B^2}{\epsilon}\right) \le KR \;;$$

$$(iv) \; \frac{8\sigma Q}{\mu^2\sqrt{M}} \ln\left(\frac{B^2}{\epsilon}\right) \le KR \;; \quad \text{and}$$

$$(v) \; \frac{4\tau}{\mu} \ln\left(\frac{B^2}{\epsilon}\right) \le R \;,$$

$$\Leftarrow \textcolor{red}{(i) \; KR \ge \frac{8\sigma}{\mu^2\sqrt{M}} \ln\left(\frac{B^2}{\epsilon}\right) \cdot \max\left\{ \frac{\tau}{B}, Q \right\}} \;; \quad \text{and}$$

$$\textcolor{red}{(ii) \; R \ge \frac{3\tau}{\mu} \ln\left(\frac{B^2}{\epsilon}\right) \cdot \left( \frac{4QB}{\mu} + \frac{5H^2}{\mu^2} \right)} \;.$$

where in (a) we used that $\tau^2/\mu^2 \le H^2/\mu^2$. Now we will upper bound $\rho_3$ as follows,

$$\rho_3$$
$$\le^{(a)} (1 - \eta\mu)^K$$
$$+ \frac{1}{R} \ln\left(\frac{B^2}{\epsilon}\right) \left( \frac{2Q^2B^2}{\mu^2} + \frac{16\sigma^2}{\mu^4K^2R^2M} \left( \frac{\tau^2}{B^2} + 1 \right) \ln^2\left(\frac{B^2}{\epsilon}\right) + \frac{3H^4}{\mu^4} \right)$$
$$\times \left( \frac{4\tau^2}{\mu^2R^2} \ln^2\left(\frac{B^2}{\epsilon}\right) + \frac{64\tau^4}{\mu^4R^4} \ln^4\left(\frac{B^2}{\epsilon}\right) \right) \;,$$

$$\le^{(b)} e^{-\eta\mu K} + \frac{1}{R} \ln\left(\frac{B^2}{\epsilon}\right) \left( \frac{2Q^2B^2}{\mu^2} + 1 + \frac{3H^4}{\mu^4} \right) \left( \frac{4\tau^2}{\mu^2R^2} \ln^2\left(\frac{B^2}{\epsilon}\right) + \frac{64\tau^4}{\mu^4R^4} \ln^4\left(\frac{B^2}{\epsilon}\right) \right) \;,$$

$$\le e^{-\eta\mu K} \left( 1 + \frac{e^{\eta\mu K}}{R} \ln\left(\frac{B^2}{\epsilon}\right) \left( \frac{2Q^2B^2}{\mu^2} + 1 + \frac{3H^4}{\mu^4} \right) \right.$$
$$\left. \times \left( \frac{4\tau^2}{\mu^2R^2} \ln^2\left(\frac{B^2}{\epsilon}\right) + \frac{64\tau^4}{\mu^4R^4} \ln^4\left(\frac{B^2}{\epsilon}\right) \right) \right) \;,$$

$$\le e^{-\eta\mu K} \exp\left( \frac{e^{\eta\mu K}}{R} \ln\left(\frac{B^2}{\epsilon}\right) \left( \frac{2Q^2B^2}{\mu^2} + 1 + \frac{3H^4}{\mu^4} \right) \left( \frac{4\tau^2}{\mu^2R^2} \ln^2\left(\frac{B^2}{\epsilon}\right) + \frac{64\tau^4}{\mu^4R^4} \ln^4\left(\frac{B^2}{\epsilon}\right) \right) \right) \;,$$

$$\le e^{-\eta\mu K} \exp\left( \frac{e^{1/R\ln(B^2/\epsilon)}}{R} \ln\left(\frac{B^2}{\epsilon}\right) \left( \frac{2Q^2B^2}{\mu^2} + 1 + \frac{3H^4}{\mu^4} \right) \left( \frac{4\tau^2}{\mu^2R^2} \ln^2\left(\frac{B^2}{\epsilon}\right) + \frac{64\tau^4}{\mu^4R^4} \ln^4\left(\frac{B^2}{\epsilon}\right) \right) \right) \;,$$

$$\leq e^{-\eta\mu K} \exp\left(\frac{1}{R}\left(\frac{B^2}{\epsilon}\right)^{1/R} \ln\left(\frac{B^2}{\epsilon}\right)\left(\frac{2Q^2B^2}{\mu^2}+1+\frac{3H^4}{\mu^4}\right)\left(\frac{4\tau^2}{\mu^2R^2}\ln^2\left(\frac{B^2}{\epsilon}\right)+\frac{64\tau^4}{\mu^4R^4}\ln^4\left(\frac{B^2}{\epsilon}\right)\right)\right) \ ,$$

where in (a) we use Bernoulli's Inequality and the choice of step-size, which implies $\eta\mu < 1$ as well as the fact that $\tau^2/\mu^2 \leq H^4/\mu^4$; and in (b) we assumed that the conditions derived above to ensure $\frac{1-(1-\eta\mu)^K}{2} \leq 1-\rho_3$ are true, which allows us to conclude $\frac{16\sigma^2}{\mu^4K^2R^2M}\left(\frac{\tau^2}{B^2}+1\right)\ln^2\left(\frac{B^2}{\epsilon}\right) \leq 1$. Raising both sides to the power $R$ gives us,

$$\rho_3^R$$

$$\leq e^{-\eta\mu KR}\exp\left(\left(\frac{B^2}{\epsilon}\right)^{1/R}\ln\left(\frac{B^2}{\epsilon}\right)\left(\frac{2Q^2B^2}{\mu^2}+1+\frac{3H^4}{\mu^4}\right)\left(\frac{4\tau^2}{\mu^2R^2}\ln^2\left(\frac{B^2}{\epsilon}\right)+\frac{64\tau^4}{\mu^4R^4}\ln^4\left(\frac{B^2}{\epsilon}\right)\right)\right) \ ,$$

$$\leq^{(a)} e^{-\eta\mu KR}\exp\left(\ln^2\left(\frac{B^2}{\epsilon}\right)\left(\frac{2Q^2B^2}{\mu^2}+1+\frac{3H^4}{\mu^4}\right)\left(\frac{4\tau^2}{\mu^2R^2}\ln^2\left(\frac{B^2}{\epsilon}\right)+\frac{64\tau^4}{\mu^4R^4}\ln^4\left(\frac{B^2}{\epsilon}\right)\right)\right) \ ,$$

$$\leq^{(b)} e^{-\eta\mu KR+1} \ ,$$

where in (a) we assume that $R \geq \frac{\ln(B^2/\epsilon)}{\ln(\ln(B^2/\epsilon))}$; and in (b) we assume $R \geq \frac{8\tau}{\mu}\ln^2\left(\frac{B^2}{\epsilon}\right)\left(1+\frac{QB}{\mu}+\frac{H^2}{\mu^2}\right)$ as well as $R \geq \frac{8\tau}{\mu}\ln^{3/2}\left(\frac{B^2}{\epsilon}\right)\left(1+\sqrt{\frac{QB}{\mu}}+\frac{H}{\mu}\right)$. These observations, along with the conditions derived so far allow us to simplify the convergence rate as follows,

$$\|\mathbb{A}(KR)\|_2$$

$$\leq e^{-\eta\mu KR+1}\sqrt{2}B^2 + \frac{2\eta\sigma^2}{\mu M} + \frac{18\eta^3\sigma^4}{\mu M^2B^2} + \left(2+\frac{4Q^2B^2}{\mu^2}+\frac{6H^4}{\mu^4}\right)\|V\|_2 \ ,$$

$$\leq 4e^{-\eta\mu KR}B^2 + \frac{2\sigma^2}{\mu^2 MKR}\ln\left(\frac{B^2}{\epsilon}\right) + \frac{18\sigma^4}{\mu^4K^3R^3M^2B^2}\ln^3\left(\frac{B^2}{\epsilon}\right) + \kappa'\left(\frac{4\tau^2\phi_\star^2}{\mu^2R^2}\ln^2\left(\frac{B^2}{\epsilon}\right)\right)$$

$$+ \kappa'\left(\frac{128\tau^4\sigma^2}{\mu^6KR^5B^2}\phi_\star^2\ln^5\left(\frac{B^2}{\epsilon}\right) + \frac{320\sigma^2\tau^2}{\mu^4KR^4B^2}\phi_\star^2\ln^4\left(\frac{B^2}{\epsilon}\right) + \frac{64\tau^4}{\mu^4B^2R^4}\phi_\star^4\ln^4\left(\frac{B^2}{\epsilon}\right)\right)$$

$$+ \kappa'\left(\frac{2H^2\zeta_\star^2}{\mu^2R^2}\ln^2\left(\frac{B^2}{\epsilon}\right) + \frac{2\tau^2\sigma^2}{\mu^4KR^3}\ln^3\left(\frac{B^2}{\epsilon}\right) + \frac{2\sigma^2\left(1+\ln(K)\right)}{\mu^2KR^2}\ln^2\left(\frac{B^2}{\epsilon}\right)\right)$$

$$+ \kappa'\left(\frac{8H^4\zeta_\star^4}{\mu^4R^3B^2}\ln^3\left(\frac{B^2}{\epsilon}\right) + \frac{88\tau^4\sigma^4}{\mu^8K^2R^5B^2}\ln^5\left(\frac{B^2}{\epsilon}\right) + \frac{160\sigma^2H^2\zeta_\star^2}{\mu^4B^2R^4}\ln^4\left(\frac{B^2}{\epsilon}\right)\right)$$

$$+ \kappa'\left(\frac{160\tau^2\sigma^4}{\mu^6KR^5B^2}\ln^5\left(\frac{B^2}{\epsilon}\right) + \frac{112\sigma^4\left(1+\ln(K)\right)}{\mu^4KB^2R^4}\ln^4\left(\frac{B^2}{\epsilon}\right)\right) \ ,$$

where we define $\kappa' := \left(2+\frac{4Q^2B^2}{\mu^2}+\frac{6H^4}{\mu^4}\right)$. We are almost done, but we need to define $\epsilon$. Our choice of $\epsilon$ is simply the maximum of all the terms (after removing the logarithmic factors) in the above convergence bound, except for the first exponential term and the target accuracy $\epsilon_{target}$, which is an input to the algorithm. Like in the previous lemmas' proofs, we recall that the term dominating in $\epsilon$ also dominates the final convergence rate. This choice of $\epsilon$ and $\eta$, proves the lemma statement. $\square$

## J  More Details on the Experiments

In this appendix we describe in full detail how we generated the synthetic data for each client and how we controlled first- and second-order heterogeneity without altering the inherent difficulty of the individual optimization problems (e.g. their condition numbers or solution norms) for the experiments in Section 6.

### J.1  Data generation for each client

We consider a linear regression problem with parameter dimension $d$. There are $M$ clients, indexed by $m = 1, \ldots, M$. For each client $m$, we generate i.i.d. data $(\beta_m, y_m) \sim \mathcal{D}_m$ with

$$\beta_m \sim \mathcal{N}(\mu_m, I_d), \qquad y_m = \langle x_m^\star, \beta_m \rangle + \varepsilon, \ \ \varepsilon \sim \mathcal{N}(0, \sigma_{\text{noise}}^2).$$

The corresponding per-sample squared loss is

$$f(x; (\beta_m, y_m)) = \tfrac{1}{2} (y_m - \langle x, \beta_m \rangle)^2,$$

and the population objective on client $m$ is

$$F_m(x) = \mathbb{E}_{(\beta, y) \sim \mathcal{D}_m} \left[ f(x; (\beta, y)) \right] = \tfrac{1}{2} (x - x_m^\star)^\top \left( \mu_m \mu_m^\top + I_d \right) (x - x_m^\star) + \tfrac{1}{2} \sigma_{\text{noise}}^2.$$

Under suitable bounds on $\|\mu_m\|$, $\sigma_{\text{noise}}$, and $\|x_m^\star\|$, these objectives satisfy Assumptions 1 to 3 for all $x$ in a bounded region.

## J.2 Controlling first-order (concept) heterogeneity

We fix the norm of each true optimizer to $\|x_m^\star\| = R_\star$. To vary the maximum pairwise distance $\max_{m,n} \|x_m^\star - x_n^\star\| = \zeta_\star$, we sample each

$$x_m^\star = R_\star \, v_m \quad \text{with} \quad v_m \in \mathbb{R}^d, \; \|v_m\| = 1,$$

where $v_m$ is drawn uniformly from the spherical cap of half-angle

$$\phi(\zeta_\star) = \arcsin\left( \frac{\zeta_\star}{2R_\star} \right)$$

around a fixed "central" random unit vector $v_0$. This ensures $\|x_m^\star\| = R_\star$ for all $m$, and $\max_{m,n} \|x_m^\star - x_n^\star\| = \zeta_\star$, so that larger $\zeta_\star$ increases concept heterogeneity purely by angular dispersion, without changing the optimizer norms. This process is illustrated in Figure 4. In our experiments we fix $R_\star = 1$.

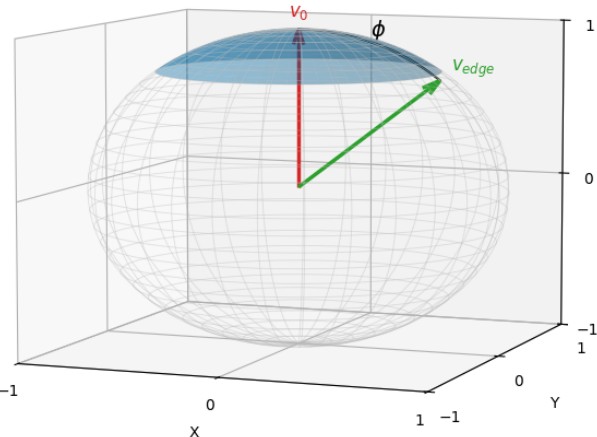

Figure 4: Illustration of sampling unit vectors from a spherical cap. We draw a cross-section of the unit sphere (circle), mark the central axis $v_0$, and show the cap of half-angle $\phi(\zeta_\star)$ (shaded blue).

## J.3 Controlling second-order (covariate) heterogeneity

Likewise, we fix each covariance matrix to $I_d$ and fix the norm of the feature mean to $\|\mu_m\| = \mu_0$. To vary the maximum pairwise mean distance $\max_{m,n} \|\mu_m - \mu_n\| = \tau$, we sample

$$\mu_m = \mu_0 \, u_m, \quad u_m \in \mathbb{R}^d, \; \|u_m\| = 1,$$

with $u_m$ drawn uniformly from the spherical cap of half-angle

$$\theta(\tau) = \arcsin\left( \tau / (2\,\mu_0) \right)$$

around the same central direction $v_0$. Again, this rotates the means without altering $\|\mu_m\|$ or the eigenvalues of the Hessians $\nabla^2 F_m = \mu_m \mu_m^\top + I_d$, whose condition number remains $1 + \mu_0^2$. In our experiments we fix $\mu_0 = 5$.

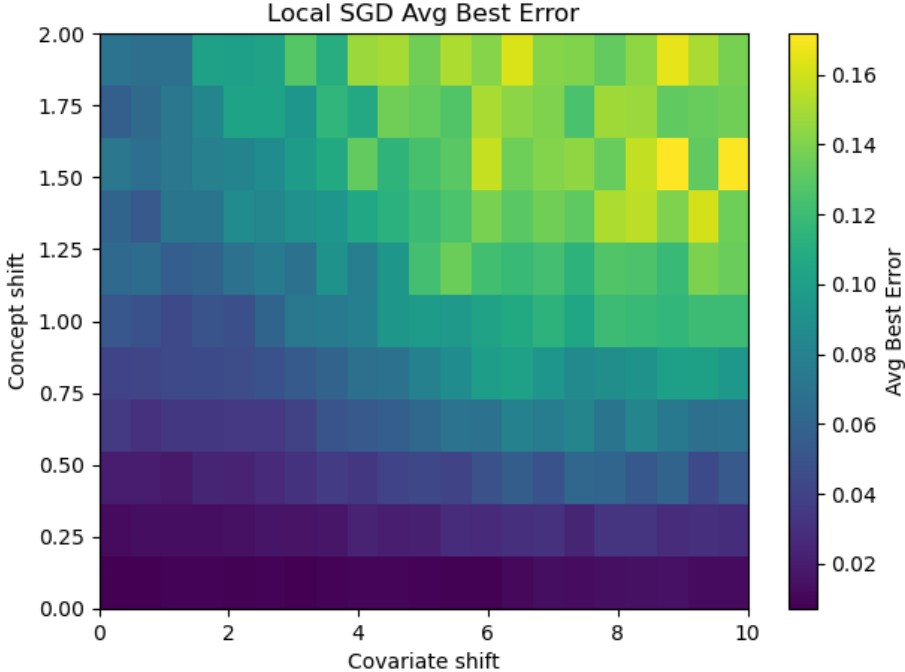

Figure 5: Same experiment as in Figure 2a but with $R = 10$, i.e., more communication rounds. Note that the heatmap colors scale changes.

## J.4 Hyper-parameter tuning and metrics

For every experimental setting $(\tau, \zeta_\star)$ (or every $\tau$ in the communication-complexity study) we first sample $v_0$ and sample $\{x_m^\star\}$, then we perform 20 independent trials with fresh draws of $\{\mu_m\}$. In each trial we search over a logarithmic grid of step-sizes $\eta \in [10^{-3}, 10^{-1}]$ and record either:

- The final $\ell_2$ error $\|x^R - \bar{x}^\star\|$ after $R$ rounds (for the heatmap in Figure 2a), or
- The minimum number of rounds $r \leq R_{\max}$ needed to reach $\|x^r - \bar{x}^\star\| \leq \epsilon$ (for the communication plot in Figure 2b).

We then average these quantities over the $n_{\mathrm{runs}}$ trials to obtain the plotted heatmaps and curves.

## J.5 Ensuring fixed problem difficulty

By sampling $\{x_m^\star\}$ and $\{\mu_m\}$ on fixed-radius spheres and using identity covariances, we keep every client's Hessian condition number and solution norm constant, so that any change in convergence or communication cost is attributable purely to the angular dispersion (i.e. heterogeneity) parameters $\tau$ and $\zeta$, not to changes in problem conditioning or scale.

## J.6 Experiments with More Machines and Communication Rounds

We include additional experiments in Figures 5 to 7 illustrating the effect of having more machines and communication rounds in the experiments of Figure 2. We note that as suspected increasing the number of communication rounds, makes the effect of data heterogeneity less drastic as we can see even at higher data heterogeneity, we are able to attain a better final error in Figure 5. On the other hand, we see that increasing the number of machines while keeping the communication rounds has a mixed effect as perhaps at this level of parallelization the SGD noise is still dominating as can be seen in Figure 6. Having said that we again see the benefit of increasing communication rounds at increased parallelization going from Figure 6 to Figure 7. We also run the analogue of Figure 2b with $M = 50$ in Figure 8.

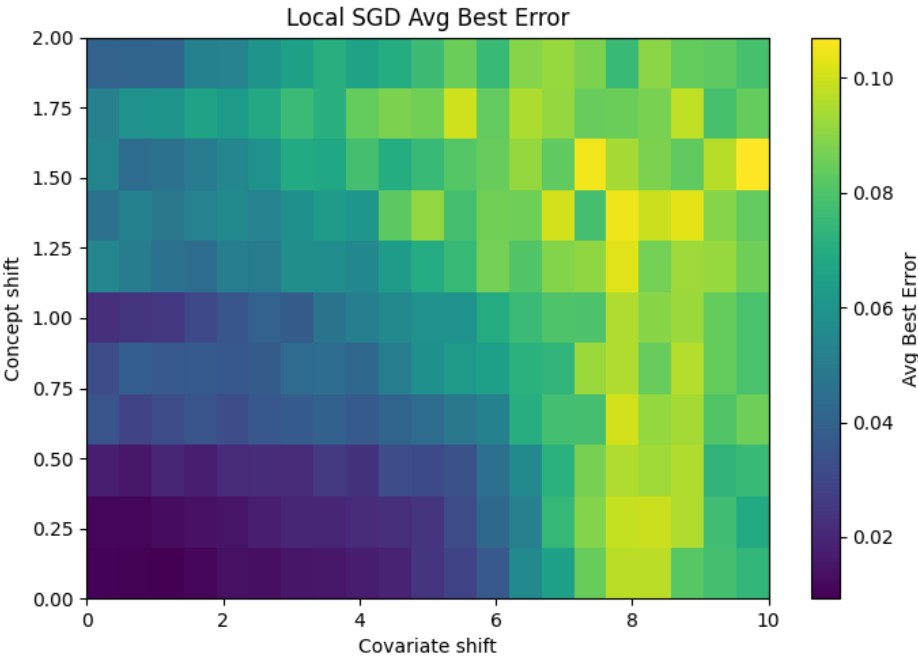

Figure 6: Same experiment as in Figure 2a but with $M = 50$, i.e., more number of machines.Note that the heatmap colors scale changes.

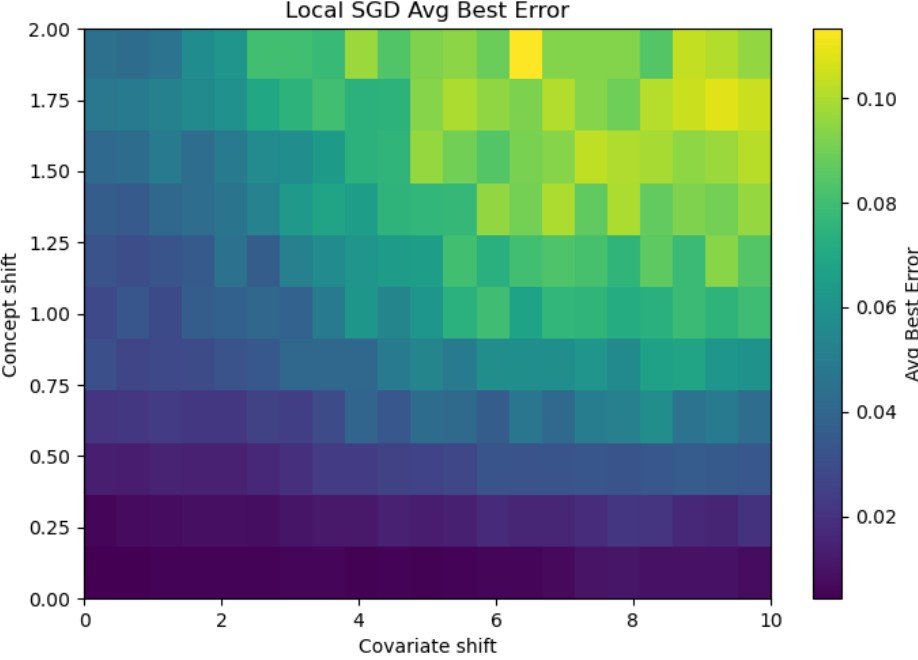

Figure 7: Same experiment as in Figure 2a but with $M = 50$ and $R = 10$. i.e., with more number of machines as well as communication rounds.Note that the heatmap colors scale changes.

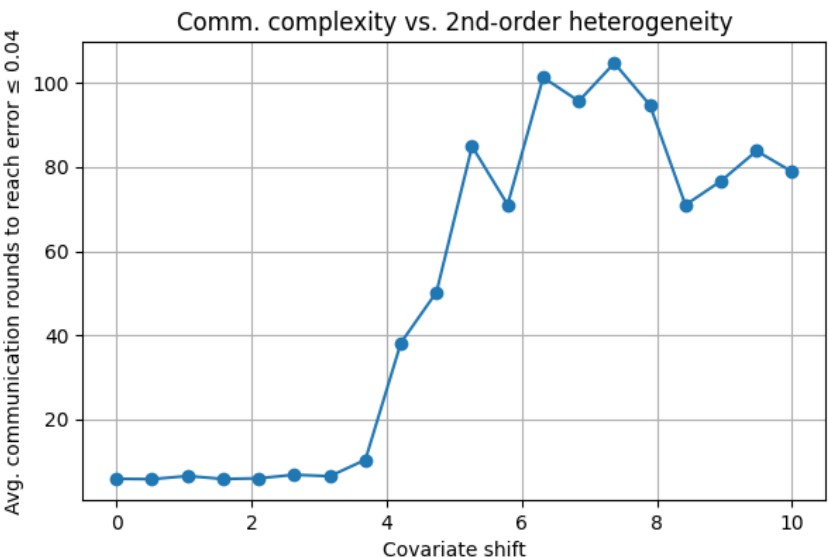

Figure 8: Same experiment as in Figure 2b but with $M = 50$.

