# OpenReview forum: "Revisiting Consensus Error: A Fine-grained Analysis of Local SGD under Second-order Data Heterogeneity"
_NeurIPS.cc/2025/Conference — NeurIPS 2025 poster_

### Official Review · Reviewer_jRiD · 2025-06-19

**Clarity:** 3
**Significance:** 3
**Originality:** 3
**Rating:** 5
**Confidence:** 4

**Summary:**

A new analysis of the convergence properties of Local SGD, a.k.a. Federated Averaging, is provided.

**Questions:**

See above

**Ethical Concerns:**

["NO or VERY MINOR ethics concerns only"]

**Final Justification:**

The authors have made a significant effort in their rebuttal to address the points raised by the reviewers, including myself, in a way that is satisfactory to me. So, I am increasing my score from 4 to 5.

**Limitations:**

Yes

**Quality:**

3

**Strengths And Weaknesses:**

Local SGD-type methods are very important and I appreciate this kind of papers with an in-depth study of the convergence properties.

I would like to see a discussion of the following papers with important insights.

1) In a series of papers [A,B,C], Mangold et al. have studied the properties of Local SGD, showing that linear speedup can be shown with a careful analysis.

2) Variance-reduction is not discussed. In my opinion the Scaffnew algorithm [D,E] is the state of the art of local training type methods. So, heterogeneity / client drift can be tackled using control variates, in the convex setting (the relevance of variance reduction in local SGD-type methods for nonconvex problems remains an open question). So, the real question is how the error of the stochastic gradients behave.

3) In the deterministic setting, the error of Local GD has been characterized in [F]. It depends on the gradients of the functions $F_m$ at the global solution $x^\star$, which is a natural measure of heterogeneity, also standard in the analysis of SGD. So, I don't think the Euclidean geometry on the primal solutions, as considered in Assumptions 3-5, is the right criterion. It should be more the first-order information on the gradients that matters.




[A] Magold et al. "Scafflsa: Taming heterogeneity in federated linear stochastic approximation and td learning" 2024

[B] Mangold et al. "Refined Analysis of Constant Step Size Federated Averaging and Federated Richardson-Romberg Extrapolation" 2025

[C] Mangold et al. "Scaffold with Stochastic Gradients: New Analysis with Linear Speed-Up" 2025

[D] Mishchenko et al. "ProxSkip: Yes! Local Gradient Steps Provably Lead to Communication Acceleration! Finally!" 2022

[E] Malinovsky et al. "Variance Reduced ProxSkip: Algorithm, Theory and Application to Federated Learning" 2022

[F] Malinovsky et al. "From local SGD to local fixed point methods for federated learning," 2020

---

> ### Author Rebuttal · Authors · 2025-07-30
>
> We appreciate the reviewer taking the time to evaluate our paper.
>
> Before we being to answer the reviewer’s concern, we would like to re-emphasize: our goal in this paper is to study Local SGD, and not other algorithms that are based on Local SGD but add additional algorithmic blocks, such as use of proximal oracles, control variates, variance reduction, or any other correction to alleviate its fixed-point discrepancy/bias. While we acknowledge that all of these are important and have their own place in the literature on federated optimization, it is clear to us that, partly due to its simplicity and partly due to its practical effectiveness, **the most commonly used algorithm in practice remains Local SGD**. And despite over a decade of work, there are clearly still many theoretical gaps in our understanding of Local SGD, as highlighted in our paper and in many other papers that preceded it. Our goal is to provide at least some more clarity on these issues.
>
> The papers the reviewer mentioned also acknowledge this gap, and papers such as those of Wang et al. [6, 8] highlight, for the same reason, that **the unreasonable effectiveness of Local SGD** is an important (if not the most important) gap in the theory of Federated Optimization. We believe that both our insights and analysis will complement the algorithms with other primitives. With this clarification about the scope of our work, we will now address the reviewer’s request to compare and contrast our work with existing works.
>
>
>
> ## On [A, C, D, E], Variance Reduction and Control Variates
>
> Of the three papers due to Manigold et al., A and C are not directly concerned with Local SGD, but with other algorithms, including SCAFFOLD and its variants. SCAFFOLD is a very interesting algorithm that popularized the use of control variates to correct for the effect of data heterogeneity, but as pointed out by the experiments in "Adaptive Federated Optimization" by Reddi et al., and further underlined by Wang et al. [8]: **SCAFFOLD and FEDAVG have roughly identical performance on many realistic federated datasets**. This is a trend we often see in the practice of machine learning, where simpler algorithmic primitives often work equally well or better (when accounting for the difference in hyperparameter tuning budgets). In particular, it is important to highlight that, at least in practice, it is not the case that
>
> > heterogeneity / client drift can be tackled using control variates
>
> Based on common wisdom (including from industry practitioners), Local SGD performs very well in practice, while SCAFFOLD (and other more complex control variate-based algorithms) tends to perform less well than expected (see Wang et al. [6, 8]). This further strengthens the case for providing a more fine-grained analysis of Local SGD, and we believe A and C are sufficiently outside the scope of our paper. Similarly, ProxSkip [D] (which we are **very familiar** with) and its follow-ups (including [E]) are interesting algorithms in their own right, but they are distinct algorithms and are nowhere near as widely adopted as Local SGD in practical settings.
>
> Regarding Variance reduction, and the reviewer’s comment:
>
> > the relevance of variance reduction in local SGD-type methods for nonconvex problems remains an open question
>
> We discuss and cite Patel et al. [30], which uses variance-reduced variants of Local SGD to provide almost min-max optimal algorithms in the non-convex setting. In fact, second-order heterogeneity is crucial for demonstrating improvement using variance reduction. This paper, along with [25], in the non-convex setting, served as motivation for our work. We acknowledge the important role of Variance reduction and control variates in Federated Learning. However, these papers [30, A, C, D, E] are not directly comparable to us and only provide interesting motivation. That said, we hope that our analyses can inspire future studies on algorithms that utilize variance reduction in even convex settings when the second-order heterogeneity is low: thus bridging the gap to Patel et al. [30].
>
> ## On [B, F], Fixed-Point Discrepancy, and the "Correct Geometry"
>
> B is a more relevant paper due to Manigold et al., because it upper bounds the fixed-point discrepancy of Local SGD and attempts to provide a new analysis using it. While the paper offers a convergence guarantee (with untuned step sizes) in the deterministic setting, **it does not tune the step sizes to achieve a convergence rate in the stochastic setting**. On the other hand, our goal is to give explicit convergence guarantees, with tuned parameters, to fill any gaps in our understanding of the min-max optimal guarantee for Local SGD.
>
> In our experience, using the fixed-point discrepancy to obtain convergence rates for Local SGD (as discussed in Appendix C) is hard. While bounding the discrepancy itself is useful and can offer important insights (as we also do in our Appendix C.4), **balancing convergence and discrepancy/bias is challenging** due to the complex terms involved in the upper bound of the discrepancy. This is why neither we nor Patel et al. [25] rely on a fixed-point-based approach, but instead use a consensus error-based analysis. It is also why Mangold et al. [B] do not provide a general guarantee for strongly convex functions explicitly: for example, their Section 5 either considers quadratic functions or the homogeneous setting, and in neither case, despite simplifications, does it provide a fully tuned convergence rate for Local SGD.
>
> Regarding Malinovsky et al. [53], we cite the paper and include a full section on the fixed-point discrepancy in Appendix C, where we also discuss it several times. The key difference with respect to their paper is that we want to emphasize the discussion on data heterogeneity assumptions, which is why, like Patel et al. [25], we also try to bound fixed-point discrepancy in terms of our data heterogeneity assumptions. Malinovsky et al. [53]'s result also suffers from the same issue as Mangold et al. [B] (as discussed above), namely, finding it difficult to provide explicit convergence guarantees in terms of data heterogeneity notions due to the need to balance discrepancy and convergence.
>
> > So, I don't think the Euclidean geometry on the primal solutions, as considered in Assumptions 3-5, is the right criterion. It should be more the first-order information on the gradients that matters.
>
> We are unsure what the reviewer means by Euclidean geometry not being the right criterion. It is well known that any convergence guarantee for SGD must rely on some version of Assumption 3, or another notion of initial sub-optimality. It is true that in some contexts, convergence rates can be provided in terms of initial function sub-optimality, such as when providing convergence rates for stationary points (cf. "The Complexity of Making the Gradient Small in Stochastic Convex Optimization" due to Foster et al.). However, at least for strongly convex problems, one cannot hope to recover the optimum without bounding their Euclidean norm.
>
> Once it becomes clear that something akin to Assumption 3 is necessary, Assumptions 4-5 are just corresponding heterogeneity conditions that bound the gap between the solution sets of different clients. Having said that, it might be possible to state some of our results in terms of assumptions about the gradients at the optima. For instance, using Assumption 5 and 1 (smoothness), note that we can get the following for any $x^\star \in \arg\min_{x}F(x)$, $$\frac{1}{M}\sum_{m\in[M]}\|\|\nabla F_m(x^\star)\|\|_2^2 \leq H^2\phi^2\enspace.$$
>
> This is essentially the first-order heterogeneity assumption studied in works such as Kolosokova et al. [18] and Patel et al. [25], among others. A similar, but not identical, version of first-order heterogeneity was studied by Wang et al. [8]. As such, our assumptions and the usual first-order assumptions at the optima are related; ours are simply more fine-grained (for instance, the demarcation between Assumptions 4 and 5, as discussed in Remark 1). Since our upper bounds are for the strongly convex setting, it is unclear if relying on the usual, more coarse first-order assumptions is any more insightful than using Assumptions 4 and 5.
>
> Similarly, since Local SGD is a first-order algorithm, it is natural to consider the second-order heterogeneity assumption in Assumption 6, which measures the difference between the "geometries" of different machines. It is possible to define a different geometry that also accounts for local update steps, but we are not trying to find the best geometry for our assumption, as ultimately, our goal is to provide explicit convergence guarantees. We need to balance **how descriptive a geometry/assumption is and our ability to use it in our analysis**. The fixed-point perspective based analyses fall short here: while they provide very explicit operators characterizing fixed points and bias, it becomes challenging to give a simple convergence rate.
>
> If the reviewer has a specific heterogeneity assumption or a geometry they would like us to comment on, we are happy to discuss it during the rebuttal phase.
>
>
> ## Summary
>
> We propose adding a remark about the techniques that rely on bounding the fixed-point discrepancy of Local SGD to give an analysis, and discuss [A, B, C, F] in this context, while deferring the reader to a more detailed discussion in Appendix C. Regarding [D, E] we will include them in our paper while discussing [30] and broadly the effect of variance reduction. **Would this be satisfactory for the reviewer?**
>
> We hope that this rebuttal, along with **our detailed discussion of these papers in our revision**, will address the reviewer’s concern and encourage them to reconsider their score. We are happy to answer any other questions the reviewer may have, and we are looking forward to engaging during the discussion phase.

---

> > ### Comment · Reviewer_jRiD · 2025-08-02
> >
> > I appreciate the efforts of the authors to reply precisely to all points raised by the reviewers. All right, I am satisfied if the authors include the details and references provided in the rebuttals in their revised paper. I am increasing my score to 5.

---

> > > ### Author Response · Authors · 2025-08-03
> > >
> > > Thanks for raising the score and for mentioning the references again! We will ensure that we include all the details mentioned in the rebuttals to all reviewers, including a discussion of additional papers. We will also add more context and discussion in the appendix where needed.

---

### Official Review · Reviewer_9Ysu · 2025-06-21

**Clarity:** 2
**Significance:** 3
**Originality:** 3
**Rating:** 5
**Confidence:** 4

**Summary:**

This paper provides a refined convergence analysis of Local SGD under the second-order heterogeneity and the third-order smoothness settings.

**Questions:**

**Questions**:

This paper gives a more fine-grained analysis of Local SGD. This may lead to different conclusions compared to the existing works in FL. Recently, [+2] showed that sequential FL is better than parallel FL (Local SGD) when the heterogeneity is relatively high, by comparing their bounds of sequential FL with the bounds of Local SGD [+1]. I am curious to know whether the new bounds (established in this paper) will cause a different conclusion compared to [+2]. Could you kindly compare the bounds between sequential FL and Local SGD? If possible (the time of rebuttal is enough), please provide some preliminary experiments (on quadratics or linear regression). I think the comparison can help improve the impact of this paper.

[+2] Sharp Bounds for Sequential Federated Learning on Heterogeneous Data, JMLR, 2025.

**Suggestions**:

1. Figures: It would be better to use PDF format for figures.

2. Comparison with the related works: It would be better to add the tables to present the differences (of the upper and lower bounds) compared to the existing works like Patel et al. [25]'s Table 1. They can be added in an additional page (if accepted) or in the appendices.

3. Assumptions: Currently, the assumption $\mathbb{E} ||\nabla f(x;z) - \nabla F_m(x) ||_2^2 \leq \sigma^2$ is seen as one part of Assumption 2. Since this assumption has appeared multiple times in the proofs, I recommend to write it as an independent assumption.

4. Lines 222 and 228: It would be better to use `\text{MB-SGD}` and `\text{L-SGD}` in LaTex to improve the readability.

5. Appendices: The current organization of the appendices are not clear, especially the sections of the upper bounds (Appendices E-J). The titles like "Double Recursion for Consensus Error" are not friendly for the readers who are unfamiliar with the settings. I recommend to use "Proof of Theorem 1" (see [+3]'s TOC of appendices) and put the current titles at the beginning of the corresponding sections to explain the purpose of the sections.

6. Proofs: Adding some sketches of the proofs can improve the readability (see [+1]).

7. Lemma 21: It would be better to add the proof of Lemma 21 for completeness.

[+3] Minibatch vs Local SGD with Shuffling Tight Convergence Bounds and Beyond, ICLR, 2022.

**Score**:

1. Currently I give the score of 4. This is only because I have not yet checked all the proofs (I mainly checked the proofs of the upper bounds), due to the very heavy workload of NeurIPS. I know this is unfair for this good paper, so I will also consider the reviews of the other reviewers.
2. However, I cannot guarantee a higher score (i.e., 6) for now, as this topic (the unreasonable effectiveness of Local SGD) has been studied for several years and this theory is still limited to strong convexity.

**Ethical Concerns:**

["NO or VERY MINOR ethics concerns only"]

**Final Justification:**

The authors' rebuttal addressed my concerns.

Considering the revisions promised by the authors during the rebuttal period, I tend to give a positive rating.

**Limitations:**

yes

**Quality:**

3

**Strengths And Weaknesses:**

**Strengths**:

1. This paper provides a more refined analysis for Local SGD compared to Patel et al. [25].
2. The topic of explaining the unreasonable effectiveness of Local SGD is important.
3. The amount of work in this paper is sufficient, and the theory is solid and rigorous.

**Weaknesses**:

I have not found any serious weaknesses. Some minor problems (and questions) are as follows:

1. 1-st line of the equation between Lines 1209 and 1210: Should "$+$" be "$-$" before $2\langle\xi_t^m,\xi_t^n \rangle$?

2. 1-st line of the equation between Lines 1213 and 1214: Same as Point 1.

3. 1-st inequality of the equation between Lines 1213 and 1214: It is well known that $(x-y)^4 = x^4 - 4x^3y + 6x^2y^2 - 4xy^3 +y^4$. Thus, is it necessary to use Cauchy Schwartz inequality for $\mathbb{E}[{(\langle\xi_t^m,\xi_t^n \rangle)^2}]$ here?

4. 3-rd line of the equation between Lines 1229 and 1230: Should "$=$" be "$\leq$"?

5. 2-rd inequality of the equation between Lines 1253 and 1254: It uses $||x_t - \eta \nabla F(x_t) - x^\star + \eta \nabla F(x^\star) || \leq (1-\eta\mu)^2 ||x_t - x^\star||^2$ here. In fact, it is well known that $||x_t - \eta \nabla F(x_t) - x^\star + \eta \nabla F(x^\star) || \leq (1-\eta\mu) ||x_t - x^\star||^2$ (see [+1]'s Lemma 6). So I want to know whether this trick helps improve the existing bound. If not, why we use it here?

6. 6-th inequality of the equation between Lines 1278 and 1279: Should "$(\nabla^2 F_m(\hat x_t^m) - \nabla^2 F_m(x_t) + \nabla^2 F_m(x_t) - \nabla^2 F(x_t)  )$" be  $(\nabla^2 F_m(\hat x_t^m) - \nabla^2 F_m(x_t) + \nabla^2 F_m(x_t) - \nabla^2 F(x_t) + \nabla^2 F(x_t) )$? Is one term $\nabla^2 F(x_t)$ missing?

7. Line 1279: What does "$\in m \mathcal{H}_t$" mean?

8. The last equality of the equation between Lines 1293 and 1294: The "$=+$" is a typo.

9. Line 1295: The $\eta$ before $\nabla F_m(x_t^m)$ is missing.

10. Line 1302: Should "Using Lemma 16" and "taking a whole square" be exchanged?

11. Line 1362: Where the fact $(1-\eta \mu)^K\leq \frac{1}{2}$ is used?

12. Line 1371: The phrase "holds the hides" seems a typo.

13. Line 1394: The definition of $\gamma_j = j- \delta(j)$ is inconsistent in the context. Should it be $\gamma_j = j-\delta(t)$? In addition, the sentence "note that the term with $1/\gamma_{t-1}$ becomes 1 when $t-1=\delta(t)$" is also unclear.

14. 3-rd inequality of the equation between Lines 1395 and 1396: Is $\sum_{j=\delta(t)}^{t-1}\frac{1}{j+1-\delta(t)}$ be upper bounded by $\ln (t-\delta(t))$? In fact, I think it is upper bounded by $1+\ln(t-\delta(t))$ if using the integrals. In addition, where the fact $..\leq \frac{1}{\eta \mu}$ before Line 1396 is used?

15. Line 1401: $\delta(j) = K(r-1)$ is unclear. Should it be "$\delta(j) = K(r-1)$ when $K(r-1) \leq j \leq Kr-1$"?

[+1] SCAFFOLD Stochastic Controlled Averaging for Federated Learning, ICML, 2020.

If the problems I raised above are wrong, please free free to point them out.

---

> ### Author Rebuttal · Authors · 2025-07-30
>
> We thank the reviewer for their time and effort in reviewing our paper. Regardless of the decision, the reviewer's comments will be very useful to us in improving our paper. Not only did the reviewer carefully read through the main body, but also several parts of the appendix. We truly appreciate it!
>
> Since the reviewer stated that they did not find any major weaknesses in the paper and did not raise any major technical issues, we will attempt to address the questions the reviewer raised about sequential FL and also respond to the issues/typos in the writing.
>
> ## Addressing Listed Weaknesses
>
> 1, 2, 3: We agree with these comments. We apologize for this error with the sign. We will correct the Lemmas. The reviewer is right, one could state a simpler proof for these lemmas.
>
> 4: Correct, we will add an upper bound after the first line in this series of equations. The conclusion should still follow.
>
> 5: We suspect the reviewer missed a whole square on the LHS in their comment (based on looking at Lemma 6 in SCAFFOLD). Having the $(1-\eta\mu)^2$ term, as opposed to $(1-\eta\mu)$, is important, as we currently do it using Lemma 16 with a specific value of $\gamma$. Note that it is easy to see (b) by just applying the mean value theorem to $\nabla F(\cdot)$ (to quickly verify that the functions were quadratic). There may be a way to utilize Lemma 6 of SCAFFOLD in conjunction with a different version of Lemma 16; however, the approach we take is effective in achieving the desired rate. If we misunderstood this comment, please let us know!
>
> 6: No, the additional term the reviewer mentioned would just be zeroed out because $\frac{1}{M}\sum_{m\in[M]}\nabla^2 F(x_t)\cdot(x_t - x_t^m) = \nabla^2 F(x_t)\cdot\frac{1}{M}\sum_{m\in[M]}(x_t - x_t^m) = 0$. It is our fault; we should have made that more explicit!
>
> 7: It means the random variable is measurable under the sigma algebra generated by $\mathcal{H}_t$. Perhaps we should have mentioned this notation before using it!
>
> 8, 9, 10: Thanks for catching these.
>
> 11: That shouldn't have been there (it's from an older version of the analysis), we are just using $1 - (1-\eta\mu)^K < 1$. Thanks for catching this!
>
> 12: Yes. It should just be: "hides the".
>
> 13: Note that for $j$ between two communication rounds, they are the same, as $\delta(j) = \delta(t)$.
>
> 14: Yes, our bad, it should be $1 + \ln(K)$, which of course only changes the numerical constant, but something we will correct. The $1/\eta\mu$ upper bound line should be removed; we were previously using it in the inequality above (a), but we ended up just upper-bounding $(1-\eta\mu)^{2(t-1-j)} - (1-\eta\mu)^{2(t-1-\delta(t))}$ by $1$.
>
> 15: Thanks, we will add that clarification about the range of $j$.
>
> ## On Suggestions about Writing
>
> 1: We will use the PDF format for potentially better load-up on readers.
>
> 2: We will include a table in the additional page. We have, in fact, created a table that is more exhaustive than that of Patel et al.
>
> 3: This is a good point. Initially, we used $\sigma$ for both assumptions for simplicity. However, after submission, we improved all our analyses to include different constants $\sigma_2$ and $\sigma_4$. One crucial takeaway from this analysis (which we would include in our revision) is that the $\sigma_4$ terms actually improve more from increasing $K$ than the terms with $\sigma_2$ do, something our current analysis ignores due to loosely upper-bounding the $\sigma_4$ terms. This is actually an interesting insight, because in practice, the fourth moment of the noise can be significantly larger than the second moment.
>
> 4: We will do that.
>
> 5: We agree with this. In hindsight, we did not expect the appendix to be this long, and the current nomenclature might have been sufficient for a shorter proof. Still, our hope was that Section E would provide some structure and support for a reader unfamiliar with the proof style. Perhaps the fact that this is not the first appendix section makes this harder already!
>
> 6: Again, we will restructure the current Section E so that it does a better job of providing a proof sketch.
>
> 7: We will do that.
>
> ## On Comparison to the Sequential Setting
>
> Thank you for bringing this work [+2] to our attention; it is indeed interesting, and we were not previously aware of it.
>
> However, hopefully the reviewer recognizes that in many settings where Parallel FL is used, sequential FL (at least in the most naive sense) cannot be implemented. We are considering cross-device settings (or even cross-silo settings with many servers), where the total physical time would be too extensive for Sequential FL. Having said that, it is likely that many cross-device settings with partial client participation are already utilizing a hybrid approach between Sequential and Parallel FL, due to the structure of client sampling around the globe. So the comparison between these extremes is interesting.
>
> Let us compare the strongly convex rates in Corollary 4 of [+2]. The function sub-optimality they provide (translated to our notation) is given by (up to poly-logarithmic problem-dependent factors) $$\text{\textbf{SFL (L-SGD):}}\quad \mu B^2 e^{-R/\kappa} + \frac{\sigma^2}{\mu MKR} + \frac{H\sigma^2}{\mu^2 MKR^2} + \frac{H^3\phi_\star^2}{\mu^2 MR^2}\enspace.$$
>
> Compared to this, our guarantee (in Theorem 3) gets the following rate:
> $$\text{\textbf{PFL (L-SGD):}}\quad \mu B^2 e^{-KR/\kappa} + \frac{\sigma^2}{\mu MKR} + \frac{\tau^2 H\phi_\star^2}{\mu^2R^2} + \frac{H^3\zeta_\star^2}{\mu^2R^2} + \frac{H\tau^2\sigma^2}{\mu^4KR^3}  + \frac{H\sigma^2}{\mu^2 KR^2} \quad \text{for} \quad R = \tilde\Omega(\tau\sqrt{\kappa}/\mu)\enspace.$$
>
> Depending on the heterogeneity parameters, **some terms in our rate may be better than theirs**: for instance, when $\ tau$ and $ \zeta_\star$ are small but $\phi_\star$ is large. But the opposite could also be true, because (quite surprisingly, perhaps due to their step-size) their heterogeneity term improves with larger $M$. This was not true previously (i.e., before our paper): as they highlighted in their paper, because their rates were strictly better than those of Kolosokova et al. for Local SGD.
>
> To make the comparison a bit simpler, let us compare their and our communication complexities in the regime with a very large $K\to\infty$:
>
> $$\text{\textbf{SFL:}}\enspace R = \tilde\Omega\left(\kappa + \kappa\sqrt{\frac{H\phi_\star^2}{M\epsilon}}\right) \quad \text{v/s} \quad \text{\textbf{PFL (our):}}\enspace R = \tilde\Omega\left(\frac{\tau}{\mu}\cdot\sqrt{\kappa} + \frac{\tau}{\mu}\cdot\sqrt{\frac{H\phi_\star^2}{\epsilon}} + \kappa\cdot\sqrt{\frac{H\zeta_\star^2}{\epsilon}}\right)\enspace.$$
>
> The above rate clearly re-emphasizes that in the regime with a large $\kappa$ and $\phi_\star$ but a smaller $\zeta_\star$ and $\tau$ (cf. Remark 1), our communication complexity is better. Is it possible that their rates can benefit from a lower $\tau, \zeta_\star$? **Yes, we think so (at least with strong convexity).**
>
> Here is what we propose: we will include a remark about the sequential FL setting, highlighting how our results alter the comparison to parallel FL presented in [+2]. Additionally, we are willing to include a section in our appendix that provides variants of the results of [+2] (both upper and lower bounds) while incorporating second-order heterogeneity. Upon reviewing their proofs, we believe this is possible (for a subset of their results) and will be a valuable addition to our paper. This should require approximately 6-7 pages in the appendix, as we can utilize some parts of our theoretical framework already. **Would the reviewer be more satisfied with this change?**
>
> Unfortunately, due to the new rebuttal guidelines, we cannot include any new experiment plots. However, we are running experiments in the setup of Figure 1 and will report the qualitative results to the reviewers during the rebuttal phase. We are also curious if SFL similarly benefits from having a lower second-order heterogeneity.
>
> ## On the Need for Strong Convexity
>
> > However, I cannot guarantee a higher score (i.e., 6) for now, as this topic (the unreasonable effectiveness of Local SGD) has been studied for several years, and this theory is still limited to strong convexity.
>
> We would like to emphasize that we have made our best effort to incorporate results in the general convex setting; however, the simplest trick of using a general-convex-to-strongly-convex reduction does not work (see lines 244-254). We provide a more detailed explanation of why strong convexity was so useful in our coupled recursions in our response to reviewer un1E. So far, all our attempts to rely solely on convexity to obtain a variant of Lemma 2 have not been successful. We understand the reviewer's disappointment that our results are limited to the strongly convex setting, but it is worth noting that our proofs are already quite complex. Given the attention the analysis of Local SGD has received, this only underscores that extending it to the convex setting is not trivial, and that a gap may exist here.
>
> ## Closing Remarks
>
> > Currently I give the score of 4. This is only because I have not yet checked all the proofs (I mainly checked the proofs of the upper bounds), due to the very heavy workload of NeurIPS. I know this is unfair for this good paper, so I will also consider the reviews of the other reviewers.
>
> We appreciate the reviewer's honesty and empathize with them about the reviewing load at NeurIPS! However, we hope the reviewer will consider both other reviews and **our detailed rebuttals** to their concerns before forming an unbiased personal opinion on the paper. This is especially important because some of the reviews contain major misunderstandings about our setting and results, which we do not believe applies to the reviewer, given that they have read our analyses carefully. Since our paper is currently borderline, the reviewer's final score could influence the decision.

---

> ### Comment · Reviewer_9Ysu · 2025-08-04
>
> Thanks for the hard work. The authors' rebuttal addressed my concerns. Overall, I am satisfied with the responses, even for the comparison with sequential FL.
>
> **Further suggestions**:
>
> 1. I hope the authors revise the paper according to the responses as they have promised.
> 2. I hope the authors check the proof more carefully in the revised version. Try to avoid some simple typos, especially in the appendix.
> 3. It is hard for the reviewers to confirm the correctness of this paper given its complexity, so I hope the authors try their best to respond if future researchers raise concerns about their work.
>
> **Rating**:
>
> The final rating will be made in Reviewer-AC Discussions, after I read the reviews and responses of other reviewers. Considering that I have already given a positive rating for this paper, I need to speed more time on other papers.

---

> > ### Author Response · Authors · 2025-08-04
> > **Thanks!**
> >
> > We thank the reviewer again for their comments, which greatly helped us improve our paper. We are pleased that we were able to address the reviewer's concerns in the rebuttal phase. We will indeed include all promised additions to the paper in our revision.

---

### Official Review · Reviewer_MhSC · 2025-06-22

**Clarity:** 1
**Significance:** 3
**Originality:** 2
**Rating:** 3
**Confidence:** 5

**Summary:**

This paper investigates the convergence behavior of Local SGD in the presence of heterogeneous data across clients. The authors focus on understanding how second-order heterogeneity affects the performance and communication complexity of Local SGD. The main contributions are:

1. A new lower bound that quantifies how convergence degrades with increasing second-order heterogeneity.

2. A new convergence rate for local SGD.

3. A synthetic regression experiment validating the theoretical predictions.

**Questions:**

Some minor concerns are listed here.

1. In line 15, the authors state that "$F_m$ is the stochastic objective". This is not accurate: there is no stochasticity in $F_m$.

2. In line 110, the authors state that "without this condition, some clients may not benefit from collaboration". This argument is vague to me. In terms of convergence rate, distributed/decentralized methods are almost always slower than the vanilla counterpart. "Collaboration" is necessary for other (realistic) reasons.

3. In lines 133-136, the authors mention "distributed proximal-point methods" and cite [50]. But [50] proposes a Newton-type distributed method.

4. In line 329, what does it mean by "monotonic relationship"? The plot is not monotonically increasing.

**Ethical Concerns:**

["NO or VERY MINOR ethics concerns only"]

**Final Justification:**

This paper contains some nontrivial contributions, but I believe substantial improvements in writing are needed, ideally under close supervision. The work attempts to present three contributions within a single paper, but lacks cohesion and coherence. I suggest the authors improve the overall flow and structure, aiming to present a more complete and unified narrative.

It is regrettable that the authors expressed a strong perspective in their response to my concerns, yet may have deliberately chosen not to submit it as an official rebuttal. This approach risks undermining the transparency and fairness of the review process, and, even if the paper were accepted, could diminish the perceived credibility and impact of the work.

**Limitations:**

yes

**Paper Formatting Concerns:**

Some LaTeX conventions in the paper are not standard. For example, $<<$ should be $\ll$.

**Quality:**

2

**Strengths And Weaknesses:**

**Strength**

I like section 6 which presents simple synthetic experiments to show the motivation.

**Major concerns**:
My major concerns lie in the theoretical results.

1. Theorem 1 presents a lower bound for **convex quadratic** functions.

(a) Does the same lower bound hold for **any** functions satisfying the blanket assumptions? If not, the authors may consider to state their first contribution more precisely in Section 1 to avoid overclaiming.

(b) In line 151, the authors state that "when $\phi_\star$ is small and $K$ is large, the lower bound is dominated by the term $\tau B^2 / R$". I cannot totally agree with it. In the extreme case where $\phi_\star = 0$, all the local quadratic functions have the same optimum, so communication is not needed at all. Then, without stochasticity, the convergence rate is lower bounded by the worst conditional number of all the local functions, not $\tau$, and the convergence rate should be linear, not $\mathcal{O}(1/R)$. So the statement is somewhat pointless or inaccurate.

Another case is $\phi_\star$ is small but nonzero. Then when $K \to \infty$, agents never communicate and the algorithm never converges. This phenomenon, again, is not captured by the term $\mathcal{O} (\tau B^2 / R)$.

(c) In line 158, the authors state that "we extend the construction of Patel et al. by introducing an additional dimension". I was only able to understand this sentence after reading the appendix, which indicates a need for clearer exposition in the main text. In addition, this sentence does not justify why we need $\tau$ instead of other quantities, e.g., the worst condition number of all local quadratic functions; it seems the choice of $\tau$ is arbitrary...

2. The authors claim that Lemma 2 is novel and distinct from existing literature, but I have reservations about this assertion. In a recent line of research on decentralized optimization [1,2], a similar proof technique has been introduced. In particular, those papers consider using graph sequences with finite-time consensus in decentralized optimization. So consensus error will somehow "disappear" periodically due to the suitable choice of communication networks. The same proof technique has also been utilized in local updates (see, e.g., [3] and references therein). The authors may consider to compare the current work with this line of research.

(a) Lemma 2 requires $\eta < 1/H$, but in line 213, the authors state that "our bound does require setting $\eta = \mathcal{O}(1/K)$".

(b) In line 214, the authors state that "an alternative bound in Appendix I.3 that avoids this ($\eta = \mathcal{O}(1/K)$)". However, I find the bound in Lemma 32 not informative at all: the consensus error will never vanish under the stated stepsize condition. Lemma 32 simply says that the difference between $x_m^t$ and $x_n^t$ is bounded by that between $x_m^\star$ and $x_n^\star$, which is trivially true because $x_m^t$ converges to $x_m^\star$ as $K \to \infty$. If the algorithm does not converge with this stepsize choice, it is unclear what value these results provide.

3. Theorem 2 states the convergence rate of local SGD.

(a) How should I compare Theorem 1 and Theorem 2? Does the lower bound in Theorem 1 match the upper bound Theorem 2? If they are incomparable, then what is the connection between Sections 3 and 4?

(b) Assumption 1 says $\mu \geq 0$, but Theorem 1 certainly fails when $\mu=0$, as pointed out in lines 244-249.

(c) Assumption 1 imposes $Q$-Lipschitz continuity of the Hessian, which, I believe, is referred to as “third-order smoothness” in the paper. However, Theorem 2 does not involve $Q$ at all. Does this imply that $Q$ has no impact on the convergence rate? If so, then what is the purpose of Section 5 then?

(d) In lines 226-227, the authors claim that "with a small $\tau$, Local SGD can converge much faster than mini-batch SGD". I am not convinced by this claim and believe it requires further justification. First, the dominating term in both rates is $\sigma^2 / (\mu^2 M K R)$ rather than the exponential term.  As such, the two rates appear essentially the same. the parameter $K$ has different interpretations in Local SGD and mini-batch SGD, so I do not believe the two are directly comparable. Therefore, the additional factor of $K$ in the exponential term of Theorem 2 does not carry much significance in my view.

(e) In lines 227-228, the authors claim that "when $K \to \infty$, the communication complexity of Local SGD for target accuracy $\epsilon$ and large $K$ satisfies ..." However, this is misleading. As $K \to \infty$, communication effectively ceases, and Local SGD will not converge to the global optimum. Therefore, the limit $K \to \infty$ does not reflect a meaningful or practical convergence behavior.

4. In Section 5, the authors restrict the attention to quadratic objectives and provide an upper bound involving $Q$ (defined in Assumption 1).

(a) How should we compare Theorem 2 and Theorem 4. Logically speaking, the bound in Theorem 4 should be no larger than that in Theorem 2. Is that correct?

(b) How should we compare Theorem 1 and Theorem 4? Do the lower and upper bounds match? At least on the surface, they do not appear to align. How, then, should these two results be interpreted? Is one of them not tight, or are both potentially loose?

(c) I do not see the motivation to include $Q$ in the analysis. In other words, why should $Q$ be the correct measure, rather than, e.g., the condition numbers of all the quadratic functions?

In addition to the technical issues discussed above, I strongly encourage the authors to improve the overall clarity and presentation of the manuscript.

1. The introduction is difficult to follow, as it attempts to present too much information at once, which makes the narrative feel overloaded. I also recommend that the authors present the motivation of the work more clearly without too many technical details, so that readers can easily understand the problem setting and the significance of the results. A clearer and more focused introduction would greatly improve the readability and impact of the paper.

2. On first reading, I expected Section 3 to start presenting the main results of the paper. However, the section opens with an extensive discussion of related literature, which detracts from the focus and makes me really confused about the purpose of this section.

3. The authors frequently reference forward to materials that appear later in the paper. While occasional forward referencing is acceptable, excessive use can hinder readability and is generally discouraged in academic writing.

(a) For example, in the very first paragraph of Section 1, the authors refer to Figure 2, which appears only in the appendix. This raises confusion: if the figure is essential for understanding the setup, it should be included in the main text; otherwise, if it is non-essential, the reference should be removed to avoid disrupting the flow of reading.

(b) A similar issue arises in the second paragraph, where the authors refer to Equation (2) on page 2 before establishing sufficient context. Forward references like this can be disorienting for readers and should be used sparingly. I recommend reordering the content to introduce necessary concepts before referencing them.

[1] Ying et al. Exponential graph is provably efficient for decentralized deep training. NeurIPS, 2021.

[2] Nguyen et al. On graphs with finite-time consensus and their use in gradient tracking. SIAM J. on Opt., 2025.

[3] Alghunaim. Local exact-diffusion for decentralized optimization and learning. IEEE TAC, 2024.

---

> ### Author Response · Authors · 2025-07-31
> **Rebuttal 1/2**
>
> We appreciate the reviewer’s valuable feedback on our paper. We would like to address their concerns regarding the theoretical results in our work.
>
> ## Regarding Concern 1 and the Min-max Optimal Value
>
> Please note that the goal of our paper is to try to give upper and lower bounds for optimizing the worst function in a given problem class (determined by convexity, smoothness, heterogeneity, etc.) when solving using the best algorithm (in an algorithm class, which here is intermittently communication algorithms with first-order oracle access), i.e.,
>
> $$\min_{A}\max_{f, D_1,\dots,D_M} \frac{1}{M}\sum_{m\in[M]}\mathbb{E}_{z\sim D_m}[f(x^A;z)]\enspace.$$
>
> Min-max optimality is a useful framework to argue about worst-case guarantees for optimization algorithms. For more context, we urge the reviewer to review the PhD Thesis titled "The Minimax Complexity of Distributed Optimization" by Blake Woodworth.
>
> As such, to provide lower bounds for this **min-max optimal value**, it is **sufficient** just **to find one hard instance within the problem class**, as no algorithm that provides a convergence guarantee on the entire class can do better than what it can do on this specific hard instance. This is a confusion that appears in multiple places in the reviewer's comments, and hopefully, this clarifies many of the issues. We will now address how this answers specific reviewer concerns.
>
> > Does the same lower bound hold for any functions satisfying the blanket assumptions? If not, the authors may consider to state their first contribution more precisely in Section 1 to avoid overclaiming.
>
> No, it won't hold for every quadratic function satisfying these assumptions. But that is **not the goal of providing convergence lower bounds**. The lower bound, of course, uses a specific hard instance, but it essentially sets the expectation for what the upper bound can be (which applies to all functions that satisfy these assumptions). As such, we are not overclaiming anything in Section 1.
>
> > In line 151, the authors state that.....
>
> No, there is a significant misunderstanding about our assumption here, and we urge the reviewer to make sure **they are not thinking about strongly convex functions when reading Theorem 1 (which is for general convex functions)**. Even if **phi_\star=0**, it implies that the clients share an optimum, not that Local SGD will quickly converge to this shared optimum. As such, the assumption that an optimum is shared is insufficient to show that mini-batch SGD can be beaten by Local SGD due to the result of Patel et al. [25]. Hopefully, this addresses the reviewer's concern. We understand that this confusion may have been caused by using the same assumption 1 to discuss both convexity and strong convexity; we will clarify this by splitting them into separate discussions.
>
> > In line 158....
>
> First of all, many existing works already consider the second-order heterogeneity assumption, so that alone is a good enough reason to study it. Note that the worst condition number doesn't capture the heterogeneity of the problem. In particular, the problems across the machines could be very poorly conditioned, while the Hessians are all the same (which would imply $\tau=0$). However, our upper bounds (e.g., see Theorem 2 when $\tau=0$) indicate that this poor conditioning can be mitigated by employing more local updates. $\tau$  inherently measures the difference in the "geometry" between the machines, capturing both the difference in possible rotations and magnitudes of eigenvalues between the machines, which in turn affects the local updates. Hopefully, this clarifies why $\tau$ and $\kappa$ measure **different sources of hardness**: of which one local updates can remedy.
>
> We will add more exposition about the proof of the lower bound, and add the above discussion.
>
> ## Regarding Lemma 2
>
> We weren't aware of the works on decentralized optimization that the reviewer mentioned. After reviewing them, it is indeed evident that there are moral similarities (although they are set in very different contexts). We will ensure that we tone down any language discussing this issue and cite these works in our revision.
>
> Regarding the bound in I.3, we understand that the consensus error does approach zero with this choice of step size. The goal of this section (which we don't use in our bounds) is to **underline that between rounds consensus error doesn't blow up either**. This is to reconcile with the work of Wang et al. [8] who claimed that the existing consensus error bounds all blow up with large $K$. In the Appendix of the paper, where additional context should be, we simply formalize the intuition the reviewer has. Regardless, this is a side comment for completeness, which we never claimed to be our main contribution.

---

> > ### Comment · Reviewer_MhSC · 2025-08-05
> > **Official comment**
> >
> > ## Regarding Concern 1 and the Min-max Optimal Value
> >
> > I appreciate the authors' "urge" for me to read the PhD thesis. However, my understanding on the "min-max optimal value" is based on the foundational work of Nemirovski and Yudin [1], which is now commonly referred to as "information-theoretic complexity lower bounds". I believe such a seminal book carries more authority and credibility than a PhD thesis (with no offense intended to Blake Woodworth). According to [1], a complexity lower bound must hold for **every algorithm in a specified oracle model** (see, e.g., Assumption 2.1.4 in [2]).
> >
> > By contrast, Theorem 1, if I am not mistaken again, describes an *instance-wise* lower bound for a specific algorithm (Local SGD). In the spirit of Nemiroski, Yudin, and Nesterov, a fixed algorithm---such as Local SGD with different stepsizes---should not be interpreted as a **class** of algorithms.
> >
> > The authors also state that "it is sufficient just to find one **hard** instance". Could the authors clarify what they mean by "hard instance" here? In modern complexity analyses of first-order methods, it is common to provide both a worst-case upper bound and a concrete problem instance that attains it. To me, such an instance qualifies as "hard" in a meaningful and rigorous sense. In case I "may have overlooked" the authors' primary objective, I would like to ask whether the authors provide a matching upper bound in the general convex setting.
> >
> > [1] Nemirovski and Yudin. Problem complexity and method efficiency in optimization. Wiley. 1983.
> > [2] Nesterov. Lectures on Convex Optimization. 2nd Edition. Springer. 2018.

---

> ### Author Response · Authors · 2025-07-31
> **Rebuttal 2/2**
>
> ## Regarding Theorem 2
>
> Since this was perhaps a point of confusion, note that **Section 3 gives the lower bound for convex functions, and Section 4 gives upper bounds for strongly convex functions**. Both of these sections provide new results and missing pieces in the puzzle to understand the effectiveness of Local updates.
>
> > Assumption 1 says...
>
> The theorem doesn't fail; it just gives a **vacuous upper bound**. It is mathematically correct.
>
> > Asumption 1 imposes...
>
> **We explicitly state the purpose of Section 5 in the section itself**. Theorem 2 is in section 4, and we never claimed that the results in this section can capture the effect of $Q$. This is because the results in Section 4 do not rely on fourth-moment bounds on consensus error, which are necessary to exploit third-order smoothness. Although our statements are all mathematically correct, we understand that the confusion arises from combining both notions of smoothness in a single assumption; we will break this assumption down.
>
> > In lines 226-227, the authors claim...
>
> Note that $\sigma$ could potentially be very small if not zero. Even then, our statement is not overclaiming: we say that Local SGD "can converge" faster than mini-batch, not that it always would.
>
> Even though the parameter $K$ has different interpretations for both algorithms, they are entirely comparable. Please read Section 2. Mini-batch and Local SGD use local computation differently, but they are both algorithms in the **intermittent communication setting**, nonetheless. It is not just our work, but **dozens of previous works** that have made this comparison.
>
> The additional exponent is extremely important because it enables the algorithm to be highly communication-efficient, i.e., it allows for fewer communication rounds by utilizing more local work (a larger $K$). This is precisely the type of result that several existing works have attempted to prove (see Woodworth et al. [20], Glasgow et al. [23], Kolosokoa et al. [20], and Patel et al. [25], among others), but have only demonstrated thus far in the presence of bounded gradient heterogeneity (Assumption 7).
>
> > In lines 227-228, the authors claim tha....
>
> It is entirely rigorous to take two quantities to infinity in a convergence rate. But if that limiting behaviour is confusing (due to perhaps not looking at the limit as a sequence of $K_n$'s), it is certainly enough to make $K$ a function of $\epsilon$, and make it large enough, so that the terms with $K$ in the denominator are no longer dominating.
>
> ## Regarding Section 5
>
> **We do not focus solely on quadratic functions in this section, but rather on all third-order smooth functions**. That's the whole point of the section: to underline the role of $Q$ and bridge the gap to quadratic functions.
>
> > How should we compare Theorem 2 and Theorem 4....
>
> Yes, the reviewer's intuition is correct, because \tau can always be upper bound by 2H. Thus, Theorem 4 is better than Theorem 2.
>
> > How should we compare Theorem 1 and Theorem 4.....
>
> As we said above, the lower bound is for convex functions, while the upper bound is for strongly convex functions. We would have liked to give upper bounds for general convex functions, but that is currently not attainable using our techniques. See the response to Reviewer un1E.
>
> > I do not see the motivation to include....
>
> This comment is **fundamentally wrong**. Theorem 6 is stated for non-quadratic functions as well. For quadratic functions, $Q$ is just 0. It appears that the reviewer may have overlooked our primary objective in this section, and we sincerely request that they take another look. We're hopeful they will see the purpose more clearly after a second reading.
>
>
> ## On writing
>
> We will take into account the very useful suggestions the reviewer has provided. We apologize for so many forward references. It was a technical paper, and in hindsight, we could have written the introduction to appeal to a broader audience. If the paper is accepted, among other changes, we will make sure to clarify our assumptions in more detail (which were sources of confusion for the reviewer while interpreting our results). We will also attempt to add more background material on mini-batch versus local SGD, highlighting how they are both intermittent communication algorithms and thus comparable.
>
> We also agree with the minor concerns (1, 3, 4) the reviewer raised in the Questions section, and we will rectify them to improve clarity. Regarding (2), we understand that there are many reasons clients might want to collaborate. What we want to say is that for consensus algorithms (as opposed to personalized ones), approximately sharing an optimizer across clients is essential; otherwise, choosing a single consensus model makes no sense.
>
> We hope the reviewer will reconsider their score after reading our review, with a particular focus on clarifying any misunderstandings regarding convexity and third-order smoothness in our results.

---

> > ### Comment · Reviewer_MhSC · 2025-08-05
> > **Official comment**
> >
> > ## Regarding Theorem 2
> >
> > My question:
> > > Assumption 1 imposes $L$-Lipschitz continuity of the Hessian, which, I believe, is referred to as “third-order smoothness” in the paper. However, Theorem 2 does not involve at all. Does this imply that has no impact on the convergence rate? If so, then what is the purpose of Section 5 then?
> >
> > Authors' response:
> > > **We explicitly state the purpose of Section 5 in the section itself.** ...
> >
> > As the authors "understand that the confusion arises from combining both notions of smoothness in a single assumption", I suppose the authors also understand my intention of this question, which **is not about the purpose of Section 5**.
> >
> > > Even though the parameter has different interpretations for both algorithms, they are entirely comparable. Please read Section 2. Mini-batch and Local SGD use local computation differently, but they are both algorithms in the intermittent communication setting, nonetheless. It is not just our work, but dozens of previous works that have made this comparison.
> >
> > I searched the contents of Section 2 and my search engine tells me Section 2 does not contain the word "mini-batch SGD". Could the authors clarify where the justification is given for why the parameter $K$ is comparable between the two algorithms?
> >
> > > The additional exponent is extremely important because it enables the algorithm to be highly communication-efficient, i.e., it allows for fewer communication rounds by utilizing more local work (a larger $K$).
> >
> > Could the authors explain more on this point? When $\sigma=0$, the dominant term is in the order or $1/R^2$, which appears independent from $K$.
> >
> > > It is entirely rigorous to take two quantities to infinity in a convergence rate. But if that limiting behaviour is confusing (due to perhaps not looking at the limit as a sequence of $K_n$'s), it is certainly enough to make a function of $\epsilon$, and make it large enough, so that the terms with in the denominator are no longer dominating.
> >
> > By "two quantities", do the authors refer to $K$ in mini-batch SGD and in local SGD, respectively?  If so, why can we take $K \to \infty$ in mini-batch SGD, since the number of mini-batches (or workers) is inherently finite in practice. Could the authors clarify the interpretation or mathematical rationale behind this limiting argument?
> >
> > My question:
> > > I do not see the motivation to include in the analysis. In other words, why should be the correct measure, rather than, e.g., the condition numbers of all the quadratic functions?
> >
> > Authors' response:
> > > This comment is **fundamentally wrong**. Theorem 6 is stated for non-quadratic functions as well. For quadratic functions, is just 0. It appears that the reviewer may have overlooked our primary objective in this section, and we sincerely request that they take another look. We're hopeful they will see the purpose more clearly after a second reading.
> >
> > This response is **fundamentally misguided**. My question concerned the motivation or intuition behind using third-order smoothness in the analysis. More specifically, I was asking why third-order smoothness is more suitable for analyzing Local SGD. Phrased differently: what makes third-order smoothness the "right" lens for this analysis? Or taken to the extreme—should we expect a follow-up paper next year that uses fourth-order smoothness to analyze Local SGD?
> >
> > It appears that the authors may have misunderstand the main intent of my question, and I respectfully urge them to revisit it.

---

> ### Comment · Reviewer_MhSC · 2025-08-01
> **Quick question**
>
> Let me ask a quick question: Does Assumption 1 assume general convexity or strong convexity?
>
> In Theorem 1, the authors state "there exists a quadratic problem satisfying Assumptions 1 to 6" and then "urge" me to **not thinking about strongly convex functions when reading Theorem 1 (which is for general convex functions)**.
>
> In Theorem 2, the authors make the same set of assumptions: "Assume a problem instance satisfies Assumptions 1 to 6" and say **Section 4 gives upper bounds for strongly convex functions**.
>
> So I should interpret Assumption 1 as general convexity in Section 3 and strong convexity in Section 4?

---

> > ### Author Response · Authors · 2025-08-01
> > **Clarification about Assumption 1**
> >
> > We apologize for the confusion that Assumption 1 caused. We also apologize if our urge to the reviewer sounded disparaging in any way; that was not our intention! We appreciate the reviewer's prompt response and willingness to engage with us.
> >
> > That aside, we originally intended Assumption 1 to capture both convexity and strong convexity, and disambiguate by simply denoting the choice of $\mu$ in the result we were using it in. **We should have stated $\mu=0$ in the theorem statement of Theorem 1**. Missing this was our fault, and we apologize for the error.  While the discussion above the Theorem provides the context of the lower bound of Patel et al. [25] and implies that Theorem 1 applies to the general convex setting, we understand that our theorem statement should have been rigorous and self-contained.
> >
> > For all the upper bound results, as we mentioned in our response, making this clarification does not matter, as those results would just be vacuous as $\mu$ approaches zero. That doesn't mean Local SGD would necessarily diverge, but that those upper bounds would be too large. When $\mu=0$, then guaranteeing convergence in iterates is not possible. We instead need to report function sub-optimality, which some of our results do, but even those become vacuous when $\mu=0$. One should rely on other upper bounds for the general convex setting (such as those provided by several works we have cited) when $\mu=0$.
> >
> > As mentioned in response to Reviewer un1E, we initially intended to present all our upper bounds in the general convex setting in the main paper, by relying on a convex-to-strongly-convex reduction, but doing that reduction led to choices of $\mu$ and $\eta$, that wouldn't satisfy the other constraints on them (in terms of the problem dependent parameters and $\epsilon$). As of now, indeed, the lower bound and the upper bound are not directly comparable, but we don't directly compare them either: in a term-by-term fashion. We only refer to their behaviour with regard to $\tau$: both of them demonstrate a communication complexity that depends on $\tau$ (besides first-order heterogeneity terms). This characterization was only conjectured in previous work (and is well known in the non-convex setting, as noted by Patel et al. [30]). Hopefully, future work can address this remaining gap about the general convex setting (which we do believe presents significant technical challenges when using the coupled recursion framework).
> >
> > Hopefully, this will address the confusion around convexity and strong convexity. In our revision, where we will incorporate the writing changes suggested by the reviewer to enhance clarity, we will simply split Assumption 1 to avoid any confusion.

---

> ### Author Response · Authors · 2025-08-06
> **On our Clubbing the Assumptions Together**
>
> We would like to thank the reviewer again for engaging with us and working to address their concerns about the paper.
>
> We would like to start by acknowledging what we could have done better to prevent the confusion surrounding our theoretical results. As we mentioned above, we should have explicitly stated $\mu=0$ in Theorem 1 because we used the same assumption to include convexity, strong convexity, and smoothness. Since the same assumption was used across all our theorems, it can be confusing which properties the upper bound depends on and which it does not. In principle, this is acceptable because we never claimed the upper bound is tight; what we did is similar to providing upper bounds for a class of functions by giving bounds for a super-class of those functions, which may not have all the nice properties implied by the assumptions. Nonetheless, **it is clear to us that this was confusing**.
>
> When we first wrote the paper, we believed that the surrounding text, the discussion of relevant results, and the problem-dependent constants in the upper bounds would clarify which function class the result concerned. However, we overlooked the fact that a valid way to read the theorem statements is simply as self-contained mathematical statements without considering the surrounding text. This was a blind spot in our writing. **We sincerely apologize for this!** To address this, we will ensure that we split Assumption 1 and clearly state that:
> 1. The lower bound in Theorem 1 applies to general convex functions.
> 2. Theorems 2-3 concern strongly convex functions and only require second-order smoothness.
> 3. Theorems 4-6 also concern strongly convex functions, but they additionally assume third-order smoothness. Moreover, Theorems 4-5 focus on the special case of quadratics where $Q=0$.
>
> Hopefully, this will resolve many of the concerns the reviewer raised in their initial review. We appreciate the reviewer taking the time and effort to read our rebuttal and respond with additional questions. If any of our comments about reconsidering our results sounded disparaging, we sincerely apologize for that.
>
>
> > As the authors "understand that the confusion arises from combining both notions of smoothness in a single assumption", I suppose the authors also understand my intention of this question, which is not about the purpose of Section 5.
>
> We apologize again for this! Hopefully, the writing changes we suggest above will resolve this for future readers.
>
> In the following comments, we will address the more specific concerns raised by the reviewer.

---

> ### Author Response · Authors · 2025-08-06
> **MB-SGD in the IC Setting**
>
> # Mini-batch v/s Local SGD
>
> > Could the authors clarify where the justification is given for why the parameter $K$ is comparable between the two algorithms?
>
> Sorry, we meant to say Section 1, not Section 2. We begin the paper by describing the intermittent communication setting and citing several papers that discuss mini-batch SGD in this context. It is a pretty common, if not the most important, baseline in this literature. Having said that, we should have included pseudocode in the paper for mini-batch SGD in the intermittent communication (IC) setting. **We will include the following pseudocode in our revision**:
>
> $$\text{for }r\in[R],\enspace x_r = x_{r-1} - \frac{\eta}{M K}\sum_{m\in[M], k\in[K]}\nabla f(x_{r-1}; z_{r,k-1}^m\sim \mathcal{D}_m) \enspace,$$
>
> Comparing this to Local SGD in equation (2) (which can also be written in a round-wise manner without relying on mod $K$ notation), mini-batch SGD uses all the local computation to compute stochastic gradients at the same point. This is precisely what comes to mind when thinking of mini-batch SGD; however, the crucial point is that in the IC setting, the iteration complexity and communication complexity for mini-batch SGD are the same. This aligns with the description in lines 19-21, as well as the illustration in Figure 2, where, during the communication round, the clients send the aggregate stochastic gradients to the server. Sometimes, this algorithm is called "large" mini-batch SGD, because when $K$ is very large, this essentially reduces to running mini-batch SGD on the average objective $F$ with a mini-batch of size $K$, because,
>
> $$\mathbb{E}\left[ \|\|\nabla F(x_{r-1}) - \frac{\eta}{M K}\sum_{m\in[M], k\in[K]}\nabla f(x_{r-1}; z_{r,k-1}^m\sim \mathcal{D}_m) \|\|_2^2 \right]\leq \frac{\sigma^2}{MK}\enspace .$$
>
> Starting from the early works on Local SGD, such as Stich [15] and Woodworth et al. [20], one of the goals of this line of research has been to identify **when Local SGD can beat mini-batch SGD**. In practice, we often observe this to be the case (Wang et al. [6], Lin et al. [10]). However, demonstrating this domination theoretically has been challenging (Wang et al. [8]). One of the key motivations of our work is to identify heterogeneity assumptions under which we can prove this domination, and more specifically, demonstrate that Local SGD can better utilize local computation. Essentially, Local SGD and mini-batch SGD present two different extremes of collaborative IC algorithms: **prioritizing synchrony v/s prioritizing progress**.
>
> We will include this background, as well as the pseudocode for mini-batch SGD, in our revision, so that it is explicitly stated that we are comparing two algorithms with the **same communication and computation budget**. We hope this clarifies why the parameter $K$ is comparable between the two algorithms.

---

> ### Author Response · Authors · 2025-08-06
> **Comparing Convergence Rates, Part 1**
>
> > Could the authors explain more on this point? When $\sigma=0$, the dominant term is in the order or $1/R^2$, which appears independent from $K$.
>
> Given the above context, let us now delve into a bit more detail about the implications of Theorem 2 regarding mini-batch SGD. Please also see the correspondence with Reviewer un1E.
>
> In the setting of Theorem 2 (i.e., for optimizing second-order smooth and strongly convex functions that satisfy our heterogeneity assumptions), we can get the following convergence guarantee for mini-batch SGD (upper bounding $\mathbb{E}[\|\|\hat x - x^\star\|\|_2^2] \leq$):
>
> $$\textcolor{green}{e^{-\frac{\mu R}{H}} B^2} + \textcolor{red}{\frac{\sigma^2}{\mu^2 MKR}}\enspace.$$
>
> The best known guarantee for Local SGD (before our paper) is due to Kolosokova et al. [18], whose analysis can not benefit from $\tau \ll H$, because they don’t differentiate between functions with a high v/s a low second-order heterogeneity. Their convergence rate is as follows (cf. Table 2, Woodworth et al. [20]),
>
> $$\textcolor{red}{\frac{\sigma^2}{\mu^2 MKR}} + \textcolor{blue}{\frac{H^3\phi_\star^2}{\mu^3R^2}} + \textcolor{red}{\frac{H\sigma^2}{\mu^3 KR^2}}\enspace,\enspace  \text{for}\enspace \textcolor{green}{R = \tilde\Omega\left(\frac{H}{\mu}\right)}\enspace.$$
>
> Finally, the convergence guarantee in Theorem 2 is as follows,
>
> $$\textcolor{green}{e^{-\mu KR/2H}B^2} +  \textcolor{red}{\frac{\sigma^2}{\mu^2 MKR}} + \textcolor{blue}{\frac{\tau^2H^2\phi_\star^2}{\mu^4R^2}} + \textcolor{blue}{\frac{H^4\zeta_\star^2}{\mu^4R^2}} + \textcolor{blue}{\frac{\tau^2H^2\sigma^2}{\mu^6KR^3}} + \textcolor{red}{\frac{H^2\sigma^2}{\mu^4 KR^2}}\enspace,\enspace  \text{for}\enspace \textcolor{green}{R = \tilde\Omega\left(\frac{H \tau}{\mu^2}\right)}\enspace.$$
>
> As the reviewer noted, the noise term, $\sigma^2/(\mu^2 MKR)$ (which essentially captures the sample complexity of optimization with $MKR$ total samples), is the same across all these convergence guarantees. Moreover, we know it is tight due to existing lower bounds for mean estimation. This means that indeed, when this term is dominating, there is no difference between the theoretical convergence rates for mini-batch SGD and Local SGD. Additionally, we can not improve upon the existing guarantee provided by Koloskova et al. [18].
>
> The catch is that in FL settings, we want to parallelize as much as we can. We also aim to minimize the number of communication rounds, as communication is often the bottleneck in massively distributed optimization. Thus, we are interested in a regime where both $M$ and $K$ are very large. In such regimes, naturally $\sigma^2/(\mu^2 MKR)$ will stop dominating at some point. This is not just a hypothesis; in the past decade, as larger and larger batch sizes have been used for machine learning tasks, this has been observed on a variety of problems (for instance, see “Accurate, Large Minibatch SGD: Training ImageNet in 1 Hour” by Goyal et al. and follow-up works). This is also around the time when it was experimentally demonstrated that with such massive levels of parallelism, Local SGD can empirically outperform mini-batch SGD.
>
> So the question we want to answer is: in a regime where $K$ is large enough so to ensure that any terms monotonic in $K$ can be made smaller than the target accuracy $\epsilon$, can we show that Local SGD has a **strictly** better communication complexity than mini-batch SGD? Upon comparing the convergence rates above, it is clear that the rate of Kolosokova et al. [18] can not imply this strict improvement. On the other hand, we see that our algorithm’s communication complexity can indeed improve over mini-batch SGD when data heterogeneity is small. We will make this even more explicit in the next comment.

---

> ### Author Response · Authors · 2025-08-06
> **Comparing Convergence Rates, Part 2**
>
> Let us make the observation in the previous comment about the convergence rates more explicit (as we do in eq. (4) in the paper). The reviewer is correct that, in the large $K$ regime (or when $\sigma = 0$), the dominant terms are those involving $R$ and the heterogeneity constants. In this regime of large $K$ (we need not make it $\infty$, but can just assume it is large enough), we can combine the "implicit" requirements on $R$ with those that arise from the convergence rate (to ensure that the convergence rate is smaller than the target accuracy $\epsilon$). Upon doing this, we get the following communication complexities (ignoring any polylogarithmic factors in $1/\epsilon$.) for mini-batch SGD and Local SGD (where $\kappa = H/\mu$):
>
> $$R^{MB-SGD}(\epsilon) = \tilde\Omega\left( \textcolor{green}{\kappa}\right)\enspace, \quad R^{L-SGD, [18]}(\epsilon) = \tilde\Omega\left( \textcolor{green}{\kappa} + \textcolor{blue}{\kappa^{3/2}\cdot\sqrt{\frac{\phi_\star^2}{\epsilon}}}\right)\enspace, \quad R^{L-SGD, Our}(\epsilon) = \tilde\Omega\left( \textcolor{green}{\frac{\kappa\tau}{\mu}} + \textcolor{blue}{\frac{\kappa\tau}{\mu}\cdot\sqrt{\frac{\phi_\star^2}{\epsilon}}} + \textcolor{blue}{\kappa^2\sqrt{\frac{\zeta_\star^2}{\epsilon}}}\right)\enspace.$$
>
> The green terms in the above complexities are due to implicit constraints on $R$ in Theorem 2, as well as the constraints in the result of Kolosokova et al. [18].
>
> First of all, note that ours is the only communication complexity that improves with a smaller $\tau$. Moreover, the communication complexity due to Koloskova et al. [18] can never improve over mini-batch SGD. For our communication complexity to improve over that of mini-batch SGD, the following two conditions need to be satisfied (up to numerical constants):
> - $\tau \leq \mu \cdot \min \left ( 1, \frac{\sqrt{\epsilon}}{\phi_\star} \right )$ ;
> - $\zeta_\star \leq \frac{\sqrt{\epsilon}}{\kappa}$ .
>
> These conditions essentially characterize a regime of low data heterogeneity. Hopefully, this addresses the reviewer’s original question about lines 226-227. For more context on why this low data heterogeneity is relevant, we note that our result bridges the gap with the existing theory in the homogeneous setting (see discussion on third-order smoothness).
>
> > By "two quantities", do the authors refer ….. Could the authors clarify the interpretation or mathematical rationale behind this limiting argument?
>
> The quantities we were referring to were $K$ and $R$, as the reviewer’s original question was how one could take $K$ to infinity while maintaining non-zero communication rounds. We were not referring to different $K$’s for mini-batch v/s local SGD.
>
> Hopefully, the above discussion about communication complexities clarifies that we don’t even need to make a limiting argument, i.e., send $K\to\infty$. We only need to make sure that $K$ is large enough in the convergence rates (for mini-batch or local SGD), such that any terms that are monotonic in $K$ can be made as small as the target accuracy $\epsilon$. Note that, in practice, for some problems, it is true that $K$ can’t be made too large. However, we are interested in massively parallel settings, such as those of Goyal et al. or Federated Learning, where mini-batch SGD is pushed to its extremes anyway, so $K$ is already quite large. Our goal is to compare whether Local SGD can be shown to be more communication-efficient for such a large $K$. And the message of our theory is that while existing results, which rely only on Assumptions 4-5, can not show such an improvement, it is possible to demonstrate this improvement along with Assumption 6. Please also see our next comment, that will provide some context about the homogeneous setting.

---

> ### Author Response · Authors · 2025-08-06
> **Motivation for Third-order Smoothness, Part 1**
>
> # On third-order smoothness
>
> > This response is fundamentally misguided. My question concerned the motivation or intuition behind using third-order smoothness in the analysis. More specifically, I was asking why third-order smoothness is more suitable for analyzing Local SGD. Phrased differently: what makes third-order smoothness the "right" lens for this analysis? Or taken to the extreme—should we expect a follow-up paper next year that uses fourth-order smoothness to analyze Local SGD?
>
> It appears that we indeed misunderstand the reviewer’s question. When the reviewer said that,
>
> > In other words, why should $Q$ be the correct measure, rather than, e.g., the condition numbers of all the quadratic functions?
>
> We were very confused because for quadratic functions, it does not make sense to talk about $Q$ as it is just $0$. This is because quadratic functions have the same Hessian everywhere, so they are the most “third-order-smooth” functions in a sense.
>
> That aside, let us provide some context on why third-order smoothness is a well-motivated assumption, rather than something we arbitrarily came up with (see also the start of Section 5). The following discussion will be presented in the general convex setting for simplicity, but the same motivation applies to the strongly convex setting as well.
>
> Woodworth et al. [11] (and some of the works cited therein) showed that when all the machines have the same data distribution, i.e., when we are in the homogeneous setting, and when the objective functions $F_m$’s are quadratic, then Local SGD is **extremely communication efficient**. This means that in the large $K$ regime we discussed above, Local SGD can use a single communication round, i.e., make $R=1$ by using a sufficiently large $K$. To make this more explicit, note that they provide the following convergence guarantee for quadratics:
>
> $$\mathbb{E}[F(\hat x)] - F(x^\star) \leq \frac{HB^2}{KR} + \frac{\sigma B}{\sqrt{MKR}}\enspace.$$
>
> The above convergence guarantee has a symmetric dependence on $K$ and $R$, and as such, as long as $KR$ is large enough, we can make $R$ as small as we want. This result, in a sense, is the best we can hope for Local SGD to achieve on convex functions. Woodworth et al. [11] demonstrated that upon acceleration, Local SGD is indeed min-max optimal.
>
>
> The natural question is, **whether this benefit is unique to quadratic functions or can be extended to convex functions as well**. Unfortunately, when we move to the larger class of convex functions, we cannot prove this exact guarantee. In particular, Woodworth et al. [11] and Glasgow et al. [23] showed that the following guarantee is tight for Local SGD when optimizing convex functions,
>
> $$\mathbb{E}[F(\hat x)] - F(x^\star) \leq \frac{HB^2}{KR} + \frac{\sigma B}{\sqrt{MKR}} + \frac{H^{1/3}\sigma^{2/3}B^{4/3}}{K^{1/3}R^{2/3}} \enspace.$$
>
> While the above guarantee also implies extreme communication efficiency, it does not have a symmetric dependence in $K$ and $R$. Furthermore, Woodworth et al. [22] in a follow-up work showed that the min-max optimal algorithm for convex functions is just the best of accelerated mini-batch SGD and accelerated SGD on a single machine. Thus, **the worse dependence on $K$ has a consequence**: that Local SGD on homogeneous convex functions can beat mini-batch SGD, but only in the trivial regime when even SGD on a single machine can also beat it.
>
> This raises a very natural theoretical question: **Are there function classes bigger than quadratics but smaller than all convex functions, where Local SGD could be better than mini-batch SGD in non-trivial regimes**? This is precisely what the third-order smoothness assumptions attempt to quantify. When $Q=0$, we recover quadratic functions; thus, we could hope that for $Q$ small enough, we would still recover the fast convergence rate of Local SGD over quadratics. Mini-batch SGD would not be able to benefit from a small $Q$, because the worst case instance for mini-batch SGD is quadratic, while the worst case instance for all IC algorithms (as proposed by Woodworth et al. [22]) is non-quadratic. In a sense, the third-order smoothness assumption attempts to avoid this non-quadratic, hard instance. Yuan and Ma [21] showed that Local SGD does indeed benefit from a small $Q$ by providing the following convergence guarantee,
>
> $$\mathbb{E}[F(\hat x)] - F(x^\star) \leq \frac{HB^2}{KR} + \frac{\sigma B}{\sqrt{MKR}} + \frac{H^{1/3}\sigma^{2/3}B^{4/3}}{K^{1/3}R^{2/3}} + \frac{Q^{1/3}\sigma^{2/3}B^{5/3}}{K^{1/3}R^{2/3}} \enspace.$$
>
> When $Q$ is small, the above rate can recover the fast rate of Local SGD over quadratics. All of this context suggests two things:
> - Quadratic functions are special in the context of Local SGD, and we can hope to obtain faster convergence guarantees for them, even in the heterogeneous setting.
> - The third-order smoothness assumption can help us interpolate between the quadratic and convex functions even in the heterogeneous setting.

---

> ### Author Response · Authors · 2025-08-06
> **Motivation for Third-order Smoothness, Part 2**
>
> The two insights in the previous comment precisely motivate our Section 5; as Lemma 3 highlights, $Q$ impacts how the fourth moment of the consensus error affects the final convergence guarantee. Using this, we first prove Theorems 4 and 5 for strongly convex quadratic functions, obtaining better rates than Theorems 2 and 3, respectively (which are over all strongly convex functions). Then Theorem 6 provides a convergence rate for all third-order smooth functions (i.e., when $Q>0$). While the rate of Theorem 6 is quite complicated, and it doesn’t recover the quadratic convergence rates (like Yuan and Ma [21] do in the homogeneous setting), it does improve as $Q$ decreases. We hope to address this gap concerning quadratics (which is pretty non-trivial) in future work.
>
> Hopefully, this provides the reviewer with some context on why the third-order smoothness result is not an arbitrary assumption and is well-motivated within the existing literature. To our knowledge, in the heterogeneous setting, ours is the first rate that can both remove the need for Assumption 7 and incorporate third-order smoothness. Patel et al. [25] also studied third-order smoothness, but by relying on Assumption 7, which significantly simplified their analysis.
>
> **Promised revision**: We understand that the discussion in lines 256-267 may be insufficient for readers unfamiliar with the references mentioned above. And towards that end, we will try to include more discussion in Section 5 in our revision, as well as dedicate an entire appendix section to discussing the results in the homogeneous setting. We also have illustrative figures that can summarize about 5-6 papers, which motivate the need for third-order smoothness. We will include those in the appendix as well.
>
> > Or taken to the extreme—should we expect a follow-up paper next year that uses fourth-order smoothness to analyze Local SGD?
>
> We don’t believe the fourth-order smoothness assumption, or even higher-order smoothness assumptions, would be very useful. That said, in the homogeneous setting, existing works have considered other higher-order smoothness assumptions such as Quasi self-concordance (Bullins et al. [26]). We believe that, for first-order algorithms, especially those that rely on a consensus-error-based analysis, second-order heterogeneity and third-order smoothness assumptions already provide a fine-grained characterization.
>
> We hope that this response, as well as the response previously, has underscored that second-order heterogeneity and third-order smoothness capture different sources of hardness compared to the worst condition number of any client’s objective. It is the results of Yuan and Ma [21], as well as our Theorem 1 (along with the conclusions of Patel et al. [25, 30]), that highlight this. In fact, we believe the following are (almost) orthogonal sources of hardness when optimizing using Local update algorithms:
> - The worst condition number of clients, as captured by $\kappa$, which is not unique to distributed optimization, but also present in serial optimization with a single client.
> - The difference in the (local) geometry between the clients, which for local update algorithms is essentially captured by Assumption 6, i.e., $\tau$.
> - The sizes and relative distance between the optima of different clients, which affects which solutions are recoverable by Local SGD: Assumptions 3-5 capture this.
> - The cost of **non-quadraticity**, which essentially exacerbates the effect of asynchrony between communication rounds due to changing geometries on each machine; this even affects local update algorithms when there is no heterogeneity!

---

> ### Author Response · Authors · 2025-08-06
> **On Theorem 1, Part 1**
>
> > I appreciate the authors' "urge" for me to read the PhD thesis. However, my understanding on the "min-max optimal value" is based on the foundational work of Nemirovski and Yudin [1], which is now commonly referred to as "information-theoretic complexity lower bounds". I believe such a seminal book carries more authority and credibility than a PhD thesis (with no offense intended to Blake Woodworth).
>
> We, of course, agree with the reviewer that Nemirovski and Yudin played a seminal role in characterizing min-max complexity in the context of optimization algorithms. Naturally, this concept originates from game theory and predates even their characterization. Our aim for pointing to the thesis was that it formalizes the notion of min-max optimality for **intermittent communication algorithms** (using a graph oracle framework), which is more relevant to our setting of distributed optimization where besides oracle access, one needs to rigorously define the communication model, what information is exchanged between the clients, and how that gets used in oracle queries. Additionally, the thesis we referred to provides a helpful background and contains numerous results (as mentioned above) that help appreciate the need for the assumptions in our paper.
>
> > According to [1], a complexity lower bound must hold for every algorithm in a specified oracle model (see, e.g., Assumption 2.1.4 in [2]).
>
> We now better understand the reviewer’s confusion about our result. It is indeed true that usual lower bounds are **algorithm-independent**, in that they are given for a class of algorithms. For instance, one could have provided a theorem for the class of distributed zero-respecting algorithms (see Patel et al. [25, 30]), of which Local SGD is a single member algorithm. However, it is entirely acceptable to discuss min-max optimality in the context of a single algorithm, which can also be viewed as a family of algorithms where every hyperparameter choice (in this case, the step size) represents a single algorithm. This is precisely what we, and many previous works (Woodworth et al. [11, 20, 22], Kolsokova et al. [18], Glasgow et al. [23]) do. This is because we are trying to underline the tight rate for Local SGD. **“Algorithm-dependent”** lower bounds, like ours, are then helpful in zooming in on the particular algorithm to understand whether it can benefit from certain assumptions and also in determining if its convergence rate is tight.
>
> Our goal in this paper is to understand the unreasonable effectiveness of Local SGD (Wang et al. [8]). Therefore, “algorithm-dependent” lower bounds are more useful to us than attempting to provide lower bounds for the entire class of distributed zero-respecting algorithms. Naturally, any lower bounds on the entire class will also apply to Local SGD, but (i) providing such lower bounds is harder;  and (ii) they give limited insight about Local SGD itself.
>
> It is worth noting that mini-batch SGD, the baseline we care most about, cannot improve with low data heterogeneity, i.e., it achieves the same rate in both homogeneous and heterogeneous settings (see Section 3.1 and Remark 10 in Patel et al. [25]). Thus, the critical question is: **Can Local SGD improve with a lower data heterogeneity?** And our Theorem 1 (which complements Theorem 1 of Patel et al. [25]), shows that when $\tau$ is small, Local SGD **could** (not that it would) indeed be **extremely communication efficient**. As mentioned before, while we can’t provide upper bounds in the convex setting, our strongly convex upper bounds further confirm that Local SGD can indeed improve with a lower $\tau$, matching morally what Theorem 1 suggests.
>
> > By contrast, Theorem 1, if I am not mistaken again, describes an instance-wise lower bound for a specific algorithm (Local SGD). In the spirit of Nemiroski, Yudin, and Nesterov, a fixed algorithm---such as Local SGD with different stepsizes---should not be interpreted as a class of algorithms.
>
> Hopefully, the above context clarifies why we give an algorithm-dependent lower bound. It is indeed interesting to identify the min-max optimal algorithm for the entire class of distributed zero-respecting IC algorithms. But we believe that question is beyond the scope of our work.

---

> ### Author Response · Authors · 2025-08-06
> **On Theorem 1, Part 2**
>
> > The authors also state that "it is sufficient just to find one hard instance". Could the authors clarify what they mean by "hard instance" here? In modern complexity analyses of first-order methods, it is common to provide both a worst-case upper bound and a concrete problem instance that attains it. To me, such an instance qualifies as "hard" in a meaningful and rigorous sense. In case I "may have overlooked" the authors' primary objective, I would like to ask whether the authors provide a matching upper bound in the general convex setting.
>
> As we mentioned before, we do not provide a matching upper bound for Theorem 1. **And the reviewer is correct in doubting Theorem 1**, in that it could very well be a loose lower bound and not “quantify” the hardness of optimizing with Local SGD. We now understand better what the reviewer meant to say in their original review: they weren’t referring to the fact that we are examining a specific quadratic function, but rather to the fact that we were examining a specific algorithm (right?). We will make sure to highlight that **Theorem 1 may be just a loose lower bound**, and adjust the statement of our first contribution accordingly.
>
> **We will also avoid the use of the phrase, “hard instance”**, because we agree that a loose lower bound might not characterize the true hardness. To clarify, **our use of “hardness” is not in the same context as “NP-hardness”** but in the colloquial sense of the word, which is to mean that some problem instance is a particularly “difficult” one for Local SGD to optimize.  And given that Theorem 1 highlights a **pathology of Local SGD**, we used that phrase in the discussion.
>
> Having said that, we conjecture that Theorem 1 is more or less tight. Our reasons to believe that (which we mention in **lines 154-170**) are as follows:
> - In the homogeneous setting, when $\tau, \zeta_\star, \phi_\star$ are zero, then the lower bound of Theorem 1 is tight for Local SGD (Glasgow et al. [23]).
> - When $\tau = H$, then the lower bound in Theorem 1 is tight (Kolosokova et al. [18]).
>
> Thus, more likely than not, the only slackness might be in quantifying the effect of $\zeta_\star$ v/s $\phi_\star$. We don’t suspect that the lower bound would be loose in some intermediate regime such as $\tau = \theta(\sqrt{H})$. Having said this, one can of course not rule this out until we have a matching general convex upper bound. Getting such an upper bound, unfortunately, appears non-trivial at least when relying on our techniques (see lines **244-254**).
>
> We again thank the reviewer for pointing out the issues in our writing, as well as for their comments on the discussion of our results. We sincerely hope that our response can address your concerns. We are happy to answer any other questions the reviewer might have.

---

> > ### Comment · Reviewer_MhSC · 2025-08-08
> > **Official comment**
> >
> > Thank you to the authors for their detailed reply. Since the discussion phase is almost over, I’ll avoid introducing any new questions at this stage. I’ll collaborate with the area chair and fellow reviewers to reach a final decision on the evaluation.

---

> > > ### Author Response · Authors · 2025-08-08
> > >
> > > Thank you so much for engaging with us and reading our detailed responses!
> > >
> > > We appreciate your careful review of our paper, which has helped us identify areas for improvement in our writing. Regardless of the final decision, this will greatly improve the presentation of our results in the revision.

---

### Official Review · Reviewer_ag4U · 2025-06-27

**Clarity:** 3
**Significance:** 3
**Originality:** 2
**Rating:** 4
**Confidence:** 4

**Summary:**

This manuscript studies distributed optimization using Local SGD with full node participation. The authors present a refined analysis for convex objective functions with high-order smoothness by introducing the second-order data heterogeneity / similarity, providing both upper and lower bounds on convergence complexity. The results emphasize that Local SGD performs well in homogeneous data scenarios.

**Questions:**

- Comparing Figures 1a and 5, it seems that as R increases, the average best-final error gap narrows. If R grows further, will the convergence errors of Local SGD continue to converge?

- The algorithm assumes full node participation, which limits scalability. What challenges would arise if node sampling were incorporated?

**Ethical Concerns:**

["NO or VERY MINOR ethics concerns only"]

**Final Justification:**

Most of my technical concerns have been addressed. I raise my score to 4.

**Limitations:**

Yes

**Quality:**

3

**Strengths And Weaknesses:**

__Strengths__

- The paper presents a new theoretical lower bound for Local SGD under second-order similarity conditions and includes sufficient technical details.

__Weaknesses__

- The significance of the contribution is unclear. The main conclusion—that Local SGD performs well in homogeneous data settings—is intuitive, as model divergence across nodes is minimal in such scenarios, reducing the need for frequent communication. While the analysis is refined, it does not yield fundamentally new insights compared to prior work.

- There are several overstatements. For instance, in line 25, the claim that Local SGD “consistently outperforms other first-order methods” is inaccurate. In heterogeneous settings, methods like SCAFFOLD [19] have been shown to outperform Local SGD. Similarly, in line 33, SCAFFOLD is proven to have advantages over SGD.

- The comparison between Assumption 6 (second-order similarity) and Assumption 7 (first-order similarity) is not entirely fair, as Assumption 6 additionally requires twice differentiability, making it more restrictive in a different sense.

- This work only analyzes the case when the second-order similarity is small, but does not discuss what happens when it is large. For example, in the comparison between Equation (3) and Lemma 2, when \tau is large, the consistency error bound given by Lemma 2 may not be tighter than that of Equation (3), due to its dependence on critical parameters such as 1/\mu and K. Similar issues also appear in the discussion of the main results. This one-sided analysis further limits the significance of the work—especially given that data heterogeneity is common in federated learning scenarios.

- Theorem 2 (and other theorems) does not clearly display the dependency on \tau. Upon closer inspection, the term R (related to \tau) appears in the denominators of multiple terms, implying that when data heterogeneity is small, the steady-state error actually increases. However, the authors only highlight the positive effect on convergence speed.

- The experiments only show the communication efficiency of Local SGD, without evaluating its impact on sampling or iteration complexity—particularly relevant since it may increase the steady-state error.

- The experiments are conducted only on synthetic settings, lacking real-world datasets to adequately support the theoretical claims.

---

> ### Author Rebuttal · Authors · 2025-07-29
>
> We thank the reviewer for their feedback. However, we believe there are some major misunderstandings about our setting and results, and we urge the reviewer to reconsider their score after reading our response.
>
> ## Allowed Values of $\tau$
>
> > This work only analyzes the case when the second-order similarity is small........ This one-sided analysis.......
>
> **This is not true**. Note in Assumption 6, we say that $\tau\leq 2H$, to highlight that $\tau$ is trivially upper bounded by smoothness. Thus, in the worst case, $\tau$ will just become $2H$ in our results. With this in mind, the reviewer's objection that the bound in Lemma 2 might not be useful is incorrect. Even if $\tau=2H$, the bounds in Lemma 2 and equation (3) are **not comparable**, because (3) requires the much stronger and opaque Assumption 7 where it is unclear how $\zeta$ in the assumption will even depend on $\mu, K, \eta$, something Lemma 2 makes very explicit.
>
> With this in mind, **we disagree with the assessment that our analysis is "one-sided"**, because even if $\tau=2H$, our results are not worse than existing results in the relevant regime when $K$ is large. We have discussed this issue extensively in our response to Reviewer un1E, comparing our results with those of Kolosokova et al. [18]. To summarize our main point in that response, it is well known what to expect when second-order heterogeneity is high, as existing analyses do not differentiate between functions with high or low second-order heterogeneity (at least not without assuming restrictive gradient similarity). The interesting case is indeed what happens when it is small, and to our knowledge, ours is the only analysis that can highlight the benefit of local updates when $\tau \ll H$ without any restrictive assumptions.
>
> ## $R$ in Theorem 2
>
> > Theorem 2 (and other theorems) does not clearly display the dependency on \tau.......
>
> **Note that the smallest value for $R$, the number of communication rounds, is $1$; it is a positive integer, and does not approach $0$ even if $\tau=0$!** We discuss this explicitly in the first paragraph of our introduction, and Figure 2 illustrates the intermittent communication setting. With this in mind, the issue the reviewer raised is completely unfounded: our results do not deteriorate when data heterogeneity is small. While Theorem 2 provides a lower bound for $R$, as soon as this lower bound is smaller than $1$, it becomes vacuous. We wonder if the reviewer confused the asymptotic notation $\Omega$ with $\theta$? If so, **we hope they will change their review upon realizing this**.
>
> Essentially, as detailed in response to Reviewer un1E, our constraint on $R$ (in Theorem 2) improves when $\tau\to 0$, as opposed to mini-batch SGD, which must always communicate $\tilde\theta(\kappa)$ times, no matter how small $\tau$ is. Along with Theorem 1, this highlights the positive role of a low second-order heterogeneity on the required number of communication rounds. In fact, as we show in Theorem 4, when $\tau=0$, by choosing a $K$ large enough, we can indeed get away with a single communication round, something well known for the case of homogeneous quadratics, but not for Local SGD in the heterogeneous setting. This characterization is our main contribution.
>
> ## SCAFFOLD
>
> > There are several overstatements. For instance, in line 25........
>
> We assure the reviewer that we are **very familiar** with the SCAFFOLD paper. Having said that, none of the existing analyses show that SCAFFOLD can theoretically improve over mini-batch SGD even when there is no second-order heterogeneity (see Table 2 in Woodworth et al. [20]). Furthermore, as pointed out by the experiments in the paper titled "Adaptive Federated Optimization" by Reddi et al., and further underlined by Wang et al. [8], **SCAFFOLD and FEDAVG have roughly identical performance** on many realistic federated datasets. Thus, there is both theoretical and empirical evidence to support the claims made in the paper, and Local SGD remains the simplest and most widely adopted federated optimization algorithm. This is why we choose to address the theoretical gap of explaining its "unreasonable effectiveness". We will include a remark about SCAFFOLD in our revision.
>
> ## Experiments
>
> > The experiments only show the communication........
>
> To clarify, in our setting, iteration and sampling complexity are equivalent, and both are $K$ times the communication complexity, as each stochastic gradient is computed using a new sample. This means that in the transition from Figure 1a to Figure 5, these complexities increase. And in Figure 1b, only $K$ is fixed, and we plot $R$ as a function of increasing co-variate shift, thereby precisely showing how all of these complexities would increase with co-variate shift.
>
> > Comparing Figures 1a and 5......
>
> Indeed, in the regime where $R$ is large, heterogeneity has less impact on Local SGD. This is because, with a very large communication budget, one can set the step size very low, essentially making Local SGD resemble mini-batch updates. This is not the most interesting setting, which is why we report experiments with $R=5$ (a low communication budget) in the main paper. We observe that as the complexity of our task increases, for instance, by increasing the dimensionality of our problem to **d=50**, we obtain a similar figure to Figure 1a, even with a larger communication budget of $R=10$ (as in Figure 5); we will include these experiments in our revision. Also note that Figure 1b in the current paper illustrates how the communication budget must increase with rising $\tau$.
>
> > The experiments are conducted only on synthetic......
>
> There are dozens of papers, including several we have cited, that conduct experiments on other "tasks." Whether academic papers truly use "real" datasets is questionable, and many experiments in Federated Learning are performed in industry using proprietary datasets. That said, we believe it is already well established (for instance, due to the work by Reddi et al. mentioned above) that Local SGD is a highly effective algorithm in practice. Are there any specific datasets that the reviewer thinks we should test on, especially where we can make a novel point by controlling both the covariate and concept shift (like we do in Figure 1a)? We will try to investigate whether we can artificially control covariate and concept heterogeneity on the FMNIST dataset, and if successful, we will report those results in our final version.
>
>
> ## Partial Participation
>
> > The algorithm assumes full node participation.....
>
> **This is a significant extension for our work**. Let us consider the case of sampling each client from some meta-distribution over clients, i.e., $m\sim P$, and assume we are interested in optimizing $$F(x) := \mathbb{E}_{m\sim P}[F_m(x)]\enspace.$$
>
> This is arguably the simplest setting for studying partial participation and has been explored in Karimireddy et al. [28], Patel et al. [30], etc. These papers analyze the client sampling with a new set of clients sampled i.i.d. from $P$ each communication round, by assuming that the inter-client variance of gradients is uniformly bounded at every point, i.e., $$\mathbb{E}_{m\sim P}[\|\|\nabla F_m(x) - \nabla F(x)\|\|_2^2]\leq G^2\enspace,\forall x\in\mathbb{R}^d\enspace.$$
>
> While we can also provide all our results under this assumption, the primary objective of our paper is to avoid something akin to Assumption 7, and this assumption on inter-client variance is precisely that. One alternative is to only assume the inter-client variance assumption at the optimizers of the average objective $F$, analogous to Assumptions 4-5. However, this relaxation makes the analysis very challenging, as we must carefully handle the randomness in sampling a client and the randomness involved in sampling the stochastic gradients for that client. Due to multiple local updates, the stochastic gradients "leak" information about the sampled client, thus breaking a naive independence argument often used in the existing papers. We believe that extending our results to the partial participation setting is possible, as Lemma 2 essentially allows us to unroll the local updates to the last communication round, at which point we can utilize the i.i.d. property of client sampling. However, this extension would require significant additional work, which is why we leave this for the future.
>
> ## Our Contribution
>
> > The significance of the contribution is unclear.....
>
> First, our paper goes well beyond the homogeneous setting—which is well understood (see Woodworth et al. [11], Glasgow et al. [23]) but unrealistic in practice—and analyzes the exact forms of heterogeneity that govern Local SGD’s performance. Prior work (e.g., Wang et al. [6,8], Patel et al. [25]) either aggregates heterogeneity into coarse metrics or imposes strong gradient‐similarity assumptions. We close this gap by highlighting a necessary and sufficient characterization, which we believe is intricately linked to covariate shift and concept shift (Figure 1), thus providing an important insight for practitioners who seek to balance local computation against communication (discussed in lines 334-342).
>
> And as far as the need for theory. Suppose one were teaching Local SGD in a graduate class. Would stating merely “Local SGD works well in homogeneous settings” suffice? Would we hold the convergence theory for, say, SGD, to the same standard? Perhaps not. Heterogeneity is what makes Federated Learning scientifically interesting. Thus, understanding precisely what heterogeneity does to Local SGD, an algorithm almost synonymous with Federated Learning, is a fundamental question at the core of the field. While there are still many open questions, we believe our results deliver both rigorous theory and actionable guidelines, thus filling a critical theoretical gap and offering clear, practical, scientific, and pedagogical value.

---

> > ### Comment · Reviewer_ag4U · 2025-08-03
> >
> > Thank you for the response. Some of my concerns have been addressed. However, I suggest the authors carefully read the review before speculating on the reviewer’s intent. The following concerns remain:
> >
> > - Regarding $\tau$, in Lines 211–213, the authors only discussed the case where $\tau \to 0$, but did not consider the case where $\tau$ increases. This regime is more significant and warrants greater attention.
> >
> > - Regarding $R$, my point is that Theorem 2 does not clearly reflect the dependency on hyperparameters, including $\tau$. For example, the right-hand side depends on $R$, and $R$ in turn depends on $\tau$ with an asymptotic notation $\mathbf{\Omega}$. This kind of nested dependency makes the influence of $\tau$ unclear. It seems that if we substitute the dependency of $R$ on $\tau$, then in terms 2 and 3–6 on the right-hand side, $\tau$ appears in the denominators, meaning that decreasing $\tau$ makes these terms larger. If this is not the case, the expressions should be modified.
> >
> > - Regarding SCAFFOLD, at least its experiments show that in some scenarios it performs better than Local SGD/FedAvg. Based on this, the statement in the paper that it “consistently outperforms” other methods is inaccurate.
> >
> > - Regarding the experiments, for work on federated learning, validating the algorithm on real datasets is both worthwhile and necessary, even on small datasets like FMNIST or CIFAR-10/100. These are more convincing than synthetic data. I agree that it may be difficult to control certain parameters quantitatively in such tasks, but qualitative control may still be possible.
> >
> > - The response does not address my question about the comparison between Assumptions 6 and 7.
> >
> > I will consider adjusting the scores based on the authors' subsequent responses.

---

> > > ### Author Response · Authors · 2025-08-04
> > > **On Experiments**
> > >
> > > We thank the reviewer for engaging with us. We will do our best to address any remaining concerns the reviewer may have.
> > >
> > > # Experiments
> > >
> > > > Regarding SCAFFOLD, at least its experiments show that in some scenarios it performs better than Local SGD/FedAvg. Based on this, the statement in the paper that it “consistently outperforms” other methods is inaccurate.
> > >
> > > We understand that this particular statement might seem like an overclaim, and **we will remove it**. We agree with the reviewer that, in particular tasks, SCAFFOLD could outperform Local SGD/FedAvg. However, as pointed out and thoroughly evaluated by some of the authors of the original SCAFFOLD paper in their follow-up works (e.g., Reddi et al. and Wang et al. [6,8]), SCAFFOLD and FedAvg exhibit very similar performances across many FL benchmarks (e.g., Figure 1 in Reddi et al.). Thank you again for pointing out this inaccuracy. We will add a more detailed discussion in the revision to clarify the relationship between SCAFFOLD and Local SGD/FedAvg.
> > >
> > > > Regarding the experiments, for work on federated learning, validating the algorithm on real datasets is both worthwhile and necessary, even on small datasets like FMNIST or CIFAR-10/100. These are more convincing than synthetic data. I agree that it may be difficult to control certain parameters quantitatively in such tasks, but qualitative control may still be possible.
> > >
> > > Firstly, we would like to emphasize that the primary contribution of our work is theoretical in nature. As such, the synthetic experiments we already include do help emphasize the point our theory is trying to make.
> > >
> > > That said, we agree with the reviewer that additional experiments could further elucidate our key insights. Since many papers already include experiments on FMNIST and CIFAR-10/100 for FedAvg, SCAFFOLD, and other algorithms, repeating those has limited utility. Instead, we propose including experiments that can highlight the impact of covariate heterogeneity. To achieve this, we plan to conduct experiments using the Celeb-A dataset, where it is easier to induce "isolated" distribution shifts, and these shifts are reasonably interpretable. We have the following in mind:
> > >
> > > - **Context**: CelebA consists of face images labeled with 40 binary attributes and is naturally partitioned by identity. To systematically study distribution shifts, we introduce two independent parameters:
> > >   - $\alpha \in [0, 1]$ to control **covariate shift** by varying the client-wise distribution over inputs, i.e., $P(X)$,
> > >   - $\beta \in [0, 1]$ to control **label shift** by varying the client-wise conditional $P(Y \mid X)$.
> > >
> > > - **Covariate shift ($\alpha$)**: To vary $P(X)$ while holding $P(Y \mid X)$ "fixed", we use a **two-step sampling process**:
> > >   1. Cluster the dataset based on input features (e.g., ArcFace embeddings), without using label information.
> > >   2. For each client:
> > >      - When $\alpha = 0$, sample uniformly from all clusters (shared $P(X)$).
> > >      - When $\alpha \to 1$, assign each client a disjoint cluster (disjoint $P(X)$).
> > >      - Then, from the selected images, **subsample to enforce a balanced label distribution per client** (e.g., equal numbers of `Smiling=1` and `Smiling=0`).
> > >
> > > - **Label shift ($\beta$)**: To vary $P(Y \mid X)$ while holding $P(X)$ "fixed", we assign all clients images from the same visual cluster (i.e., a fixed region of input space), and then alter the label proportions:
> > >   - When $\beta = 0$, all clients receive a balanced label distribution.
> > >   - When $\beta \to 1$, clients receive increasingly skewed label proportions (e.g., 90% positive for one, 10% for another), creating variation in $P(Y \mid X)$ over a fixed $P(X)$.
> > >
> > > - **Joint variation ($\alpha, \beta$)**: By varying both parameters, we assign each client a different visual cluster (via $\alpha$) and a different label distribution over their local $X$ (via $\beta$). At $(\alpha, \beta) = (0, 0)$, the data is i.i.d. across clients. At $(\alpha, \beta) = (1, 1)$, clients differ in both input distribution and conditional label semantics.
> > >
> > > This procedure does not completely disassociate these two shifts, as, for instance, the sub-sampling based on labels after fixing the feature-based clusters can re-weight the relative mixing of label clusters. But this qualitatively matches what we did in Figure 1a. Something similar might also be feasible on the FMNIST dataset. If the reviewer has suggestions for other datasets, we are happy to include them under this setup as well.

---

> > > ### Author Response · Authors · 2025-08-04
> > > **On Assumption 6 v/s Assumption 7**
> > >
> > > # On Assumption 6 v/s Assumption 7
> > >
> > > We apologize for missing this question before.
> > >
> > > First, we would like to note that Assumption 6 is not something we came up with, but an assumption that has been used extensively in this literature (for instance, Karimireddy et al. [28], Murata and Suzuki [29], Patel et al. [25, 30], among many other works). This alone is a good enough reason to study the properties of Local SGD under this assumption.
> > >
> > > Second, for convex loss functions that are common in learning problems, twice differentiability is not a significant concern. The most notable loss functions, namely the square loss and logistic regression, are both twice differentiable. As noted in Remarks 1 and 2 of Patel et al. [25], Assumption 6 is significantly less restrictive than Assumption 7, as it allows for interesting data heterogeneity among clients. In contrast, the latter is nearly equivalent to assuming homogeneity for quadratic problems. Thus, for highly relevant problems such as solving linear systems across machines, Assumption 6 (along with Assumptions 4 and 5) characterizes the inherent heterogeneity more effectively and transparently than Assumption 7 (cf. our Section 6).
> > >
> > > Thirdly, we don't necessarily need twice-differentiability. We only assume twice-differentiability to make the application of Assumption 6 simple in an already complex analysis. For instance, we can view Assumption 6 as a Lipschitz condition on the gradients of the difference between the functions on any two machines. In particular, we only need to assume that:
> > >
> > > $$\|\| \nabla F_m(x) - \nabla F_n(x) - \nabla F_m(y) + \nabla F_n(y)\|\|_2 \leq \tau \|\| x-y \|\|_2\enspace, \forall m,n\in[M],\enspace \forall x,y \in \mathbb{R}^d\enspace.$$
> > >
> > > To see why this is sufficient in our analysis, note that in the two key places where we apply Assumption 6, it is possible to rely on gradient differences and not apply the mean-value theorem to go to Hessians (see the equations between lines 1390-1391 and the red terms between 1261-1262). We will add a remark to highlight that this weaker version of the assumption is sufficient. Still, we emphasize that this additional complexity in our analysis does not necessarily yield much more insight and makes the proof less readable.
> > >
> > > Finally, when we assume third-order smoothness (as in Theorem 6), we must anyway assume twice-differentiability. While it is possible to relax the third-order smoothness assumption slightly as well (such as viewing it as a quadraticity gap), assuming any sort of third-order smoothness will always imply second-order differentiability. Since bridging the gap to quadratic functions and exploiting third-order smoothness was a key motivation for our paper, we did not focus extensively on relaxing the twice-differentiability assumption.
> > >
> > > Hopefully, this addresses the reviewer's concern.

---

> > > ### Author Response · Authors · 2025-08-04
> > > **All Possible Values of \tau**
> > >
> > > # On Different Regimes of $\tau$
> > >
> > > In our response below, we will strive to be even more explicit about the regimes of $\tau$ and how the implications of Theorem 2 compare to the best comparable existing result (due to Koloskova et al. [18]).
> > >
> > > Note that, as argued in the paper and the rebuttal, the goal of most of the theoretical results in this area is to identify assumptions where we can minimize the communication complexity $R$ by putting in more local work, i.e., increasing $K$. There are several practical, systems-motivated reasons to do this (see Wang et al. [6]), and as such, communication is most often the bottleneck in massively distributed optimization.
> > >
> > > Towards this end, and to very explicitly discuss different regimes of $\tau$, we will compare the communication complexities of mini-batch SGD and Local SGD (as implied by our results as well as Kolosokova et al. [18]) in a regime where $K$ is large enough so that we can ensure that any terms that are monotonically decreasing in $K$ can be made smaller than the target accuracy $\epsilon$. This will also allow us to combine the "implicit" requirements on $R$ with those that arise from the convergence rate (to ensure that the convergence rate is smaller than the target accuracy $\epsilon$). This perspective on communication reduction (as opposed to writing explicit convergence guarantees) is ubiquitous in distributed optimization literature (for instance, see Patel et al. [30]).
> > >
> > > Upon doing this, we get the following communication complexities (ignoring any polylogarithmic factors in $1/\epsilon$.) for mini-batch SGD and Local SGD (where $\kappa = H/\mu$):
> > >
> > > $$R^{MB-SGD}(\epsilon) = \tilde\Omega\left( \kappa\right)\enspace, \quad R^{L-SGD, [18]}(\epsilon) = \tilde\Omega\left( \textcolor{green}{\kappa} + \textcolor{blue}{\kappa^{3/2}\cdot\sqrt{\frac{\phi_\star^2}{\epsilon}}}\right)\enspace, \quad R^{L-SGD, Our}(\epsilon) = \tilde\Omega\left( \textcolor{green}{\frac{\kappa\tau}{\mu}} + \textcolor{blue}{\frac{\kappa\tau}{\mu}\cdot\sqrt{\frac{\phi_\star^2}{\epsilon}}} + \textcolor{blue}{\kappa^2\sqrt{\frac{\zeta_\star^2}{\epsilon}}}\right)\enspace.$$
> > >
> > > The green terms in the above complexities are due to implicit constraints on $R$ in Theorem 2, as well as the constraints in the result of Kolosokova et al. [18] (we discuss this further in the next official comment).
> > >
> > > First of all, note that the only communication complexity that improves with a smaller $\tau$ is ours. Moreover, the communication complexity due to Koloskova et al. [18] can never improve over mini-batch SGD. For our communication complexity to improve over that of mini-batch SGD, the following two conditions need to be satisfied (up to numerical constants):
> > > - $\tau \leq \mu \cdot \min \left ( 1, \frac{\sqrt{\epsilon}}{\phi_\star} \right )$ ;
> > > - $\zeta_\star \leq \frac{\sqrt{\epsilon}}{\kappa}$ .
> > >
> > > This gives us two regimes to discuss the effect of $\tau$:
> > > 1. When $0 \leq \tau \leq \mu \cdot \min \left ( 1, \frac{\sqrt{\epsilon}}{\phi_\star} \right )$ and $\zeta_\star \leq \frac{\sqrt{\epsilon}}{\kappa}$, then our rate implies that the communication complexity of Local SGD is better than mini-batch SGD.
> > > 2. When $\mu \cdot \min \left ( 1, \frac{\sqrt{\epsilon}}{\phi_\star} \right ) \leq \tau \leq 2H$ or $\zeta_\star \geq \frac{\sqrt{\epsilon}}{\kappa}$, then mini-batch SGD has a better communication complexity than Local SGD (based on our rate).
> > >
> > > The fact that in the second regime of high data heterogeneity, our results do not imply an improvement over mini-batch SGD is expected, and **morally matches the theoretical predictions by every other paper on Local SGD's convergence to date** (see Woodworth et al. [20]). Finally, it is worth noting that when $\tau \approx H$, then our communication complexity is worse than Kolosokova et al. [18] (due to additional powers of $\kappa$). However, the best communication complexity for Local SGD should always be the minimum of our rate and theirs, as both of us provide valid convergence upper bounds, albeit under different assumptions. This is why in our submission, we focused on the regime where our results provide novel insights, i.e., when $\tau\ll H$. Last but not least, we want to emphasize that Theorem 4 provides even better communication complexity than Theorem 2, for the special case of quadratic functions.
> > >
> > > We will incorporate all of this into our revision, discuss what happens for all values of $\tau \in [0, 2H]$, and ensure that our discussion is more balanced, in contextualizing our results in the existing literature. Hopefully, this will address the reviewer's concern about both any potential "one-sidedness" in our discussion and how the implicit dependence of $R$ on $\tau$ influences the final trade-off.

---

> > > ### Author Response · Authors · 2025-08-04
> > > **Interpreting Theorem 2**
> > >
> > > # Clarification about Theorem 2 and Implicit Requirements on $R$
> > >
> > > Hopefully, the alternative perspective of communication complexity provided above addresses the reviewer's concern about interpreting the $R$ terms in our convergence rate. However, we also wanted to directly address what we think might be a misinterpretation of the result of Theorem 2. In the theorem, when the reviewer takes the lower bound on $R$ and plugs it into Theorem 2 to obtain an upper bound, although this results in a valid upper bound, **the upper bound would be very loose**. This is because, as we clarified in our rebuttal before, while $\tau$ can be $0$, $R\geq 1$, so the constraint in the theorem should be interpreted as $R\geq \min \left(1, \tilde \Omega\left(\frac{H\tau}{\mu^2}\right)\right)$. Because of this, when $\tau$ approaches $0$ and becomes smaller than $\mu/\kappa$, then the constraint on $R$ is not binding. On the other hand, the convergence rate in Theorem 2, and all our other theorems, keep improving even as $\tau$ gets smaller.
> > >
> > > This implicit constraint on $R$ is not unique to our paper; a similar and more restrictive constraint already exists in the result of Kolosokova et al. [18], which to date offers the best explicit rate that does not rely on any variant of Assumption 7. With a little bit of work, their rate (i.e., an upper bound on $\mathbb{E}[\|\|\hat x - x^\star\|\|_2^2]$ up to polylogarithmic factors) in our setting is given as (cf. Table 2, Woodworth et al. [20]),
> > >
> > > $$\textcolor{red}{\frac{\sigma^2}{\mu^2 MKR}} + \textcolor{blue}{\frac{H^3\phi_\star^2}{\mu^3R^2}} + \textcolor{red}{\frac{H\sigma^2}{\mu^3 KR^2}}\enspace,\enspace  \text{for}\enspace \textcolor{green}{R = \tilde\Omega\left(\frac{H}{\mu}\right)}\enspace.$$
> > >
> > > On the other hand, our convergence rate in Theorem 2 is given by:
> > >
> > >  $$\textcolor{green}{e^{-\mu KR/2H}B^2} +  \textcolor{red}{\frac{\sigma^2}{\mu^2 MKR}} + \textcolor{blue}{\frac{\tau^2H^2\phi_\star^2}{\mu^4R^2}} + \textcolor{blue}{\frac{H^4\zeta_\star^2}{\mu^4R^2}} + \textcolor{blue}{\frac{\tau^2H^2\sigma^2}{\mu^6KR^3}} + \textcolor{red}{\frac{H^2\sigma^2}{\mu^4 KR^2}}\enspace,\enspace  \text{for}\enspace \textcolor{green}{R = \tilde\Omega\left(\frac{H \tau}{\mu^2}\right)}\enspace.$$
> > >
> > > We color-coded different terms to make the comparison easier and relate them to the communication complexity discussion above. We can note a few things when we compare these two convergence rates:
> > > - Both rates have an **implicit requirement on communication rounds**. When discussing the communication complexities, we can incorporate this quite easily in our comparison. But even if we want to compare the convergence rates directly, then both these constraints are akin to the constraints one would get while **"hiding the exponential decay term"** in the convergence. For instance, in Kolsokova et al. [18]'s result, ensuring $e^{-HR/\mu}B^2 < \epsilon$ will lead to a similar constraint on $R$. In our case, ensuring $e^{-H\tau R/\mu^2}B^2 < \epsilon$ leads to the same constraint (cf. Lemma 34 in our Appendix). But our constraint gets better with a lower $\tau$. We wanted to be highly transparent about every dependence, which is why we did not simplify away this constraint in our informal statement in the main paper (something Kolsokova et al. [18] do).
> > > - The convergence rate due to Kolosokova et al. [18] **can never improve over mini-batch SGD** (cf. Table 2, Woodworth et al. [20]) because the convergence rate for mini-batch SGD in the same setting just includes an exponential term: $e^{-HR/\mu}B^2$ and the noise term: $\sigma^2/(\mu^2 MKR)$. This conclusion aligns with the general convex setting, where Patel et al. [25] demonstrated that, under Assumptions 4 and 5, Local SGD cannot outperform mini-batch SGD. On the other hand, our convergence rate, which additionally assumes Assumption 6, can improve over mini-batch SGD, as discussed in the previous official comment.
> > > - We also note that Kolosokova et al. [18]'s rate can **never imply the "extreme communication efficiency" of Local SGD**, which we do observe in the homogeneous setting (see Woodworth et al. [11, 22]). In particular, in the regime where $K$ is large and data heterogeneity is low, we cannot choose $R$ to be very small. This is because even if $\phi_\star=0$, the convergence rate of Kolosokova et al. implies that we need $R$ to grow with the condition number of the problem $H/\mu$ (as highlighted in the previous official comment). As such, their convergence rate can not bridge the gap to the homogeneous setting in regimes of low data heterogeneity. Our rate can bridge this gap.
> > >
> > > We hope this addresses any concerns the reviewer had about interpreting Theorem 2. We reiterate that the **implicit constraint in Theorem 2 does not conceal any information**.
> > >
> > > We are happy to address any other concerns the reviewer might have and thank them again for engaging with us.

---

> > > > ### Comment · Reviewer_ag4U · 2025-08-05
> > > >
> > > > Thank you for the detailed response. Most of my technical concerns have been addressed. I will make my final rating decision during the Reviewer–AC discussion phase.

---

> > > > > ### Author Response · Authors · 2025-08-05
> > > > > **Thank you!**
> > > > >
> > > > > We would like to thank the reviewer for engaging with us during the discussion period and for giving us the opportunity to address their concerns. We truly appreciate the reviewer's patience in reviewing our detailed response. The reviewer's comments have highlighted some areas for improvement in our writing and have helped us refine our paper. In our revision, we will ensure that we incorporate these changes, such as re-emphasizing that $R \geq 1$ in our theorem statements, to prevent future readers from encountering the same issues.
> > > > >
> > > > > > Most of my technical concerns have been addressed.
> > > > >
> > > > > We are happy to address any other questions that the reviewer may still have, or any new ones that may have arisen.
> > > > >
> > > > > Thanks again!

---

> ### Comment · Area_Chair_g2SR · 2025-08-01
> **Authors rebuttal**
>
> Dear reviewer,
>
> The authors have posted their rebuttal and they raised some concerns. In particular, the authors feel that there are some misunderstanding regarding the paper. Please read their rebuttal and respond as soon as possible so that there can be some back and forth communication with the reviewers. Thanks a lot.
>
> Best,
>
> AC

---

### Official Review · Reviewer_un1E · 2025-07-03

**Clarity:** 3
**Significance:** 3
**Originality:** 3
**Rating:** 4
**Confidence:** 4

**Summary:**

This paper explores the advantages of Local SGD over mini-batch SGD in heterogeneous settings. The authors derive new upper and lower bounds on the convergence of Local SGD and demonstrate its superior communication complexity, particularly in settings with low second-order heterogeneity.

**Questions:**

One of the paper's key contributions is presented as removing the assumption of uniformly bounded first-order heterogeneity. However, prior work in similar convex settings, such as Koloskova et al. (2020), also avoids this assumption, instead using bounded first-order heterogeneity at the optimum.
Given that this paper also appears to rely on a similar condition (Assumptions 4 and 5), could you clarify how your contribution differs from what could be derived from these existing results?

I would be happy to reconsider my score after clarification in this regard.

**Ethical Concerns:**

["NO or VERY MINOR ethics concerns only"]

**Limitations:**

Yes.

**Paper Formatting Concerns:**

NA.

**Quality:**

3

**Strengths And Weaknesses:**

Strengths:
This paper makes a valuable theoretical contribution by clarifying why Local SGD can outperform mini-batch SGD—a phenomenon widely observed in practice but not well-established in theory. The work is also clearly written.

Weaknesses:
some assumptions are quite restrictive, such as requiring the objective function to be strongly convex (μ>0).

---

> ### Author Rebuttal · Authors · 2025-07-27
>
> We thank the reviewer for taking the time and effort to provide feedback on our work. We are glad that the reviewer found our paper to be clear and appreciated the key technical contribution: identifying conditions under which Local SGD can beat mini-batch SGD.
>
> We would like to answer the two concerns that the reviewer has raised.
>
> ### **Regarding Kolsokova et al. [18]**
>
> To summarize our argument, **the results in Koloskova et al. [18] analysis can not differentiate between functions with a high v/s a low second-order heterogeneity, while our more fine-grained analysis shows that Local SGD converges faster under low second-order heterogeneity.** We elaborate in more detail below:
>
> We agree with the reviewer that Kolsokova et al. [18] considered a similar assumption that bounds the gradient heterogeneity at the optimal point. Specifically, the assumption they considered in their paper is a bound on $\frac{1}{M}\sum_{m\in[M]}\|\|\nabla F_m(x^\star)\|\|_2^2$ where $x^\star$ is an optimizer of the average objective $F$. Their assumption is weaker than ours, but overall, given second-order smoothness and Assumption 5 in our paper, it is straightforward to derive the condition needed for their assumption, so our assumptions are comparable. The main reason we used Assumptions 5 and 6 is to highlight the pathology mentioned in Remark 1 (see Appendix B).
>
> Kolsokova et al. [18] consider both the convex and strongly convex settings, but they **cannot show an improvement over mini-batch SGD**. Patel et al. [25] demonstrated that the general convex bounds in Kolsokova et al. are tight, indicating that additional assumptions are needed beyond their setting to demonstrate the benefit of local updates. Patel et al. [25] also conjectured (see their Conjecture 1) that second-order heterogeneity (Assumption 6) might be the needed assumption, providing preliminary evidence to support this.
>
> We have cited these works (and many more), and we provide guarantees that are strictly better than all of them. Specifically, we prove **the conjecture of Patel et al.**, showing that assumptions about optima (akin to first-order assumptions), along with the second-order heterogeneity assumption (Assumptions 4-6), enable Local SGD to improve over mini-batch SGD. This is something Kolsokova et al. (and all the previous papers) were unable to accomplish.
>
> Let us be more precise. We will use the notation from our paper, except that we will use $\phi$ and $\zeta$ instead of $\phi_\star$ and $\zeta_\star$, as the latter do not compile correctly with exponents in open review. In the strongly convex setting, translating our assumptions to the setting of Kolsokova et al. [18] gives the following convergence guarantee (ignoring polylogarithmic factors in problem-dependent parameters):
>
> $$\mathbb{E}[\|\|x_{KR} - x^\star\|\|_2^2] \leq \textcolor{red}{\frac{\sigma^2}{\mu^2 MKR}} +  \textcolor{blue}{\frac{H^3\phi^2}{\mu^3R^2}} +  \textcolor{red}{\frac{H\sigma^2}{\mu^3KR^2}}\enspace,  \text{for}\enspace \textcolor{green}{R= \tilde\Omega\left(\frac{H}{\mu}\right)}\enspace.$$
>
> Translating their guarantee requires some work (cf. Table 2 in Woodworth et al. [20]). Note that the requirement for communication rounds in the above rate stems from the exponential term and, in particular, matches the requirement for mini-batch SGD. Furthermore, the first term in the above rate is identical to the "noise term" for mini-batch SGD. Both of these together mean that **local updates do not provide a provable improvement over mini-batch SGD based on the result of Kolsokova et al. [18]**. Even when $\phi_\star=0$ and $K\to\infty$, their result implies that Local SGD needs at least $\Omega(\kappa)$ communications rounds, as opposed to constant communication rounds, which is what Local SGD needs in the homogeneous setting (see Theorem 2 due to Woodworth et al. [11] and let $K\to\infty$).
>
> On the other hand, the rate implied by our Theorem 2 in the same setting is:
>
> $$\mathbb{E}[\|\|\hat x - x^\star\|\|_2^2] \leq e^{-\frac{\mu KR}{2H}}B^2 +  \textcolor{red}{\frac{\sigma^2}{\mu^2 MKR}} + \textcolor{blue}{\frac{\tau^2H^2\phi^2}{\mu^4R^2}} + \textcolor{blue}{\frac{H^4\zeta^2}{\mu^4R^2}} + \textcolor{blue}{\frac{H^2\tau^2\sigma^2}{\mu^6KR^3}} +  \textcolor{red}{\frac{H^2\sigma^2}{\mu^4KR^2}}\enspace, \quad \text{for}\enspace \textcolor{green}{R = \tilde\Omega\left(\frac{H\tau}{\mu^2} \right)}\enspace.$$
>
> As opposed to Kolsokova et al. [18], our rate's requirement on communication rounds reduces to a constant when $\tau\to 0$. This is because Kolsokova et al. [18]'s analysis **cannot benefit from $\tau \ll H$**. Furthermore, note that when heterogeneity is low ($\zeta, \tau, \phi\to 0$) and $K\to\infty$, then our convergence rate may become arbitrarily small, meaning that we can also prove the benefit of local update steps over mini-batch SGD in regimes of low data heterogeneity. This matches the expectation in the homogeneous setting. The results of Woodworth et al. [20] and Patel et al. [25] also predict that this should occur in regimes of low data heterogeneity; however, they require the restrictive assumption of uniformly bounded gradient heterogeneity.
>
> Furthermore, since we make more fine-grained assumptions, note that in the regime where $\tau=0$ and $\zeta\ll\phi$, our rate is significantly better than theirs, even if we ignore our improved communication complexity. This is why we demarcate between $\tau, \zeta, \phi$, to delineate all relevant sources of data heterogeneity. We also want to underline how our paper fills the quadratic v/s non-quadratic gap by studying rates under third-order smoothness.
>
> We hope this addresses the reviewer's concern. We assure the reviewer that we are **very familiar** with Kolsokova et al. [18]'s paper as well as all the follow-up work. If our paper is accepted, we will ensure that it includes a table summarizing the convergence rates from the seminal papers in this area. Currently, we have some discussion after Theorem 2, but we will make it more explicit by converting it to a remark.
>
>
> ### **Why We Need Strong Convexity**
>
> Firstly, we would like to point out that our lower bound is intended and stated for general convex functions, so not all our results assume strong convexity.
>
> Our upper bounds do assume strong convexity, but we believe that understanding the role of data heterogeneity for strongly convex functions is worthwhile, and many existing works (including Kolsokova et al. [18]) also consider strongly convex functions for that reason. From a practical point of view, convex optimization problems often exhibit strong convexity due to regularization (for instance, consider logistic regression with an L2 penalty).
>
> Furthermore, we believe understanding the precise rates for strongly convex functions is an essential step towards extending our analyses to general convex functions. As the reviewer can note about our analyses, they are extremely non-trivial and require considerable care in writing the correct recursions, tuning the step size, and adjusting other hyperparameters. We attempted to use our strongly convex analysis as a stepping stone, hoping to perform a convex-to-strongly-convex reduction —a common approach in the optimization literature. Unfortunately, the parameter dependence we obtained was not ideal, which is why we did not report those results (see lines 244-254).
>
> We believe that removing the strong convexity assumption is a non-trivial task. For instance, we cannot state a version of Lemma 2 when we do not have strong convexity, as the machines may converge to solutions that are far apart from each other between communication rounds. Strong convexity and some control on first-order heterogeneity precisely avoid this problem. There is also no existing work that provides similar guarantees in the convex setting. For instance, the rate due to Patel et al. [25] (see their Theorem 3) requires the more stringent uniform bounded first-order heterogeneity. In the non-convex setting, Patel et al. [30] provide a rate that improves with a small $\tau$, but for a more complicated **variance-reduced version of Local SGD**. It is unclear what led to improved rates in their paper: local updates or the particular variance reduction.
>
> That said, we have a few ideas to extend our work to the convex setting. This would involve deriving a version of Lemma 2 in terms of the gradient norms on each machine. However, so far, we have not been able to make this coupled recursion work. **It is possible that, without strong convexity, the kinds of improvements shown in our paper may not be achievable for Local SGD**.  We are also unsure about how to prove such a hardness result, as incorporating strong convexity into our construction for Theorem 1 does not imply this. Another clue as to why this may be challenging comes from the quadratic setting. As discussed in Appendix C.3 in Proposition 2, for convex quadratic functions, the fixed point of Local SGD is the minimum norm interpolator for the problem under a different geometry. It is unclear whether controlling just second- and first-order heterogeneity will reduce the fixed-point discrepancy in the directions "not shared" or "mis-aligned" (due to non-commutativity of Hessians) between the machines.
>
> Hopefully, this discussion highlights that our strong convexity assumption is not a careless oversight, and the general convex setting is a significant technical extension. It remains an open problem whether such rates can even be obtained, and as such, our paper only partly answers the conjecture of Patel et al. [25]. We are actively investigating how to close this critical gap in the literature.
>
> ### **Summary**
>
> We hope that we have addressed the reviewer's concern. Since, besides these issues, the reviewer had a positive impression, we sincerely hope they will consider raising their score and vouch for accepting our paper.

---

> > ### Author Response · Authors · 2025-08-04
> >
> > Dear Reviewer,
> >
> > Please let us know if our clarification about the result of Kolosokova et al. [18] addresses your concern. We also provide additional discussion on Kolosokova et al. [18]'s communication complexity in our response to Reviewer ag4U.
> >
> > We are happy to address any other concerns the reviewer may have and thank them again for their feedback.

---

> > > ### Comment · Reviewer_un1E · 2025-08-05
> > >
> > > Thank you for the clarification that addresses my concern. I will make my final rating decision during the Reviewer–AC discussion phase.

---

> ### Comment · Area_Chair_g2SR · 2025-08-05
> **Response to rebuttal**
>
> Dear reviewer,
>
> As you know, the authors have submitted their rebuttals. Please take some time to carefully read and respond to them. It is crucial to engage in the discussion with the authors before acknowledging that you have read the rebuttal.
>
> Regards,
> AC

---

### Decision · Program_Chairs · 2025-09-17

**Decision:**

Accept (poster)

**Comment:**

This paper investigates distributed optimization using Local SGD with full node participation, focusing on its advantages over mini-batch SGD in heterogeneous settings. The authors present a refined theoretical analysis for convex objectives with higher-order smoothness by introducing a second-order measure of data heterogeneity. They establish matching upper and lower bounds on convergence complexity, showing that Local SGD achieves superior communication efficiency, particularly in low-heterogeneity regimes. The work provides a valuable theoretical explanation for the empirically observed benefits of Local SGD and is clearly written. The analysis offers meaningful insights and has the potential to be extended to other algorithms. Overall, the paper brings new theoretical understanding to a widely studied method.

One final remark for the authors: I strongly recommend including a table that compares prior Local SGD rates with the current improvements and the benefits of local steps. Such a table would make the contribution of this work more transparent and accessible to readers.